# Proteogenomics of clear cell renal cell carcinoma response to tyrosine kinase inhibitor

Hailiang Zhang [1,2,6], Lin Bai [1,6], Xin-Qiang Wu[1,2,6], Xi Tian[1,2,6], Jinwen Feng [1,6], Xiaohui Wu [1,6], Guo-Hai Shi[1,2,6], Xiaoru Pei[1,6], Jiacheng Lyu[1,6], Guojian Yang[1], Yang Liu [1], Wenhao Xu[1,2], Aihetaimujiang Anwaier[1,2], Yu Zhu[1,2], Da-Long Cao[1,2], Fujiang Xu [1], Yue Wang[1,2], Hua-Lei Gan[2,3], Meng-Hong Sun[2,3], Jian-Yuan Zhao [4,5] ✉, Yuanyuan Qu[1,2] ✉, Dingwei Ye [1,2] ✉ & Chen Ding [1] ✉

The tyrosine kinase inhibitor (TKI) Sunitinib is one the therapies approved for advanced renal cell carcinoma. Here, we undertake proteogenomic profiling of 115 tumors from patients with clear cell renal cell carcinoma (ccRCC) undergoing Sunitinib treatment and reveal the molecular basis of differential clinical outcomes with TKI therapy. We find that chromosome 7q gain-induced mTOR signaling activation is associated with poor therapeutic outcomes with Sunitinib treatment, whereas the aristolochic acid signature and *VHL* mutation synergistically caused enhanced glycolysis is correlated with better prognosis. The proteomic and phosphoproteomic analysis further highlights the responsibility of mTOR signaling for non-response to Sunitinib. Immune landscape characterization reveals diverse tumor microenvironment subsets in ccRCC. Finally, we construct a multi-omics classifier that can detect responder and non-responder patients (receiver operating characteristic–area under the curve, 0.98). Our study highlights associations between ccRCC molecular characteristics and the response to TKI, which can facilitate future improvement of therapeutic responses.

In 2020, 431,288 people were newly diagnosed with kidney tumors and 179,368 patients with kidney tumors died worldwide[1]. Clear cell renal cell carcinoma (ccRCC) represents approximately 70% of kidney tumor cases in adults[2]. Analyses of The Cancer Genome Atlas (TCGA) and the Clinical Proteomics Tumor Analysis Consortium (CPTAC) indicate that chromosome 3p loss, chromosome 5q gain, and chromosome 7 gain are the main copy number alterations of ccRCC[3,4]. Functional loss of

von Hippel Lindau (*VHL*) (located in chromosome 3p) in ccRCC results in the accumulation of hypoxia-inducible factor (HIF)[5], which results in the upregulation of vascular endothelial growth factor (VEGF) and platelet-derived growth factor (PDGF). Many studies have further revealed that ccRCC is a hyper-angiogenic tumor due to *VHL* inactivation-induced overexpression of VEGF. Based on this molecular mechanism, tyrosine kinase inhibitors (TKIs) targeting the VEGF

[1]Department of Urology, Fudan University Shanghai Cancer Center, State Key Laboratory of Genetic Engineering, Collaborative Innovation Center for Genetics and Development, School of Life Sciences, Qingdao Institute, Institutes of Biomedical Sciences, and Human Phenome Institute, Fudan University, Shanghai 200433, China. [2]Department of Oncology, Shanghai Medical College, Shanghai Genitourinary Cancer Institute, Shanghai 200032, China. [3]Tissue Bank & Department of Pathology, Fudan University Shanghai Cancer Center, Shanghai 200032, China. [4]Institute for Developmental and Regenerative Cardiovascular Medicine, MOE-Shanghai Key Laboratory of Children's Environmental Health, Xinhua Hospital, Shanghai Jiao Tong University School of Medicine, Shanghai 200092, China. [5]Department of Anatomy and Neuroscience Research Institute, School of Basic Medical Sciences, Zhengzhou University, Zhengzhou 450001, China. [6]These authors contributed equally: Hailiang Zhang, Lin Bai, Xin-Qiang Wu, Xi Tian, Jinwen Feng, Xiaohui Wu, Guo-Hai Shi, Xiaoru Pei, Jiacheng Lyu. ✉e-mail: zhaojy@fudan.edu.cn; quyy1987@163.com; dwyeli@163.com; chend@fudan.edu.cn

signaling pathway have been broadly used to treat metastatic and recurrent RCC, and have significantly prolonged the overall survival (OS) and progression-free survival (PFS) of patients[6]. However, not all patients do respond to TKI therapy or benefit from these treatments, and most patients finally develop resistance. It is thus necessary to further clarify the association between TKI efficacy and *VHL* mutations.

Memorial Sloan Kettering Cancer Center (MSKCC) risk and International Metastatic RCC Database Consortium (IMDC) risk models, based on clinical features, have been applied for the prognostic prediction of advanced RCC in the last decade[7,8]. However, at the molecular level, the impact of these risk categories on prognosis remains unclear. Beyond these, many studies have explored the predictive response markers and unraveling resistance mechanisms of TKI therapy, predominantly using genomic and transcriptional approaches[9–12]. Clinical feature-based prognostic models and omics studies have been conducted in Western populations; however, similar research is lacking in Chinese populations, and there are rare studies using comprehensive proteogenomic characterization to investigate biomarkers for predicting TKI treatment response.

Aristolochic acid (AA) is prevalently used in traditional herbal medicine in Asia[13–15]; AA exposure may cause mutagenesis characteristic of predominant T > A transversions, which match COSMIC SBS22[16–19]. Several studies have demonstrated the potential association between AA exposure and ccRCC oncogenesis[20,21]. Thus, exploring the influence of AA exposure on TKI efficacy in ccRCC might be beneficial for Asian patients.

In this study, comprehensive proteogenomic characterization of treatment-naïve tumors and paired normal adjacent tissues is performed to elucidate the association between clinical features, genomic alterations, proteomic features, and TKI therapeutic efficacy in a Chinese population. We also elucidate the associations of clinical features with tumor molecular phenotypes and patient prognosis. Meanwhile, we find that mTOR signaling and inflammatory response were the prevalent features correlated with Sunitinib resistance. A robust response classifier is constructed based on multi-omics data and its stability is validated in another cohort. Our findings shed light on personalized therapy in metastatic ccRCC.

## Results

### Proteogenomic analysis of ccRCC response to sunitinib

We retrospectively collected paired tumor and tumor-adjacent ccRCC samples based on strict criteria from 115 Chinese patients treated with Sunitinib, comprising 68 advanced and 47 recurrent tumors for proteogenomic analysis (Fig. 1a). The 47 recurrent cases were localized ccRCC at the time of surgery and were enrolled due to the subsequent development of metastatic disease. Pathological examinations were performed to determine the tumor and tumor-adjacent tissue regions (Methods). Comprehensive clinicopathologic data were available for the patients (Fig. 1b and Supplementary Data 1). We divided the patients into responders (complete response [CR] and partial response [PR], $n = 27$) and non-responders (stable disease [SD] and progressive disease [PD], $n = 88$), based on the response evaluation criteria in solid tumors (RECIST). Examination of patients' clinical parameters showed that responders and non-responders presented highly significant differences in both progression-free survival (Fig. 1c, log-rank $p < 0.0001$) and overall survival (Fig. 1c, log-rank $p < 0.0001$). The fresh frozen tissues were then dissected, used for sample preparation, and submitted for whole exome sequencing (WES) ($n = 113$), transcriptome sequencing ($n = 94$), proteome ($n = 115$), and phosphoproteome ($n = 66$) identification (Fig. 1a).

For genomic analysis, 6487 somatic mutations were identified by comparison of the tumors with tumor-adjacent samples (Supplementary Data 2). Among 113 patients, 19 significantly mutated genes were identified, including the most frequently mutated genes, *VHL* (65%), *PBRM1* (35%), *BAP1* (16%), and *SETD2* (14%) (Supplementary

Fig. 1a). We compared the mutation frequencies of frequently mutated genes in ccRCC in eight ccRCC datasets, including this study, another two Chinese cohort[22,23], Japanese cohort[24], JAVELIN Renal 101 cohort[25], IMmotion151 cohort[12], European cohort[26] and TCGA cohort[3] (Fig. 1d). The results showed that the mutation frequencies of these genes were similar in three Chinses cohorts, while TCGA cohort and Japanese cohort had relatively lower mutation frequencies of these genes.

As for the transcriptome, we identified 12,276 protein-coding genes with median fragments per kilobase of transcript per million fragments mapped (FPKM) of more than 1 (Supplementary Data 3). Label-free proteomics identified 12,310 proteins with an average of 6585 proteins in the tumor samples and 5753 proteins in the tumor-adjacent tissues (Supplementary Fig. 1b). The transcriptome and proteome showed moderate correlation with sample-wise median Spearman's correlation coefficient (SCC) of 0.39 in this study (Supplementary Fig. 1c), consistent with the previous study[27]. There was no obvious difference in the proteomic coverage between the responders and non-responders. In total, 9641 proteins were identified in both responders and non-responders, and 442 and 2226 proteins were identified specifically in the non-responders and responders, respectively (Supplementary Fig. 1d). Proteins were identified in at least 25% of the samples were used for analysis (Supplementary Data 4). Proteins exclusively detected in responders and non-responders failed to pass the criterion of 25% cut-off, and were not included in the further analysis. We identified 4862 phosphosites in each sample on average, and 37,055 phosphosites in total, corresponding to 7502 phosphoproteins (Supplementary Fig. 1e). Among these phosphosites, 6749 phosphosites were identified in more than 25% of samples was used for further analysis. In addition, to evaluate whether the associations of protein abundances and prognosis were independent of therapy, we collected 37 cases of advanced ccRCC samples who have not received any treatment after surgery, as control (Supplementary Data 4). The expression levels of 270 proteins were significantly associated with patient survival in the Sunitinib treatment cohort, and the expression levels of 630 proteins were significantly associated with patient survival in the non-treated control group. Compared to these 630 prognostic markers in a non-treated group with 270 prognostic markers in the Sunitinib treatment group, only 19 proteins were found in both groups (Supplementary Fig. 1f), indicating the prognostic values of protein expression levels were profoundly dependent on whether the patients have ever been prescribed Sunitinib treatment after surgery.

### The clinical features associated with sunitinib treatment outcomes

We first conducted a differential analysis of clinical baseline features between responders and non-responders. It was observed that the tumor, nodes, and metastasis (TNM) stage (stage I-III were recurrent RCCs, stage IV were metastatic RCCs) and International Society of Urologic Pathologists (ISUP) grade and tumor size did not show any significant difference between responders and non-responders (Supplementary Fig. 2a). Other clinical information, including chronic disease status such as hypertension, diabetes, Karnofsky performance score (KPS, a clinical assessment tool used to assess the overall health of patients, Methods), neutrophilic granulocyte (GRAN) count, lymphocyte (LYM) count, eosinophil (ESO) count, platelet (PLT) level, hemoglobin (Hb) level, serum lactate dehydrogenase (LDH) level, serum calcium (Ca) level, serum creatinine (CRE) level, blood urea nitrogen (BUN) level, urine protein state, estimated glomerular filtration rate (eGFR) (Methods) were also collected. None of these clinical parameters was significantly different between the responder and non-responder groups (Supplementary Fig. 2b, c). However, we found that tumor size was differentially distributed in the responder and non-responder groups (Supplementary Fig. 2d). There were more

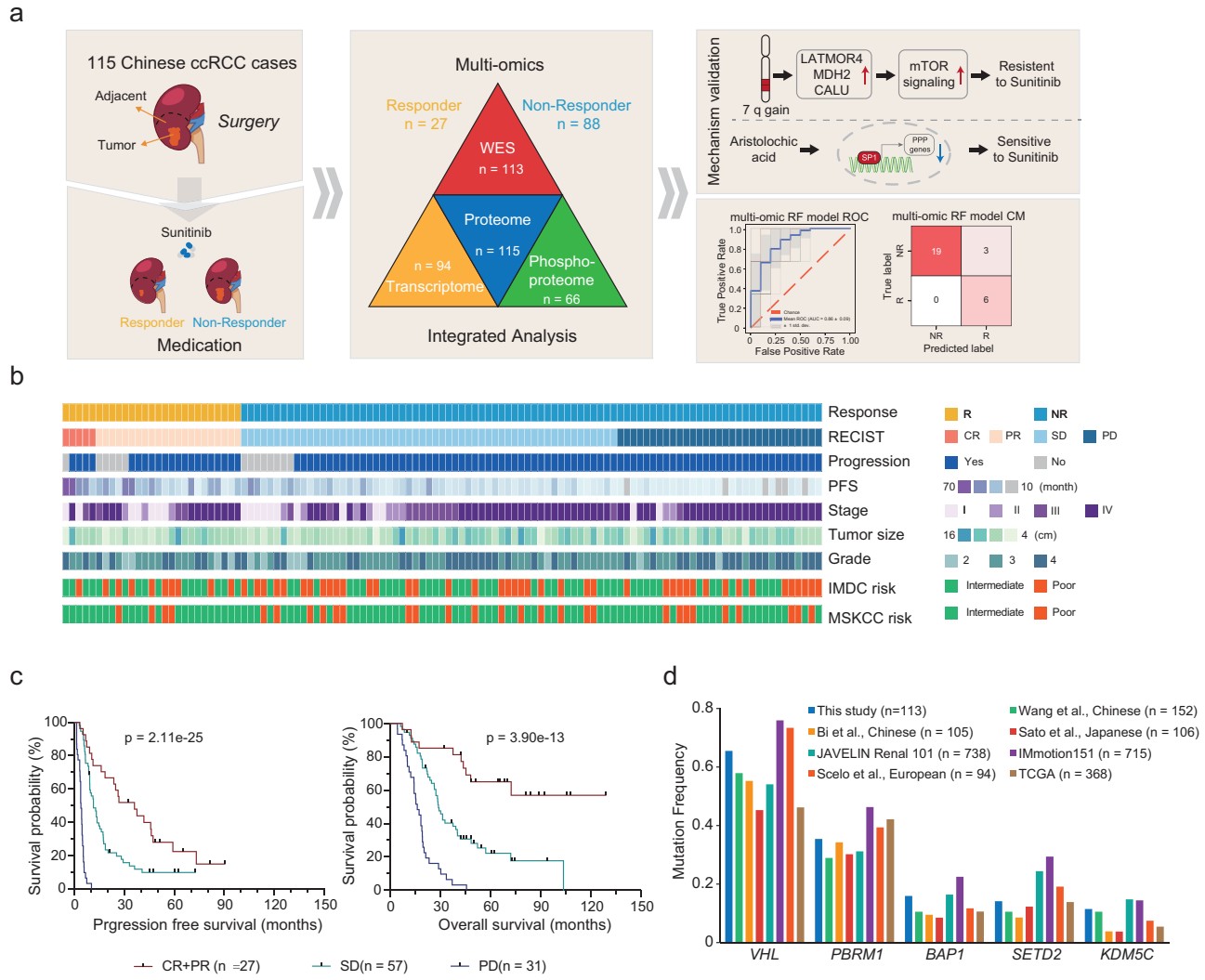

**Fig. 1 | Proteogenomic analysis of ccRCC response to sunitinib. a** Schematic representation of the multi-omics analyses of ccRCC, including sample preparation, protein identification, WES, and function verification. **b** The cohort includes 27 responders and 88 non-responders undergoing sunitinib treatment. Their clinical parameters are shown in the heatmap. **c** Kaplan–Meier curves of progression-free survival (PFS) and overall (OS) for patients with distinct clinical responses (two-sided log-rank test). *P* values were described in the figure. **d** Comparison of frequently mutated genes among ccRCC cohorts. Source data are provided as a Source Data file.

responders compared to non-responders when the tumor size was smaller than 5 cm, whereas there were more non-responders than responders when the tumor size was larger than 9 cm (Supplementary Fig. 2d). Therefore, we divided the patients into three groups based on tumor size: small (tumor size <5 cm; *n* = 16), median (tumor size >5 cm and <9 cm; *n* = 61), and large (tumor size >9 cm; *n* = 38) (Supplementary Fig. 2e). Interestingly, the distribution of different tumor size groups was significantly different in responders and non-responders (Chi-square test, *p* = 0.007) (Supplementary Fig. 2e). Moreover, the PFS of three tumor size groups was significantly different (*p* = 0.006), among which large tumors showed significantly poor prognosis (Supplementary Fig. 2f).

Univariate analysis based on PFS and OS was performed, to explore the associations between other clinical features and patient clinical outcomes. We found that tumor size, PLT level, and LDH level were significantly correlated with PFS/OS (Supplementary Data 1). To elucidate the biological basis, we calculated SCC between clinical features and the ssGSEA score of gene sets (Hallmark, KEGG, and Reactome databases). As a result, the proteomic pathways, complement and coagulation cascades (SCC = 0.24, *p* = 0.01), epithelial-mesenchymal transition (SCC = 0.23, *p* = 0.01), and inflammatory

response (SCC = 0.21, *p* = 0.02) were positively correlated with tumor size. An inflammatory response (SCC = 0.22, *p* = 0.02), innate immune system (SCC = 0.27, *p* < 0.01), neutrophil degranulation (SCC = 0.27, *p* < 0.01), and RAB regulation of trafficking (SCC = 0.25, *p* < 0.01) were positively correlated with the PLT level. Energy-dependent regulation of mTOR by LKB-AMPK (SCC = 0.24, *p* < 0.01), and fatty acid metabolism (SCC = 0.24, *p* < 0.01) were positively correlated with the LDH level (Supplementary Fig. 3a). As a complementary approach, we built a network of proteins with significant correlations of these clinical features (Supplementary Fig. 3b). The defining proteomic pathway, positively correlated with tumor size, was complement cascades. Vesicle-mediated transport was recurrently identified as a positively correlated pathway with plasma PLT levels (Supplementary Fig. 3b). Tumor size-related proteins were involved in angiogenesis (VCAN, VTN, NRP1), EMT (FN1, CD44, THBS2), and translation (EIF1AX, EIF5), further indicating the biological basis of ccRCC growth (Supplementary Fig. 3c). In addition, acute phase proteins (ORM1, CRP, SAA1) and complement cascade components (SERPINA1, C5, FGA) showed an obviously significant correlation with tumor size, reflecting a positive connection between inflammatory response and tumor size (Supplementary Fig. 3a, d).

Furthermore, we found that CRP abundance was significantly correlated with inflammatory response scores (SCC = 0.31, $p$ = 8.1E-4) and could act as a prognostic indicator in Sunitinib treatment (log-rank test, $p$ = 0.021) (Supplementary Fig. 3e, f). Consistently, CRP abundance in the non-responders was significantly higher than those in the responders (Supplementary Fig. 3g). Although the LDH levels of the responder and non-responder groups showed no significant differences, higher LDH levels were associated with poor prognosis (Supplementary Data 5). Higher LDH levels were associated with a higher level of fatty acid degradation (ACOT4, ECHS1) and OXPHOS (NDUFB5, NUDFS2, SDHB) (Supplementary Fig. 3h). Interestingly, abundances of CD8A and CD274 (PD-L1) were significantly correlated with the LDH level (Supplementary Fig. 3i), suggesting enhanced immune evasion in ccRCC patients with high LDH levels. To test whether the plasma LDH levels were associated with the potential immune evasion of ccRCC tumors, we collected 72 ccRCC tumor samples and conducted PD-L1 immunohistochemistry. Comparing the preoperative plasma LDH level between PD-L1 positive and negative patients, we found that PD-L1 positive patients had higher plasma LDH levels (Supplementary Fig. 3j), further supporting the association between plasma LDH level and the potential immune evasion of ccRCC tumors. Further, higher expression of CD274 was associated with poor prognosis in Sunitinib treatment (log-rank test, $p$ = 0.005) (Supplementary Fig. 3k), consistent with a previous report[28]. Based on these results, we considered that patients with high LDH levels might not benefit from sunitinib therapy, but could benefit from immune checkpoint blockade (ICB) therapy.

### MSKCC and IMDC risk models to prognosticate in advanced RCC

MSKCC and IMDC risk models were frequently used for prognosis prediction of advanced RCC. Despite there was no significant difference in MSKCC risk and IMDC risk between Responders and Non-Responders, MSKCC risk and IMDC risk were significantly associated with patient survival (Supplementary Fig. 4a). Gene set enrichment analysis (GSEA) revealed that poor MSKCC risk patients showed upregulation of complement cascade and lipoprotein assembly, and downregulation of collagen formation and glycolysis compared to intermediate-risk patients (Supplementary Fig. 4b). Similarly, upregulated complement cascade and lipoprotein assembly and downregulated glycolysis and apoptosis were observed in poor IMDC risk patients (Supplementary Fig. 4c). Leibovich scoring system was also widely applied to predict patient prognosis with clear cell metastatic renal cell carcinoma[29]. The results showed that Non-Responders had higher Leibovich scores than Responders (Supplementary Fig. 4d). In addition, the Leibovich scoring system could predict patients' prognosis in this cohort, and a higher score was significantly associated with worse OS and PFS (Supplementary Fig. 4e).

### Impacts of mutation signatures on sunitinib therapeutic outcomes

Mutational spectra revealed that the frequency of T > A transversions was higher in the non-responder group than in the responder group (14.4 vs. 20.3%) (Supplementary Fig. 5a). AA exposure may cause mutagenesis characteristic of predominant T > A transversions[16,18], which matched the Catalog of Somatic Mutations in Cancer (COSMIC) SBS22. Thus we further decomposed the mutation spectra using the COSMIC database[30]. The results showed that Responder and Non-Responder groups had similar single-base substitution (SBS) signature (SBS1, SBS5, SBS22, and SBS40) distribution (Supplementary Fig. 5b). Interestingly, patients with SBS22, were associated with aristolochic acid (AA) exposure, showed better survival in Sunitinib treatment (GB-Wilcoxon test, $p$ = 0.038) (Fig. 2a). Consistently, compared with the non-AA group, patients with the AA signature showed smaller tumor size ($t$-test, $p$ = 0.0062) (Fig. 2b), suggesting the specific mechanism of AA exposure.

Further investigation of the proteomic impact of AA signature showed that, compared with the patients without AA signature, proteins involved in the immune system (MRC1, CSK), focal adhesion-PI3K-AKT-mTOR-signaling pathway (ITGB2, RPTOR), DNA replication (MCM5, MCM6), and pentose phosphate metabolism (TKT, PGD) were significantly downregulated, whereas the proteins functioning in biological oxidation (MAT2A) and glycolysis (PGAM2, PFKM) were upregulated in patients with AA signature (Fig. 2c). The downregulated DNA replication might account for the smaller tumor size of patients with AA signature. Interestingly, we found that the enhanced glycolysis and attenuated pentose phosphate pathway (PPP) in patients with AA signature, manifested as upregulation of PFKP, PFKL, GPI ($t$-test, $p$ < 0.05) and downregulation of G6PD, PGD, TKT ($t$-test, $p$ < 0.05) both on proteome and transcriptome level (Fig. 2d). Further, low expression of PGD and TKT was associated with better survival in Sunitinib treatment group, which was not observed in the non-treated control group (Supplementary Fig. 5c, d). Thus, we supposed that AA improved the prognosis of ccRCC patients under Sunitinib treatment by downregulation PPP.

To further validate the impacts of AA on Sunitinib therapy, we performed in vitro experiments in 786-O and ACHN cells. We found that in vitro treatment of ccRCC cells with proper concentration of AA had no direct effect on the proliferation of ccRCC cells, but could enhance the inhibitory effect of sunitinib on ccRCC cells (Fig. 2e, f, Methods). The transwell assay showed that AA treatment did not impact cell invasiveness directly, but enhanced the inhibition of Sunitinib to cell invasiveness (Supplementary Fig. 5e). Moreover, we verified that treating 786-O, and ACHN cells with AA caused notably decreased PPP enzymes (G6PD, PGD, and TKT) in time- and dose-dependent manners (Fig. 2g, h). The levels of ribose-5-phosphate, which is an important product in both oxidative and non-oxidative PPP, decreased notably in AA-treated cells (Fig. 2i).

We next investigated how AA inhibited the expressions of those genes, by predicting the potential transcriptional factors (TFs) which were involved in the regulation of the transcription of G6PD, PGD, and TKT. We obtained 84 TFs that can bind to G6PD, PGD, and TKT gene promoter regions by the JASPAR[31] database and 200 potential TFs by the ChEA3[32] database (Methods). To improve the confidence of predicted TFs, we take the intersection of the 84 TFs in JASPAR and 200 TFs in ChEA3 databases. Ultimately, we obtained 18 TFs including TFE3, SP1, SPI1, GATA1, NR2C1, KLF4, TCF3, TEAD4, TFAP2A, KLF5, GATA3, SMAD4, ISX, YY1, SNAI3, SREBF2, MEF2A, and VDR. In this study, there were 8 out of 18 TFs identified in this ccRCC cohort. As the bioinformatic analysis showed the inhibition of AA on the pentose phosphate pathway (PPP), we hypothesized the expression level of TFs that regulated PPP affected by AA was downregulated in tumor tissue. Therefore, we assessed the expression of these 8 TFs in RCC, the results showed four out of eight TFs were lower expressed, including SP1, YY1, SREBF2, and SMAD4 (Supplementary Fig. 5f). Hence, we selected these four TFs for further in vitro experiments. By knocking down these four candidate transcriptional factors using siRNA in cultured ccRCC cancer cells (Supplementary Fig. 5g–k), we found the mRNA levels of G6PD and PGD decreased in SP1 knocking down cells (Supplementary Fig. 5g). However, decreased mRNA levels of SMAD4, YY1, and SREBF2, did not affect the transcription of G6PD, PGD, and TKT (Supplementary Fig. 5h–j). Consistently, we observed the downregulation of G6PD and PGD proteins in SP1 knocking down cells (Fig. 2j, k). Furthermore, we overexpressed SP1 in cultured cells and found SP1 overexpression led to increased levels of mRNA of G6PD, PGD, and TKT (Supplementary Fig. 5k) and protein levels of G6PD and PGD (Fig. 2j, k). When treated with AA, we found that the SP1 mRNA level was not changed, but the SP1 protein level was downregulated, indicating AA might lead to SP1 degradation in the post-translation stage (Fig. 2l, m). We verified this result by the reduction of SP1, which was caused by AA exposure, was blocked by

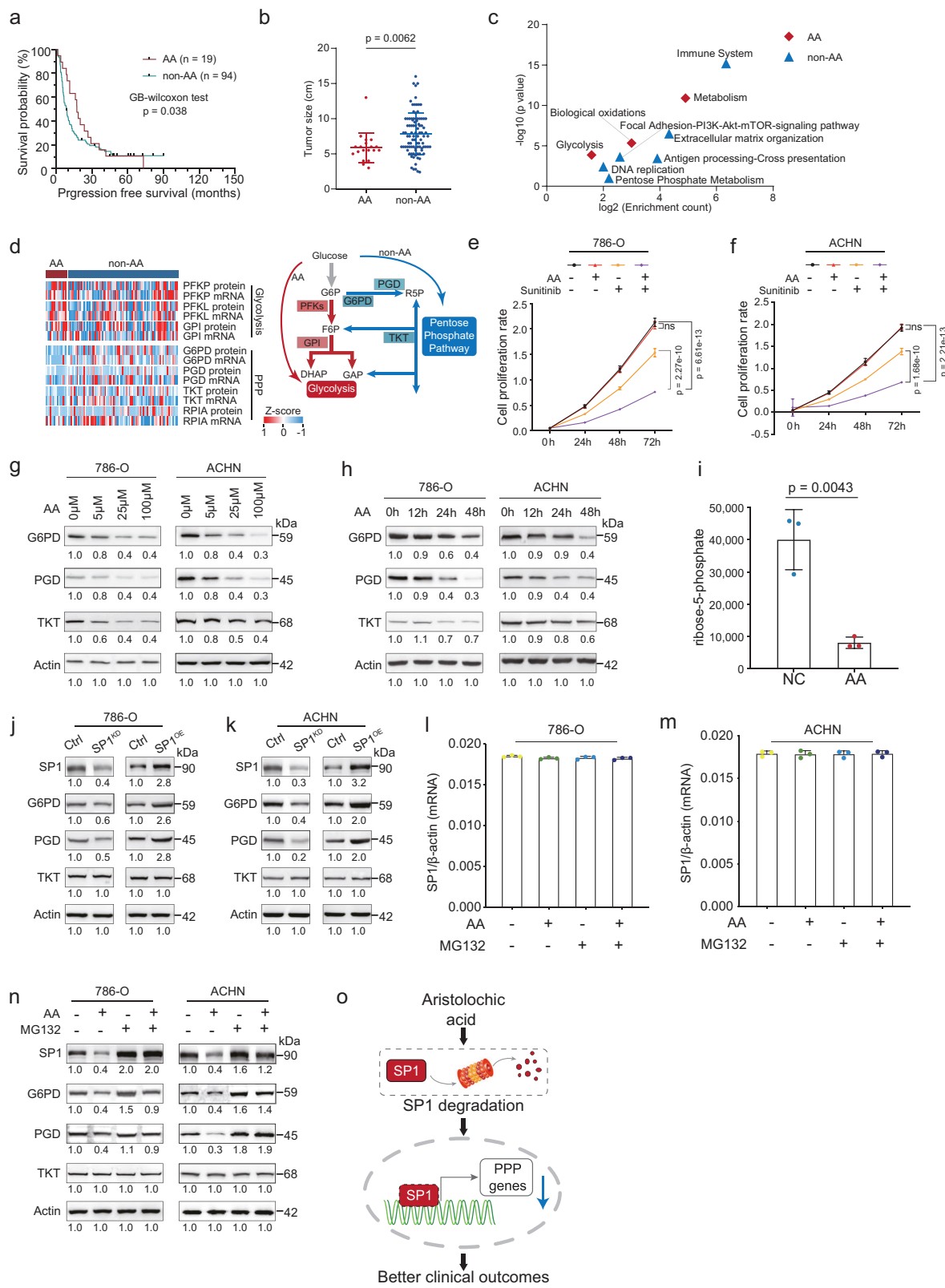

the proteasome inhibitor MG132 (Fig. 2n). We used the target genes of SP1 from DoRothEA[33] to infer the SP1 activities in ccRCC tumor using VIPER[34]. To evaluate the SP1 signature between responder (R) and non-responder (NR), we split all samples into two groups by the median score of SP1 signature (SP1 signature (high) and SP1 signature (low)) and performed the Fisher exact test. The results showed there's no significant difference between R and NR, but a significantly

different between AA and non-AA, which indicated the association between SP1 and AA signature (Supplementary Fig. 5l). Taken together, these results indicated that there were linkages among AA exposure, Sunitinib treatment outcome, and the downregulation of pentose phosphate pathway. More importantly, we verified that SP1 was the key transcription factor of AA regulating the pentose phosphate pathway (Fig. 2o).

**Fig. 2 | Impacts of AA exposure on sunitinib therapeutic outcomes.**
**a** Kaplan–Meier curves of progression-free survival (PFS) for patients with ($n = 19$) or without ($n = 94$) the AA signature (two-sided GB-Wilcoxon test). **b** Comparisons of tumor size between patients with ($n = 19$) or without ($n = 94$) the AA signature (two-sided $t$-test). Data were shown as mean ± SD. **c** Pathways enriched by differentially expressed proteins between patients with or without the AA signature. **d** Left panel: heatmap showed the expression of glycolysis-related and PPP-related proteins at the transcriptome and proteome levels, respectively. The values were transformed by $z$-score. Right panel: Glucose metabolism alteration caused by the AA signature. **e, f** Proliferation of 786 O and ACHN cells was detected by CCK-8 assay (AA-100 μM, Sunitinib-200 nM, $n = 9$ independent experiments, two-sided $t$-test). $P$ values were described in the figure. ns not significant. Shown are the average values with SD. **g, h** AA treatment inhibited the expressions of G6PD, PGD, and TKT in 786 O and ACHN cells in time-and dose-dependent manners. Numerical values below the gels indicate the quantification of the bands relative to the control (hereinafter). **i** Ribose-5-phosphate concentrations in cells treated with AA or not ($n = 3$ independent experiments, two-sided $t$-test). Data were shown as mean ± SD. $P$ values were described in the figure. **J, k** Left panel, SP1 knockdown downregulated PPP enzymes at the protein level. Right panel, SP1 overexpression upregulated PPP enzymes at the protein level, in 786 O and ACHN cells, respectively. Numerical values below the gels indicate the quantification of the bands relative to the control. **l, m** AA treatment did not affect the transcription of SP1 in the 786-O cell and ACHN cell, respectively ($n = 3$ independent experiments). Data were shown as mean ± SD. (AA-100 μM, 24 h). **n** MG132 abrogated the AA-mediated PPP enzymes downregulation by inhibiting the degradation of SP1. (AA-100 μM, 24 h). Numerical values below the gels indicate the quantification of the bands relative to the control. **o** Schematic diagram showing AA regulated the pentose phosphate pathway. Source data are provided as a Source Data file.

## The impacts of copy number alterations on sunitinib therapeutic outcomes

Somatic copy number alterations (SCNAs) in ccRCC were identified using GISTIC[35] (Supplementary Data 2). The most frequently deleted chromosomal regions in the Chinese cohort were 3p (75%), 14q (45%), 9p (44%), and 9q (44%). The most frequently amplified regions were 5q (44%), 7q (32%), 7p (29%), and 20q (28%). Comparing the responder group with the non-responder group, there was no significant difference in significant arm-level events (Supplementary Data 2). When Cox regression analysis was used to determine the correlation between arm-level CNAs and clinical outcomes, we found that gains of 3q (HR = 1.8, $p = 0.025$), 7p (HR = 2, $p = 2.48E-3$), 7q (HR = 2.1, $p = 7.12E-4$), and 8q (HR = 1.7, $p = 0.031$) were associated with poor survival (Fig. 3a). We performed multivariate analysis using 3q gain, 7p gain, 7q gain, and 8q gain as covariates. We found the co-occurrence of these four CNA events (gains of 3q, 7p, 7q, and 8q). Notably, 7p gain and 7q gain were even totally overlapped (Supplementary Fig. 6a). We calculated variance inflation factors (VIFs) of these four covariates to assess multicollinearity. The results showed that there was multicollinearity in this model (7p gain, VIF = 9.276; 7q gain, VIF = 9.826) (Supplementary Data 2). Therefore, we performed a multivariate analysis of 3q gain, 8q gain, and 7p or 7q gain. The results showed 7p gain (HR = 1.71, $p = 0.045$) and 7q gain (HR = 1.91, $p = 0.015$) were the dominant arm-level CNA events associated with patient survival (Supplementary Fig. 6b).

Comparing the CNA landscape at the gene level between the responder and non-responder groups, we found that genes, more frequently amplified in the non-responder group than in the responder group, were located in 7q (Fisher's exact test, $p < 0.05$) (Fig. 3b). To find out the effects of 7q copy number (CN) on sunitinib therapy at the proteome level, we calculated the SCC between 7q CN and proteome data. Totally, there were 485 proteins that showed significant positive correlations with the 7q CN (SCC > 0, $p < 0.05$). As shown in Fig. 3c, these proteins were mainly enriched in pathways including lysosome (LARP1. LARP2), innate immune system (PTGES), mTOR signaling (RRAGB, RRAGD, LAMTOR2, LAMTOR4, MLST8), TCA Cycle (CS, SDHA, MDH2), oxidative phosphorylation (COX6C, NDUFS6), and fatty acid biosynthesis (ACACA, MCAT) (Fig. 3c and Supplementary Fig. 6c). Notably, there were 29 out of 485 proteins encoded by 7q genes. Genomic alterations that affect gene expression levels at the same locus are defined to act in *cis*, whereas the impacts of another locus are defined as a *trans* effect[36,37]. The diagonal patterns in Supplementary Fig. 6d represent the *cis* effects of CNAs, whereas vertical patterns indicate the *trans* effects. We observed 3 (LAMTOR4, MDH2, CALU) out of 29 proteins showed *cis* effects both on proteome and transcriptome levels with their encoding genes on 7q (Fig. 3d), in which LAMTOR4, participating mTOR signaling, showed the most significant *cis* effects on 7q ($p = 1.88E-4$).

To further validate the impacts of chromosome 7q gain on Sunitinib therapy, we overexpressed five *cis*-regulated genes including LAMTOR4, MDH2, CALU, and two other random genes, TBL2 and POR2, as a control to mimic chromosome 7q gain in the in vitro cell experiments, respectively (Fig. 3e and Supplementary Fig. 6e). We found overexpressed LAMTOR4, MDH2, and CALU, rather than TBL2 and POR2, increased the phosphorylation level of S6K in the 786-O cell line, indicating activation of mTORC1 (Fig. 3e and Supplementary Fig. 6e). We investigated the TCGA ccRCC data and found that 7q gain tumors had a higher level of pS6K (P70S6KP T389, Wilcoxon rank-sum test, $p = 0.03$) and mTOR signaling scores than 7q WT tumors (Wilcoxon rank-sum test, $p < 0.01$) (Supplementary Fig. 6f). These results indicated the 7q gain was associated with the activation of mTOR signaling in ccRCC patients. We also performed immunohistochemical (IHC) staining for pS6K in responders and non-responders. The results showed that non-responders expressed higher levels of pS6K than responders (Fig. 3f). Next, we treated the candidate genes overexpressed cells and normal cells with or without Sunitinib (Methods) in 786-O and 769-P cell lines, respectively. By monitoring the cell proliferation rate and transwell assay, we found increased expression of LAMTOR4, MDH2, or CALU abrogated the effects of Sunitinib, while TBL2 and POR2 not (Fig. 3g and Supplementary Fig. 7a–c). In conclusion, chromosome 7q gain would activate mTOR signaling and link to the poor sunitinib treatment effectiveness (Fig. 3h).

## Gene mutation in responders and non-responders associated with therapy outcomes

In total, we identified ten genes (*VHL, KMT2C, SFT2D1, COL5A3, DYNC2H1, EYS, STAG3, HEATR1, PABPC5,* and *HLA-B*) (Supplementary Data 2) that were significantly differentially mutated between the responder and non-responder groups (Fisher's exact test, $p < 0.05$) (Fig. 4a). Interestingly, all ten genes showed higher mutation rates in the responders than in the non-responders. Further, we investigated the impact of mutations on the OS in our cohort. Mutations of eight genes with a mutation frequency >2% (*VHL, BAP1, ACAN, CNTNAP4, HUWE1, ZNF236, ZMYM4,* and *MYH1*) were associated with prognosis (log-rank test, $p < 0.05$). Among them, *BAP1* mutations were associated with poor survival in both the TCGA cohort and this study (Fig. 4b). In contrast, the association of *VHL, ACAN, CNTNAP4, HUWE1, ZNF236, ZMYM4,* and *MYH1* mutations with overall survival was only observed in our cohort (Fig. 4b). Notably, we found that *VHL* mutation, considered as a truncal genetic alteration event[3], was significantly associated with good prognostic outcomes (log-rank test, $p = 0.0135$, HR = 0.58) in sunitinib treatment (Fig. 4c). To prove the correlation between *VHL* mutation and sunitinib treatment outcomes, we surveyed the data from IMmotion151[12]. Consistently, it was observed responders were significantly overrepresented in patients with *VHL* mutations (Fisher's exact test, $p = 0.013$) (Supplementary Fig. 8a), and patients with *VHL* mutation have longer PFS than patients without *VHL* mutation (log-rank test, $p = 0.012$) in the IMmotion151 study (patients not evaluated were not included) (Supplementary Fig. 8b). We also found that

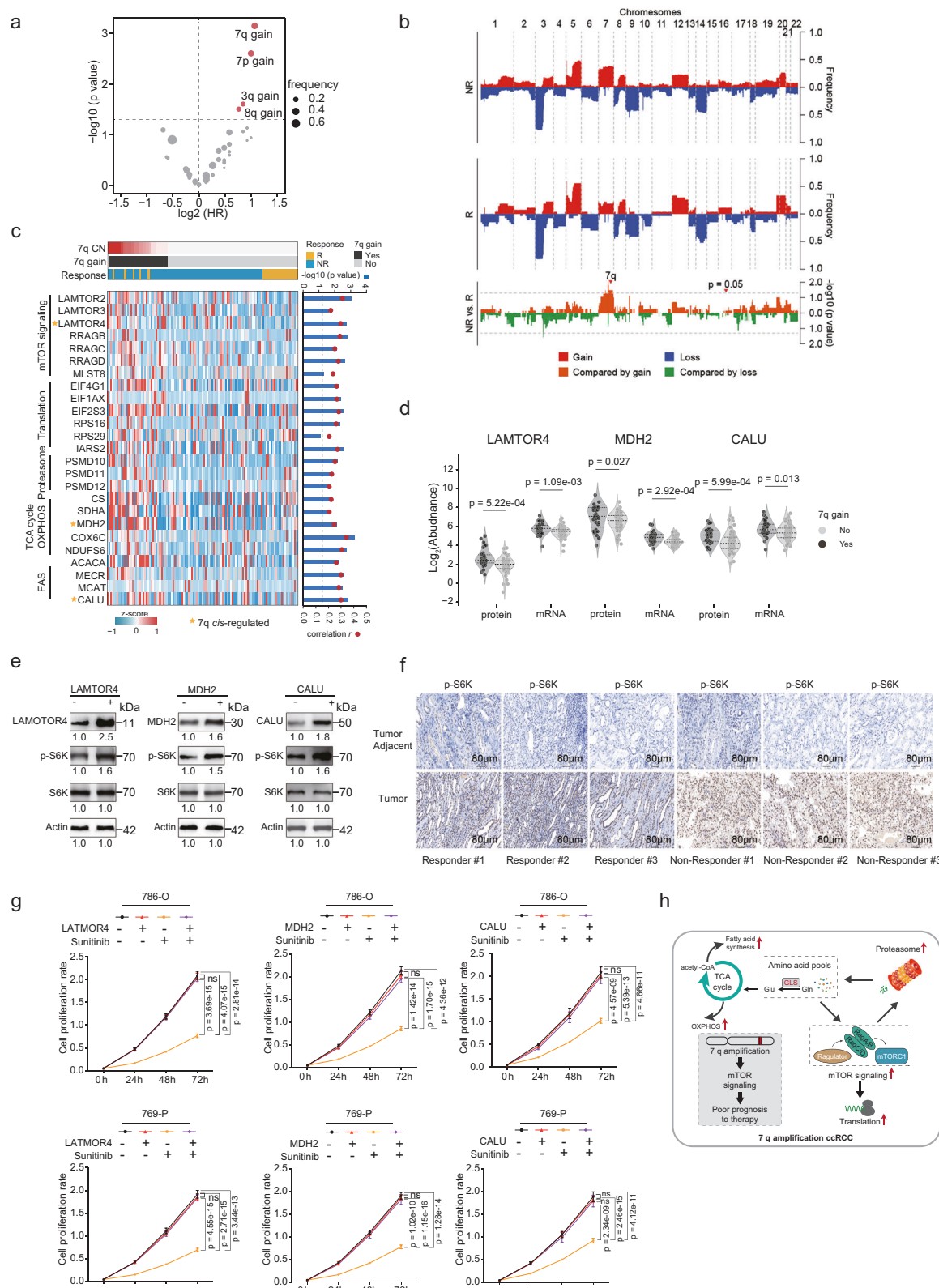

tumors with *VHL* mutation were significantly smaller (*t*-test, *p* = 0.0039) (Supplementary Fig. 8c).

To further investigate the impacts of *VHL* mutations on protein expressions and related biological functions, we examined the significantly altered proteins in patients with or without *VHL* mutation (FC >1.5, *p* < 0.05) both on proteome and transcriptome level (Fig. 4d and Supplementary Fig. 8d). Proteins in pathways such as glycolysis/

gluconeogenesis (ENO1, PGK1, and ALDOC), HIF-1 signaling (CA9, PLIN2), Ras signaling (RASAL1, RALB), and necroptosis and apoptosis (BCAP31, DFFB), were found to be elevated in patients with *VHL* mutation (Fig. 4e). *VHL* is an important tumor suppressor that is lost in the majority of ccRCC cases, acting as a component of E3 ligase mediating HIF degradation[38]. In *VHL* mutation tumors, accumulated HIF led to upregulation of glycolysis and downregulation of oxidative

**Fig. 3 | Impacts of copy number alterations on sunitinib therapeutic outcomes.** **a** Cox regression analysis of significant arm-level CNA events, based on the PFS. **b** Comparison of gene-level CNAs between Responders and Non-Responder in this cohort. The upper plot illustrates the frequency of CNA events and the lower plot illustrates the −log10 (p value) of each gene for the comparison of Responders and Non-Responder (two-sided Fisher's exact test). **c** Heatmap depicting the protein expression levels positively correlated with the Chromosome 7q copy number. Two-sided Spearman's correlations are shown in the right panel. The values were transformed by z-score. **d** Boxplots depicted the expression of LAMTOR4 (proteome level: 7q gain $n = 30$, non-7q gain $n = 55$; transcriptome level: 7q gain $n = 31$, non-7q gain $n = 62$), MDH2 (proteome level: 7q gain $n = 31$, non-7q gain $n = 62$; transcriptome level: 7q gain $n = 31$, non-7q gain $n = 62$), and CALU (proteome level: 7q gain $n = 31$, non-7q gain $n = 62$; transcriptome level: 7q gain $n = 31$, non-7q gain

$n = 62$) between 7q gain and non-7q gain cohort at transcriptome and proteome level, separately (two-sided Wilcoxon rank-sum test). The line represents the mean with SEM and upper and lower quartiles, respectively. P values were described in the figure. **e** Overexpression of LAMTOR4, MDH2, and CALU increased the phosphorylation of S6K. Numerical values below the gels indicate quantification of the bands relative to control. **f** The immunohistochemical (IHC) for pS6K in the tumor and tumor-adjacent tissue of responders and non-responders (analyzed patients: $n = 3$). The scale bar indicates 80 μm. **g** CCK-8 detected the effect of LAMTOR4, MDH2, and CALU overexpression and Sunitinib treatment on cell proliferation. (Sunitinib-200 nM, $n = 9$ independent experiments, two-sided t-test). Data were presented as mean values ± SD. P values were described in the figure. ns not significant. **h** Proposed model explaining the 7q gain-induced Sunitinib treatment non-response. Source data are provided as a Source Data file.

phosphorylation (OXPHOS), indicating activation of the Warburg effect (Fig. 4e, f). Moreover, to provide experimental evidence, we knocked down *VHL* in ACHN cells (80% knockdown efficiency, Supplementary Fig. 8e), which belong to *VHL*-proficient cells, mimicking *VHL* loss-of-function mutations. We tested the glycolysis levels by monitoring lactate production. The results showed that lactate levels increased in *VHL* deficiency cells (Fig. 4g), suggesting that *VHL* loss-of-function mutations caused enhanced glycolysis. In addition, through the transwell assay and cell proliferation assay, we found that *VHL* knockdown distinctly enhanced the inhibition of cell proliferation and invasiveness of Sunitinib to cancer cells (Fig. 4h and Supplementary Fig. 8f). These results indicated that the *VHL* mutation, leading to Warburg effect, increased the vulnerability to Sunitinib treatment in ccRCC.

*KMT2C* was another differentially mutated gene between the Responders and Non-Responders (Fig. 4a). Further evaluation of the proteomic impacts of KMT2C mutations showed that 922 proteins impacted by KMT2C mutation were downregulated ($p < 0.05$) (Fig. 4i), enriched in platelet aggregation (ITIH3, APBB1IP), signaling by VEGF (SRC, NCF4), vesicle-mediated transport (SEC23IP, VPS37B), and apoptosis through dr3/4/5 death receptors (RIPK1, DFFB) (Fig. 4j). Based on the mutation status of *VHL* and *KMT2C*, we divided patients into four genotypes ($VHL^{Mut}/KMT2C^{Mut}$, $VHL^{Mut}/KMT2C^{WT}$, $VHL^{WT}/KMT2C^{Mut}$, and $VHL^{WT}/KMT2C^{WT}$). Notably, the different genotypes showed different tumor sizes (Supplementary Fig. 8g) and distinct clinical outcomes (Fig. 4k, l). $VHL^{Mut}/KMT2C^{Mut}$ had the highest proportion of responders (83.33%) (Fig. 4k) and the best survival among these four genotypes (log-rank test, $p = 0.0158$) (Fig. 4l). Conversely, all the $VHL^{WT}/KMT2C^{Mut}$ cases were non-responders (Fig. 4k). Analysis of differentially expressed proteins comparing the four genotypes showed that proteins involved in PI3K-AKT-mTOR pathway (RRAGB, EIF4E2) and apoptosis (CASP3, CASP7) were downregulated in the *VHL* and *KMT2C* co-mutation group (Fig. 4m, n), suggesting that the PI3K-AKT-mTOR pathway was a resistance pathway in Sunitinib treatment (Fig. 4m, n).

Considering the latent impact of the interactions of genetic alterations on the prognosis of Sunitinib therapy, we comprehensively analyzed the mutational signatures, CNAs, and somatic mutations. The integrative analysis revealed a co-occurrence of the AA signature, 3p loss, and *VHL* mutation (Supplementary Fig. 8h). To further explore the interactions of these genetic alterations, we divided our cohort into 5 groups based their alteration status (Group1: AA/3p loss/$VHL^{Mut}$; Group2: AA/3p loss/$VHL^{WT}$; Group3: non-AA/3p loss/$VHL^{WT}$; Group4: non-AA/3p loss or $VHL^{Mut}$; Group5: non-AA/non-3p loss/$VHL^{WT}$) (Supplementary Fig. 8h). Significantly, these five groups were different in tumor size, among which Group1 showed the smallest tumor burden (Supplementary Fig. 8i). Further, the five types were significantly associated with prognosis and showed an ascending trend in PFS from Group1 to Group5 (log-rank for trend, $p = 0.031$) (Supplementary Fig. 8j). These results suggested that the AA signature, 3p loss, and *VHL* mutation were functionally superimposed. Differentially expressed

protein analysis among the five types of events showed the attenuation of glycolysis (PFKL, LDHA) and HIF pathways (VEGFA), and enhancement of pathways such as glucose transport (SLC2A3), the pentose phosphate pathway (TKT, PGD), and inflammatory response (C1QC, C9, S100A8) from Group1 to Group5 (Supplementary Fig. 8k). Single-sample gene set enrichment analysis (ssGSEA) complementarily correlated genomic alteration groups with these pathways (Supplementary Fig. 8l). To validate whether AA could enhance 3p loss and *VHL* mutations caused enhanced glycolysis, we measured the intracellular lactate when treated with AA in $VHL^{KD}$ ACHN cells. The results showed that AA treatment significantly upregulated the *VHL* deficiency-caused lactate increase (Fig. 4g). Overall, the integrated proteogenomic analysis demonstrated that AA signature was contributed to 3p loss and *VHL* mutation caused enhanced glycolysis, which associated with sunitinib therapeutic response.

## Differential analysis between sunitinib therapeutic responders and non-responders at proteome and phosphoproteome levels

As FLT1, FLT3, FLT4, KDR, KIT, PDGFRA, and PDGFRB were targets of Sunitinib, we surveyed their abundances in the responder and non-responder groups and found no significant difference between the two groups (Fig. 5a). We further evaluated the global activities of Sunitinib-targeted receptor tyrosine kinases (RTK), by comparing the phosphorylation levels of all substrates of these kinases. We found that the global activities of sunitinib-targeted RTKs also showed no significant differences between responders and non-responders (Fig. 5b). These results suggested that the abundances and activities of the targeted proteins might not be effective indicators for the TKI response.

Next, we used GSEA to compare the proteome of the responder and non-responder groups. It was observed that G2M checkpoint, antigen processing and presentation, Th17 cell differentiation, and NF-kappa B signaling pathway were enhanced in the responders, while mTOR signaling pathway, neutrophil degranulation, and platelet activation signaling, and aggregation were upregulated in the non-responders (Fig. 5c). The differentially expressed proteins (t-test <0.05, FC >1.5) between responder and non-responder groups in tumor tissues were shown in the Supplementary Fig. 9a (Supplementary Data 5).

Kinase-substrate enrichment analysis (KSEA) was conducted to probe the differentially activated kinases between responders and non-responders (Supplementary Data 5). We found that MAP2K1 (MEK1) and MTOR were activated in non-responders, while CDK1/2 were activated in responders (Fig. 5d). Notably, both MTOR and CDK2's protein expression level were no significant difference between Responders and non-responders (Supplementary Fig. 9b). Evaluation of kinase activities by single-sample gene set enrichment analysis (ssGSEA) further confirmed that MTOR was activated in non-responders while CDK2 was activated in Responders (Fig. 5e). The activity of MAP2K1 was significantly associated with poorer survival (Fig. 5f). Further investigating the differentially expressed phosphosites between responders and non-responders, we found that mTOR

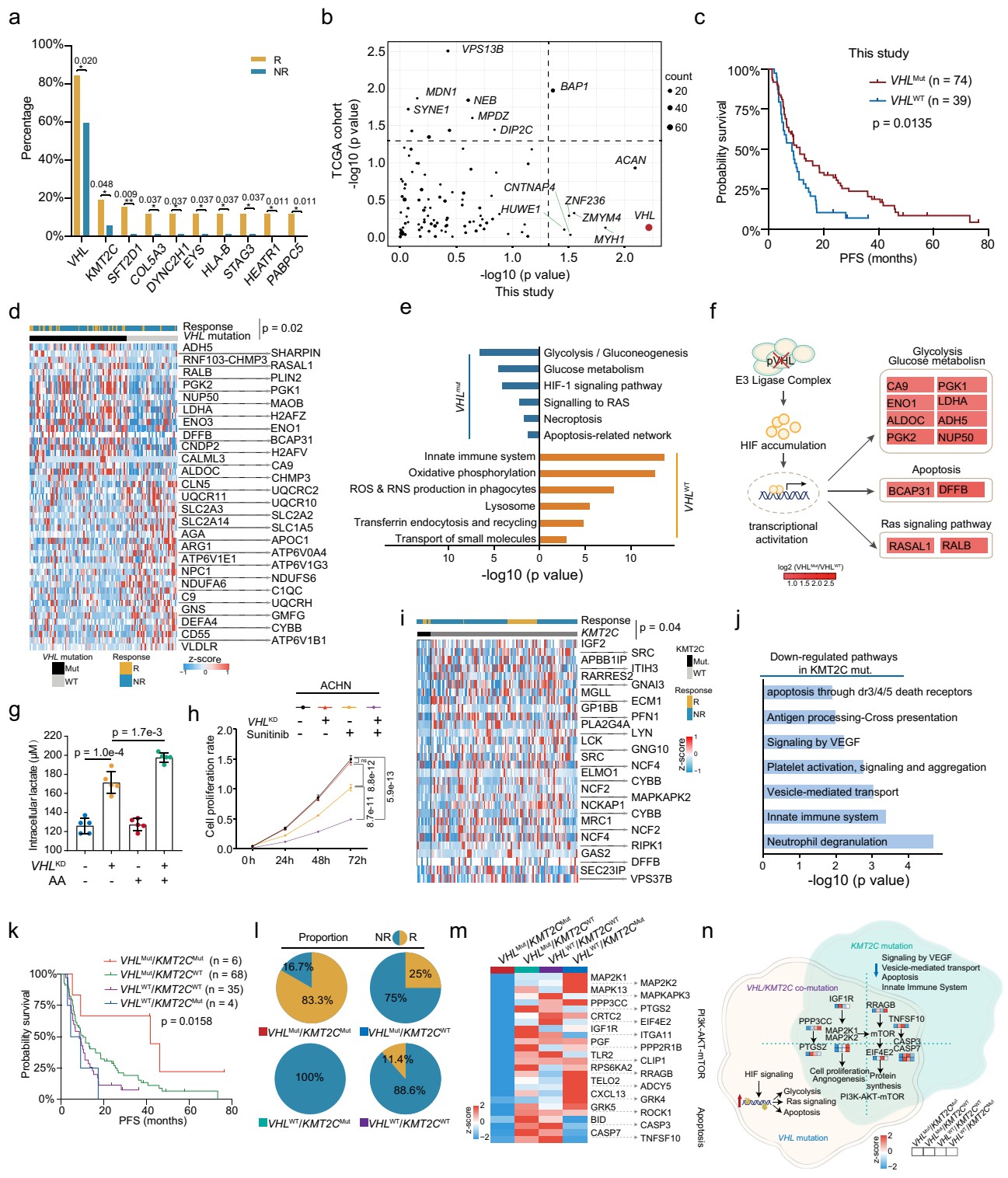

and MAPK signaling pathways associated substrates phosphorylation were elevated in non-reponders, such as DEPTOR at S265, MAP-K1(ERK2) at Y187, whlie SNRNP70 at S226 involved in RNA splicing was increased in reponders (Supplementary Fig. 9c). Consistently, abundances of phosphosites DEPTOR pS265, MAPK1 pY187, and SNRNP70 pS226 were significantly correlated with MTOR, MAP2K1, and CDK2 activities, respectively (Fig. 5g). Previous studies reported that sunitinib treatment did not affect the phosphorylation of ERK[39] and MEK inhibition abrogated Sunitinib resistance in RCC PDX-xenograft model[40], which provided supports for our findings. In conclusion, our phosphoproteome data demonstrated that the activation of

mTOR and MAPK signaling pathways was associated with Sunitinib treatment resistance.

## Associations between tumor heterogeneity and immune infiltration of ccRCC and therapy outcomes

We performed the consensus clustering algorithm to identify three proteomic subtypes with distinct features, containing 51, 37, and 27 patients, respectively (Supplementary Fig. 10a), reflecting the inter-tumoral heterogeneity of ccRCC. Notably, the different subtypes showed different proportions of Responders and Non-Responders, and tumor subtypes significantly differed in PFS (Supplementary

**Fig. 4 | Gene mutation in responders and non-responders associated with therapy outcomes. a** Significant differentially altered mutated genes in responder and non-responder groups (from left to right, the $p$ values are: 0.0201, 0.0486, 0.0097, 0.0375, 0.0375, 0.0375, 0.0375, 0.0375, 0.0111, and 0.0111; two-sided Fisher's exact test). $P$ values were described in the figure. **b** Comparison of the effect of mutations on the OS among this study and TCGA cohorts (log-rank test). **c** Kaplan–Meier curves of overall survival (OS) for patients with or without *VHL* mutations (log-rank test). **d, e** Differentially expressed proteins in the *VHL* mutation and WT groups and their associated biological pathways. The values were transformed by *z*-score. **f** A brief model depicting the functional impact of *VHL* mutation. **g** The impact of *VHL* knockdown and AA treatment on intracellular lactate ($n = 5$ independent experiments, data were presented as mean values ± SD, two-sided *t*-test, AA-100μM). $P$ values were described in the figure. **h** Cell proliferation assay

detected the effect of *VHL* knockdown and Sunitinib treatment on cell proliferation in ACHN cells. (Sunitinib-200 nM, two-sided *t*-test). Shown are the average values with ±SD. $P$ values were described in the figure. ns not significant. **i, j** Differentially expressed proteins in the *KMT2C* mutation and WT groups and their associated biological pathways. The values were transformed by *z*-score. **k** Kaplan–Meier curves of PFS for patients with different genotypes of *VHL* and *KMT2C* (log-Prank test). **l** Pie charts representing the distribution of different genotypes in the responder and non-responder groups. **m** Heatmap of protein expression abundances of PI3K-AKT-mTOR pathway and apoptosis pathway among the four genotypes. The values were transformed by *z*-score. **n** A brief model depicting the functional impact of *VHL* and *KMT2C* co-mutation. \*$p < 0.05$, \*\*$p < 0.01$. Source data are provided as a Source Data file.

Fig. 10b). Subtype3 had the highest proportion of non-responders (96%) (Fisher's exact test, $p = 0.004$) and the worst survival among the three subtypes (Supplementary Fig. 10b, c), while Subtype1 had the best survival (log-rank test, $p = 0.037$) (Supplementary Fig. 10b). After performing GSEA, Subtype1 was characterized by the upregulation of spliceosome and metabolism pathways including in glycolysis and valine/leucine/isoleucine degradation. The Subtype2 exhibited enrichment of MYC targets, antigen processing and presentation, and MTORC1 signaling. Subtype3 showed the upregulation of complement cascade, angiogenesis, and EMT (Supplementary. Fig. 10d).

We used xCell[41] to perform cell type deconvolution analysis, inferring the relative abundance of different cell types in the tumor microenvironment (TME), and performed ESTIMATE analysis[42] to determine the immune and stromal scores (Fig. 6a). Consensus clustering based on the inferred cell proportion identified three sets of tumors defined as T-cell infiltrated, cold, and progenitor-cell infiltrated, comprising 45, 40, and 30 cases respectively (Fig. 6a and Supplementary Fig. 10e). Notably, the immune clusters differed significantly in PFS (log-rank test, $p = 0.027$, Fig. 6b). Moreover, among the three subtypes, we observed higher responder proportions in the T-cell infiltrated the group and lower responder proportions in the progenitor-cell infiltrated group (Fig. 6c), which could be leveraged to predict therapeutic response.

The T-cell infiltrated group was characterized by high degrees of CD8$^+$ and CD4 + T-cell infiltration (Kruskal–Wallis test, $p < 0.05$) (Fig. 6a). The progenitor-cell infiltrated group contained the lowest responder proportion and the lowest frequency of chromosome 3p loss and *VHL* mutation (Fig. 6d), further emphasizing the importance of pVHL inactivation in sunitinib therapy. Proteomic analysis showed upregulation of proteins involved in inflammasomes and the intrinsic apoptosis pathway in the T-cell infiltrated tumors (Kruskal–Wallis test, $p < 0.05$) (Fig. 6e). The cold cluster was named so due to the lowest immune, stromal scores (Kruskal–Wallis test, $p < 0.05$) (Supplementary Fig. 10f, g). Compared with other tumors, the cold group showed the activation of the FGFR signaling and the Citrate cycle/TCA cycle (Kruskal–Wallis test, $p < 0.05$) (Fig. 6e).

The progenitor-cell infiltrated group was characterized by the progenitors of multiple types of immune cells, including common myeloid progenitor (CMP) cells and multipotent progenitor (MPP) cells (Fig. 6a), manifesting as the upregulation of platelet aggregation formation, the intrinsic pathway of fibrin clot formation, formation of fibrin clotting cascade, and complement cascade at the proteome level (Kruskal–Wallis test, $p < 0.05$) (Fig. 6e). Consistently, we found that patients in the progenitor-cell infiltrated group showed the highest expression of platelet marker CD321 (Kruskal–Wallis test, $p < 0.01$) (Supplementary Fig. 10h) and an elevated level of plasma PLT (Wilcoxon rank-sum test, $p = 0.033$) (Fig. 6f). These results revealed the connection of platelet activation with TME-mediated sunitinib resistance. Consistently, almost all patients with thrombocytosis (PLT count > normal upper limit) were non-responders

(Fisher's exact test, $p = 0.023$) (Fig. 6g). Common myeloid progenitor (CMP) cells are platelet precursors, which produce platelets due to the requirements for coagulation, showed significantly elevated levels in the progenitor-cell infiltrated group (Kruskal–Wallis test, $p = 6.6e-11$) (Fig. 6h). Indeed, CMP cells showed the highest correlation with platelet aggregate (plug formation) among all the xCell inferred cell types (SCC = 0.53, $p = 1.49E-09$) (Fig. 6i). To further explore the influence of TME of Progenitor-cell infiltrated tumors, we evaluated the differentially expressed proteins among three clusters, especially those involved in platelet aggregate and coagulation (Fig. 6j). Abundance of transforming growth factor beta 1 (TGFB1), a growth factor associated with multiple oncogenic pathways such as tumor proliferation, EMT, angiogenesis, immune evasion, and metastasis, was elevated in the Progenitor-cell infiltrated cluster and well-correlated with the Sunitinib response (Fig. 6k). We observed that TGFB1 was co-expressed with proteins involved in angiogenesis and tumor immune escape in ccRCC (Fig. 6l), indicating that the alternative angiogenesis driven by TGFB1 resulted in Sunitinib resistance. Moreover, by adding TGFB1 in the culture medium to simulate the impact of TME-derived TGFB1 on cancer cells, we found that TGFB1 could enhance the invasiveness and cell proliferation of ccRCC cancer cell (786-O cell and ACHN cell) and abrogate the impact of Sunitinib on ccRCC cancer cells (Fig. 6m, n and Supplementary Fig. 10i). In summary, we considered that the connection of platelet activation with TME, comprising abundant progenitors, which might lead to insufficient sunitinib therapy response.

## Construction of a response classifier based on proteome data

Above, we identified the clinical, genomic, transcriptomic, proteomic, and phosphoproteomic features that are associated with response to TKI therapy. This motivated the use of a machine learning framework (Fig. 7a) to integrate all the features into a predictive model of the response.

For the feature selection, based on the model construction pipeline (Methods), the F test and chi-square test were performed to select the features. As a result, 18 proteins that showed robust performance were selected for the predictive model construction including CCDC132, COTL1, EIF3C, EPB41L3, GPR89C, HEATR3, HNRNPH3, HNRNPU, HOGA1, LAMTOR4, NBEAL2, NPM1, PMM1, RPS7, SMARCA5, SNRPE, TNS1, and TRIO. Next, we used the ensemble random forest (RF) model algorithm to build the predictive model with the above 18 proteins as the input features. As shown in Fig. 7b, the RF model showed good performance with receiver operating characteristic–area under the curve (ROC-AUC) = 0.85 on the training cohort.

It had ROC-AUC = 0.85, specificity = 0.85, and sensitivity = 0.75 on the test cohort (Fig. 7e and Supplementary Fig. 11a). Furthermore, we calculated the feature importance for this RF model (Supplementary Fig. 11b).

To construct an advanced classifier for improving the predicted performance, we enrolled the multi-omics features rather than the

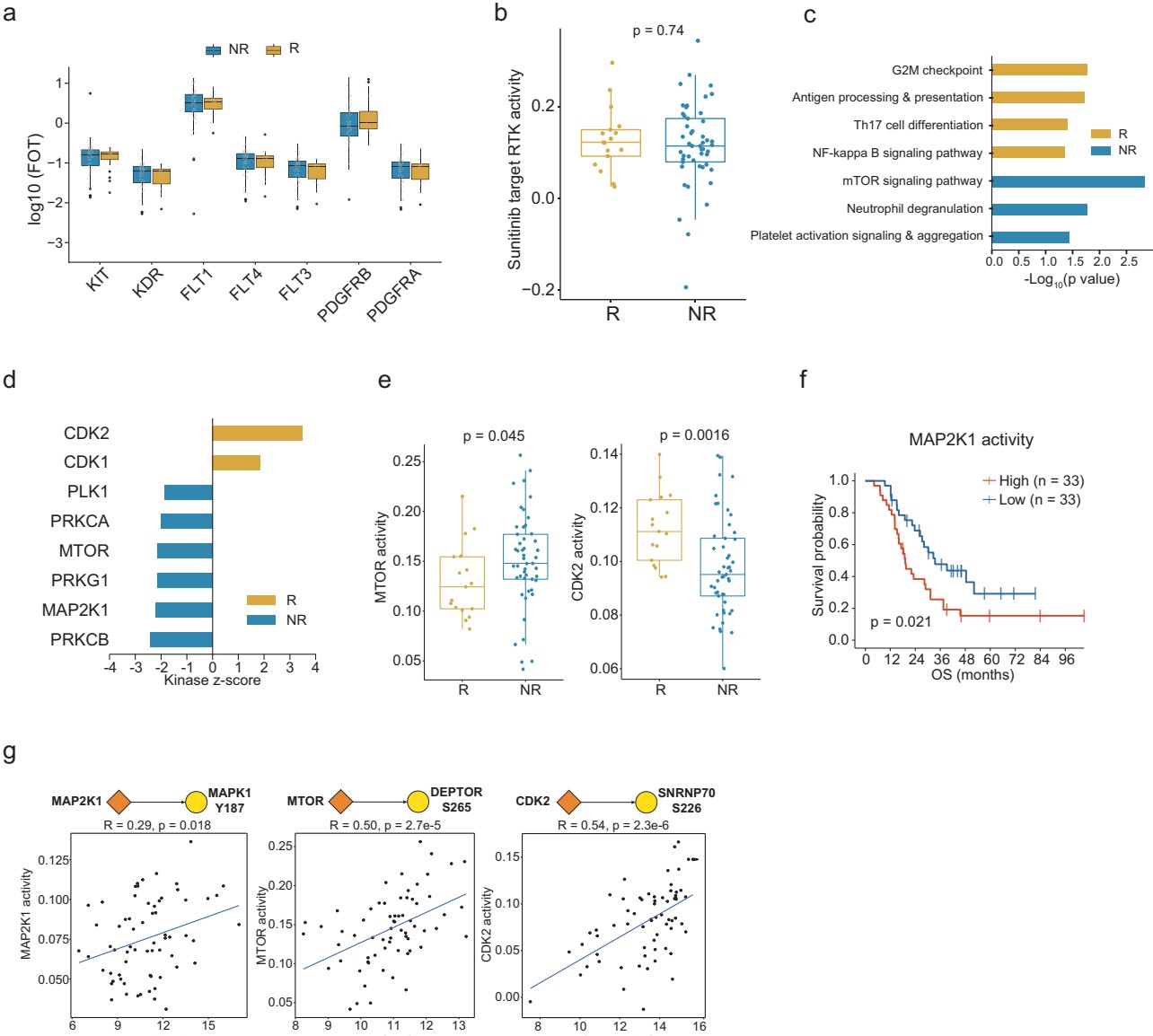

**Fig. 5 | Proteomic and clinical features associated with sunitinib therapeutic outcomes. a** Abundance of sunitinib targets in responders (*n* = 27) and non-responders (*n* = 88). *P* values are derived from the two-sided Wilcoxon rank-sum test. Boxplots show the median (central line), the 25–75% IQR (box limits), and the ±1.5×IQR (whiskers). **b** Boxplot showing the inferred Sunitinib-targeted RTK activities between responders (*n* = 17) and non-responders (*n* = 49). *P* values are derived from the two-sided Wilcoxon rank-sum test. Boxplots show the median (central line), the 25–75% IQR (box limits), and the ±1.5×IQR (whiskers). **c** Proteins involved in pathways correlated with sunitinib response. **d** Differential analysis of kinase

activities by KSEA between responders and non-responders. **e**, Comparison of inferred kinase activities of MTOR and CDK2 between Responders (*n* = 17) and Non-Responders (*n* = 49) (two-sided Wilcoxon rank-sum test). Boxplots show the median (central line), the 25–75% IQR (box limits), and the ±1.5×IQR (whiskers). *P* values were described in the figure. **f** Kaplan–Meier curves of OS for patients with different inferred MAP2K1 activities (log-rank test). **g** Scatterplot showing inferred kinase activity (y-axis) versus the phospho-abundance of the targeted substrates (x-axis) (two-sided Spearman's correlation test). *P* values were described in the figure. Source data are provided as a Source Data file.

only proteome features and built the RF-based classifier (Fig. 7a). In detail, clinical features (LDH, tumor size, and PLT), mutation features (*VHL* and *KMT2C*), mutational signatures (AA signature), copy number alteration features (7q, 3p), transcriptome features (MIR3939, ALDH1A3, LPAR1, FBLN5, and C7), and proteome features (same as the proteome-based model) were selected for the multi-omics classifier construction. As shown in Fig. 7c, the multi-omics classifier showed ROC-AUC = 0.86 for the repeated cross-validation on the training cohort. The sensitivity and specificity of the test cohort were 1 and 0.86, with a great improvement to the proteome-based model (Fig. 7d). Compared to the proteome-based RF model, the multi-omics RF model showed better performance with ROC-AUC = 0.98 on the test cohort (Fig. 7e). Furthermore, we evaluated the balanced accuracy,

precision, recall, and F1 score on the test cohort and observed good generalized performance on the test cohort (Supplementary Fig. 11c). Proteomic and transcriptomic features, i.e., protein/phosphoprotein abundance and gene expression levels, contribute most to the success of the prediction model, as revealed by the feature importance analysis included in Fig. 7f.

In summary, we used an ensemble approach that input multi-omics features to derive predictors of the TKI Responders. The predictive model offers an opportunity to expedite the translation of basic research to more precise diagnosis and treatment in the clinic. In addition, this framework highlights the importance of data integration in machine learning models for response prediction and could be utilized in other cancers.

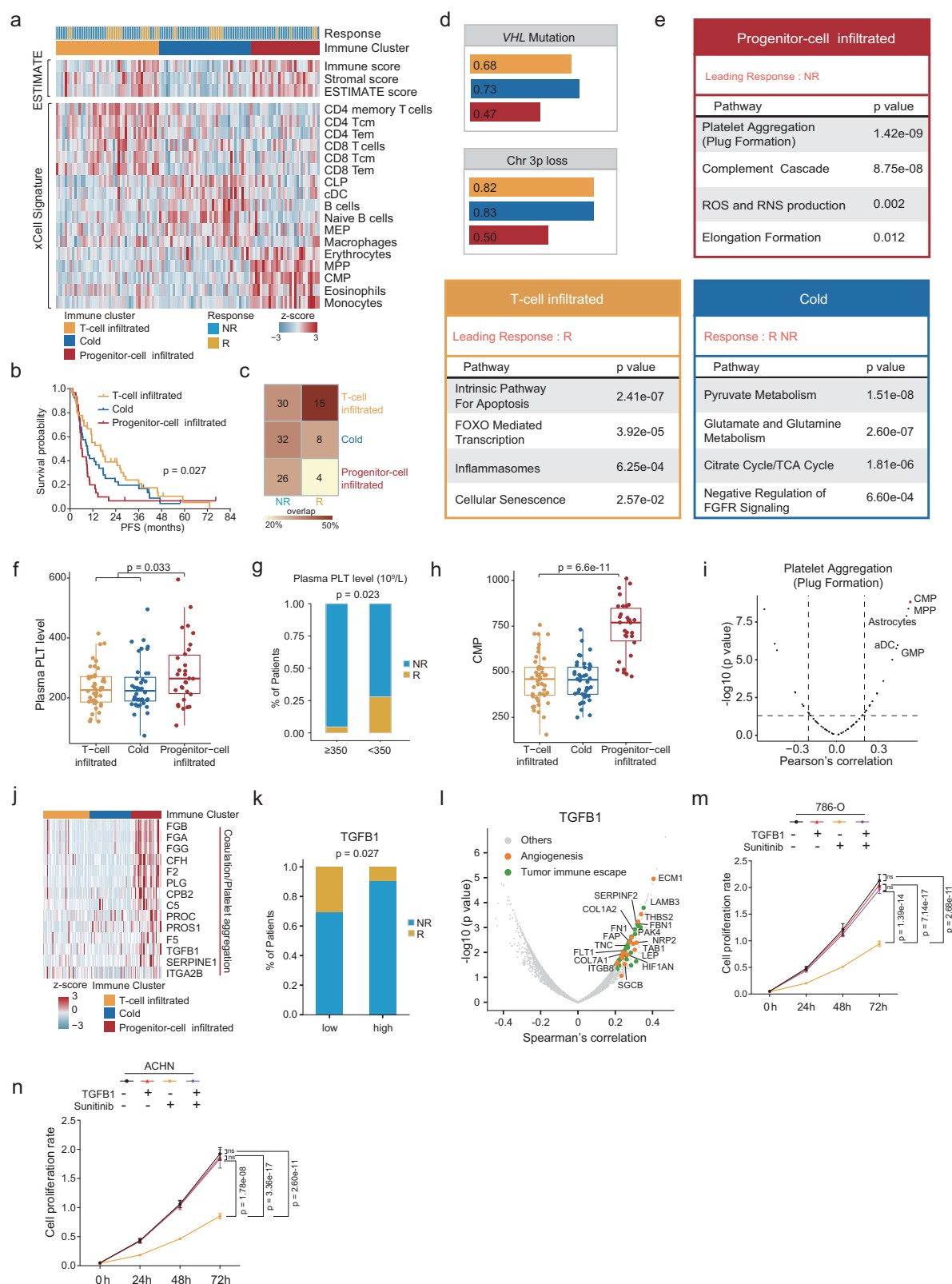

## Discussion

In this study, we established a comprehensive landscape of Sunitinib drug response in ccRCC patients at both the genome and proteome levels. Mutation signatures revealed that the AA signature was associated with smaller tumors and better survival. The results showed strong regional characteristics, indicating the necessity of considering the life habits and environmental factors of the Chinese population in

ccRCC treatment. It is well-established that AA exposure impacts the mutagenesis characteristic of predominant T > A transversions. Here, based on the multi-omics data and bioinformatical analysis, in addition to the genomic impact of AA, we described the proteomic differences between AA patients and non-AA patients, and observed attenuated pentose phosphate pathway and enhanced glycolysis alteration, which connected to the Sunitinib treatment. The changes of proteomic level

**Fig. 6 | Associations between immune infiltration of ccRCC and therapy outcomes. a** Heatmap of immune signatures in three ccRCC immune clusters. **b** Kaplan–Meier curves of PFS for the three immune clusters (log-rank test). *P* values were described in the figure. **c** Proportions of responders and non-responders among the three clusters. **d** Proportions of *VHL* mutation and chromosome 3p loss in immune groups. **e** Pathways enriched in the three immune subtypes. **f** Comparison of plasma PLT counts among three immune subtypes (*n* (T-cell infiltrated cluster) = 51, *n* (Cold cluster) = 37, *n* (Progenitor-cell infiltrated cluster) = 27 biologically independent samples examined). *P* values are derived from the two-sided Wilcoxon rank-sum test. Boxplots show the median (central line), the 25–75% IQR (box limits), and the ±1.5×IQR (whiskers). **g** Distribution of thrombocytosis in responders and non-responders (two side Fisher's exact test). *P* values were described in the figure. **h** Comparison of CMP scores among three immune subtypes (*n* (T-cell infiltrated cluster) = 51, *n* (Cold cluster) = 37, *n* (Progenitor-cell

infiltrated cluster) = 27 biologically independent samples examined). *P* values are derived from Kruskal–Wallis test. Boxplots show the median (central line), the 25–75% IQR (box limits), and the ±1.5×IQR (whiskers). **i** Correlations between platelet aggregate (plug formation) scores and xCell inferred cell components (two-sided Spearman's correlation test). **j** Heatmap of proteins involved in platelet aggregate and coagulation in the three subtypes. **k** Responders and non-responders include different proportions of high (*n* = 57) or low (*n* = 58) expression of TGFB1 (two-sided Fisher's exact test). *P* values were described in the figure. **l** Proteins involved in tumor immune escape and angiogenesis were co-expressed with TGFB1. **m, n** CCK-8 detected the effect of TGFB1 intervention and sunitinib treatment on cell proliferation in 786-O and ACHN cells, respectively. (sunitinib-200 nM, **\**p* < 0.01, two-sided *t*-test). Shown are the average values with SD. *P* values were described in the figure. ns not significant. Source data are provided as a Source Data file.

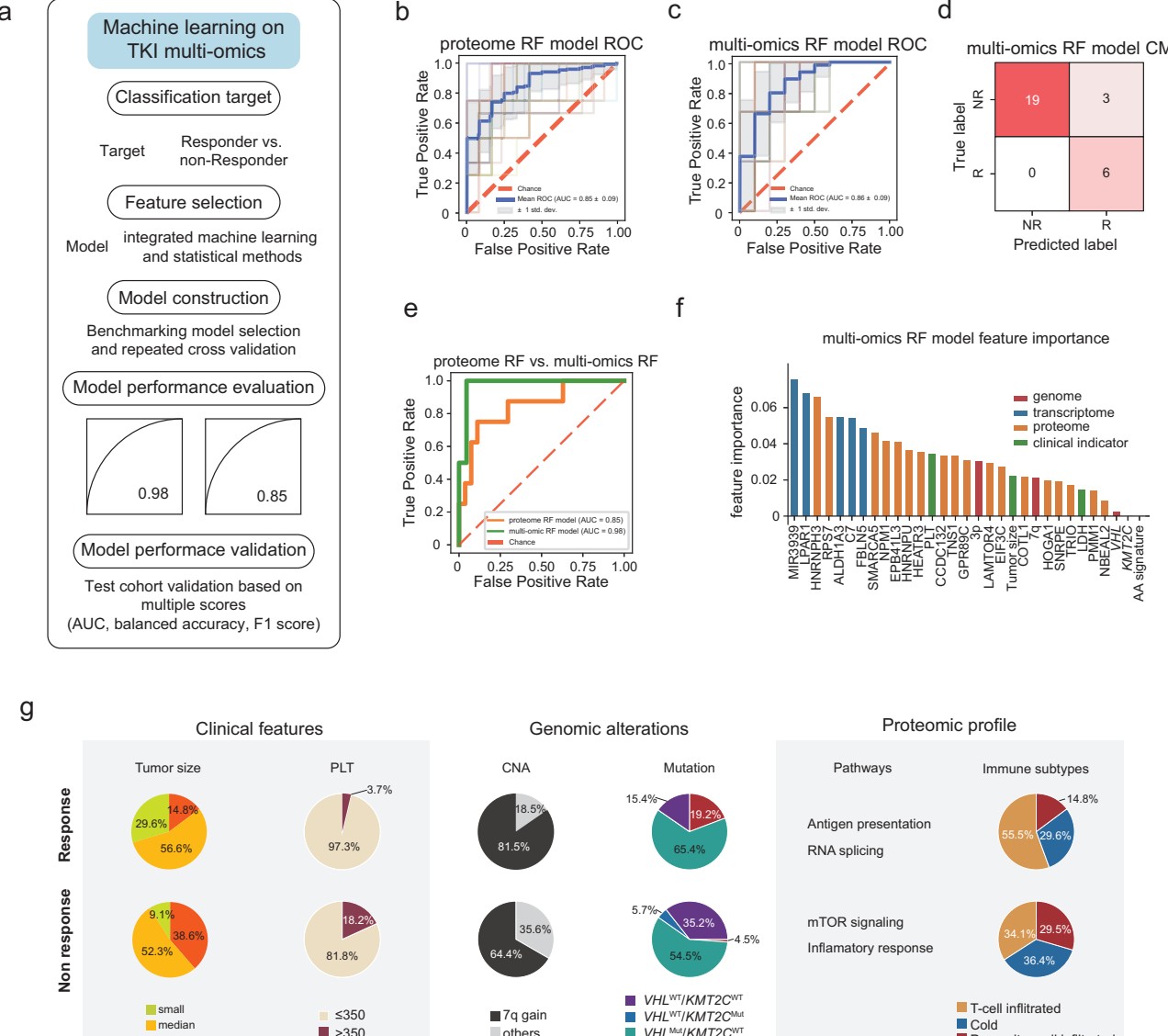

**Fig. 7 | Proteomic classifier to predict sunitinib response. a** Machine learning-based model construction pipeline for the TKI treatment response, including classification target, feature selection, model construction, model performance evaluation, and model performance validation. **b, c** The five repeatedly cross-validated ROC-AUC on the train cohort for proteome-based random forest (RF) and multi-omics-base RF, separately. **d** The confusion matrix of test cohort for multi-

omics-base RF. **e** The comparison of ROC-AUC on the test cohort for proteome-based RF and multi-omics-base RF. **f** The feature importance of multi-omics-based RF model. The blue, orange, green, and red rectangles indicated proteome, transcriptome, clinical, and genomic features, respectively. **g** Summary of clinical and molecular characteristics in sunitinib responders and non-responders. Source data are provided as a Source Data file.

caused by AA signature still needs further investigation using experimental models or in clinical practice. However, the presence of an AA mutation signature does not necessarily mean that the patients have significant AA levels in their plasma during sunitinib treatment. Our study is retrospective and the tissue samples from patients we used in the current study are collected in the past several years. Therefore, we cannot quantify the concentration of AA in the blood of our cohort of patients, which is really a regret of this research. Further studies with AA close to physiological concentration in animal primary cells need to be conducted in the future.

In this cohort, 19 out of 115 patients were found to carry the AA signature. We compared the frequency of T > A transversions between tumor and tumor-adjacent tissues. The data showed that the mutation signatures of tumor tissues were higher than the mutation signatures of tumor-adjacent tissues, demonstrating the strong mutagenic effect of AA in tumor tissues. Besides, in our cohort, there were 15 out of 19 patients also carried T > A transversions in their normal tissue; the frequency of T > A transversions ranged from 0–11 in the normal tissue, which was far lower than the 238–5802 in the tumor tissue. These results indicated that AA signature might also affect the tumor-adjacent tissue to a certain extent, which was consistent with Li et al. study[43] reported that AA signatures were found in normal human tissues. Overall, although the number of mutations in normal tissues was far less than in the tumor tissues, it is necessary to study the somatic mutation accumulation in normal cells, which is essential for understanding tumorigenesis and development.

We identified ten significantly differentially altered genes in the responders and non-responders in our cohort. Notably, *VHL* mutations, the most frequent mutation events in ccRCC[3,4], occurred more frequently in Responders and were associated with improved survival in Sunitinib treatment. This finding was also validated in the IMmotion151 study[12]. We validated that *VHL* deficiency cells showed increased glycolysis and were more vulnerable to sunitinib treatment. It was noted that *VHL* mutation frequencies varied in different studies, based on different populations (Fig. 1a). There might be two reasons for the lower *VHL* mutation frequencies in TCGA and Japanese cohorts. One was differences across regions and ethnic backgrounds, and the other was insufficient sequencing depth. The genetic ancestry analysis of the TCGA cohort found there were racial differences in ccRCC, with lower levels of *VHL* and *PBRM1* mutations in tumors from African versus European patients[44]. The higher depth of the sequencing increased the chance to detect the mutations[45]. Advances in sequencing technology and reduction in cost improved sequencing output, resulting in a higher probability to detect the *VHL* mutation.

Proteins overexpressed in *VHL* mutant tumors were enriched in the HIF1A transcription factor network and glycolysis/gluconeogenesis, consistent with the expectation of pVHL inactivation. We further found the co-occurrence of AA signature, 3p loss, and *VHL* mutations and that the AA signature could enhance 3p loss/*VHL* mutation-induced enhanced glycolysis flux. This phenomenon indicated that glucose metabolism in RCC was vulnerable and associated with the response to anti-angiogenesis therapy. The divergent mutational effect of *KMT2C* was also observed. Co-mutation of *VHL* and *KMT2C* was associated with a better response to Sunitinib. Further, an association of copy number amplifications in 3q, 7p, 7q, and 8p with poor survival was observed in this sunitinib treatment cohort but not in the all-stage ccRCC cohort, indicating that the association depended on sunitinib therapy. The comparison of CNAs in the responder and non-responder groups revealed the importance of 7q amplification in the sunitinib therapy response.

LAMTOR4, MDH2, and CALU on 7q showed a cis effect in our cohort. In the in vitro validation experiments, we found that over-expressed LAMTOR4, MDH2, and CALU increased the phosphorylation of S6K, indicating activation of mTORC1[34]. LAMTOR4, as a part of the Ragulator complex[46], is directly involved in amino acid sensing and

activation of mTORC1. MDH2, utilizing the NAD/NADH cofactor system in the citric acid cycle, participates in mitochondrial metabolism. Mitochondrial metabolism is closely related to the mTOR pathway[47]. We found MDH2 cis-regulated by 7q gain in our cohort and overexpression of MDH2 increased the phosphorylation of S6K. Based on the published studies and the experiment results, we speculate MDH2 might participate in mTOR signaling activation by mitochondrial metabolism. The product of CALU is a calcium-binding protein localized in the endoplasmic reticulum (ER) and is involved in such ER functions as protein folding and sorting. It is rarely reported that CALU is directly connected to mTOR signaling in the previously published study. The specific mechanism needs further research in the future. Overall, the 7q amplification was confirmed to upregulate mTOR signaling pathways and amino acid turnover in ccRCC. This result supported mTOR inhibitors, such as Everolimus, as the second-line treatment after sunitinib failure, especially in patients with 7q gain. In conclusion, metabolic reprogramming induced by genomic alterations in ccRCC impacted the anti-angiogenesis therapy. sunitinib-resistant ccRCC was less dependent on the nutrients provided by angiogenesis.

The high tumor purity (more than 90%) and a relatively broad range of variant allele frequency (VAF) indicated a high intra-heterogeneity of a tumor. In our study, by applying the NMF algorithm on the mutation spectrum based on the COSMIC, we identified different mutation signatures and found the characteristics of genomic signatures were distinctive from each other. Furthermore, consensus clustering identified three ccRCC proteomic subtypes with distinct features. Moreover, immune landscape characterization also revealed diverse tumor microenvironment subsets in ccRCC patients. In summary, the high tumor purity and a relatively broad range of VAF are possibly induced by the heterogeneity of tumor gene variation.

Sensitivity and resistance to sunitinib therapy were influenced by many factors and showed diversity and heterogeneity. As proteins are direct executors of biological functions, it is necessary to illuminate the biological basis underlying the differential TKI response in the proteome context. Our study revealed RNA splicing and antigen presentation-related proteins were associated with Sunitinib response, whereas proteins involved in complement cascades and the mTOR signaling pathway were associated with Sunitinib resistance. We also indicated that the association of RNA splicing and T-cell response might be a potential way to make anti-angiogenesis therapy more effective[48]. More interestingly, we found that the LDH level was positively correlated with the abundance of CD8A and PD-L1, indicating that patients with high-level LDH might benefit from ICB therapy.

In recent years, studies on TKIs combined with immune checkpoint inhibitors for advanced RCC have made great breakthroughs, improving the objective response rate to 50–60%. The success of combination therapy has suggested that TME greatly affects anti-angiogenesis therapy. Wang et al.[49] established an inflamed pan-RCC subtype (IS), characterized by infiltration of large amounts of immune cells, including NK cells, B cells, and CD8⁺ T cells, and found IS was a strong predictor of poor survival. Coincidentally, Hakimi et al.[10] also identified four subsets of ccRCC, among which a cluster characterized by high immune infiltration was associated with poor response to Sunitinib. By investigating the proteomic profiles of tumors, we proposed a rational stratification of ccRCC patients based on immune signatures as T-cell infiltrated, cold, and progenitor-cell infiltrated. The progenitor-cell infiltrated group exhibited upregulation of the platelet aggregate formation pathway and complement cascades, and high expression of the platelet marker CD321, suggesting that the platelets in the TME were responsible for sunitinib failure. Higher PLT counts were associated with poorer survival and thrombocytosis was associated with sunitinib resistance, further supporting this finding. Further, we found that TGFB1, overexpressed in a progenitor-cell infiltrated group, was an important component for alternative

angiogenic signaling in ccRCC, indicating that TGFB1 inhibitors might increase the sunitinib response.

Functional studies have confirmed that miRNA dysregulation is causal in many cases of cancer[50]. Insights into the roles of miRNAs in development and disease, particularly in cancer, have made miRNAs attractive tools and targets for novel therapeutic approaches. In our study, we analyzed the miRNAs which might regulate the proteins associated with the response to sunitinib. Based on the transcriptome sequencing data in our cohort, we identified 348 miRNAs in this cohort, among which four miRNAs showed significant differences between responders and non-responders. Specifically, MIR3939, MIR4635, and MIR578 showed higher expression levels in Responders, while MIR27A was higher in non-responders (Supplementary Fig. 12).

As previously reported, models for predicting the response of TKIs[9,10] or combination therapy[12] were almost constructed based on genome and transcriptome data, constructing a classifier to predict the drug response based on multi-omics data provides an alternative perspective. Our research provided a model for predicting Sunitinib response and could facilitate the precise use of sunitinib. The multi-omics or even the multimodal data could provide different information for one topic. Hence, for the machine learning model, integrating the multi-omics features could improve the model performance in a feature-distribution-orthogonal manner, which would benefit precision medicine development. However, this single-center study was retrospective, and was subject to the inherent limitations associated with retrospective analyses. Thus, prospective studies are still needed to confirm our findings.

In conclusion, we first delineated the proteogenomic landscape of Sunitinib response in Chinese patients with ccRCC. We found that *VHL* mutation and the AA signature synergistically improved the clinical outcomes of sunitinib treatment in Chinese patients with ccRCC. Multiple results repetitively showed that mTOR signaling was an intrinsic pathway for sunitinib resistance. Our study further defined three immune subsets as T-cell infiltrated, cold, and progenitor-cell infiltrated, and showed that the progenitor-cell infiltrated cluster was significantly correlated with Sunitinib resistance, which may be caused by activation of platelet signaling and secretion of TGFB1. We summarized the features of responders in multi-dimension (Fig. 7g). We also constructed a model for predicting the sunitinib response and validated the robustness of the predictive model in an independent dataset. Overall, our multi-level omics analysis identified the molecular mechanisms underlying the sunitinib response and defined the genomic, proteomic, and immune signatures to stratify patients with ccRCC to develop more rational therapeutic interventions.

## Methods

### Clinical sample collection

The study was compliant with the ethical standards of Helsinki Declaration II and was approved by the institutional review board of Fudan University Shanghai Cancer Center (FUSCC) (050432-4-1911D). Written informed consent was obtained from each patient before any study-specific investigation was conducted.

This study included 115 patients with advanced ccRCC, who were treated with TKIs at the Department of Urology of FUSCC from Jan 2008 to Dec 2019. All electronic medical records were screened retrospectively. Among the 115 cases, 47 cases were localized ccRCC at the time of surgery and were included due to the subsequent development of metastatic disease. The median follow-up was 28.4 months (range, 4.2–127.5 months). At the last follow-up, 102 patients (88.7%) had progressive disease and 84 patients (73.0%) had died of ccRCC. To evaluate whether prognostic markers are associated with survival independently of therapy, we collected another independent cohort with 37 cases of ccRCC patients, who were not received any treatment after surgery. These 37 cases had complete follow-up information (Supplementary Data 4).

Clinicopathological indicators, including clinical manifestations, laterality, tumor size, chronic disease status, TNM stage (stage at the time of surgery), and ISUP grade, are summarized in Supplementary Data 1. All samples were collected during radical nephrectomy and all patients were treatment-naïve before surgery and received Sunitinib treatment after surgery. The initial dose of Sunitinib was the same between patients and adjusted according to side effects and the intervention was set as 50 mg orally taken daily for 4 weeks and off treatment for 2 weeks until progression or unacceptable toxicity.

Tumor and adjacent non-tumor tissue samples were available from the FUSCC tissue bank. The specimens were collected according to the following criteria: (1) tumor-adjacent tissues were collected at a distance of >2 cm from the tumor margin; (2) each tumor/adjacent sample was checked by an expert pathologist to confirm sample quality. We use the matched tumor-adjacent tissue DNA as the background to call the somatic mutations and copy number alteration; therefore, the samples with tumor cells in the adjacent tissues were excluded from this study. Tumor and paired tumor-adjacent tissues were collected within 30 min after resection, immediately transferred into sterile freezing vials, and snap frozen in liquid nitrogen, cut into ~0.5 cm³ pieces under −40 °C, then split and stored at −80 °C until being used. H&E-stained sections were reviewed by an experienced genitourinary pathologist to determine the ISUP grade, and frozen sections were reviewed to determine the tumor cell rate of the ccRCC tissues.

### DNA extraction and WES

WES was conducted at Life Healthcare Clinical Laboratory (China). DNA isolated from frozen tumor tissue samples was used for WES, and matched germline DNA was obtained from tumor-adjacent tissues. DNA was isolated from fresh frozen using DNeasy Blood & Tissue Kit (Qiagen, 69504) according to the manufacturer's instructions. Purified DNA was quantified using a Qubit 3.0 Fluorometer (Life Technologies). For matched tumor and tumor-adjacent tissues, 100 ng of DNA was sheared to 200–300-bp fragments using a Covaris M220 system. Tumor and matched tumor-adjacent DNA libraries were constructed using Accel-NGS 2 S HYB DNA LIBRARY KIT (Swift Biosciences, 23096) and Accel-NGS 2 S MID S1-S4 (Swift Biosciences, 279384). xGen Exome Research Panel v1.0 (IDT, 1056115) and xGen Lockdown reagents (IDT, 1072281) were used for exome enrichment. Dynabeads M-270 Streptavidin (Thermo Fisher, 65306) was used for library purification, P5/P7 primers (Nanodigmbio, ND10010), and HotStart ReadyMix (KAPA, KK2612) were used for library amplification. The amplified libraries were purified using SPRISELECT (Beckman, B23319). DNA quality was assessed using a Bioanalyzer High Sensitivity DNA Analysis kit (Agilent Technologies, 5067-4626). Samples underwent paired-end sequencing on a Nextseq CN500 platform (Illumina), with a 150-bp read length. The WES target region was 39 M. Detailed information on the number of reads per sample, and coverages of exomes are also provided in Supplementary Data 2.

### Somatic variant detection

Read-depth statistics were calculated using the DepthOfCoverage function in the Genome Analysis Toolkit (GATK v3.8.1.0)[51]. Paired-end reads in the Fastq format were aligned to a reference human genome[52] (UCSC Genome Browser, hg38) using Burrows–Wheeler Aligner. Variant calling was conducted following the GATK best practices. Somatic single-nucleotide variations and small insertions and deletions were detected using MuTect2 (GATK v4.1.2.0) and were annotated using ANNOVAR[53] based on UCSC known genes. The two longest genes, *TTN* and *MUC16*, were excluded as they tended to acquire numerous chance mutations in large-scale genome/exome sequencing experiments. The Maftools R package[54] was used to display mutant genes with non-synonymous mutations (Supplementary Fig. 1d).

## Mutational signature

SBSs are defined as the replacement of a certain nucleotide base. There are six possible substitutions: C > A, C > G, C > T, T > A, T > C, and T > G. Considering the nucleotide context, these SBS classes can be further expanded to 96 possible mutation types. The frequencies of these 96 mutation types were estimated for each sample. The non-negative matrix factorization algorithm of SigProfiler[55] was used to estimate the minimal components that could explain the maximum variance among samples. De novo mutation signatures were decomposed using COSMIC v3[30]. After decomposing a matrix of the 96 substitution classes of the samples into five signatures, the contribution of each signature in each sample was estimated.

## CNA calling

CNAs were called following the somatic CNA best practice, using the CalculateTargetCoverage function in GATK (v4.1.2.0). We applied Genomic Identification of Significant Targets in Cancer (GISTIC2.0)[35] to identify significantly amplified or deleted focal-level and arm-level events, with $q < 0.05$ considered significant. The following parameters were used: amplification threshold = 0.1; deletion threshold = 0.1; cap value = 1.5; broad length cut-off = 0.90; remove X-chromosome = 0; confidence level = 0.95; join segment size = 4; arm-level peel off = 1; maximum sample segments = 2000; gene GISTIC = 1. Each gene in each sample was assigned a threshold (0.1) copy number that reflects the magnitude of its deletion or amplification. (Supplementary Data 2).

## RNA extraction and RNA-seq

Total RNA from each tissue sample was isolated using TRIzol Reagent (Invitrogen). All RNA samples were assayed for RNA purity and integrity. After RNA samples were qualified, the RNA was reverse-transcript into cDNA and constructed library, and conducted sequencing. The clustering of the index-coded samples was performed on a cBot Cluster Generation System using TruSeq PE Cluster Kit v3-cBot-HS (Illumina), according to the manufacturer's instructions. After cluster generation, the libraries were sequenced on an Illumina HiSeq 4000 platform and 125 bp paired-end reads were generated. After trimming the adapters and removing low-quality tags, sequencing reads were mapped onto the hg19. The mapped reads were assembled into transcripts or genes by using StringTie software (v2.1.4)[56]. For quantification purposes, the relative abundance of the transcript or gene was measured by a normalized metric, FPKM (Fragments Per Kilobase of transcript per Million mapped reads). Transcripts with median FPKM >1 were retained (Supplementary Data 3).

## Protein extraction and trypsin digestion

Samples were minced and lysed in lysis buffer (8 M urea, 100 mM Tris hydrochloride, pH 8.0) containing protease and phosphatase inhibitors (Thermo Scientific) and then sonicated for 1 min (3 s on and 3 s off, amplitude 25%, SONICS, VCX130). The lysates were centrifuged at 14,000×$g$ for 10 min and supernatants were collected as whole-tissue extracts. Protein concentrations were determined using the Bradford protein assay (TaKaRa, T9310A). Extracts (50 μg protein) were reduced with 10 mM dithiothreitol at 56 °C for 30 min and alkylated with 10 mM iodoacetamide at room temperature in the dark for 30 min. The samples were digested with trypsin using a filter-aided sample preparation method[57]. Tryptic peptides were separated using a home-made reverse-phase C18 column. Peptides were eluted, vacuum-dried (Concentrator Plus, Eppendorf), and analyzed by liquid chromatography-tandem MS (LC-MS/MS).

## Enrichment of phosphopeptides

Phosphopeptides were enriched by High-Select™ Fe-NTA Phosphopeptide Enrichment Kit (Thermo Fisher, A32992), according to the manufacturer's instruction. Briefly, 1 mg peptides were resuspended in 200 μL binding/wash buffer and loaded to the equilibrated spin column with Fe-NTA resin. The samples were mixed with resin by gently tapping and then incubated for 30 min. The mixture was centrifuged at 1000×$g$ for 30 s to discard the flowthrough and then washed with 200 μL of binding/wash buffer three times and washed with 200 μL of LC-MS grade water for one additional time. The enriched phosphopeptides in NTA resin were eluted by adding 100 μL of elution buffer and centrifuged at 1000×$g$ for 30 s for two times and vacuum-dried.

## LC-MS/MS

Samples were analysed on a Q Exactive HF-X mass spectrometer (Thermo Fisher Scientific, Rockford, IL, USA) coupled with a high-performance liquid chromatograph (EASY-nLC 1200 System, Thermo Fisher). Dried peptide samples were dissolved in solvent A (0.1% formic acid in water) and loaded onto a trap column (100 μm × 2 cm, home-made; particle size, 3 μm; pore size, 120 Å; SunChrom) with a maximum pressure of 280 bar using solvent A, and were then separated on a home-made 150 μm × 12 cm silica microcolumn (particle size, 1.9 μm; pore size, 120 Å; SunChrom) with a gradient of 5–35% mobile phase B (acetonitrile and 0.1% formic acid) at a flow rate of 600 nL/min for 75 min. MS analysis was conducted with one full scan (300–1400 m/z, R = 120,000 at 200 m/z) at an automatic gain control (AGC) target of 3e6 ions, followed by up to 20 data-dependent MS/MS scans with higher-energy collision dissociation (target 5e4 ions, max injection time 20 ms, isolation window 1.6 m/z, normalized collision energy of 27%). Detection was performed using Orbitrap (Q Exactive HF-X mass spectrometer, Thermo Fisher Scientific) and data were acquired using Xcalibur software (Thermo Fischer Scientific).

## Proteome identification and quantification

Raw files were processed in Firmiana[58] and searched against the National Centre for Biotechnology Information (NCBI) RefSeq human protein database (updated on 04-07-2013, 32,015 entries) using the Mascot 2.4 search engine (Matrix Science Inc). Mass tolerances were 20 ppm for the precursor and 50 mmu for product ions. Up to two missed cleavages were allowed. Cysteine carbamidomethylation was set as a fixed modification, with methionine N-acetylation and oxidation as variable modifications. For phosphoproteomic samples, phosphorylation at Ser/Thr/Tyr was set as an additional variable modification. Precursor ion score charges were limited to +2, +3, and +4. The data were also searched against a decoy database so that peptide and protein identifications were accepted at an FDR of 1%. Label-free protein quantifications were calculated using the intensity-based absolute quantification (iBAQ) approach[59]. Matching between runs[60] was used to improve the parallelism between tumor/adjacent samples. Specifically, we built a dynamic regression function based on those commonly identified peptides in grouping experiments. According to correlation value $R^2$, Firmiana will choose a linear or quadratic function for regression to calculate the RT of corresponding hidden peptides, and check the existence of the extracting ion current (XIC) based on the m/z and calculated RT. Finally, the program could evaluate the peak area values of those existed XICs. These peak area values should be considered as parts of corresponding proteins. The scripts were uploaded to GitHub: https://github.com/FirmianaPlatform/SourceCode/blob/master/Firmiana%20Frontend/gardener/views.py (line 3948- line 4588). The same MBR strategy was also implemented in the published study[61,62].

The FOT was used to represent the normalized abundance of proteins across samples. FOT was defined as a protein's iBAQ divided by the total iBAQ of all proteins identified in each sample. FOT values were multiplied by $10^5$ for ease of presentation and missing values were assigned n.a (Supplementary Data 4). The abundances of phosphosites were applied with the median centroid correction to adjust for sample-specific biases. Phosphosites with a missing value of less than 25% were selected, and missing values were then imputed with the K-nearest

neighbor (KNN) algorithm using the R package DreamAI[63] (Supplementary Data 4).

## Protein and pathway alterations in responders vs. non-responders

In total, 7451 proteins identified in >25% of tumor samples were used for subsequent analysis. Volcano plots were used to display DEPs in responders and non-responders by applying the thresholds of fold change >1.5 and $p < 0.05$. Among the DEPs, 97 proteins were significantly upregulated and 105 proteins were significantly downregulated in the responders. The DEPs were then subjected to overrepresentation analysis in ConsensusPathDB (http://cpdb.molgen.mpg.de/)[64].

## GSEA

GSEA was conducted using the GSEA 4.0.3 software (http://software.broadinstitute.org/gsea/index.jsp)[65]. KEGG, Reactome, and HALLMARK gene sets downloaded from the MSigDB v7.1 were set as the background. $P$ value <0.05 was used as the cut-off.

## Immune, stromal, and pathway scores

Immune and stromal scores were inferred using the R package, ESTIMATE v1.0.11[42]. Although the ESTIMATE algorithm was designed to analyze transcriptome data, some studies have used it for proteome analysis[4,66]. These results indicate the feasibility of evaluating the engagement of each subtype of immune cells. Pathway gene sets were obtained from MSigDB (MSigDB v7.1, http://software.broadinstitute.org/gsea/msigdb/index.jsp) and pathway scores were computed using single-sample GSEA (ssGSEA)[67].

## Associations between clinical characteristics and the ccRCC proteome

The specific clinical information is presented in Supplementary Data 1. Only variables associated with prognosis were included in the analysis. Categorical variables, including MSKCC risk and IMDC risk, were analyzed using GSEA. Spearman's correlation was conducted on continuous variables, including tumor size, PLT level, and LDH level. Proteins significantly ($p$ value <0.05) associated with these clinical characteristics were subjected to overrepresentation analysis using ConsensusPathDB (http://cpdb.molgen.mpg.de/)[64]. The clinical characteristic-associated pathways are listed in Supplementary Data 5.

## Effects of CNAs

Spearman's correlations between CNA values (gene level and arm level) and protein abundances were calculated using 12,310 genes quantified at both CNA and proteome levels. CNAs with a significant correlation with proteins were selected based on $p$ values <0.05. Correlations were visualized using the R package multiOmicsViz. Genomic alterations that affect gene expression at the same locus are said to act in *cis* (vertical patterns in Supplementary Fig. 6d), whereas the impact of other locus was defined as a *trans* effect (diagonal patterns in Supplementary Fig. 6d).

## Survival analysis

The Kaplan–Meier method was used for survival analysis, and groups were compared using the log-rank test. Progression-free survival was analysed first, followed by overall survival. The R survival package 3.2-3[68] and Survminer 0.4.8 were used for statistical tests and visualization. The hazard ratio was calculated by Cox proportional hazards regression analysis.

## Immune subtype identification

To evaluate the impact of the tumor immune microenvironment on TKI therapy, the abundance of 64 different cell types was computed via xCell, based on the tumor proteomic profiles (Supplementary Data 6).

Consensus clustering was performed using the R package ConsensusClusterPlus based on the z-score normalized Raw enrichment scores of tumor samples. Specifically, 80% of the original ccRCC tumor samples were randomly subsampled without replacement and were partitioned into 3 major clusters using the Partitioning Around Medoids (PAM) algorithm, which was repeated 2000 times.

## Prediction of TFs

We obtained TFs that can bind to target gene promoter regions by JASPAR (http://jaspar.genereg.net) database and ChEA3[32] database.

## Cell culture

Human ACHN (ATCC, CRL-1611; RRID: CVCL_1067) cell were cultured in high-glucose Dulbecco's modified Eagle's medium (DMEM; HyClone) supplemented with 10% fetal bovine serum (FBS; Invitrogen), 100 units/mL penicillin (Invitrogen), and 100 μg/mL streptomycin (Invitrogen). 769-P (ATCC, CRL-1933; RRID: CVCL_1050) and 786-O cells (ATCC, CRL-1932; RRID: CVCL_1051) were maintained in RPMI 1640 medium (Invitrogen) containing 10% FBS. Cells were incubated in 5% $CO_2$ at 37 °C. Cells were transfected using Lipofectamine 2000 (Invitrogen).

## AA concentration used in vitro assay

To determine the optimal concentration of AA in vitro experiments, based on the product manual, a gradient of AA (including 0, 5, 25, and 100 μM) in the in vitro experiments was performed with renal cell lines. When the concentration of AA was set as 100 μM, we observed the dramatic inhibition effect of the protein expression. Once the concentration exceeds 100 μM, it will adversely impact the cell state. Hence, the concentration of AA in our experiment was set as 100 μM and this AA concentration is consistent with the previous study[69] about the AA on renal epithelial cells.

## Sunitinib concentration used in vitro assay

According to the in vitro cell assay experimental concentration recommended by the product manual (1–500 nM), the doses of sunitinib we used in treating cells were 200 nM. Specifically, to determine the optimal concentration of sunitinib in the in vitro experiments, based on the product manual and previous studies, we did the gradient experiments (including 0, 100, 200, and 300 nM). When the concentration of sunitinib was set to 300 nM, it can cause great damage to the cells. Finally, the concentration of Sunitinib in our experiment was set as 200 nM, which was consistent with the previous studies[70].

## Plasmid transfection

The overexpression plasmid of SP1 was purchased from GeneChem, and overexpression plasmids of LAMTOR4, CALU, MDH2, TBL2, and POR were purchased from Tsingke Biotechnology. For transient transfection, 1 μg of each plasmid was transfected using Lipofectamine 2000 (Invitrogen) according to the manufacturer's instructions.

## RNA interference

Synthetic oligos were used for siRNA-mediated silencing of SP1 (5′-CCAGGUGCAAACCAACAGAUU-3′), YY1 (5′- CGAUGGUUGUAAUAAGA AGUU −3′), SMAD4 (5′- UACCAUACAGAGAACAUUGGA −3′), SREBF2 (5′- CCUGAGUUUCUCUCUCCUGAA −3′), VHL (5′- GCUCUACGAAGAU CUGGAAUU −3′; 5′- GGCUCAACUUCGACGGCGA −3′; 5′- CUGCCAGUG UAUACUCUGA −3′), and scramble siRNA was used as a control. Cells were transfected with siRNAs using Lipofectamine 2000 according to the manufacturer's protocol. Knockdown efficiency was verified by qRT-PCR or western blotting.

## Transwell assay

Before the experiment, the matrigel matrix (BD, #356234) was melted overnight at 4 °C. The pipette tips were precooled on ice for 30 min,

and the matrigel matrix was diluted to a working concentration of 300 µg/mL with precooled serum-free medium. The diluted matrigel matrix was added into transwell chambers (Corning, #3422) and incubated in an incubator at 37 °C for 1 h. The pretreated cells were digested with trypsin, resuspended with serum-free medium, counted, and concentrated to $2.5 \times 10^5$ cells/mL. Take out the transwell chamber where the basement membrane has been formed, suck out the residual liquid in the chamber, add medium containing 20% FBS to the lower chamber of the chamber, and add cell suspension with different pretreatment to the upper transwell chamber. Then it was incubated in a cell incubator at 37 °C and 5% $CO_2$ for 24 h. Take out the chamber and wash it three times with PBS. Each chamber was fixed with 100% methanol at room temperature for 30 min and then stained with 0.1% crystal violet (Solarbio, #G1063) at room temperature for 20 min. Rub gently with a cotton ball to remove non-invasive cells from the upper compartment. Cells were observed in random five fields under a microscope, counted, and analyzed.

## Cell proliferation assay
Cell proliferation was assessed using the Cell Counting Kit-8 (Dojindo Laboratories). In brief, cells were seeded in a 96-well plate at $4 \times 10^3$ cells/well and allowed to adhere. Cell Counting Kit-8 solution (10 µL) was added to each well, and the cells were incubated in 5% $CO_2$ at 37 °C for 2 h. Cell proliferation was determined by measuring the absorbance at 450 nm.

## Western blot analysis
Cultured cells were lysed with 0.5% NP-40 buffer containing 50 mM Tris-HCl (pH 7.5), 150 mM NaCl, 0.5% Nonidet P-40, and a mixture of protease inhibitors (Sigma-Aldrich). After centrifugation at 13,80×g and 4 °C for 15 min, supernatants were collected for western blotting according to standard procedures. Antibodies against G6PD (#25413-1-AP, 1:1000), PGD (#14718-1-AP, 1:1000) were purchased from Proteintech. Antibodies against TKT (#PA5-56165, 1:1000), LAMTOR4 (#PA5-54301, 1:1000) were purchased from Invitrogen. Antibodies against MDH2 (#A13516, 1:1000), CALU (#A6538, 1:1000) were purchased from Abclonal. Antibodies against S6K (#5707, 1:1000) and pS6K (#9209, 1:1000) were purchased from Cell Signaling Technology. Antibody against VHL was purchased from Abcam (#ab270968, 1:5000). Antibody against Actin was purchased from Genscript (#A00702, 1: 800). Chemiluminescence was measured on a Typhoon FLA 9500 instrument (GE Healthcare). Uncropped and unprocessed scans of all the blots in this paper were provided in the Source Data file.

## IHC
We randomly collected three non-responders and three responder samples for pS6K IHC validation additionally. Sections of ccRCC and adjacent tissues were obtained from formalin-fixed, paraffin-embedded tissue blocks (not enrolled in the proteogenomic cohort). Immunostaining was carried out as reported previously[71,72]. Immunohistochemistry staining was performed using the avidin-biotin-peroxidase technique. The 4-µm-thick sections from representative FFPE blocks were deparaffinized in a series of xylene and rehydrated in a graded series of ethanol solutions. Endogenous peroxidase was quenched in 3% hydrogen peroxide in absolute methanol for 15 minutes at 37 °C Heat-induced antigen retrieval was carried out in 10 mM citrate buffer solution (pH 6.0) and then incubated with the primary antibody at 4 °C overnight. The following primary antibodies was assayed: rabbit anti-phospho-p70 S6 Kinase monoclonal antibody (Abclonal, #AP0502, 1:100). Immunostaining was performed with the EnVision system (Dako, Cytomation, Glostrup, Denmark). All slides were counterstained with hematoxylin, dehydrated, and mounted. Immunostaining was quantified based on the number of immunoreactive cells (quantity score) and the staining intensity (intensity score), as reported[71,72].

## Lactate measurement
Cultured cells were collected and washed with PBS. Add homogenization medium to the cells according to the ratio of 500 µL/10⁶ cells, and mechanically homogenize the cells to fully break. Centrifuge at 4 °C, 1000×g for 10 min, and collect the supernatant. The lactate concentration in the supernatant was determined utilizing the L-lactic acid/lactate (LA) Colorimetric Assay Kit (Elabscience, #E-BC-K044-S), according to the manufacturer's instructions.

## eGFR measurement
CKD-EPI equation[73] was used to estimate the eGFR and the equation is listed as below: eGFR = 141 × min(Cr/κ,1)α × (max(Cr/κ,1)−1.209) × (0.993Age) × 1.018 [if female] × 1.159 [if black]. Cr is serum creatinine in mg/dL, κ is 0.7 for females and 0.9 for males, α is −0.329 for females and −0.411 for males, min indicates the minimum of Cr/κ or 1, and max indicates the maximum of Cr/κ or 1.

## Karnofsky performance score (KPS)
Karnofsky performance score (KPS) is a clinical assessment tool used to assess the overall health of patients. KPS assesses an individual's overall functional status on an 11-point scale, in increments of 10, where a score of 0 for death and 100 for normal function, higher scores signified better functional status[74]. KPS of patients in this cohort range from 70 to 100. A score of 70 shows patients who, despite the inability to normal activity, can take care of themselves; a score of 80 shows patients who are able to carry on normal activity with effort, and have some signs or symptoms of disease; a score of 90 shows patients who are able to carry on normal activity and have minor signs or symptoms; the score 100 shows patients who are no complaints and no evidence of disease.

## Classifier construction
The machine learning framework was built on Python (version 3.9.0) using the following libraries: scikit-learn (version 1.2.1), numpy (version 1.16.4), scipy (version 1.3), and pandas (version 1.5.2).

## Feature selection
For the feature selection of the predictive model, we applied the following pipeline: the first step removed all features with a mutual Pearson correlation above 0.8, retaining only the one with the highest correlation with the response variable. The second step used the chi-square test and F test, which is embedded in SelectKBest function of scikit-learn library to select the best proteome features. The third step applied z-score scaling to the remaining features.

## Model construction
Considering the model's complexity and interpretability, we chose the random forest model to construct the predictive model. We split the dataset into the train cohort (70%) and test cohort (30%). Hyperparameters were optimized using fivefold cross-validation in the training set to maximize the area under the receiver operating characteristic (AUC ROC) curve. As for the proteome-based model, after applying the model construction pipeline, the Random forest model was implemented with class_weight was balanced, max_features was 1, min_samples_leaf was 14, min_samples_split was 14, n_estimators was 150. As for the multi-omics-based random forest model class_weight was balanced, max_features was 2, min_samples_leaf was 8, min_samples_split was 5, and n_estimators was 90.

## Model performance evaluation
The model performance was evaluated by the different evaluation scores, including precision, recall, ROC-AUC, F1 score, and balanced accuracy for the classification.

## Statistical analysis

Quantification methods and statistical analysis methods for proteomic and integrated analyses were mainly described and referenced in the respective Method Details subsections. Additionally, standard statistical tests were used to analyze the clinical data, including but not limited to the Wilcoxon rank-sum test, Fisher's exact test, Kruskal–Wallis test, and log-rank test. Statistical significance was considered when $p$ value <0.05. Kaplan–Meier plots (log-rank test) were used to describe survival. Variables associated with overall survival were identified using univariate Cox proportional hazards regression models. Significant factors in univariate analysis were further subjected to a multivariate Cox regression analysis. All the analyses of clinical data were performed in R, Python, and GraphPad Prism.

## Reporting summary

Further information on research design is available in the Nature Portfolio Reporting Summary linked to this article.

## Data availability

The proteome and phosphoproteome raw datasets have been deposited to the ProteomeXchange Consortium (dataset identifier: PXD042844) via the iProX partner repository (https://www.iprox.cn/)[75] under Project ID: IPX0002932000. The raw WES and RNA data are available in the Genome Sequence Archive (GSA) under restricted access HRA003490. The raw sequencing data are available under controlled access due to data privacy laws related to patient consent for data sharing and the data should be used for research purposes only. According to the guidelines of GSA-human, all non-profit researchers are allowed access to the data, and the Principal Investigator of any research group can apply for Controlled-access of the data. The user can register and log in to the GSA database website (https://ngdc.cncb.ac.cn/gsa-human/) and follow the guidance of "Request Data" to request the data step by step. The approximate response time for accession requests is about 2 weeks. The access authority can be obtained for Research Use Only. The user can also contact the corresponding author directly. Once access has been granted, the data will be available to download for 3 months. The remaining data are available within the Article, Supplementary Information, or Source Data file. Source data are provided with this paper.

## Code availability

No special code was used in this study, and code for all figures in the study are available for research purposes from the corresponding authors on request.

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

## Acknowledgements

This work is supported by National Key R&D Program of China (2022YFA1303200 [C.D.], 2017YFA0505102 [C.D.], 2016YFA0502500 [C.D.], 2018YFA0507501 [C.D.], 2017YFC0908404 [C.D.], 2020YFE0201600 [C.D.], 2018YFE0201603 [C.D.], 2017YFA0505101 [C.D.], 2019YFC1316005 [H.Z.]); The National Natural Science Foundation of China (31770886 [C.D.], 31972933 [C.D.], 31700682 [C.D.], 82172817 [Y.Q.], 82172741 [D.Y.]); The Shanghai Municipal Science and Technology Major Project (2017SHZDZX01 [C.D.]; The Major Project of Special Development Funds of Zhangjiang National Independent Innovation Demonstration Zone (ZJ2019-ZD-004 [C.D.]); Supported by the Shuguang Program of Shanghai Education Development Foundation and Shanghai Municipal Education Commission (19SG02 [C.D.]); Sponsored by Program of Shanghai Academic/Technology Research Leader; Project supported by the Young Scientists Fund of the National Natural Science Foundation of China (32201215 [J.W.F]); Sponsored by Shanghai Sailing Program (J.F.); The Fudan original research personalized support project (C.D.); Natural Science Foundation of Shanghai (20ZR1413100 [H.Z.]); Shanghai "Science and Technology Innovation Action Plan" medical innovation research Project (22Y11905100 [Y.Q.]); Shanghai Municipal Health Bureau Project (2020CXJQ03 [D.Y.]); Beijing Xisike Clinical Oncology Research Foundation (Y-HR2020MS-0948 [H.Z.]); and Shanghai Anti-Cancer Association Eyas Project (SACA-CY21A06 [W.X.] and No. SACA-CY21B01 [A.A]).

## Author contributions

H.Z., J.-Y.Z., Y.Q., D.Y., and C.D. conceived and planned the project. H.Z., X.T., G.-H.S., W.X., A.A., Y.Z., D.-L.C., Y.W., X.-Q.W., H.-L.G., and M.-H.S. were responsible for sample and clinical information collection. L.B., X.W., X.P., and G.Y. contributed to sample preparation. H.Z., X.-Q.W., and F.X. planned and carried out the validation experiments. H.Z., L.B., X.-Q.W., J.F., X.W., G.Y., J.L., and Y.L. analysed the data and contributed to the interpretation of the results. C.D. took the lead in writing the manuscript. All authors provided critical feedback and helped shape the research, analysis, and manuscript.

## Competing interests

The authors declare no competing interests.
