## [Peer Review File · Nature Communications]

REVIEWER COMMENTS

Reviewer #1 (Remarks to the Author): Expert in renal cell carcinoma genomics

Qu et al. report their results from a combined proteomic, whole genome sequencing and clinicopathological analysis of a cohort of clear cell renal cell carcinoma (ccRCC) patients that were treated with the commonly used receptor tyrosine inhibitor sunitinib. The study is a nice addition to the already significant series of large-scale profiling studies on ccRCC, with the novelty coming from the unique patient cohort of Chinese patients that all are under the same therapy, and the use of proteomics for molecular profiling. The fact that the mutational signature caused by aristolochic acid (AA) is particularly prevalent in Chinese patients, due to AA being prevalent in traditional medicine in Asia, highlights the need for cancer profiling studies from different populations. Overall, the manuscript is well written and the data are presented clearly. I have the following specific comments.

1. The difference between predictive and prognostic biomarkers should be made clear throughout the manuscript. For example on line 63, the authors refer to biomarkers that would predict TKI efficiency. However, without having a control group it is impossible to know whether the markers are associated with prognosis independently of therapy. The issue comes up several times in the text and should be addressed throughout data presentation and discussion.
2. In several places the text applies circular logic when identifying clinical or molecular correlates. For example, the authors find that PFS is different between responders and non-responders. Then they find that smaller tumors tended to respond better, after which they show that tumor size is associated with survival. Other examples come from the mutational signature and mutation analyses. The AA signature is linked to smaller tumors and more mutations, after which it is stated that smaller tumors have more mutations. Similarly, the authors first identify mutations that correlate with survival and then split patients using mutational data into groups, which are then stated to have different survival. While some of these observations are worth mentioning explicitly, presenting them as “findings” is misleading.
3. Related to point #2, the authors refer to “superimposition” when suggesting that different molecular features might be similarly affecting patient prognosis. The authors try to deal with this by doing several univariate tests. The correct approach would, however, be to use multivariate models to test the independence of each molecular parameter with patient survival.
4. The authors state that the transcriptome and proteome don't correlate (line 268). It would be interesting to test this more formally. Is it possible to use other molecular data sets (e.g. Wang et al. Nat Commun 2020, <https://doi.org/10.1038/s41467-020-14601-9>) for example to compare the AA signature

at the protein and RNA level? Also, how does the protein-level clustering agree with other ccRCC proteome data sets and immunophenotyping (Clark et al. Cell 2019; Chevrier et al. Cell 2017).

5. In some ccRCC cohorts the fraction of VHL inactivated tumors is significantly higher than in the TCGA cohort (e.g. the TRACERx renal cohort, Turajlic et al. Cell 2018). This should be discussed.

Minor

Line 37, the incidence numbers appear to be from the US only. This should be clarified. Also, the 2020 diagnoses have not yet resulted in 13,780 deaths. The wording should be corrected.

Line 142, "... suggesting the specific mechanism..." What mechanism are the authors referring to?

Line 168, "... poor prognosis caused..." Would be more accurate to talk about associations.

Line 193, Is there evidence of curative effects of sunitinib?

Line 256, what data is there that VHL mutations cause glycolysis addiction?

Line 389, should it be "in any of the samples"?

Line 401, "we established a comprehensive landscape of Sunitinib drug response...", but there is no non-treated control group.

Figure 1B, tumour size units?

Reviewer #3 (Remarks to the Author): Expert in clear-cell renal cell carcinoma clinical research and genomics

The authors can be congratulated to their very extensive and carefully performed research. They have established a comprehensive landscape of Sunitinib drug response in renal cell carcinoma both at the genome and proteome levels, which has the potential to make a significant clinical impact.

Major Comments

1. A number of patients have reduced kidney function, diabetes and hypertension. Therefore this subgroup might well have chronic kidney disease with histological alterations (e.g. hypertensive/diabetic nephropathy) in the peritumoral tissue. This may influence the results of this investigation.
2. Inherent tumor heterogeneity, a well-known phenomenon in ccRCC, has not been assessed.
3. Analyses of (selected) miRNA regulating the response classifier could be interesting from a possible therapeutic point of view to ameliorate Sunitinib response.
4. Median follow-up was 28.4 months. This is relatively short since metastases, esp. in the lower risk patients, might well occur after this point of time.

Minor comments

1. Patient characteristics: eGFR would be better than creatinine/BUN. Furthermore, not entirely clear if all patients were subjected to full or partial nephrectomy.
2. How long and variable was the intraoperative ischemia time prior to tissue harvesting?
3. The authors may comment why Leibovich scoring system was not included.
4. Concomitant (anti-tumor/anti-inflammatory) therapy? Sunitinib dose identical among patients?

Reviewer #4 (Remarks to the Author): Expert in cancer proteomics and proteogenomics

Qu and co-workers performed whole exome sequencing and DDA-based mass spectrometry analysis of samples from human ccRCC tumors and adjacent tissue, from 68 patients with advanced tumor and from another 47 patients who developed metastatic disease after surgery and treatment with the TKI sunitinib. They identified a number of differentially expressed proteins in responders and non-responders to the TKI, and several somatic mutations and copy number alterations. Interestingly, they associate a mutational signature with some herb that is applied Chinese traditional medicine and contains the compound aristolochic acid (AA), with better survival of ccRCC patients. This herb, and the effective agent (AA) have previously been associated with increased development of ccRCC. Here the authors find that the mutational signature that is induced by AA seems to correlate with better response to sunitinib treatment. The authors further conclude that the mutations having been caused by the drug lead to an addiction of tumor cells to glycolysis and that this is beneficial for the patients. Another interesting findings appears to be that mutated VHL was found to correlated with better patient outcome and with smaller tumor sizes. These findings are surprising since hypoxia and angiogenesis are factors that commonly associated with progressive disease and metastasis. Along the lines of this, the authors might consider PMID:33996558. There, VHL is described as a 'main gene (...) involved in ccRCC carcinogenesis [however, is] not the most relevant for assessing survival.'

I have a number of concerns with this study.

The first sentence of the abstract suggests that the authors performed an integrated proteogenomic characterization of patients' tumors who were under ('undergoing') therapy. They further claim that they revealed the molecular basis of differential clinical outcomes with tyrosine kinase inhibitor (TKI) therapy. Since the tumors had been surgically removed before therapy (also adjacent tissue was removed with the tumor), the molecular analysis of all samples revealed the proteomic status of the tumors prior to therapy and does not necessarily reveal molecular causes of drug failure – not until the hypothesized mechanisms have been experimentally proven. The authors performed a large number of bioinformatics analyses and present a similarly large number of data, however, all this is bioinformatics and imputed from literature (which indeed is not always along the same lines – e.g., angiogenesis has been associated with progressive disease on sunitinib PMID:33593885, PMID:31268155).

No information on the whole-exome sequencing is provided (enrichment protocol used, numbers of reads sampled, coverage of exomes, numbers of variants detected, availability of sequencing data). For example, the graphs in Figure 7F do not provide any numbers of CNAs or mutations having been found. It is interesting to note that the authors did not apply RNA-sequencing, which would have been a complementary approach to their total proteomic experiments. The authors comment on the low level of concordance between RNA-seq and proteomic data. This level of concordance/discordance is a matter of debate in the community. The authors could have added to this with solid RNA-seq data.

I am not sure about the clinical regimen that were applied. The patients in the cohort are reported to having been 'treated with TKIs' between January 2008 and December 2019 (Materials and Methods). 1. The authors need to make clear which TKIs were given (only sunitinib?). 2. ccRCC is still mostly treated in the adjuvant setting. In this way, Figure 1A would be at least misleading as tumor sampling was very likely prior to treatment. This point is important, for example as the authors are surprised that they did not see effects of sunitinib on the abundance of some target proteins (lines 260ff). The authors should realize that the activities of signaling proteins has higher relevance on pathway activities than the abundance of individual components, particularly when these TKIs are wildtype in sequence and thus require activation by respective ligands to become activated themselves. In this way, a phospho-proteomic analysis of tumors might have been an idea.

In the proteomic experiment, 10,075 proteins are reported to having been identified in responders and non-responders. The total number of proteins that was identified is 12,310 (R, R+NR, NR). The authors should be congratulated to these high numbers of identified proteins, as DDA in combination with the HF-X mass spectrometer commonly detects much fewer proteins. How many of those 10,075 proteins were identified in 90% of the samples? How many of the 1208 and 1027 proteins that were identified specifically in the non-responders and responders, respectively, were recurrently identified? The authors write in lines 129f that proteins identified in at least 25% of the samples were used for the analysis. How many proteins were detected in at least 25% of the samples and, consequently, used in the analysis (Line 584 describes 7451 proteins, which would still be many compared to comparable proteomic studies)? Some figures (e.g., Figure 2K, 2I, 3B) suggest that the indicated proteins were detected in all patient samples that are on display. For how many of those patients has the data been imputed? Given that the patient subgroups were sometimes rather small, how many patients were included in any such subgroup that met this criterion of 25%?

In line 587f, the authors write that among the DEPs, 1296 proteins were significantly upregulated and 699 proteins were significantly downregulated in the responders. Were all these proteins detected also in the non-responders? If the criterion of identification in >25% of patients was applied, how many of the proteins were detected/not detected in the non-responders at levels similar to the ones in responders? What was the range of expression? Where is the raw data?

The authors highlight some proteins and pathways that are up- or down-regulated in one or the other condition. It should be interesting to learn how many genes/proteins were positively identified (i.e., with data) to come to these lists.

The authors write that glycolysis addiction was caused by VHL mutations and was associated with sunitinib therapeutic response, and that the AA signature could enhance glycolysis addiction. The addiction of tumor cells to glycolysis has been long recognized as a driving factor in tumorigenesis and progression. Here, the authors add a strong point of concern speaking against the application of AA in humans.

In lines 319ff the authors write that tumor size and LDH levels were associated with diverse proteomic features, the revealing the molecular mechanisms underlying the influence of clinical parameters on sunitinib therapeutic response (Figure 4M). This and some other claims should be tested (in vitro/in vivo) until suggested mechanisms are potentially verified.

In lines 612f, spearman's correlations between CNA values (gene level and arm level) and protein abundances were calculated using 12,310 genes quantified at both CNA and proteome levels. How reliable is this analysis when quite some of the proteomic data seems to be unreliable (was not regarded in other analysis)?

Other points.

Line 82: what are 'strict criteria' for patient sampling?

In lines 134ff the authors write that the mutational signature SBS22, associated with aristolochic acid (AA) exposure, was higher in the non-responder than in the responder group (14.4% vs. 20.3%) and showed better overall survival upon sunitinib treatment. 1. In the text, the authors write that the AA signature would be correlated with better survival. Am I correct to assume that there is a swap of non-responders and responders in that sentence in lines 134f. 2. Did the authors find this signature also in the tissue adjacent to the tumor, or is this mutational signature confined to the tumor cells? What was purity of the tumors and what was allele frequency of the mutations and mutational signatures? AA has been associated with DNA damage and somatic mutations (reference Ng et al., 2017). Did cells of the TME harbor the same signatures or were these cells wildtype for the respective positions? If the TME was affected by AA, why not the adjacent tissue? 3. Looking at Figure S2B, the numbers of responders and non-responders showing this AA signature appear to be quite low (4 responders, 15 non-responders), which fits the percentages given in the text. These tumors carry quite variable ratios of other mutational signatures (SBS1, SBS5 and SBS40 are shown – how about tobacco-related signatures that might be present in smokers?): how were these other signatures treated in the bioinformatics analysis to exclude any other potential confounding factors?

Some of the regression lines that are shown in Figure 4J/L are interesting. The line in the right panel of Figure 4L is way below the cloud that has the most data points. The graph in the left panel of this figure has some outliers, but the majority of data points seem to be independent of LDH levels.

In line 341 the citrate cycle and the TCA cycle are mentioned to be upregulated in the Cold group. What is difference between citrate and TCA cycle?

In line 370 the authors speak of redundant angiogenesis (driven by TGFB1 resulted in sunitinib resistance). What is 'redundant' angiogenesis?

In lines 404f the authors write that strong regional characteristics would indicate the necessity of considering the genetic background of the Chinese population in ccRCC treatment. The authors have not really found differences in the genetic background of Chinese vs. other populations. Rather, they found the environmental factor aristolochic acid and the genetic (better: somatic) alterations that are induced by this drug to be necessary to be taken into account. The authors do not distinguish between true genetic and environmental factors in their study.

The finding that this AA signature is caused by aristolochic acid strongly speaks against ingestion of this herb. While the AA signature seems to be correlated with better survival in ccRCC, it is very likely correlated with worse survival in other entities, many of which are addicted to glycolysis even in normoxia (i.e., Warburg effect).

In line 424f the authors write that the 7q amplification was confirmed to upregulate mTOR signaling pathways and amino acid turnover in ccRCC. I could not find the data that would have confirmed causality of the correlation that is presented in the manuscript. Later, in lines 477f, the authors claim to have identified the molecular mechanisms underlying sunitinib response. This claim is not sufficiently backed up with experimental data.

In lines 451ff the authors suggest that the platelets in the TME were responsible for sunitinib failure. Several causes of poor treatment-response have been identified/published (compare e.g., PMID:25446042). The authors should consider testing their hypothesis, such mechanism could indeed be interesting, if validated.

Reviewer #5 (Remarks to the Author): Expert in renal cell carcinoma sunitinib therapy and immunotherapy

The paper deals with “proteogenomics” analyses on over 100 clear-cell mRCC top come up with associations with “clinical outcomes”. The paper, from a clinical standpoint, is confusing especially with multiple definitions on what constitutes activity. In the abstract alone, we see terms such as “treatment failure”, “treatment benefit”, “therapeutic outcome”, “prognosis” and “treatment prediction”. While it is quite interesting to focus on the Chinese patients as most of biomarkers of RCC responses have been in Caucasians, this interesting aspect of the study is dampened by the fact that TKI, as single agents, are not standard of care in mRCC anymore. In addition, and despite FDR of 1% is used for Proteome identification and quantification, it is not clear where else it is used down the road especially with this “unsupervised and unplanned”. What about other clinical metrics like PFS and OS and why responses (since they are the ones used as a primary correlate for efficacy) were not centrally reviewed? It is also interesting that responses did not differ by MSKCC/IMDC criteria that are known for a long time to prognosticate well in mRCC. Figure S1G (patients’ baseline characteristic. Also what is the AA signature’s could enhance 3p loss/VHL mutation-induced glycolysis addictions) do not have data on exposure to AA. So how do we assume these patients had any exposure? There is a claim that AA signature could enhance 3p loss/VHL mutation-induced glycolysis addiction. But what is the MOA? This is a premature statement. Finally the “validation” cohort needs more information. How independent is this cohort or did you split your cohort into training and “validation”. In summary, several problems with this report that need to be addressed in great details.

Reviewer #1 (Remarks to the Author): Expert in renal cell carcinoma genomics

Qu et al. report their results from a combined proteomic, whole genome sequencing and clinicopathological analysis of a cohort of clear cell renal cell carcinoma (ccRCC) patients that were treated with the commonly used receptor tyrosine inhibitor sunitinib. The study is a nice addition to the already significant series of large-scale profiling studies on ccRCC, with the novelty coming from the unique patient cohort of Chinese patients that all are under the same therapy, and the use of proteomics for molecular profiling. The fact that the mutational signature caused by aristolochic acid (AA) is particularly prevalent in Chinese patients, due to AA being prevalent in traditional medicine in Asia, highlights the need for cancer profiling studies from different populations. Overall, the manuscript is well written and the data are presented clearly. I have the following specific comments.

Response: We appreciate the reviewers for the positive evaluation and constructive comments. We have revised the manuscript according to the comments. Especially, we modified the terms used in our manuscript and made a logical combing to our findings in the revision. The point-to-point responses were as follows.

Q1. The difference between predictive and prognostic biomarkers should be made clear throughout the manuscript. For example on line 63, the authors refer to biomarkers that would predict TKI efficiency. However, without having a control group it is impossible to know whether the markers are associated with prognosis independently of therapy. The issue comes up several times in the text and should be addressed throughout data presentation and discussion.

Response:

Thank you for your professional suggestions. We sincerely thank the reviewer for careful reading. In this study, we defined prognostic biomarkers as molecular signatures which was significantly associated with patient survival. The Kaplan–Meier method and Cox proportional hazards regression analysis were used to identified prognostic biomarkers. The predictive biomarkers in this study were defined as the molecular signatures showed alteration frequencies and differential expression levels between Responders and Non-Responders.

The review is correct that without having a control group it is impossible to know whether the markers are associated with prognosis independently of therapy. There were 270 prognostic biomarkers identified in this Sunitinib treatment cohort. To further evaluate whether prognostic markers are associated with survival independently of therapy, we collected another independent cohort with 37 cases of ccRCC patients, who were not received any treatment after surgery, as control (**Methods** in revision). These 37 cases had complete follow-up information. We further evaluated the associations between protein expression levels and patient survival. The results showed that the expression levels of 630 proteins were significantly associated with patient survival in the non-treated control group. Compared these 630 prognostic markers in non-treated group with 270 prognostic markers in Sunitinib treatment group, only 19 proteins were found in both groups (**Figure RL1A**), indicating the prognostic values of protein expression levels were profoundly dependent on whether or not the patients have ever been prescribed Sunitinib treatment after surgery. As the prognostic biomarkers mentioned in the original manuscript, CD274, CRP, C8A, NOVA2, RRAGB, NUP50, prognostic biomarkers in treatment group, were not associated with patient survival in the non-treated control group (**Figure RL1B**). Higher expression of BID was associated with better survival in both Sunitinib treatment group and non-treated control group (**Figure RL1B**). We added these results in **Supplementary Figure 1C** the revision and proofread the descriptions about predictive and prognostic biomarkers throughout the manuscript.

Figure RL1.

A, Venn plot of prognostic significance of protein expression levels in Sunitinib treatment group and non-treated group.

B, Kaplan–Meier curves of PFS for patients with different protein abundances in non-treated control group (log-rank test).

Q2. In several places the text applies circular logic when identifying clinical or molecular correlates. For example, the authors find that PFS is different between responders and non-responders. Then they find that smaller tumors tended to respond better, after which they show that tumor size is associated with survival. Other examples come from the mutational signature and mutation analyses. The AA signature is linked to smaller tumors and more mutations, after which it is stated that smaller tumors have more mutations. Similarly, the authors first identify mutations that correlate with survival and then split patients using mutational data into groups, which are then stated to have different survival. While some of these observations are worth mentioning explicitly, presenting them as “findings” is misleading.

Response:

We thank the reviewer’s comments. The reviewer is correct that there were logic cycles in our manuscript.

About the associations among responders/non-responders, tumor sizes, and PFS

As mentioned in the original manuscript, we disclosed the associations among responders/non-responders, tumor sizes, and PFS. To avoid circular logic, we made a logical combing to our findings in the revision. In detail, we found that Responders and Non-Responders had no significant difference in tumor size in general (**Figure RL2A**). However, when tumor sizes were above or below certain thresholds, Non-Responders or Responders were significantly enriched respectively (Fisher’s exact test, $p = 0.019$) (**Figure RL2B-C**). Specifically, Responders were overrepresented in patients with tumor size < 5 cm (small tumors), while Non-Responders were overrepresented in patients with tumor size > 9 cm (large tumors) (**Figure RL2C**). Therefore, we divided the patients into three groups based on the tumor size: small (tumor size < 5 cm; $n = 16$), median (tumor size > 5 cm and < 9 cm; $n = 61$), and large (tumor size > 9 cm; $n = 38$). The survival curves of patients with these three groups indicated that tumor size (less than 5cm or more than 9 cm) could affect the response of patients to Sunitinib and further manifested as the PFS benefit or not (**Figure RL2D**).

Figure RL2.

A, Tumor sizes between responders and non-responders (t test).

B-C, Tumor sizes distributions of Responders and Non-Responders (Fisher's exact test).

D, Kaplan–Meier curves of progression-free survival (PFS) for patients with different tumor sizes (log-rank test).

About the associations between AA signature, tumor sizes, and TMBs

We revised our descriptions about the associations between AA signature, tumor sizes, and tumor mutation burdens (TMBs). It was interesting that patients with SBS22, associated with aristolochic acid (AA) exposure, showed better PFS in Sunitinib treatment (GB-Wilcoxon test, $p = 0.038$) (**Figure RL3A**). To find out the underlying mechanisms of which patients with AA signature were more likely to benefit from Sunitinib treatment, we compared the clinical features and protein expressions between patients with and without AA signature. We found that patients with AA signature showed smaller tumor sizes compared with patients without AA signature (t test, $p = 0.0062$) (**Figure RL3B**). Consistently, we found that DNA replication (MCM5, MCM6) was significantly downregulated in tumors with AA signatures (**Figure RL3C**). Moreover, we found that pentose phosphate metabolism was significantly downregulated, whereas glycolysis was upregulated in tumors with the AA signature, indicating that glucose was shunted away from the pentose phosphate pathway into glycolysis in tumors with AA signature (**Figure RL3D**). We further conducted *in vitro* validation experiments, verifying AA exposure downregulated pentose phosphate pathway (**Figure RL3E**) and decreased the ribose-5-phosphate (**Figure RL3F**), which is the precursor of nucleotide. Moreover, AA exposure enhanced the effect of Sunitinib in inhibiting proliferation and invasiveness of ccRCC cancer cells (**Figure RL3G-H**). To make our point clear,

we removed the discussions about TMB in the revision, due to TMB was beyond our storyline.

These modifications were added in **Figure 2** the revision.

Figure RL3.

A, Kaplan–Meier curves of PFS for patients with and without AA signature (GB-Wilcoxon test).

B, Tumor sizes between tumors with and without AA signature (t test).

C, Pathways enriched by differentially expressed proteins between patients with or without the AA signature.

D, Glucose metabolism alteration caused by the AA signature.

E, AA treatment inhibits the expressions of G6PD, PGD and TKT in time-and dose-dependent manners.

F, The ribose-5-phosphate concentrations in cells treated with AA or not.

G, The cell proliferation measured by MTT assay.

H, Transwell detected the effect of AA and Sunitinib treatment on cell invasiveness.

About the associations among responders/non-responders, mutations, and PFS

To uncover the mutations might affect Sunitinib treatment, we compared the associations of mutations and patients' survival between the TCGA cohort and our cohort (Sunitinib treatment cohort) (**Figure RL4A**). The results showed that the associations of *VHL*, *ACAN*, *CNTNAP4*, *HUWE1*, *ZNF236*, *ZMYM4*, and *MYH1* mutations with survival were only observed in the Sunitinib treatment cohort but not in the TCGA cohort (**Figure RL4A**), indicating the associations between these mutations and patient survival was dependent on Sunitinib treatment. *VHL* mutation, as the most frequent mutation in ccRCC, was associated with patient survival in Sunitinib treatment, which caught our attention. To delineate the association about *VHL* mutation and survival, we showed the Kaplan–Meier curves for patients with and without *VHL* mutations (**Figure RL4B**). It showed that patients with *VHL* mutations had longer survival in Sunitinib treatment. Moreover, we knocked down *VHL* in ACHN cells which belong to *VHL*-proficient cells, to mimic *VHL* loss-of-function mutations (**Figure RL4C**). We found *VHL*-knockdown cells had reduced invasiveness under the treatment of Sunitinib, indicating *VHL* deficiency enhanced the antitumor effect of Sunitinib (**Figure RL4D**).

Figure RL4.

A, Comparison of the effect of mutations on the OS among this study and TCGA cohorts (log-rank test).

B, Kaplan-Meier curves of PFS for patients with or without *VHL* mutations (log-rank test).

C, Transfection efficiency of *VHL* siRNA detected by qRT-PCR and western blot.

D, Transwell detected the effect of *VHL* knock down and Sunitinib treatment on cell invasiveness.

Q3. Related to point #2, the authors refer to “superimposition” when suggesting that different molecular features might be similarly affecting patient prognosis. The authors try to deal with this by doing several univariate tests. The correct approach would, however, be to use multivariate models to test the independence of each molecular parameter with patient survival.

Response:

Thanks for reviewer’s constructive suggestion. In the original manuscript, we performed Cox regression analysis to determine the correlation between arm-level CNAs and clinical outcomes. The results showed that gains of 3q, 7p, 7q, and 8q were associated with poorer PFS of patients (**Figure RL5A**). According to the reviewer’s suggestion, we performed multivariate analysis using 3q gain, 7p gain, 7q gain, and 8q gain as covariates. We found that occurrence of these four CNA events (gains of 3q, 7p, 7q, and 8q) (**Figure RL5B**), and 7p gain and 7q gain were even totally overlapped. We calculated variance inflation factors (VIFs) of these four covariates to assess multicollinearity. The results showed that there was multicollinearity in this model (7p gain, VIF = 9.276; 7q gain, VIF = 9.826). Therefore, we performed multivariate analysis of 3q gain, 8q gain and 7p or 7q gain. The results showed 7p gain (HR = 1.71, p = 0.045) and 7q gain (HR = 1.91, p = 0.015) were the dominant arm-level CNA events associated with PFS (**Figure RL5C**). Correspondingly, as the associations between 3q/8q gains and patient survival were dependent on the gains of 7p/7q, we removed the part about the proteomic alterations of these four gain events (see Figure 2I in the original manuscript) and focused on the proteomic consequences of 7q gain instead (see **Figure 3** in the revision).

Figure RL5.

A, Cox regression analysis of significant arm-level CNA events, based on the PFS.

B, Co-occurrence analysis of gains of 3q, 7p, 7q, and 8q.

C, Multivariate analysis of 3q gain, 7p/7q gain and 8q gain.

Q4. The authors state that the transcriptome and proteome don't correlate (line 268). It would be interesting to test this more formally. Is it possible to use other molecular data sets (e.g. Wang et al. Nat Commun 2020, <https://doi.org/10.1038/s41467-020-14601-9>) for example to compare the AA signature at the protein and RNA level? Also, how does the protein-level clustering agree with other ccRCC proteome data sets and immunophenotyping (Clark et al. Cell 2019; Chevrier et al. Cell 2017).

Response:

Thank the reviewer for the comments. To better response to reviewer's concerns, we would like to response in two parts.

First, to further assess the correlation between transcriptome and proteome, we performed transcriptome sequencing in 94 samples in this cohort (**Methods** in the revision). We identified 12,276 protein-coding genes with median fragments per kilobase of transcript per million fragments mapped (FPKM) of more than 1 (**Supplementary Table 3** in the revision). Our data showed that the mRNA-protein correlation was moderate with sample-wise median Spearman's correlation of 0.36 (**Figure RL6A**), consistent with previous report (PMID: 32649875). According to reviewer's suggestion, we compared the AA signature at protein and mRNA level. The results showed that differentially expressed proteins and mRNA were overlapped in a small portion (**Figure RL6B**). Specifically, 14 upregulated genes (84 upregulated protein and 203 upregulated mRNA) and 10 downregulated genes (470 upregulated protein and 105 upregulated mRNA) were commonly identified at both mRNA and protein levels in tumors with AA signature. Interestingly, we found that metabolism related pathways were upregulated in patients with AA signature, while immune related pathways were downregulated at both protein and mRNA levels (**Figure RL6C-D**). These results indicated that although mRNA and protein did not show high correlation, mRNA and protein exhibited concordance in bioprocess regulation.

Figure RL6.

A, Boxplot showing sample-wise mRNA-protein correlation.

B, Venn plots showing the identified differential expressed genes between tumors with and without AA signature at mRNA and protein levels.

C-D, Pathways enriched by differentially expressed mRNA between tumors with or without the AA signature.

Second, according to reviewer's suggestion, we conducted immune subtyping using proteome data of CPTAC ccRCC samples (PMID: 31923397) in the same way as we used in this study. We performed consensus clustering using xCell scores, stratifying CPTAC samples into 3 subtypes (**Figure RL7A**). Specifically, the subtype with high T-cell immune infiltration was considered as the T-cell infiltrated group, the subtype contained high level of progenitor-cell (including CMP, MPP) was considered as progenitor-cell infiltrated subtype, and the rest subtype was considered as Cold tumors. The CD8+ inflamed tumors in the CPTAC study were mainly classified to T-cell infiltrated group (n = 22, 88.0%); CD8- inflamed tumors in the CPTAC study were mainly grouped into Progenitor-cell infiltrated subtype (n = 16, 76.2%); VEGF immune-desert tumors in the CPTAC study were largely assigned to Cold subtype (n = 26, 83.8%); Metabolic immune-

desert tumors in the CPTAC study were assigned to Cold (n = 10, 38.5%) and T-cell infiltrated (n = 46.2%) groups (**Figure RL7B**). These results showed that our proteome-derived immune subtyping was highly consistent with CPTAC ccRCC proteome data. More importantly, the concordance of our data and CPTAC data in proteome-based immune subtyping indicated that this strategy was also applicable for clinical treatment decisions in other cohorts.

As for Chevrier *et al.*'s study, the authors deposited the mass cytometry data at CytoBank (<https://premium.cytoBank.org/cytoBank/projects/875>), and we could not access the data.

Meanwhile, there was not sufficient metadata in the Supplementary information of Chevrier *et al.*'s study.

Figure RL7.

A, Heatmap showing the representative cell type of different immune subtypes.

B, Proteome-derived immune subtyping of the CPTAC ccRCC cohort.

Q5. In some ccRCC cohorts the fraction of *VHL* inactivated tumors is significantly higher than in the TCGA cohort (e.g. the TRACERx renal cohort, Turajlic et al. Cell 2018). This should be discussed.

Response:

Thanks for the constructive comment. Following reviewer's suggestion, we analyzed the *VHL* mutation frequencies in ccRCC from 5 different clinical datasets, including this study, another Chinese cohorts (PMID: 32029730), TCGA cohort (PMID: 23792563), European cohort (PMID: 25351205), and TRACERx Renal cohort (PMID: 29656894). The *VHL* mutation frequencies were

determined to be 65% (This study, 74 of 113), 58% (Wang *et al.* Chinese cohort, 88 of 152), 46% (TCGA cohort, 170 of 368), 73% (European cohort, 69 of 94), 73% (TRACERx Renal cohort, 77 of 106), respectively (**Figure RL8A**). The results showed that *VHL* mutation frequency in TCGA cohort was the lowest among these cohorts, while there was no significant difference of *VHL* mutation rate between two Chinese cohorts and between two European cohorts (**Figure RL8A**).

There might be two reasons of the lower *VHL* mutation rate in TCGA cohort. First, the *VHL* mutation frequency was reported to be varied across regions and ethnic background (PMID: 25078357, 32396860, 31620170, 27010573). Baoan *et al.* reported that higher frequency of *VHL* mutation in East Asian ccRCC patients than in the Western ccRCC patients (PMID: 31620170). The ethnic analysis of TCGA-KIRC cohort found there were racial differences in ccRCC, with lower levels of *VHL* mutation in tumors from African American versus white patients (PMID: 27010573). Second, we speculated that the reason might be due to insufficient sequencing depth in TCGA cohort. We evaluated the raw data of above-mentioned cohorts. The mean coverage of sequence depth for TCGA cohort was 20×, which were much lower than other cohorts (55×-612×, **Figure RL8B**). The higher depth of the sequencing increased the chance to detect the mutations (PMID: 31552176). Advances in sequencing technology and reduction in cost improved sequencing output, resulting the higher probability to detect the *VHL* mutation. We added the discussion in the revised manuscript.

Figure RL8.

A, Mutation frequencies of *VHL* in different studies.

B, Depth of the sequencing in different studies.

Minor

Q6. Line 37, the incidence numbers appear to be from the US only. This should be clarified. Also, the 2020 diagnoses have not yet resulted in 13,780 deaths. The wording should be corrected.

Response:

Thank you again for your comments and valuable suggestions to improve the quality of our manuscript. We have made revisions in the manuscript as you suggested: “In 2020, 431,288 people were newly diagnosed with kidney tumors and 179,368 patients with kidney tumor died worldwide”. We have also modified the reference as appropriate (GLOBALCAN, PMID: 33538338, <https://gco.iarc.fr/>).

Q7. Line 142, “... suggesting the specific mechanism...” What mechanism are the authors referring to?

Response:

We thank the reviewer for the question. In the revised manuscript, we elaborated the mechanism of AA promoting the efficiency of Sunitinib in killing ccRCC tumors through inhibiting pentose phosphate pathway. In detail, we verified that treating 786-O cells with AA caused notably decreased G6PD, PGD and TKT, in time- and dose- dependent manners, at both mRNA and protein levels (**Figure RL9A-D**). In addition, the levels of ribose-5-phosphate, which is an important product in both oxidative and non-oxidative PPP, decreased notably in AA treated cells (**Figure RL9E**). Therefore, it is the inhibitory effect of AA on pentose phosphate pathway that leads to the reduction of ribose 5-phosphate, an important raw material for DNA synthesis. Since inhibition of DNA synthesis directly inhibits tumor growth, the size of tumors in the AA exposed group is smaller. We have added these results in **Figure 2** in the revision.

Figure RL9.

A-B, AA treatment inhibits the expressions of G6PD, PGD and TKT in a dose-dependent manner.

C-D, AA treatment inhibits the expressions of G6PD, PGD and TKT in a time dependent manner.

E, The ribose-5-phosphate concentrations in cells treated with AA or not.

Q8. Line 168, "... poor prognosis caused..." Would be more accurate to talk about associations.

Response:

Thank the reviewer for the comment. The reviewer is correct that it was more accurate to use the "associations" instead of "caused", due to lack of evidence for causality. In addition, the results of this part were revised based on the reviewer's suggestion in Q3. We performed multivariate analysis to test the independence of CNA events with patient survival. The results showed that 7p gain (HR = 1.71, p = 0.045) and 7q gain (HR = 1.91, p = 0.015) were the dominant arm-level CNA events associated with PFS. Thus, we revised this part to: "We conducted multivariate analysis using poor prognosis-associated CNA events as covariates, and found that 7p/7q gain were the dominant events associated with patient survival".

Q9. Line 193, Is there evidence of curative effects of sunitinib?

Response:

We thank the reviewer for the question. As for the relation between chromosome 7q gain and curative effects of Sunitinib, in the revision, we overexpressed LAMTOR4, MDH2 and CALU, located in chromosome 7q, to mimic chromosome 7q gain in 786-O cells, respectively (**Figure RL10A**). We found overexpressed LAMTOR4, MDH2 and CALU increased the phosphorylation of S6K, indicating activation of mTORC1 (**Figure RL10B**). Next, we treated the candidate genes overexpressed cells and normal cells with or without Sunitinib. By monitoring the invasion of cancer cells with transwell assay, we found increased expression of LAMTOR4, MDH2 or CALU abrogated the effects of Sunitinib. We provided these results in **Figure RL10C**.

As for the role of AA exposure, we treated 786-O cells with AA or Sunitinib or not. We found that AA treatment did not impact cell proliferation directly, but enhanced the inhibition of Sunitinib to cell proliferation (**Figure RL10D**). The transwell assay showed that AA treatment did not impact cell invasiveness directly, but enhanced the inhibition of Sunitinib to cell invasiveness (**Figure RL10D**).

These results provided the evidences of AA and chromosome 7q gain might influence the curative effects of Sunitinib. We had added these results in **Figure 3** the revision.

Figure RL10.

A, Transfection efficiency of LAMTOR4, MDH2 and CALU overexpression plasmids detected by qRT-PCR and western blot.

B, The overexpression of LAMTOR4, MDH2 and CALU increased the phosphorylation of S6K.

C, Transwell detected the effect of LAMTOR4, MDH2, CALU overexpression and Sunitinib treatment on cell invasiveness.

D, The cell proliferation measured by MTT assay.

E, Transwell detected the effect of AA and Sunitinib treatment on cell invasiveness.

Q10. Line 256, what data is there that VHL mutations cause glycolysis addiction?

Response:

We thank the Reviewer for the question. First, based on the proteomics data, we found the protein

levels of glycolysis enzymes such as CA9, PGK1, ENO1 and LDHA were increased in *VHL* mutations carriers (**Figure RL11A**). Next, to mimic *VHL* loss-of-function mutations, we knocked down *VHL* in ACHN cells (**Figure RL11B**), which belong to *VHL*-proficient cells, and tested the glycolysis levels by monitoring the lactate production. The result showed that lactate levels increased in *VHL* deficiency cells (**Figure RL11C**), suggesting that *VHL* loss-of-function mutations caused enhanced glycolysis. We added these results in **Supplementary Figure 5** the revision.

Figure RL11.

A, The heatmap showing the upregulated glycolytic enzymes in the *VHL* mutant tumors, compared with *VHL* wildtype tumors.

B, Transfection efficiency of *VHL* siRNA detected by qRT-PCR and western blot.

C, The lactate concentrate in cells after *VHL* knockdown.

Q11. Line 389, should it be “in any of the samples”?

Response:

We apologize for the misleading in the previous manuscript and thank the reviewer for pointing out.

We have revised the sentence as “in any sample of the non-response group”

Q12. Line 401, “we established a comprehensive landscape of Sunitinib drug response...”, but there is no non-treated control group.

Response:

Thanks for the valuable suggestion. According to reviewer’s suggestion, we collected another 37 cases of metastatic or recurrent ccRCC tumor samples and conducted proteome analysis as a non-treated control group, which was also described in **Q1**. It is important to know whether the markers are associated with prognosis independently of therapy by collecting a non-treated control group. There were 270 prognostic biomarkers identified in this Sunitinib treatment cohort. To further evaluate whether prognostic markers are associated with survival independently of

therapy, we collected another independent cohort with 37 cases of ccRCC patients, who were not received any treatment after surgery, as control (**Methods** in revision). These 37 cases had complete follow-up information. We further evaluated the associations between protein expression levels and patient survival. The results showed that the expression levels of 630 proteins were significantly associated with patient survival in the non-treated control group. Compared these 630 prognostic markers in non-treated group with 270 prognostic markers in Sunitinib treatment group, only 19 proteins were found in both groups (**Figure RL12A**), indicating the prognostic values of protein expression levels were profoundly dependent on whether or not the patients have ever been prescribed Sunitinib treatment after surgery. As the prognostic biomarkers mentioned in the original manuscript, CD274, CRP, C8A, NOVA2, RRAGB, NUP50, prognostic biomarkers in treatment group, were not associated with patient survival in the non-treated control group (**Figure RL12B**). Higher expression of BID was associated with better survival in both Sunitinib treatment group and non-treated control group (**Figure RL12B**). We added these results in the revision and proofread the descriptions about predictive and prognostic biomarkers throughout the manuscript.

Figure RL12.

A, Venn plot of prognostic significance of protein expression levels in Sunitinib treatment group and non-treated group.

B, Kaplan–Meier curves of PFS for patients with different protein abundances in non-treated control group (log-rank test).

Q13. Figure 1B, tumour size units?

Response:

Thank reviewer for the comments. We apologize for not explaining it clearly. The tumor size units were cm. We made this clear in the Figure legend in the revision (**Figure RL13, also see in Figure 1B**).

Figure RL13. The summary of the clinical parameters in this cohort. This cohort includes 27 Responders and 88 Non-Responders undergoing Sunitinib treatment. Their clinical parameters are shown in the heatmap.

Reviewer #3 (Remarks to the Author): Expert in clear-cell renal cell carcinoma clinical research and genomics

The authors can be congratulated to their very extensive and carefully performed research. They have established a comprehensive landscape of Sunitinib drug response in renal cell carcinoma both at the genome and proteome levels, which has the potential to make a significant clinical impact.

Response: We appreciate the reviewer for the constructive and insightful comments, which help to improve the quality of this manuscript. The point-to-point responses are as follows.

Major Comments

Q14. A number of patients have reduced kidney function, diabetes and hypertension. Therefore this subgroup might well have chronic kidney disease with histological alterations (e.g.

hypertensive/diabetic nephropathy) in the peritumoral tissue. This may influence the results of this investigation.

Response:

Thank you again for your positive comments and valuable suggestions to improve the quality of our manuscript. In this study, 37/115 patients were diagnosed with hypertension and 13/115 patients were diagnosed with diabetes before Sunitinib therapy. We found that hypertension, diabetes and eGFR were not significantly associated with both OS and PFS in our cohort by Cox regression analysis as listed in **Table RL1**. Similarly, previous studies suggested that renal impairment or hypertension was not significantly associated with Sunitinib response among patients with metastatic renal cell carcinoma (PMID: 26702763, 21244613). Thus, we think hypertension, diabetes or renal impairment do not affect efficacy of Sunitinib. Sincere thanks should be given to the reviewer for the constructive comments and suggestions.

Table RL1. Results of Cox regression analysis

Risk factors	OS [HR (95%CI)]	OS (P-value)	PFS [HR (95%CI) PFS]	PFS (P-value)
Hypertension	0.8 (0.48-1.32)	0.378	1.07 (0.68-1.69)	0.768
Diabetes	1.08 (0.51-2.28)	0.839	1.13 (0.59-2.17)	0.716
eGFR	0.74 (0.34-1.58)	0.431	0.9 (0.44-1.84)	0.775

Q15. Inherent tumor heterogeneity, a well-known phenomenon in ccRCC, has not been assessed.

Response:

Response: We would like to thank the reviewer for the constructive comments, which made the article more accurate. To assess the inherent tumor heterogeneity of ccRCC, consensus clustering based on ConsensusClusterPlus (CCP) was performed, identifying three sets of tumors with distinct features (**Figure RL14A**). Remarkably, the tumor subtypes significantly differed in progression-free survival, Subtype1 had the best survival, while subtype3 had the worst clinical outcome (log-rank test, $p = 0.037$) (**Figure RL14B**). Notably, the different subtypes showed different proportion of Responders and Non-responders. Subtype 3 had the highest proportion of Non-responders (96%) (Fisher's exact test, $p = 0.004$) and the worst survival among the three subtypes (**Figure RL14C**). We performed GSEA to reveal proteomic differences among these subtypes. The result showed that Subtype1 was characterized by the upregulation of spliceosome and metabolism pathways including

in glycolysis and valine/leucine/isoleucine degradation. The Subtype2 exhibited enrichment of MYC targets, antigen processing and presentation, and MTORC1 signaling. The Subtype3 showed the upregulation of complement cascade, angiogenesis and EMT (**Figure RL14D**). These results indicated that the tumor heterogeneity of ccRCC was associated with Sunitinib treatment outcomes. We have re-written the manuscript to present tumor heterogeneity of ccRCC and added these results in **Supplementary Figure 7** in the revision.

Figure RL14.

A, Consensus matrices of the 115 samples from k = 2 to k = 6. Consensus clustering was conducted for the top 500 most-variant proteins.

B, Kaplan–Meier curves of PFS for the three subtypes (log-rank test).

C, Proportions of Responders and Non-Responders among the three proteomic subtypes.

D, Gene set enrichment analysis of three proteomic subtypes.

Q16. Analyses of (selected) miRNA regulating the response classifier could be interesting from a possible therapeutic point of view to ameliorate Sunitinib response.

Response:

Thanks for the suggestions. Following reviewer’s suggestions, we analyzed the miRNAs which might regulate the proteins associated with the response to Sunitinib. We performed

overrepresentation analysis, using the differentially expressed proteins between Responders and Non-Responders, in ConsensusPathDB using the TargetScan v7.2, miRTarBase v8.0, and miRDB v6.0 as the background to investigate the miRNAs might regulated Sunitinib response. Totally, 27 miRNAs were enriched by the proteins upregulated in Responders, and 11 miRNAs were enriched by the proteins upregulated in Non-Responders. However, the enrichment ratios of these miRNAs were less than 10%, indicating the differential expressed proteins of Responders and Non-Responders were not the main targets regulated by these miRNA (**Figure RL15A**).

To further investigate the impact of miRNAs on the response of Sunitinib, during the revision, we performed transcriptome sequencing in 94 samples in this cohort. We identified 348 miRNA in this cohort, among which 4 miRNA showed significant differences between Responders and Non-Responders. Specifically, MIR3939 (FC = 1.88), MIR4635 (FC = 1.96), and MIR578 (FC = 1.32) showed higher expression levels in Responders, while MIR27A (FC = 0.95) was higher in Non-Responders (**Figure RL15B**).

Figure RL15.

A, Scatter plot showing the overrepresented miRNAs which regulated the differential expressed proteins of Responders and Non-Responders.

B, Volcano showing the differential analysis of miRNA between Responders and Non-Responders.

Q17. Median follow-up was 28.4 months. This is relatively short since metastases, esp. in the lower risk patients, might well occur after this point of time.

Response:

Thank you again for your positive comments and valuable suggestions to improve the quality of our manuscript. According to reviewer's suggestion, we updated the follow-up information. In the latest follow-up, there was 1 patient died from kidney cancer. As 87.8% of the patients (n = 101) got disease progression in the previous follow-up, the updated follow-up did not increase the median follow-up time and did not influence the result of prognostic analysis. We have added the latest follow-up in the **Supplementary Table 1**. Moreover, the median follow-up of this study was similar with the median follow-up of SU011248 study (PMID: 19487381) (median follow-up: 26.4 month), which opened the era of targeted therapy. In conclusion, we think the follow-up time would not impact the results of our analysis.

Minor comments

Q18. Patient characteristics: eGFR would be better than creatinine/BUN. Furthermore, not entirely clear if all patients were subjected to full or partial nephrectomy.

Response:

Many thanks for pointing this out. We have calculated the eGFR due to the standard formula and we found that renal impairment did not affect efficacy of Sunitinib (**Table RL1**). In our study, all the patients underwent radical nephrectomy. We have added this information in **Methods** in the revision. Thank you again for your suggestions.

Q19. How long and variable was the intraoperative ischemia time prior to tissue harvesting?

Response:

Many thanks for pointing this out. In our study, all the patients underwent radical nephrectomy. We have added this information in **Methods** in the revision. In addition, tumor and paired tumor adjacent tissues were collected within 30 min after resection, immediately transferred into sterile freezing vials and snap frozen in liquid nitrogen and then split and stored at -80 °C (cold ischemia time, PMID: 23134551) until being used. The cold ischemia time of this study was consistent with other large-scale proteogenomic studies (PMID: 31675502; PMID: 31730861). Thus the quality of our samples for proteogenomic analysis was guaranteed. Thank you again for your professional suggestions.

Q20. The authors may comment why Leibovich scoring system was not included.

Response:

Thank you for your professional suggestions. Leibovich et al., developed a clinically useful scoring algorithm to predict cancer-specific survival for patients with clear cell metastatic renal cell carcinoma (PMID: 16217278). We thank the reviewer for pointing out that we forgot to mention this point and we calculated the score to validate the scoring algorithm. The results showed that Non-Responders had lower Leibovich scores than Responders (**Figure RL16A**). In addition, Leibovich scoring system could predict patients' prognosis in this cohort and a higher score was significantly associated with worse overall survival (**Figure RL16**). Thank you again for your professional suggestion. We have added these results in the **Supplementary Figure 2** in the revision.

Figure RL16.

A, Comparing Leibovich scores between Responders and Non-Responders (Fisher's exact test).

B, Kaplan-Meier curves of OS and PFS for patients different Leibovich scores (log-rank test).

Q21. Concomitant (anti-tumor/anti-inflammatory) therapy? Sunitinib dose identical among patients?

Response:

Thank you for your professional suggestions. In this research, no patients underwent concomitant anti-tumor or anti-inflammatory therapy. The initial dose of Sunitinib was same between patients and adjusted according to side effects and the intervention was set as: 50-mg orally taken daily for 4 weeks and off treatment for 2 weeks until progression or unacceptable toxicity. We added this information in the revision. Thank you again for your professional suggestions.

Reviewer #4 (Remarks to the Author): Expert in cancer proteomics and proteogenomics

Qu and co-workers performed whole exome sequencing and DDA-based mass spectrometry analysis of samples from human ccRCC tumors and adjacent tissue, from 68 patients with advanced tumor and from another 47 patients who developed metastatic disease after surgery and treatment with the TKI sunitinib. They identified a number of differentially expressed proteins in responders and non-responders to the TKI, and several somatic mutations and copy number alterations. Interestingly, they associate a mutational signature with some herb that is applied Chinese traditional medicine and contains the compound aristolochic acid (AA), with better survival of ccRCC patients. This herb, and the effective agent (AA) have previously been associated with increased development of ccRCC. Here the authors find that the mutational signature that is induced by AA seems to correlate with better response to sunitinib treatment. The authors further conclude that the mutations having been caused by the drug lead to an addiction of tumor cells to glycolysis and that this is beneficial for the patients.

Response:

We, the authors, sincerely thank the reviewer for the careful read and thoughtful comments on previous manuscript. We have carefully taken reviewer's comments into consideration in preparing our revision. Importantly, we provided additional transcriptome and phosphoproteome data to make our dataset more valuable. In addition, we carried out a series of experiments to validate our hypothesized mechanisms from our analysis. We believed that our revision addressed all the points raised by the reviewer, which will lead to a clearer and more compelling manuscript.

Q22. Another interesting findings appears to be that mutated VHL was found to correlated with better patient outcome and with smaller tumor sizes. These findings are surprising since hypoxia and angiogenesis are factors that commonly associated with progressive disease and metastasis. Along the lines of this, the authors might consider PMID:33996558. There, VHL is described as a 'main gene (...) involved in ccRCC carcinogenesis [however, is] not the most relevant for assessing survival.'

Response:

Thanks for the comments. Florent collected the ccRCC prognostic expression markers from 249 published articles between 2003 and 2018. Florent's study identified a total of 341 distinct markers and 13 multiple-marker models. Comparing the cohort in the Florent et al with our study, Florent et al did not distinguish patients with TKI treatment or not. The reviewer is correct that there was no correlation between the *VHL* alteration and overall survival in the entire ccRCC groups (PMID: 28103578). However, when we focused on ccRCC patients treated with TKI, the PFS showed significantly difference between *VHL* mutant and wild-type groups. We further evaluated the association between *VHL* mutation and patient outcomes with Sunitinib treatment in IMmotion151 (PMID: 33157048). A total of 356 patients treated with Sunitinib were recruited in the study, including 82 patients carrying *VHL* mutation and 274 patients with wild-type *VHL* gene. Consistently, Responders (CR+PR) were significantly overrepresented in patients with *VHL* mutations (Fisher's exact test, $p = 0.013$) (**Figure RL17A**), and patients with *VHL* mutation have longer PFS than patients without *VHL* mutation (log-rank test, $p = 0.012$) (**Figure RL17B**).

To validate the correlation of *VHL* mutation and Sunitinib response, we knocked down *VHL* in ACHN cells which belong to *VHL*-proficient cells, to mimic *VHL* loss-of-function mutations (**Figure RL17C**). We tested the relation between *VHL* mutation and Sunitinib treatment in cultured cells. By testing the invasion of cells with transwell assay, we found *VHL* deficiency enhanced the effects of Sunitinib (**Figure RL17D**). We provided this information in **Supplementary Figure 5** the revised manuscript.

Figure RL17.

A, Stacked graph bars represent the number of responses for patients treated with Sunitinib in IMmotion151 study; complete response (CR), partial response (PR), stable disease (SD), progression disease (PD) or not evaluated (NE).

B, Kaplan–Meier curves of progression-free survival (PFS) for patients with or without *VHL* mutation.

C, Transfection efficiency of *VHL* siRNA detected by qRT-PCR and western blot.

D, Transwell detected the effect of *VHL* knock down and Sunitinib treatment on cell invasiveness.

Q23. I have a number of concerns with this study. The first sentence of the abstract suggests that the authors performed an integrated proteogenomic characterization of patients' tumors who were under ('undergoing') therapy. They further claim that they revealed the molecular basis of differential clinical outcomes with tyrosine kinase inhibitor (TKI) therapy. Since the tumors had been surgically removed before therapy (also adjacent tissue was removed with the tumor), the molecular analysis of all samples revealed the proteomic status of the tumors prior to therapy and does not necessarily reveal molecular causes of drug failure – not until the hypothesized mechanisms have been experimentally proven. The authors performed a large number of bioinformatics analyses and present a similarly large number of data, however, all this is bioinformatics and imputed from literature (which indeed is not always along the same lines – e.g., angiogenesis has been associated with progressive disease on sunitinib PMID:33593885, PMID:31268155).

Response:

We thank the reviewer for the comments. The reviewer was correct that only proteomic data analysis was not sufficient to reveal the molecular causes of TKI response. In the revision, we gave evidences for the hypothesized mechanisms that we proposed at cell level.

About the relation between chromosome 7q gain and curative effects of Sunitinib

First, in the previous manuscript, we found that 7q gain was associated with shorter survival in this Sunitinib treatment cohort (**Figure RL18A-B**). In additions, genes that were more frequently amplified in the Non-Responder group than in the Responder group were located in 7q (Fisher's exact test, $p < 0.05$) (**Figure RL18C**). To further evaluated the proteomic consequences of 7q gain

in ccRCC, we performed *cis-/trans*-effect analysis. Proteins showed significantly positive correlation with the 7q copy number (CN) (Spearman's correlation, $p < 0.05$) were enriched in pathways including lysosome (LARP1, LARP2), mTOR signaling (RRAGB, RRAGD, LAMTOR2, LAMTOR4, MLST8), TCA Cycle (CS, SDHA, MDH2), oxidative phosphorylation (COX6C, NDUFS6), and fatty acid biosynthesis (ACACA, MCAT) (**Figure RL18D**). We supposed that 7q gain activated mTOR signaling and upregulated amino acid turnover.

In the revision, we overexpressed LAMTOR4, MDH2 and CALU, located in chromosome 7q and *cis*-regulated by 7q gain, to mimic chromosome 7q gain in 786-O cells, respectively (**Figure RL18E**). We found overexpressed LAMTOR4, MDH2 and CALU increased the phosphorylation of S6K, indicating activation of mTORC1 (**Figure RL18F**). Next, we treated the candidate genes overexpressed cells and normal cells with or without Sunitinib. By monitoring the invasion of cancer cells with transwell assay, we found increased expression of LAMTOR4, MDH2 or CALU abrogated the effects of Sunitinib (**Figure RL18G**). These results were added in the **Figure 3** in the revision.

Figure RL18.

A, Cox regression analysis of significant arm-level CNA events, based on the PFS.

B, Kaplan–Meier curves of PFS for patients with or without chromosome 7q gain (log-rank test).

C, Comparison of gene-level CNAs between Responders and Non-Responder in this cohort. The upper plot illustrates the frequency of CNA events, the lower plot illustrates the $-\log_{10}(p \text{ value})$ of each gene for the comparison of Responders and Non-Responder (Fisher's exact test).

D, Heatmap of proteins positively correlated with the Chromosome 7q Copy number. Spearman's correlations are shown in the right panel.

E, Transfection efficiency of LAMTOR4, MDH2 and CALU overexpression plasmids detected by qRT-PCR and western blot.

F, The overexpression of LAMTOR4, MDH2 and CALU increased the phosphorylation of S6K.

G, Transwell detected the effect of LAMTOR4, MDH2, CALU overexpression and Sunitinib treatment on cell invasiveness.

About the relation between AA (aristolochic acid) exposure and Sunitinib treatment outcomes.

Secondly, we described that patients with AA signatures had longer PFS (**Figure RL19A**) and showed upregulated glycolysis and downregulated pentose phosphate pathway (PPP) at proteomic level (**Figure RL19B**). To make the role of AA exposure clear, we treated 786-O cells with AA or Sunitinib or not. We found that AA treatment did not impact cell proliferation directly, but enhanced the inhibition of Sunitinib to cell proliferation (**Figure RL19C**). The transwell assay showed that AA treatment did not impact cell invasiveness directly, but enhanced the inhibition of Sunitinib to cell invasiveness (**Figure RL19D**). Moreover, we verified that treating 786-O cells with AA caused notably decreased G6PD, PGD and TKT, in time- and dose- dependent manners (**Figure RL19E**). The levels of ribose-5-phosphate, which is an important product in both oxidative and non-oxidative PPP, decreased notably in AA treated cells (**Figure RL19F**). More importantly, we verified that SP1 is the key transcription factor of AA regulating pentose phosphate pathway (**Figure RL19G-L**). These results were added in the **Figure 3** in the revision.

Figure RL19.

A, Kaplan–Meier curves of PFS for patients with and without AA signature (GB-Wilcoxon test).

B, Glucose metabolism alteration comparing tumors with and without AA signature.

C, The cell proliferation measured by MTT assay.

D, Transwell detected the effect of AA and Sunitinib treatment on cell invasiveness.

E, AA treatment inhibits the expressions of G6PD, PGD and TKT in time-and dose-dependent manners.

F, The ribose-5-phosphate concentrations in cells treated with AA or not.

G-H, SP1 knockdown downregulated PPP enzymes at mRNA and protein levels.

I-J, SP1 overexpression upregulated PPP enzymes at mRNA and protein levels.

K, AA treatment did not affect the transcription of SP1.

L, MG132 abrogated the AA mediated PPP enzymes downregulation by inhibiting the degradation of SP1.

About the *VHL* mutation and Sunitinib response.

In the previous manuscript, we found that Responders had higher *VHL* mutation frequency than Non-Responders, and patients with *VHL* mutation possessed longer PFS (**Figure RL20A-B**). Proteomic data revealed that patients with *VHL* mutation showed higher expressions of glycolytic enzymes, indicating the glycolysis addiction of *VHL* mutated tumors (**Figure RL20C**).

In the revision, the associations between *VHL* mutation and patient outcomes with Sunitinib treatment were further verified in the IMmotion151 study (PMID: 33157048). A total of 356 patients treated with Sunitinib were recruited in the study, including 82 patients carrying *VHL* mutation and 274 patients with wild-type *VHL* gene. Consistently, Responders (CR+PR) were significantly overrepresented in patients with *VHL* mutations (Fisher's exact test, $p = 0.013$) (**Figure RL20D**), and patients with *VHL* mutation have longer PFS than patients without *VHL* mutation (log-rank test, $p = 0.012$) (**Figure RL20E**). Moreover, to provide more experimental evidences, we knocked down *VHL* in ACHN cells (**Figure RL20F**) which belong to *VHL*-proficient cells, mimicking *VHL* loss-of-function mutations. We tested the glycolysis levels by monitoring the lactate production. The results showed that lactate levels increased in *VHL* deficiency cells (**Figure RL20G**), suggesting that *VHL* loss-of-function mutations caused enhanced glycolysis. In addition, the knockdown of *VHL* distinctly enhanced the inhibition the invasiveness of Sunitinib to cancer cells (**Figure RL20H**).

Figure RL20.

A, Stacked graph bars represent the number of responses for patients treated with Sunitinib in this study (Fisher's exact test).

B, Kaplan–Meier curves of progression-free survival (PFS) for patients with or without *VHL* mutation in this study.

C, Pathways enriched by the differentially expressed proteins between patients with or without *VHL* mutation.

D, Stacked graph bars represent the number of responses for patients treated with Sunitinib in IMmotion151 study (Fisher's exact test); complete response (CR), partial response (PR), stable disease (SD), progression disease (PD) or not evaluated (NE).

E, Kaplan–Meier curves of progression-free survival (PFS) for patients with or without *VHL* mutation in IMmotion151 study.

F, Transfection efficiency of *VHL* siRNA detected by qRT-PCR and western blot.

G, The lactate concentrates in cells after *VHL* knockdown.

H, Transwell detected the effect of *VHL* knock down and Sunitinib treatment on cell invasiveness.

These results indicated that the hypothesized mechanisms of TKI response, derived from proteomic data analysis, were reliable in to a great extent. Our proteomic provided important resources for TKI response mechanisms. We have added these results in **Supplementary Figure 5** the revision.

Q24. No information on the whole-exome sequencing is provided (enrichment protocol used, numbers of reads sampled, coverage of exomes, numbers of variants detected, availability of sequencing data). For example, the graphs in Figure 7F do not provide any numbers of CNAs or mutations having been found.

Response:

Thanks for your kind suggestions, and we apologize for the unclear presentation. In this study, WES was conducted at Life Healthcare Clinical Laboratory (China). DNA isolated from freshly frozen tumor tissue samples was used for WES, and matched germline DNA was obtained from adjacent non-tumor tissue samples. DNA was isolated from fresh tissues using DNeasy Blood & Tissue Kit (Qiagen, 69504) according to the manufacturer's instructions. Purified DNA was quantified using

a Qubit 3.0 Fluorometer (Life Technologies). For matched adjacent and tumor tissues, 100 ng of DNA was sheared to 200–300-bp fragments using a Covaris M220 system. Tumor and matched germline DNA libraries were constructed using Accel-NGS 2S HYB DNA LIBRARY KIT (Swift Biosciences, 23096) and Accel-NGS 2S MID S1-S4 (Swift Biosciences, 279384). xGen Exome Research Panel v1.0 (IDT, 1056115) and xGen Lockdown reagents (IDT, 1072281) were used for exome enrichment. Dynabeads M-270 Streptavidin (Thermo, 65306) was used for library purification, P5/P7 primers (Nanodigmbio, ND10010) and HotStart ReadyMix (KAPA, KK2612) were used for library amplification. The amplified libraries were purified using SPRISELECT (Beckman, B23319). DNA quality was assessed using a Bioanalyzer High Sensitivity DNA Analysis kit (Agilent Technologies, 5067-4626). Samples underwent paired-end sequencing on a Nextseq CN500 platform (Illumina), with a 150-bp read length. The WES target region was 33 M. The WES data was deposited at the NODE (<https://www.biosino.org/download/node/data/OED625358> and <https://www.biosino.org/download/node/data/OED625359>).

As for the Figure 7F (actually the Figure 6F in the previous manuscript), the percentage of pie plot demonstrate the proportion of patients with CNAs and SNPs in Responders and Non-Responders. We have revised the Figure 6F in the revised manuscript (**Figure RL21**).

Figure RL21. Summary of molecular and clinical features in the Responders and Non-Responders.

Q25. It is interesting to note that the authors did not apply RNA-sequencing, which would have been a complementary approach to their total proteomic experiments. The authors comment on the low level of concordance between RNA-seq and proteomic data. This level of

concordance/discordance is a matter of debate in the community. The authors could have added to this with solid RNA-seq data.

Response:

Thank the reviewer for the comments. The reviewer is correct that our statement was not appropriate and solid. To further assess the correlation between transcriptome and proteome, we performed transcriptome sequencing in 94 samples in this cohort. We identified 12,276 protein-coding genes with median fragments per kilobase of transcript per million fragments mapped (FPKM) of more than 1 (**Supplementary Table 3** in the revision). Our data showed that the mRNA-protein correlation was moderate with sample-wise median Spearman's correlation of 0.36 (**Figure RL22A**), similar to the previous report (PMID: 31923397). In conclusion, our statement about the poor correlation between the transcriptome and proteome was not appropriate. We apologized for our unthoughtful statement and removed the relevant statement in the revision. In addition, we compared the impacts of AA signature at protein and mRNA level. The results showed that differentially expressed proteins and mRNA were overlapped in a small portion (**Figure RL22B**). Specifically, 14 upregulated genes (84 upregulated protein and 203 upregulated mRNA) and 10 downregulated genes (470 upregulated protein and 105 upregulated mRNA) were commonly identified at both mRNA and protein levels in tumors with AA signature. Interestingly, we found that metabolism related pathways were upregulated in patients with AA signature, while immune related pathways were downregulated at both protein and mRNA levels (**Figure RL22C-D**). These results indicated that although mRNA and protein did not show high correlation, mRNA and protein exhibited concordance in bioprocess regulation. Thanks again for the considerate comments.

Figure RL22.

A, Boxplot showing sample-wise mRNA-protein correlation.

B, Venn plots showing the identified differential expressed genes between tumors with and without AA signature at mRNA and protein levels.

C-D, Pathways enriched by differentially expressed mRNA between tumors with or without the AA signature.

Q26. I am not sure about the clinical regimen that were applied. The patients in the cohort are reported to having been ‘treated with TKIs’ between January 2008 and December 2019 (Materials and Methods). 1. The authors need to make clear which TKIs were given (only sunitinib?). 2. ccRCC is still mostly treated in the adjuvant setting. In this way, Figure 1A would be at least misleading as tumor sampling was very likely prior to treatment. This point is important, for example as the authors are surprised that they did not see effects of sunitinib on the abundance of some target proteins (lines 260ff). The authors should realize that the activities of signaling proteins has higher relevance on pathway activities than the abundance of individual components, particularly when these TKIs are wildtype in sequence and thus require activation by respective ligands to become activated themselves. In this way, a phospho-proteomic analysis of tumors might have been an idea.

Response: Sincere thanks should be given to the reviewer for the constructive comments and suggestions. In this study, all patients began to use Sunitinib as first-line treatment after diagnosed with metastasis/recurrent disease. All samples were collected during radical nephrectomy and all patients were treatment-naïve before surgery and received Sunitinib treatment after surgery. The initial dose of Sunitinib was same between patients and adjusted according to side effects and the intervention was set as: 50-mg orally taken daily for 4 weeks and off treatment for 2 weeks until progression or unacceptable toxicity. We have added this information in Methods section in the revision and **revised the Figure 1A** to avoid misleading (**Figure RL23A**).

We apologized that our misleading figure and descriptions. Actually, patients received Sunitinib therapy after surgery. Thus, in lines 260ff, we showed that protein abundances of Sunitinib targeted receptor tyrosine kinases showed no significant difference between Sunitinib Responders and Non-Responders. In addition, according to the reviewer's suggestion, we performed 66 cases of phosphoproteome analysis out of these 115 ccRCC tumors (**Methods** in the revision). We identified 4,862 phosphosites in each sample on average, and 37,055 phosphosites in total, corresponding to 7,502 phosphoproteins (**Figure RL23B; Supplementary Table 4** in the revision). Among these phosphosites, 6,749 phosphosites were identified in more than 25% of samples was used for further analysis. Firstly, we evaluated the global activities of Sunitinib targeted receptor tyrosine kinases (RTK), by comparing the phosphorylation levels of all substrates of these kinases. We found that global activities of Sunitinib targeted RTKs showed no significant differences among Responders and Non-Responders (**Figure RL23C**). We next conducted kinase-substrate enrichment analysis (KSEA) and found that MAP2K1(MEK1) and MTOR were activated in Non-Responders, while CDK1/2 were activated in Responders (**Figure RL23D**). Evaluation of kinase activities by single-sample gene set enrichment analysis (ssGSEA) further confirmed that MTOR was activated in Non-Responders while CDK2 was activated in Responders (**Figure RL23E**). Further investigation into the differentially expressed phosphosites between Responders and Non-Responders, it was showed that mTOR and MAPK signaling pathways associated substrates phosphorylation were elevated in Non-Reponders, such as DEPTOR at S265, MAPK1(ERK2) at Y187, whlie SNRNP70 at S226 which was involved in RNA splicing was increased in Reponders (**Figure RL23F**). Consistently, abundances of

phosphosites DEPTOR pS265, MAPK1 pY187 and SNRNP70 pS226 were significantly correlated with MTOR, MAP2K1 and CDK2 activities, respectively (**Figure RL23G**). Previous studies reported that Sunitinib treatment did not affect the phosphorylation of ERK (PMID: 21850379) and MEK inhibition abrogated Sunitinib resistance in RCC PDX-xenograft model (PMID: 27560553), provides support for our findings. In conclusion, our phosphoproteome revealed that exceeded mTOR and MAPK signaling pathways were associated with Sunitinib treatment resistance. We added these results in the **Figure 5** and **Supplementary Figure 6** in the revision.

Figure RL23. Schematic representation ccRCC sample collection.

A, Schematic representation ccRCC multi-omics analysis in Sunitinib cohort.

B, Overview of identified phosphosites of ccRCC tumor samples in this study.

C, Boxplot showing the inferred Sunitinib targeted RTK activities between Responders and Non-responders.

D, Differential analysis of kinase activities by KSEA between Responders and Non-Responders.

E, Inferred kinase activities of MTOR and CDK2 between Responders and Non-Responders.

F, Abundances of phosphosites (DEPTOR_S265, MAPK1_Y187, SNRNP70_S226) between

Responders and Non-responders.

G, Scatterplot showing inferred kinase activity (y axis) versus the phospho-abundance of the targeted substrates (x axis).

Q27-1. In the proteomic experiment, 10,075 proteins are reported to having been identified in responders and non-responders. The total number of proteins that was identified is 12,310 (R, R+NR, NR). The authors should be congratulated to these high numbers of identified proteins, as DDA in combination with the HF-X mass spectrometer commonly detects much fewer proteins. How many of those 10,075 proteins were identified in 90% of the samples? How many of the 1208 and 1027 proteins that were identified specifically in the non-responders and responders, respectively, were recurrently identified?

Response:

Thanks for the comments. In our study, firstly, peptide identification stringency was set at a maximum 1% FDR at peptide level. Then, proteins with 1% FDR and at least one unique peptide were select for further analysis. The same cutoff strategies of FDR at protein/peptide level have been widely used in recently published researches (**PMID: 34534465**, **PMID: 33577785**). As a result, in total, we identified 12,310 proteins with an average 6,585 proteins in the tumor samples and 5,753 proteins in the tumor-adjacent tissues.

About proteins identified in 90% of the samples

As for the number of proteins identified both or exclusive in R and NR, we apologized that our misleading figure and descriptions. We updated the data in our revision, and uploaded the all data to iProX (<https://www.iprox.cn/page/PSV023.html?url=1628575800937Gsoc>, with a password 6mFE) and Firmiana (phenomics.fudan.edu.cn/firmiana). Actually, 9,641 proteins were identified in both R and NR. 442 proteins were identified exclusively in R, and 2,226 proteins were identified exclusively in NR (**Figure RL23**). Among the 9641 proteins, 4,037 proteins were identified in 90% of the samples. We investigated the Xu et al., study published in Cell (PMID:32649877), and the Jiang et al., study published in Nature (PMID:30814741). Proteins identified in 90% of the cohort samples were 4,996 (4,996/11,091) and 3,802 (3,802/9252) respectively (**Figure RL24**). The results showed that our proteomic data was reliable and comparable with the published paper.

Figure RL24. Stacked graph bars represent the number of proteins in the three cohort, respectively.

About the exclusively identified proteins in R and NR

Among the 442 proteins identified exclusively in R, there were 53 proteins were recurrently identified. Among the 2,226 proteins identified exclusively in NR, there were 903 proteins recurrently identified (**Figure RL25**). To improve the reliability of our analysis results, proteins with low identification frequency were excluded from the analysis. The analysis in our cohort focused on the proteins identified in $> 25\%$ of the samples which had been applied in the previous published studies (**PMID:33212010**, **PMID:34358469**, and **PMID:29739932**). Therefore, applying the 25% cutoff, exclusive proteins in R and NR were excluded in our analysis.

We revised the manuscript in the revision. Thank the reviewers again for pointing out.

Figure RL25.

A, Pie charts showing the identification of different categories of proteins.

B, Barplots showing the numbers of recurrently identified proteins in Responders and Non-Responders.

Q27-2. The authors write in lines 129f that proteins identified in at least 25% of the samples were used for the analysis. How many proteins were detected in at least 25% of the samples and, consequently, used in the analysis (Line 584 describes 7451 proteins, which would still be many compared to comparable proteomic studies)?

Response:

We thank reviewer for the comments. There were 7,451 proteins detected in at least 25% of the samples.

About the cutoff of 25%

The analysis focusing on the proteins identified in > 25% of the samples has been applied in the previous published studies. For instance, the breast cancer study published in Cell (**PMID: 33212010**) have demonstrated that proteins were required to have at least in > 25% of samples in order to be included in the proteome dataset. In the study of lung squamous cell carcinoma from CPTAC published in Cell (**PMID: 34358469**), proteins identified in > 25% cohort samples were used for downstream analysis. The diffuse-type gastric cancer cohort (**PMID: 29739932**) also used 25% cutoff for the downstream analysis. We compared our cohort with the Xu et al., study published in Cell (PMID:32649877), and the Jiang et al., study published in Nature (PMID:30814741), according to the number of the proteins detected in more than 25%, 35%, 45%, 55%, 65%, 75%, 85%, 95%, and 100% of samples, respectively (**Figure RL26**). The proteins detected at different cutoff of the samples in the three cohort were, 25% (7451, 8729, 7149), 35% (7184, 8042, 6578), 45% (6920, 7572, 6101), 55% (6607, 7065, 5667), 65% (6221, 6556, 5249), 75% (5765, 5984, 4743), 85% (5243, 5381, 4185), 95% (4256, 4432, 3316), and 100% (2823, 3068, 2226), respectively. The results showed that our cohort was reliable.

Figure RL26. Number of proteins identified with different criterion in our cohort, Xu et al., cohort and Jiang et al., cohort.

Q27-3. Some figures (e.g., Figure 2K, 2I, 3B) suggest that the indicated proteins were detected in all patient samples that are on display. For how many of those patients has the data been imputed? Given that the patient subgroups were sometimes rather small, how many patients were included in any such subgroup that met this criterion of 25%?

Response:

We thank reviewer for the comments. According to the reviewer’s comments, we added the number of identified samples of each protein in the Figure 2K, 2I, 3B (**Supplementary Table 5**).

About the data imputation

In detail, for the missing values, we firstly applied match between runs (MBR) algorithm (PMID: 24942700) in this study. We built a dynamic regression function based on common identified peptides in samples. According to correlation value R^2 , the function chooses linear or quadratic function for regression to calculate retention time (RT) of corresponding hidden peptides, and check the existence of the extracted ion chromatogram (XIC) based on the m/z and calculated RT. The function evaluated the peak area values of those existed XICs. These peak area values are considered as parts of corresponding proteins. MBR has been proved to be an effective technique to fill the missing values, which was widely used in other proteomic studies (PMID: 31495571). As for the rest missing values after applying MBR, to avoid artificially increasing the false discovery rate, we did not apply other data imputation algorithm. Missing values were assigned to $1e-5$, the minimum value across our proteome data. This strategy has been applied in the previous published studies,

such as the diffuse-type gastric cancer project (PMID: 29520031), the early-stage hepatocellular carcinoma (PMID: 30814741) and so on.

About the subgroups whether met the criterion of 25%

In our cohort, 7,451 proteins detected in > 25% of samples were used for analysis. The strategy has been applied in the previous published studies (also see Q27-2). According to the reviewer's comments, we evaluated the protein distribution in the subgroup (3q gain subgroup, 7p gain subgroup, 7q gain subgroup, 8q gain subgroup and R/NR subgroup). The results showed that proteins met the criterion of detected in > 25% subgroup samples had a high proportion of over 99%. The proteins used for subsequent analysis were mostly medium/high identification frequency whether in global groups or subgroups. We have revised the related Supplementary table and figures in the revision.

Figure RL27. The number of proteins identified in more than or less than 25% different subgroup samples.

Q28. In line 587f, the authors write that among the DEPs, 1296 proteins were significantly upregulated and 699 proteins were significantly downregulated in the responders. Were all these proteins detected also in the non-responders? If the criterion of identification in >25% of patients was applied, how many of the proteins were detected/not detected in the non-responders at levels similar to the ones in responders? What was the range of expression? Where is the raw data?

Response:

Thanks for the comments.

About the identification of DEPs and detected frequencies of DEPs

As for the number of DEPs of R compared with NR, we apologized that our misleading figure and descriptions. We updated the data in our revision, and uploaded the all data to iProX (<https://www.iprox.cn/page/PSV023.html?url=1628575800937Gsoc>, with a password 6mFE) and Firmiana (phenomics.fudan.edu.cn/firmiana). As mentioned in Q27-1, 7,451 proteins identified in more than 25% samples. These proteins were used to further analysis. By applying the thresholds of fold change > 1.5 and $p < 0.05$, 183 DEPs were identified between Responders and Non-Responders (**Figure RL 28**). Among the DEPs, 117 proteins were significantly upregulated and 66 proteins were significantly downregulated in the R compared to NR. As for the exclusively detected proteins in R and NR, because of these exclusively detected proteins in R and NR failed to pass the criteria of 25% cutoff, none of them were included in the analysis. As the identification of DEPs in the R/NR group, all the R upregulated proteins were also identified in the NR group and all the NR upregulated proteins were also identified in the R group. **Figure RL 26** showed the detected frequencies of DEPs (identified in $>25\%$ samples) in both R and NR group.

We displayed the identification frequency of each DEPs in R group and NR group in detail (**Figure RL 28**). Due to the strict cutoff of the data (also mentioned in Q 27-2), all the DEPs have a relative high identification frequency both in R and NR group.

Figure RL 28.

A, Overlap of proteins in different cutoff.

B, Identification frequency of DEPs in R and NR group, respectively.

About the proteins detected in > 25% of samples

As mentioned in Q27-1, stringency cutoff was set in our cohort. The low identification frequency proteins and exclusive proteins in R and NR were all excluded in our bioinformatic analysis. Finally, 7,451 proteins identified in > 25% of samples were used for subsequent analysis. The all 7,451 proteins were also detected in both R and NR. To make it clear, we added identification frequency of each protein in the update the **Supplementary Table 4**.

About the range of expression and the raw data

The dynamic range of all proteins in a sample spanned six orders of magnitude (**Figure RL27**). The dynamic range of each protein in the whole dataset also spanned six orders of magnitude (**Figure RL29**). Proteome raw datasets are available at the iProX data portal:

<https://www.iprox.cn/page/PSV023.html?url=1628575800937Gsoc>, with a password 6mFE. WES data files can be accessed at <https://www.biosino.org/download/node/data/OED625358> and <https://www.biosino.org/download/node/data/OED625359>.

We have revised the manuscript. Thanks the reviewers again for pointing out.

Figure RL29. Boxplot of the log₂ (FOT) protein expression levels for the 115 pairs of tumor and adjacent tissue samples. Green indicates tumor samples; orange indicates adjacent samples.

Q29. The authors highlight some proteins and pathways that are up- or down-regulated in one or the other condition. It should be interesting to learn how many genes/proteins were positively identified (i.e., with data) to come to these lists.

Response:

Thanks for the comment. We added the corresponding information in the revised **Supplementary Table 4**.

Q30. The authors write that glycolysis addiction was caused by VHL mutations and was associated with sunitinib therapeutic response, and that the AA signature could enhance glycolysis addiction. The addiction of tumor cells to glycolysis has been long recognized as a driving factor in

tumorigenesis and progression. Here, the authors add a strong point of concern speaking against the application of AA in humans.

Response:

Thank the reviewer for the comment. It was quite interesting that AA signature and *VHL* gene mutations showed synergistically enhanced glycolysis. Traditionally, it is believed that *VHL* is a tumor suppressor gene, and *VHL* gene mutation will lead to HIF-1 α accumulation, resulting in the up-regulation of VEGF and PDGF expression to promote tumor progression. While ccRCC is a highly angiogenic tumor, theoretically, *VHL* gene mutation can promote the progress of ccRCC. However, sunitinib is a drug developed based on the molecular mechanisms such as the up-regulation of VEGF and PDGF caused by the loss of *VHL* function. Therefore, ccRCC with *VHL* mutation will be effective for sunitinib, which is also the reason why the mutation of *VHL* is beneficial to the curative effect of sunitinib, for it accords with the therapeutic mechanism of sunitinib. Therefore, whether AA exposure or *VHL* gene mutation, the molecular mechanism of glycolysis enhancement caused by them is consistent with the anti-tumor mechanism of sunitinib, so this glycolysis addiction can be beneficial to the curative effect of sunitinib. So this may be an important discovery.

More importantly, in the revision, we further elaborated the mechanism of the influences of AA exposure to Sunitinib response. We found that AA treatment did not impact cell proliferation directly, but enhanced the inhibition of Sunitinib to cell proliferation (**Figure RL30A**). The transwell assay showed that AA treatment did not impact cell invasiveness directly, but enhanced the inhibition of Sunitinib to cell invasiveness (**Figure RL30B**). Moreover, we verified that treating 786-O cells with AA caused notably decreased G6PD, PGD and TKT, in time- and dose- dependent manners (**Figure RL28C-F**). The levels of ribose-5-phosphate, which is an important product in both oxidative and non-oxidative PPP, decreased notably in AA treated cells (**Figure RL28G**). We next investigated how AA inhibited the expressions of those genes. By predicting the potential transcriptional factors which involved in the regulating the transcription of G6PD, PGD and TKT, we identified four kinds of TF had the common binding/recognizing regions in G6PD, PGD and TKT promoters, including SP1, SMAD4, YY1, and SREBF2. By knocking down these candidate transcriptional factors using siRNA in cultured cells (**Figure RL28H-K**), we found the mRNA levels of G6PD and PGD decreased in SP1 knocking down cells (**Figure RL28H**). However, decreased mRNA levels of SMAD4, YY1 and SREBF2, did not affect the transcription of G6PD, PGD and TKT (**Figure**

RL30I-K). Protein levels of G6PD and PGD were also decreased in SP1 knocking down cells (**Figure RL30L**). Furthermore, we overexpressed SP1 in cultured cells and found increased SP1 led to increased levels of mRNA of G6PD, PGD and TKT (**Figure RL30M**) and protein levels of G6PD and TKT (**Figure RL30N**). When treated with AA, we found that SP1 mRNA level was not changed, but SP1 protein level was downregulated, indicating AA led to SP1 degradation in the post-translation stage (**Figure RL30O**). We verified this result by the reduction of SP1, which caused by AA exposure, was blocked by the proteasome inhibitor MG132 (**Figure RL30P**). In addition, to validate whether AA could enhance 3p loss and *VHL* mutations caused glycolysis addiction, we measured the intracellular lactate when treated with AA in *VHL*^{KD} ACHN cells. The results showed that AA treatment significantly upregulated the *VHL* deficiency caused lactate increase (**Figure RL30Q**).

Taken together, these results indicated that the AA enhanced the response to sunitinib therapeutic treatment by downregulating pentose phosphate pathway and decreasing the ribose-5-phosphate. More importantly, we verified that SP1 is the key transcription factor of AA regulating pentose phosphate pathway. We have added these results in the **Figure 2** in the revision.

Figure RL30.

- A, The cell proliferation measured by MTT assay.
- B, Transwell detected the effect of AA and Sunitinib treatment on cell invasiveness.
- C-F, AA treatment inhibits the expressions of G6PD, PGD and TKT in time-and dose-dependent manners.
- G, The ribose-5-phosphate concentrations in cells treated with AA or not.
- H-K, The impacts of SP1, YY1, SREBF2, SMAD4 knockdown on PPP enzymes at mRNA level.
- L, SP1 knockdown downregulated PPP enzymes at protein levels.
- M-N, SP1 overexpression upregulated PPP enzymes at mRNA and protein levels.
- O, AA treatment did not affect the transcription of SP1.
- P, MG132 abrogated the AA mediated PPP enzymes downregulation by inhibiting the degradation of SP1.
- Q, AA enhanced the *VHL* deficiency caused lactate production increase.

Q31. In lines 319ff the authors write that tumor size and LDH levels were associated with diverse proteomic features, the revealing the molecular mechanisms underlying the influence of clinical parameters on sunitinib therapeutic response (Figure 4M). This and some other claims should be tested (in vitro/in vivo) until suggested mechanisms are potentially verified.

Response:

We thank the reviewer for the comments. The reviewer is correct that some of our claims need further validation. There were great gaps between the tumor size, plasma LDH levels and the underlying mechanisms. Thus we tuned down our statement in the revision. As supplementary, in the revision, we collected independent data to test whether the plasma LDH levels were associated with the potential immune evasion of ccRCC tumors. Specifically, 72 ccRCC tumor samples were conducted PD-L1 immunohistochemistry. Comparing the preoperative plasma LDH level between PD-L1 positive and negative patients, we found that PD-L1 positive patients had higher plasma LDH level (**Figure RL31, see also Figure 5M in the revision**), further supporting the association between plasma LDH level and the potential immune evasion of ccRCC tumors.

Figure RL31. Comparison of plasma LDH levels between PD-L1 positive and negative ccRCC patients.

Q32. In lines 612f, spearman's correlations between CNA values (gene level and arm level) and protein abundances were calculated using 12,310 genes quantified at both CNA and proteome levels. How reliable is this analysis when quite some of the proteomic data seems to be unreliable (was not regarded in other analysis)?

Response:

Thanks for your kind suggestions, and we apologize for the unclear presentation. In the revision, we re-analyzed the correlation between CNA values and proteome data based using the 7,451 proteins identified in at least 25% of the samples (**Figure RL32**). A total of 296 proteins showed significantly *cis*-effect, these proteins were enriched in pathways including innate immune system, glycolysis and TCA cycle. A total of 110,221 pairs of proteins and CNAs showed significantly *trans*-effect. The CNAs with *trans*-effect were centered around 2p, 3p, 5p, 7q, 11p, 14p, and 14q. The *cis/trans* effect of 7q amplification were correlated to the non-response of the Sunitinib, which were also verified by the experiment. We overexpressed LAMTOR4, MDH2 and CALU which were three genes located within chromosome 7q, in 786-O cells, respectively. We found overexpressed LAMTOR4, MDH2 and CALU increased the phosphorylation of S6K, indicating activation of mTORC1. Next, we treated the candidate genes overexpressed cells and normal cells with or without sunitinib. By monitoring the invasion of cancer cells with transwell assay, we found increased expression of LAMTOR4, MDH2 or CALU abrogated the effects of sunitinib. Following reviewer's suggestions, we updated the Figure and data in revised manuscript.

Figure RL32. Correlations of CNA (x axes) with protein abundance (y axes). Significant ($q < 0.1$) positive (red) and negative (green) correlations are shown.

Other points.

Q33. Line 82: what are 'strict criteria' for patient sampling?

Response:

Thank reviewer for the comments. We apologize for not explaining it clearly and revised the description of sample collection. We included 115 patients with advanced ccRCC, who were treated with TKIs at the Department of Urology of Fudan University Shanghai Cancer Center (FUSCC, Shanghai, China) from Jan 2008 to Dec 2019. All electronic medical records were screened retrospectively. Among the 115 cases, 47 cases were localized ccRCC at the time of surgery and were included due to the subsequent development of metastatic disease. Tumor and adjacent non-tumor tissue samples were collected during surgery and were available from the FUSCC tissue bank. Tumor and paired tumor adjacent tissues (collected > 2 cm from the tumor margin) were collected within 30 min after resection, immediately transferred into sterile freezing vials and snap frozen in liquid nitrogen, cut into ~ 0.5cm³ pieces under -40 °C, then split and stored at -80 °C until being used. The histologic sections were obtained from top and bottom portions of tumor/adjacent tissues and Hematoxylin and eosin (H&E)-stained for review. Each tumor/adjacent sample was checked by an expert pathologist to confirm the sample quality according to the following standards: 1) histopathologically defined ccRCC tumors; 2) tumor samples with tumor cell rate (tumor purity) > 90%; 3) no tumor cells in the adjacent tissues. In conclusion, we improved the method descriptions of sample collection, sample assessment and quality control in the revised manuscript.

Q34-1. In lines 134ff the authors write that the mutational signature SBSS22, associated with aristolochic acid (AA) exposure, was higher in the non-responder than in the responder group (14.4% vs. 20.3%) and showed better overall survival upon sunitinib treatment. 1. In the text, the authors write that the AA signature would be correlated with better survival. Am I correct to assume that there is a swap of non-responders and responders in that sentence in lines 134f.

Response:

Thank you for the comments. Mutational spectra revealed that the frequency of A > T transversions was higher in the Non-Responder group than in the Responder group (14.4% vs. 20.3%) (Fisher's exact test, $p < 2.2e-16$) (**Figure RL33A**). The results showed that Responder and Non-Responder groups showed similar single-base substitution (SBS) signature (SBS1,

SBS5, SBS22, SBS40) distribution (**Figure RL33B**). Although AA signature was associated with better survival, the frequencies of patients with AA signature showed no significant difference between Responders (n = 4) and Non-Responders (n = 15) (Fisher's exact test, p = 1).

Figure RL33.

A, Mutational spectra in Non-Responder group and Responder group.

B, single-base substitution (SBS) signature (SBS1, SBS5, SBS22, SBS40) distribution in Non-Responder group and Responder group.

Q34-2. Did the authors find this signature also in the tissue adjacent to the tumor, or is this mutational signature confined to the tumor cells?

Response:

We apologize for the confusion. In this study, we used adjacent normal tissues instead of corresponding normal blood samples as the background to detect the somatic mutations in the tumor tissues. Following reviewer's comments, we re-analyzed the mutation signatures using the sequencing result from peripheral blood mononuclear cells of normal people as the background. Few mutational signatures could be identified in the tumor adjacent tissues. To compare the mutation rate AA signatures between tumor and tumor adjacent tissues, we used A>T transversion as marker of AA. The results showed signature of AA (A>T transversions) were found in the tissue adjacent to the tumor. Moreover, we compared the frequency of A>T transversions between tumor and tumor adjacent tissues. The data showed that the mutation signatures of tumor tissues were significantly higher than the mutation signatures of tumor adjacent tissues (p = 3.213e-9, **Figure RL34**), demonstrating the strong mutagenic effect of AA in tumor tissues. Li *et al.* (PMID:

33004515) reported that AA signatures were found in normal human tissues, which was consistent with our conclusion.

Figure RL34. A>T transversions in tumor adjacent tissues and tumor tissues.

Q34-3. What was purity of the tumors and what was allele frequency of the mutations and mutational signatures?

Response:

Thanks for the suggestions. The purities of tumors were higher than 90%. During the sample collection, the histologic sections were obtained from top and bottom portions of tumor/adjacent tissues and Hematoxylin and eosin (H&E)-stained for review. Each tumor/adjacent sample was checked by an expert pathologist to confirm the sample quality according to the following standards: 1) histopathological defined ccRCC tumors; 2) tumor samples with tumor cell rate (tumor purity) > 90%; 3) no tumor cells in the adjacent tissues.

Following the reviewer's suggestions, we calculated the allele frequency of the mutations, these results were listed in the revise **Supplementary Table 2**. The allele frequency of the most frequently mutated genes were 0.056-0.736 for *VHL*, 0.075-0.557 for *PBRM1*, 0.04-0.487 for *BAP1*, and 0.045-0.67 for *SETD2* (**Figure RL 35**).

Figure RL35. Variant allele frequencies of the most frequently mutant genes in ccRCC.

Q34-4. AA has been associated with DNA damage and somatic mutations (reference Ng et al., 2017). Did cells of the TME harbor the same signatures or were these cells wildtype for the respective positions? If the TME was affected by AA, why not the adjacent tissue?

Response:

Thanks for the comments. The AA mutation signature was identified in the normal urothelium tissue in the previous report (PMID: 33004515). However, it was difficult to evaluate the mutation signature of the TME components using our data. Following reviewer's comments, we calculated the variant allele frequency (VAF) of AA related-signature in the tumor samples. We used VAF of *VHL* as the proportion of tumor cell in the tissues since *VHL* mutation was thought as the early events in ccRCC carcinogenesis. We used VAF of T > A transversion as the maker of AA. We noticed that VAF of *VHL* was approximately equal or higher than the VAF of T > A transversion in most samples. The result indicated that most of the AA signature were inferred from the tumor cells. However, VAF of T > A transversion were higher than the VAF of *VHL* in two samples, which means there were proportion of the TME harbor with the AA signature.

As mentioned in Q34-2, the signature of AA (A > T transversions) was found in the tissue adjacent to the tumor. Moreover, we compared the frequency of A > T transversions between tumor and tumor adjacent tissues. The data showed that the mutation signatures of tumor tissues were significantly higher than the mutation signatures of tumor adjacent tissues ($p = 3.213e-9$, **Figure RL36**), demonstrating the strong mutagenic effect of AA in tumor tissues.

Figure RL36. VAF of *VHL* mutation and T > A transversions in the cohort.

Q34-5. Looking at Figure S2B, the numbers of responders and non-responders showing this AA signature appear to be quite low (4 responders, 15 non-responders), which fits the percentages given in the text. These tumors carry quite variable ratios of other mutational signatures (SBS1, SBS5 and SBS40 are shown – how about tobacco-related signatures that might be present in smokers?): how were these other signatures treated in the bioinformatics analysis to exclude any other potential confounding factors?

Response:

Thanks for the suggestions. Following reviewer’s suggestions, we performed the non-negative matrix factorization analysis on the mutation spectra. The results showed that 67 signature were found in the cohort (SBS1-SBS85).

Following the reviewer’s suggestions, we analyzed the mutation signatures between smokers (n = 51) and non-smokers (n = 62). The result showed that frequency of signature 15 was significantly higher in smokers than in the nonsmokers (p = 0.0091). We compared the signatures to COSMIC databases. The Pearson correlation coefficient of signature 15 and four COSMIC signatures were higher than 0.5, including SBS40, SBS3, SBS58 and SBS4 (**Figure RL37B**). The number of mutations attributed to SBS40 is correlated with age. The aetiology of SBS58 is still unknown. SBS3 is correlated to defective homologous recombination based DNA Repair and SBS4 is associated with tobacco smoking. In conclusion, we have discovered more signatures were

correlated to the patient's information. With the multivariate analysis of these signatures, none of them were correlated to efficacy of Sunitinib, except AA related signature SBS22 (**Supplementary Figure Table 2**).

Figure RL37.

A, Frequency of mutation signature 15 in smokers and non-smokers.

B, Pearson correlation coefficient of signature 15 and COSMIC signatures.

Q35. Some of the regression lines that are shown in Figure 4J/L are interesting. The line in the right panel of Figure 4L is way below the cloud that has the most data points. The graph in the left panel of this figure has some outliers, but the majority of data points seem to be independent of LDH levels.

Response:

Thank the reviewer for the comment. The points in the right panel of Figure 4L which reviewer noted were the missing values after assigned as $10e-4$. We excluded these missing value points and recalculated the correlation between LDH and CD274 expression levels. The results showed that LDH and CD274 expression levels were significantly correlated (**Figure RL38A**). As for the left panel of Figure 4L, we excluded the outliers ($LDH \geq 300$, $n = 6$) and recalculated the correlation between LDH and CD8A expression levels. We found that CD8A expression levels were still significantly correlated with LDH levels (**Figure RL38B**). In conclusion, the missing values and outliers in these two figures did not impact the results.

Figure RL38.

A, Scatter plot showing the correlation between LDH and CD274 expression.

B, Scatter plot showing the correlation between LDH and CD8A expression.

Q36. In line 341 the citrate cycle and the TCA cycle are mentioned to be upregulated in the Cold group. What is difference between citrate and TCA cycle?

Response:

Thanks for the comment. The appropriate statement was “citrate cycle / TCA cycle”. We apologized for the typo. We have revised the statement in the revision.

Q37. In line 370 the authors speak of redundant angiogenesis (driven by TGFB1 resulted in sunitinib resistance). What is ‘redundant’ angiogenesis?

Response:

Thanks for the comments. We apologize for the misleading in the previous version and thank the reviewer for pointing out. The statement in the previous manuscript that the “redundant” angiogenesis driven by TGFB1 resulted in Sunitinib resistance was inaccurate. TGFB1 was co-expressed with proteins involved in angiogenesis and tumor immune escape in ccRCC, which may indicate the resistance role of TGFB1 in Sunitinib treatment targeted therapy. To be more accurate, we revised the statement as: TGFB1 signaling might be an “alternative” pathway in angiogenesis.

To assess the immune infiltration, we performed cell type deconvolution analysis, inferring the relative abundance of different cell types in the tumor microenvironment. Consensus clustering

based on the inferred cell proportion identified three sets of tumors defined as T-cell infiltrated, Cold, and Progenitor-cell infiltrated. It's worth noting that the Progenitor-cell infiltrated group contained the lowest Responder proportion. The Progenitor-cell infiltrated group were characterized by the progenitors of types of immune cells including common myeloid progenitor (CMP) cells, manifesting as the upregulation of platelet aggregation formation, and complement cascade at the proteome level. Consistently, we found that patients in the Progenitor-cell infiltrated group showed the highest expression of platelet marker CD321 (Kruskal–Wallis test, $p = 0.035$) and an elevated level of plasma PLT, which revealed the connection of platelet-activation with TME-mediated Sunitinib resistance. To further explore the influence of such a TME, we evaluated the differentially expressed proteins among three clusters, especially those involved in platelet aggregate formation and the complement cascade. We found that TGFB1, a growth factor associated with multiple oncogenic pathways such as tumors proliferation, EMT, angiogenesis, immune evasion, and metastasis, was significantly increased in the Progenitor-cell infiltrated cluster and was well-correlated with the Sunitinib response.

In summary, we considered that the TME comprising CMPs, multipotent progenitors, involving platelet aggregation, the complement cascade and TGFB1 signal led to sunitinib therapy failure. We have revised the manuscript and thanks the reviewers again for pointing out.

Q38. In lines 404f the authors write that strong regional characteristics would indicate the necessity of considering the genetic background of the Chinese population in ccRCC treatment. The authors have not really found differences in the genetic background of Chinese vs. other populations. Rather, they found the environmental factor aristolochic acid and the genetic (better: somatic) alterations that are induced by this drug to be necessary to be taken into account. The authors do not distinguish between true genetic and environmental factors in their study.

Response:

Thank the reviewer for the comment. The reviewer is correct that we rarely discussed the differences in genetic background of different populations. To make the description more appropriate, we revised the sentence as “The results showed strong regional characteristics, indicating the necessity of considering the life habits and environmental factors of the Chinese population in

ccRCC treatment” in the revision. Thank you again for your professional comments.

Q39. The finding that this AA signature is caused by aristolochic acid strongly speaks against ingestion of this herb. While the AA signature seems to be correlated with better survival in ccRCC, it is very likely correlated with worse survival in other entities, many of which are addicted to glycolysis even in normoxia (i.e., Warburg effect).

Response:

We thank the reviewer for the question. In the previous manuscript, we found that patients with SBS22, associated with aristolochic acid (AA) exposure, showed better overall survival in Sunitinib treatment (GB-wilcoxon test, $p = 0.038$) (**Figure RL39A**). Interestingly, it was observed that glucose metabolism was shunted away from the pentose phosphate pathway (PPP) into glycolysis in tumors with AA signature at protein level (**Figure RL39B**), manifested as upregulation of PFKP (FC = 1.51, $p = 0.01$), PFKP (FC = 1.73, $p = 0.0028$), GPI (FC = 1.47, $p = 0.04$) and downregulation of G6PD, PGD, TKT ($p < 0.05$).

During the revision, we conducted a series of experiments to probe the function of AA exposure. In the experiments *in vitro*, we found that AA treatment did not impact cell proliferation directly, but enhanced the inhibition to cell proliferation of Sunitinib (**Figure RL39C**). The transwell assay showed that AA treatment did not impact cell invasiveness directly, but enhanced the inhibition of Sunitinib to cell invasiveness (**Figure RL39D**). Moreover, we verified that treating 786-O cells with AA caused notably decreased PPP enzymes (G6PD, PGD and TKT) in time- and dose- dependent manners (**Figure RL39E-H**). The levels of ribose-5-phosphate, which is an important product in both oxidative and non-oxidative PPP, decreased notably in AA treated cells (**Figure RL39I**).

We next investigated how AA inhibited the expressions of those genes, by predicting the potential transcriptional factors which involved in the regulating the transcription of G6PD, PGD and TKT, and identified four kinds of TF had the common binding/recognizing regions in G6PD, PGD and TKT promoters, including SP1, SMAD4, YY1, and SREBF2. By knocking down these four candidate transcriptional factors using siRNA in cultured ccRCC cancer cells (**Figure RL39J-M**), we found the mRNA levels of G6PD and PGD decreased in SP1 knocking down cells (**Figure RL39J**). However, decreased mRNA levels of SMAD4, YY1 and SREBF2, did not affect the transcription of G6PD, PGD and TKT (**Figure RL39K-M**). Consistently, we observed the

downregulation of G6PD and PGD proteins in SP1 knocking down cells (**Figure RL39N**). Furthermore, we overexpressed SP1 in cultured cells and found SP1 overexpression led to increased levels of mRNA of G6PD, PGD and TKT (**Figure RL39O**) and protein levels of G6PD and TKT (**Figure RL39P**). When treated with AA, we found that SP1 mRNA level was not changed, but SP1 protein level was downregulated, indicating AA led to SP1 degradation in the post-translation stage (**Figure RL39Q**). We verified this result by the reduction of SP1, which caused by AA exposure, was blocked by the proteasome inhibitor MG132 (**Figure RL39R**). In addition, to validate whether AA could enhance 3p loss and *VHL* mutations caused glycolysis addiction, we measured the intracellular lactate when treated with AA in *VHL*^{KD} ACHN cells. The results showed that AA treatment significantly upregulated the *VHL* deficiency caused lactate increase (**Figure RL39S**). Taken together, these results indicated that the AA enhanced the response to sunitinib therapeutic treatment by downregulating pentose phosphate pathway and decreasing the ribose-5-phosphate. AA exposure enhanced the glycolysis addiction caused by *VHL* deficiency. More importantly, we verified that SP1 is the key transcription factor of AA regulating pentose phosphate pathway (**Figure 2n**). The impacts of AA were described in **Figure 2** in the revision.

Figure RL39.

A, Kaplan–Meier curves of PFS for patients with and without AA signature (GB-Wilcoxon test).

B, Glucose metabolism alteration caused by the AA signature.

C, The cell proliferation measured by MTT assay.

D, Transwell detected the effect of AA and Sunitinib treatment on cell invasiveness.

E-H, AA treatment inhibits the expressions of G6PD, PGD and TKT in time- and dose-dependent manners.

I, The ribose-5-phosphate concentrations in cells treated with AA or not.

J-M, The impacts of SP1, YY1, SREBF2, SMAD4 knockdown on PPP enzymes at mRNA level.

N, SP1 knockdown downregulated PPP enzymes at protein level.

O-P, SP1 overexpression upregulated PPP enzymes at mRNA and protein levels.

Q, AA treatment did not affect the transcription of SP1.

R, MG132 abrogated the AA mediated PPP enzymes downregulation by inhibiting the degradation of SP1.

S, AA enhanced the *VHL* deficiency caused lactate production increase.

Q40. In line 424f the authors write that the 7q amplification was confirmed to upregulate mTOR signaling pathways and amino acid turnover in ccRCC. I could not find the data that would have confirmed causality of the correlation that is presented in the manuscript. Later, in lines 477f, the authors claim to have identified the molecular mechanisms underlying sunitinib response. This claim is not sufficiently backed up with experimental data.

Response: We thank the reviewer for the question. First, in the previous manuscript, we found that 7q gain was associated with shorter survival in this Sunitinib treatment cohort (**Figure RL40A-B**). In additions, genes that more frequently amplified in the Non-Responder group than in the Responder group were located in 7q (Fisher's exact test, $p < 0.05$) (**Figure RL40C**). To further evaluated the proteomic consequences of 7q gain in ccRCC, we performed *cis-/trans*-effect analysis. Proteins showed significantly positive correlation with the 7q copy number (CN) (Spearman's correlation, $p < 0.05$) were enriched in pathways including lysosome (LARP1, LARP2), mTOR signaling (RRAGB, RRAGD, LAMTOR2, LAMTOR4, MLST8), TCA Cycle (CS, SDHA, MDH2), oxidative phosphorylation (COX6C, NDUFS6), and fatty acid biosynthesis (ACACA, MCAT) (**Figure RL40D**). We supposed that 7q gain activated mTOR signaling and upregulated amino acid turnover.

In the revision, we overexpressed LAMTOR4, MDH2 and CALU, located in chromosome 7q and *cis*-regulated by 7q gain, to mimic chromosome 7q gain in 786-O cells, respectively (**Figure RL40E**). We found overexpressed LAMTOR4, MDH2 and CALU increased the phosphorylation of S6K, indicating activation of mTORC1 (**Figure RL40F**). Next, we treated the candidate genes overexpressed cells and normal cells with or without Sunitinib. By monitoring the invasion of cancer cells with transwell assay, we found increased expression of LAMTOR4, MDH2 or CALU abrogated the effects of Sunitinib (**Figure RL40G**). We have added these results in **Figure 3** the revision.

Figure RL40.

A, Cox regression analysis of significant arm-level CNA events, based on the PFS.

B, Kaplan–Meier curves of PFS for patients with or without chromosome 7q gain (log-rank test).

C, Comparison of gene-level CNAs between Responders and Non-Responder in this cohort. The upper plot illustrates the frequency of CNA events, the lower plot illustrates the $-\log_{10}(p \text{ value})$ of each gene for the comparison of Responders and Non-Responder (Fisher’s exact test).

D, Heatmap of proteins positively correlated with the Chromosome 7q Copy number. Spearman’s correlations are showed in the right panel.

E, Transfection efficiency of LAMTOR4, MDH2 and CALU overexpression plasmids detected by qRT-PCR and western blot.

F, The overexpression of LAMTOR4, MDH2 and CALU increased the phosphorylation of S6K.

G, Transwell detected the effect of LAMTOR4, MDH2, CALU overexpression and Sunitinib treatment on cell invasiveness.

Q41. In lines 451ff the authors suggest that the platelets in the TME were responsible for sunitinib failure. Several causes of poor treatment-response have been identified/published (compare e.g., PMID:25446042). The authors should consider testing their hypothesis, such mechanism could indeed be interesting, if validated.

Response:

We thank the reviewer for the question. It was noted that potential mechanisms of resistance to sunitinib in RCC including tumor microenvironment mediated cytokines secretion. Due to the close relationship between TGFB1, VEGF and FGF and platelets, which can secrete these cytokines, thus promoting the invasion and growth of tumor cells. Therefore, in order to verify the effect of platelets in TME on the efficacy of Sunitinib, we applied TGFB1, VEGF and FGF closely related to platelet function to ccRCC cancer cells. The results showed that after intervention with TGFB1, VEGF or FGF, the effect of Sunitinib, inhibiting the invasion of ccRCC, was significantly reduced (**Figure RL41**). which also indicated that platelets in TME could promote the progress of ccRCC by promoting the secretion of cytokines.

Figure RL41. Transwell detected the effect of TGFB1, VEGF and FGF intervention and Sunitinib treatment on cell invasiveness.

Reviewer #5 (Remarks to the Author): Expert in renal cell carcinoma sunitinib therapy and immunotherapy

Q42. The paper deals with “proteogenomics” analyses on over 100 clear-cell mRCC top come up with associations with “clinical outcomes”. The paper, from a clinical standpoint, is confusing especially with multiple definitions on what constitutes activity. In the abstract alone, we see terms

such as “treatment failure”, “treatment benefit”, “therapeutic outcome”, “prognosis” and “treatment prediction”.

Response:

Many thanks for pointing this out. To make it clearer, we have adjusted these definitions as you suggested. In the revised manuscript, we changed our descriptions and focused on efficacy of Sunitinib. We used “non-response” to replace “treatment failure” and “better prognosis” to replace “clinical benefit”, based on objective response and survival time after Sunitinib treatment respectively.

This manuscript has been revised extensively according to the reviewers' constructive suggestions. Thank you again for your professional suggestions.

Q43. While it is quite interesting to focus on the Chinese patients as most of biomarkers of RCC responses have been in Caucasians, this interesting aspect of the study is dampened by the fact that TKI, as single agents, are not standard of care in mRCC anymore.

Response:

Many thanks for pointing this out. Recently, targeted therapy combined with immunotherapy and double immunotherapy have been recommended as first-line therapy for advanced clear cell renal cell carcinoma respectively. However, Sunitinib is still a first-line option for patients with advanced clear cell renal cell carcinoma and non-clear cell renal cell carcinoma (NCCN Clinical Practice Guidelines in Kidney Cancer, Version 2.2022; <http://www.nccn.org>). Targeted therapy is still the cornerstone of the treatment of advanced renal cell carcinoma, and immunotherapy plays a further synergistic role. Thus, our research still has an important value and could provide some evidence for precise use of TKI in Chinese patients.

Q44. In addition, and despite FDR of 1% is used for Proteome identification and quantification, it is not clear where else it is used down the road especially with this “unsupervised and unplanned”. What about other clinical metrics like PFS and OS and why responses (since they are the ones used as a primary (Methods) correlate for efficacy) were not centrally reviewed?

Response:

Thanks for the comment. We apologize for not explaining it clearly.

As for the use of 1%FDR.

Raw files were processed in Firmiana and searched against the human NCBI RefSeq protein database using the Mascot 2.4 search engine. The data were also searched against a decoy database so that peptide and protein identifications were accepted at an FDR of 1%. Proteins with at least 1 unique peptide were selected for further analysis. As a result, 12,310 proteins were identified in this study totally.

As for the use of response, PFS and OS.

RECIST based response, PFS and OS are three important objects discussed in this study. RECIST v1.1 is currently the reference standard for evaluating cancer treatments for objective response in solid tumors. We attempted to find out the intrinsic molecular basis of resistance/sensitivity of Sunitinib by analyzing the molecular features associated with Sunitinib response. The reviewer is correct that OS and PFS are primary correlates for efficacy, thus OS and PFS were also important analysis targets in this study, which were centrally described in **Supplementary Figure 1G, 2G** and **3B** in the previous manuscript.

Q45. It is also interesting that responses did not differ by MSKCC/IMDC criteria that are known for a long time to prognosticate well in mRCC.

Response:

Thanks for the comment. Despite MSKCC/IMDC risks were not associated with Sunitinib response, MSKCC/IMDC risks was significantly associated with patient prognosis (**Figure RL42**), which were described in the **Supplementary Figure 1F** in the previous manuscript (**Supplementary Figure 2G** in the revision).

Figure RL42. Kaplan–Meier curves of PFS/OS for patients with different MSKCC/IMDC risks (log-rank test).

Q46. Figure S1G (patients’ baseline characteristic. Also what is the AA signatures could enhance 3p loss/VHL mutation-induced glycolysis additions) do not have data on exposure to AA. So how do we assume these patients had any exposure?

Response:

Thank the reviewer for the kind suggestion and we apologize for the unclear presentation.

We inferred that the patients with SBS22 were exposure to AA.

First, we calculated the mutational spectra of the cohort. The frequency of T > A transversion was the second largest in our cohort. Then we adopted the non-negative matrix factorization algorithms on the mutational spectra and mapping the mutational signature to the COSMIC database. SBS1, SBS5, SBS22, SBS40 were detected in our cohort, among which SBS22 was associated with exposure to aristolochic acid (AA). AA bonded to dA and dG residues in DNA to form aristo lactam-DNA adducts which are concentrated in the renal cortex. It is reported that aristo lactam-DNA were causally related to the initiation phase of tumorigenesis. Both dG and dA adducts block DNA replication and give rise to misincorporation of dA. When dAMP is inserted opposite the dA-AL adduct owing to misincorporation, the dA-AL is excised and replaced with dTMP leading to permanent A-to-T transversion. The repair results in a mutational pattern of marked no transcribed strand bias and the persistence of dA-AL adducts in tissues even after stopping exposure to AA for

decades. SBS22 was found in cancer samples with known exposures to aristolochic acid and the pattern of mutations exhibited by the signature is consistent with that observed in experimental systems of aristolochic acid exposure (PMID: 26443852, PMID: 29046434). In conclusion, patients with SBS22 mutational signature were supposed to be AA exposed, which was also widely used in recent studies (PMID: 32029730, PMID: 34971568, PMID: 31730861).

Q47. There is a claim that AA signature could enhance 3p loss/*VHL* mutation-induced glycolysis addiction. But what is the MOA?

Response:

We thank the reviewer for the question. In the revision, to validate whether AA could enhance 3p loss and *VHL* mutations caused glycolysis addiction, we measured the intracellular lactate when treated with AA in *VHL*^{KD} ACHN cells. The results showed that AA treatment significantly upregulated the *VHL* deficiency caused lactate increase (**Figure RL43A**).

We did not probe the specific MOA of AA exposure enhanced the glycolysis caused by 3p loss/*VHL* mutation. Instead, we investigate the impact of AA exposure on pentose phosphate pathway. Specifically, we found that AA exposure caused notably decreased PPP enzymes (G6PD, PGD and TKT) in time- and dose- dependent manners (**Figure RL43B-E**). The levels of ribose-5-phosphate, which is an important product in both oxidative and non-oxidative PPP, decreased notably in AA treated cells (**Figure RL43F**).

We further probed how AA inhibited the expressions of those genes, by predicting the potential transcriptional factors which involved in the regulating the transcription of G6PD, PGD and TKT, and identified four kinds of TF had the common binding/recognizing regions in G6PD, PGD and TKT promoters, including SP1, SMAD4, YY1, and SREBF2. By knocking down these four candidate transcriptional factors using siRNA in cultured ccRCC cancer cells (**Figure RL43G-J**), we found the mRNA levels of G6PD and PGD decreased in SP1 knocking down cells (**Figure RL43G**). However, decreased mRNA levels of SMAD4, YY1 and SREBF2, did not affect the transcription of G6PD, PGD and TKT (**Figure RL43H-J**). Consistently, we observed the downregulation of G6PD and PGD proteins in SP1 knocking down cells (**Figure RL43K**). Furthermore, we overexpressed SP1 in cultured cells and found SP1 overexpression led to increased levels of mRNA of G6PD, PGD and TKT (**Figure RL43L**) and protein levels of G6PD and TKT

(Figure RL43M). When treated with AA, we found that SP1 mRNA level was not changed, but SP1 protein level was downregulated, indicating AA led to SP1 degradation in the post-translation stage (Figure RL43N). We verified this result by the reduction of SP1, which caused by AA exposure, was blocked by the proteasome inhibitor MG132 (Figure RL43O).

Taken together, these results indicated that the AA enhanced the response to sunitinib therapeutic treatment by downregulating pentose phosphate pathway and decreasing the ribose-5-phosphate. More importantly, we verified that SP1 is the key transcription factor of AA regulating pentose phosphate pathway.

Figure RL43.

A, AA enhanced the *VHL* deficiency caused lactate production increase.

B-E, AA treatment inhibits the expressions of G6PD, PGD and TKT in time-and dose-dependent manners.

F, The ribose-5-phosphate concentrations in cells treated with AA or not.

G-J, The impacts of SP1, YY1, SREBF2, SMAD4 knockdown on PPP enzymes at mRNA level.

K, SP1 knockdown downregulated PPP enzymes at protein level.

L-M, SP1 overexpression upregulated PPP enzymes at mRNA and protein levels.

N, AA treatment did not affect the transcription of SP1.

O, MG132 abrogated the AA mediated PPP enzymes downregulation by inhibiting the degradation of SP1.

Q48. This is a premature statement. Finally, the “validation” cohort needs more information. How independent is this cohort or did you split your cohort into training and “validation”. In summary, several problems with this report that need to be addressed in great details.

Response:

Many thanks for pointing this out. In this research, we applied the 80/20 split for training and test set. A total of 96 samples, including 18 Responders and 78 Non-Responders, were used as the training set. The rest of 19 samples were used as the test set. The linear classifier of significance of drug response outliers were trained using the training cohort. The result showed the recall of 0.9 (9/10) of Responders and 1.0 (10/10) of Non-Responders in the test set (**Figure RL44A**). In addition, to validated the robustness of the classifier, we additionally collected 50 cases of ccRCC patients as an independent validation set. These 50 patients also received Sunitinib treatment and comprised 15 Responders and 35 Non-Responders. Proteomic identification was conducted using the same methods as the proteome analysis of the 115 ccRCC samples. As a result, 11 of 15 Responder patients were predicted correctly, whereas 33 of 35 Non-Responder patients were predicted correctly (**Figure RL44B**), indicating the robustness of the classifier.

Figure RL44. Classification confusion matrix of the 20% test set and an independent validation set. The number of samples identified is noted in each box.

REVIEWER COMMENTS

Reviewer #1 (Remarks to the Author):

The authors have done a significant amount of work and analyses to address my previous points. The manuscript has evolved, but some of the experiments and data raise new questions. Specific comments:

1. The authors still postulate that VHL loss causes glycolysis addiction. While the new data show increased lactate levels following VHL knock-down in ACHN cells, that does not show that the cells were not addicted to glycolysis before VHL inhibition nor that they have become addicted to glycolysis afterwards. Enhanced glycolysis is not the same as glycolysis addiction. Moreover, it is not clear how the acute effects of VHL inhibition in ACHN cells are related to the effects of VHL inactivation in cancer clones that arise from VHL mutant clones. That VHL has been knocked down is clear, but is this level of knockdown sufficient to induce the expression of hypoxia-inducible factors? It could be more straightforward to not to include these VHL knockdown data in the manuscript, the strengths of which are not in functional work but in an in-depth analysis of clinical samples.

2. In figure 3e, why are only three genes marked as cis-regulated? The text was not clear on why these three genes were picked for further analysis. Also, even though LAMTOR4 was classified as an mTOR pathway gene, all LAMTOR4, MDH2 and CALU had the same effect on pS6K level. Was this result expected? Did the authors only try these three genes, or did some genes have different effects?

3. The idea of using a trans-well migration assay to address a question about a drug being curative (my original comment) or having anti-tumor effects more generally is poorly justified. While the migratory phenotype might have something to do with drug resistance, it is not clear to the reader whether this is the case here. This comment relates to the data shown in 2f, 3g, 4h and 6m. The images are hard to interpret, and no quantification is shown. But the biggest issue is the underlying logic. Why use this assay in this context?

4. The original version of this manuscript linked AA exposure to higher mutation burden, and the authors also identified some pathway-level characteristics of the AA-exposed tumors. This seemed meaningful and the role of AA as a mutagen has been reported elsewhere too. The current version has expanded this by suggesting that AA exposure would directly affect cell signaling. While this is likely to be the case, as would with any chemical, it is less clear how the increased level of mutations is linked to the acute signaling events and possible alterations in drug responsiveness. How close are the AA concentrations used in the experimental assays to those seen in patients with dietary AA exposure? The thinking and model related to AA should be clarified.

Minor:

Fig. 1B still doesn't have units for the tumor size, PFS or age.

Reviewer #3 (Remarks to the Author):

The authors can be congratulated to their excellent study with further improvements after the review process. My comments have been fully addressed in a satisfactory fashion and the paper merits its publication.

Just some very minor comments:

- 1.) Line 138: "Other clinical information, including...": Hypertension and diabetes could be added.
- 2.) Line 142:".... level, urine protein state, estimated glomerular filtration rate (eGFR)": Formula used to calculate eGFR should be stated (e.g. CKD-EPI).
- 3.) Could not find the Supplementary Table 1 among the uploaded files.
- 4.) Obtained findings on miRNA could be briefly mentioned as data not shown in the paper also.

Reviewer #4 (Remarks to the Author):

Qu et al. have substantially enlarged the datasets in their study (e.g., phosphoproteomics, RNA-seq, more experiments). In general, this has vastly improved the rationale of their argumentation. The conclusions seem to be mostly backed up with data.

However, I have some concerns with the rebuttal letter and the data that is presented there (some of this is also part of the main figures in the manuscript).

Reviewer 1 indicated that it should not be possible to know whether the markers identified would be associated with therapy. In response, the authors state that they recruited 37 cases of ccRCC patients, who did not receive any treatment after surgery and this cohort would support their claims. While it seems ideal to have such a control-cohort, I am unsure if it was ethical to not provide patients/human beings with any treatment other than surgery in the conditions of advanced and metastasized cancer. The authors should comment on this.

In the following, I focus on the points that I had in my initial review and on the specific responses having been provided.

Q22: The authors knocked down the VHL gene and performed trans-well assays to test whether this knockdown would impinge on the migration capacity of cells. 1. The achieved knockdown efficiency of 50% is not convincing. The authors should use some other means of knocking down/out VHL and seek to achieve an efficiency of at least 70-80%. Use of two or three independent si/shRNAs should be performed to rule out off-target effects. 2. In panel D of Figure RL17/Supplementary Figure 5, the authors present the outcome of this experiment. In the photograph, they see a reduction in the number of cells particularly in the VHL-knockdown cells having been treated with sunitinib. The data should be quantified and presented together with the photograph in the supplementary figure. Quantification of image data should be done also for other figures, like effects of overexpressing LAMTOR4, MDH2 and CALU, presented in Figure 3.

Q23: The authors did a series of experiments to verify a link between VHL deficiency and metabolic phenotypes in the context of sunitinib-treatment. In panel D of Figure RL20, the authors show that the mutation status in VHL correlates with response to sunitinib in patients of the IMmotion151 study. There seem to be two groups that are most different between the two cohorts: patients with partial response (PR) and patients not evaluated (NE). Were patients who could not be evaluated included in the calculation of statistical difference between the cohorts (Fisher's exact test)? If so, this group should likely be removed prior to statistical testing.

Q24: The authors describe very well the experimental details in the sequencing experiment. However, the numbers of reads per sample, the numbers of mapped reads, and the coverages of exomes should be provided as well. In the revised manuscript the author states that : 'The WES target region was 33 M.' Do the authors mean 33 Mbp? The xGen 1.0 panel which was applied for exome enrichment had been designed in 2015 to span 39 Mb of the human genome and to provide coverage of 19,396 genes (https://genome.ucsc.edu/cgi-bin/hgTables?db=hg38&hgta_group=map&hgta_track=exomeProbesets&hgta_table=xGen_Research_Targets_V2&hgta_doSchema=describe+table+schema, accessed 22/06/03). Why are the numbers not consistent?

Q27/Figure RL25: In the original manuscript, it was not clear how many proteins were recurrently identified in the 115 samples (88 responders and 27 non-responders). If I get it right from panel B in Figure RL25, the number of proteins that were identified in more than 1 sample was 52 specifically identified in responders, and less than 10 proteins were identified in more than 2 different samples. The numbers seem to be better for proteins that were specifically identified in non-responders. Do the authors have some idea why the re-identification of proteins was so poor in the responders?

Q27-3: The authors write that they used the match-between-runs algorithm. In the manuscript, they cite Tyanova et al., 2016 (reference 52). In the rebuttal letter, they cite the original article for this algorithm (Cox et al., 2014). This should be corrected in the manuscript, even if both manuscripts come from the Mann-lab. The match-between-runs algorithm was originally implemented in the MaxQuant software. Did the authors use this software or, if not, how did they implement the algorithm?

Below, the authors write that they did not apply any other data imputation algorithm. However, this concrete and positive statement is followed by another statement in the very next sentence, where the authors write that missing values were assigned to $1e-5$, the minimum value across their proteome data. The assignment of such value is a kind of imputation. The right way to treat missing values is to annotate them as n.a. (not analyzed) or n.d. (not detected). The way the authors treat missing values is wrong even if this has been used in previous publications. This measure does not allow discrimination between real missing values/not expressed and random dropouts (compare Supplementary Data 4, and the statement on poor re-identification of proteins in different samples).

1. The impact of this assignment becomes obvious in Figure RL28. There, top up- and down-regulated proteins in the responder and non-responder groups are presented. The numbers of recurrently identified proteins were much higher in the non-responder group than in the responder group (see Q27/Figure RL25). A protein that was detected, for example, in just a few R-samples, however, in a much larger fraction in NR-samples, would automatically be much lower expressed in the R-samples as its imputed expression value of $1e-5$ was more prevalent in the R-group. A correct way out is to treat missing values as missing values.

2. Also the dynamic range of protein expression might be related to the assignment of an expression value of $1e-5$ for missing values as this could potentially explain the huge dynamic range (six orders of magnitude) that is described for each protein in the whole dataset. The lower plot in Figure RL29 suggests that many (it is not clear how many proteins are shown) proteins have been identified recurrently in a large number of samples (a span of \log_{10} fold-changes, and a very strong enrichment at some particular range on the Y-scale). How does this relate to the finding that so many proteins were found in 'just' 25% of the samples (the total number of samples having been analyzed is 115)?

3. In response to question Q35 'the points in the right panel of Figure 4L which the reviewer noted were the missing values after assigned as $10e-4$ ' points at consequences the assignment of any values to missing values could have', the authors excluded missing values and recalculated the correlation plot. This is realistic.

Q30 and revised manuscript lines 203 and the following: The authors predicted transcription factors potentially involved in the regulation of transcription of G6PD, PGD, and TKT. How was this prediction done? Which algorithm was applied? Which other transcription factors were predicted? How/why were SP1, SMAD4, YY1, and SREBF2 selected?

Q34-2: I am confused by the response. 1. Why have PBMCs from normal people been sequenced as the background? Where is the data? 2. The authors find that AA leads to a strongly enhanced AA Signature in the tumor tissue, while this is not found in the adjacent normal tissue (Figure RL34). They cite Li et al. (PMID:33004515) and write that their findings would be consistent with that previous publication. Li and coworkers had analyzed somatic clonal events in morphologically normal human urothelium with and without application of AA. These authors found drastically accelerated mutation accumulation also in

the normal urothelium and concluded that AA was a major mutagenic driving factor there. Based on the findings of Li et al., I would have expected to see indications of the AA signature (i.e., A>T transversions) also in normal adjacent tissue from patients having been treated with AA. The cited study does not really support the findings of Qu et al. The authors should discuss the apparent disparate findings.

Q34-3: The tumor samples were inspected by an expert pathologist who confirmed sample quality according to the following standards: '1) histopathological defined ccRCC tumors; 2) tumor samples with tumor cell rate (tumor purity) > 90%; 3) no tumor cells in the adjacent tissues.' This classification scheme does not make sense. I should suppose that all samples matched standard 1 – only ccRCC samples were taken into the study. All 115 tumor samples in the study should also meet criterion #2 and have >90% cellularity. #3: Why were tumors excluded that had tumor cells in adjacent tissue?

Some explanation of abbreviations, like 'KPS', should be warranted.

The span in VAFs, in combination with the high tumor purity (90%) suggests high intra-tumor heterogeneity in most patients. This should be discussed.

Q35: The relevance of the regression line in Figure RL38A could be disputed. The authors might look into this.

Other points

Materials and Methods:

Clinical sample collection

The authors write: 'At the last follow-up, 102 patients (88.7%) had progressive disease and 84 patients (73.0%) had died of ccRCC.' These numbers cannot be true for a cohort of 115 patients.

RNA extraction and RNA-seq

The authors now performed RNA-sequencing. In the materials and methods section, the library preparation protocol having been applied should be stated. In the results part, the mean number of reads/sample should be provided.

Protein extraction and trypsin digestion

The make and brand of the sonicator should be provided. The listed conditions (3s on and 3s off, amplitude 25%) do not make sense without knowing which instrument had been applied.

Reviewer #5 (Remarks to the Author):

the authors responded to their best abilities to my comments.

in total, they had 71 pages of comments. The reviewer appreciate that the authors made this huge and lauded effort

Reviewer #6 (Remarks to the Author): Expert in renal cell carcinomas and metabolism

The authors performed multi-omics analysis of primary ccRCCs from a Chinese cohort of patients that responded or did not respond to sunitinib treatment. The analysis showed activation of mTOR pathway in nonresponders and aristolochic acid signature in responders. While the study provides some interesting and new information, there are several issues that should be addressed.

Major comments:

1. The manuscript requires major reorganization, prioritization of the results, rewriting of the sections so that they integrate in a concise form all information supporting major conclusions, rather than description of the flow of experiments. Rigorous English editing is necessary. As is, the manuscript is very long, very descriptive, and includes a lot of details that confuse and distract from the main points and ultimately make the rigorous review process difficult to impossible. In general, figure legends lack sufficient information to evaluate what is shown.
2. The cell line data used for the validation of the bioinformatics data are of not very good quality. Only one cell line was used in individual experiments (786-0 in most and ACHN for VHL KDs), and the minimum should be two. The data are often not quantified, the effects are weak and there is no proper information about doses and time courses. Methods section lists 769P cell line but missed it being used.
3. While the effects of AA are interesting there is no description how exposure to AA was determined and if it was qualitative or quantitative. It is not clear how the doses of AA used in cell line experiments correspond to the AA exposures in patients.
4. The main conclusion that 7q gain induced mTOR signaling is responsible for lack of response to sunitinib is based on (i) enrichment for mTOR pathway among proteins that were positively correlated with 7q copy number gain; (ii) GSEA analysis in transcriptome; however, the results of RNA analysis are not clearly presented. (iii) increased mTOR activity measured by KSEA. The cell line validation of these conclusions is weak. A simple convincing validation would be immunocytochemistry for S6-P between the responder and non-responders.

Detailed comments:

Lanes 133-144: did the authors check combinations of the clinical parameters for differences between nonresponders and responders? These data are confusing as it is stated that clinical parameters were not different between responders and nonresponders, however, later on in the “clinical correlations of proteomic feature”, it appears that LDH and PLT have a number of correlations that are relevant for the stratification. This is confusing and perhaps all clinical correlations should be presented in one section.

What are the criteria for the definition of AA exposure? Is it possible to have a quantification of AA content in patients' tissues or any other concrete estimation of exposure?

Figure 2 and the legend require additional specific information: 2e and f - cell line and doses of the drugs; 2g duration of treatment, 2h, 2l, 2m - doses of AA; 2j, 2k – what is BC and NC; 2m – not clear what the bleached parts of the blots are. It is not clear how the doses of AA used in in vitro experiments relate to the AA exposures of patients. Cell line validation should use a minimum of two cell lines in each experiment.

Lane 187: the statement that “glucose was shunted away from pentose phosphate pathway” is not appropriate as there is no metabolomic flux data. The conclusion is implied by transcriptomic data and should be reported as such.

Was there an enrichment for SPI signature in the transcriptomics data of the responders?

Lane 221: “ results indicated that AA enhanced the response to sunitinib by downregulating ppp...” is an overstatement as there is no mechanistic causative connection shown.

What exactly is shown in Fig. 3e? is the expression of the genes based on RNA or CNA? Or are these protein expressions?

Fig. 3g – doses of sunitinib and quantification of the effects need to be provided. Overall, the effects of overexpression on S6K-P are rather minor. These data would need to be quantified. How is MDH2, mitochondrial malate dehydrogenase, connected to mTOR signaling?

Fig. 3f and 3g: It is not clear why these particular genes were selected for the in vitro analysis. Is the putative connection between 7q gain and mTOR supported in other studies or databases?

Lane 266 – HLA-B genes are highly polymorphic; how the mutation status was altered between the two groups?

Fig. 4g – it is well established that loss of VHL leads to increased glycolytic activity and lactate production; the fact that that AA appears to enhance this response is not referred to in the manuscript. Again, the doses of AA are not specified. VHL KD is very weak. 4h – needs quantification.

Lanes 328-332 – it is not specified in which group what pathways are changed.

The section “ Differential analysis between sunitinib... proteome and phosphoproteome levels” (lane 341-342) is written in a very confusing manner. It starts with phosphoproteome, then moves to the GSEA analysis of the transcriptome, and then to proteome and phosphoproteome again. If the main conclusion is that sunitinib resistance is mediated by augmented mTOR signaling, the data should be presented to support that main conclusion.

I am not sure where proper transcriptome analysis is presented.

It is not clear why only proteome classifier was determined and not classifiers that would integrate multiple omics.

The reviewer comments and our responses are marked in blue font and black font, respectively.

REVIEWER COMMENTS

Reviewer #1 (Remarks to the Author):

The authors have done a significant amount of work and analyses to address my previous points. The manuscript has evolved, but some of the experiments and data raise new questions. Specific comments:

Q1. The authors still postulate that VHL loss causes glycolysis addiction. While the new data show increased lactate levels following VHL knock-down in ACHN cells, that does not show that the cells were not addicted to glycolysis before VHL inhibition nor that they have become addicted to glycolysis afterwards. Enhanced glycolysis is not the same as glycolysis addiction. Moreover, it is not clear how the acute effects of VHL inhibition in ACHN cells are related to the effects of VHL inactivation in cancer clones that arise from VHL mutant clones. That VHL has been knocked down is clear, but is this level of knockdown sufficient to induce the expression of hypoxia-inducible factors? It could be more straightforward to not to include these VHL knockdown data in the manuscript, the strengths of which are not in functional work but in an in-depth analysis of clinical samples.

Response: Thank you for your professional suggestions. We will response this comments from the following two aspects: in-depth proteomic analysis and *VHL* loss-of-function mutations data.

About the in-depth proteomic analysis for enhanced glycolysis of *VHL* mutated tumors

Firstly, VHL is a component of E3 ligase mediating HIF degradation. It is well established that inhibition of VHL led to increased HIF expression and glycolysis (*J Clin Oncol*, 2014, PMID: 24821879; *Nat Rev Nephrol*, 2020, PMID: 32561872). Proteomic data revealed that patients with *VHL* mutation showed higher expressions of glycolytic enzymes, such as CA9, PGK1, ENO1 and LDHA, indicating the enhanced glycolysis of *VHL* mutated tumors (**Figure RL 1A**). Secondly, in

the clinic samples, we found *VHL* mutation tumors exhibited accumulated HIF and upregulation of glycolysis (**Figure RL 1B**).

Figure RL 1 A The heatmap showing the upregulated glycolytic enzymes in the *VHL* mutant tumors, compared with *VHL* wildtype tumors. **B** Differentially expressed proteins in the *VHL* mutation and WT groups and their associated biological pathways. **Please also see in the revised Fig. 4d-e.**

About the *VHL* loss-of-function mutations assay

To mimic *VHL* loss-of-function mutations, we knocked down *VHL* in ACHN cells (**Figure RL 2A**), which belong to *VHL*-proficient cells, and tested the glycolysis levels by monitoring the lactate production. In the cultured cells, when we knockdown *VHL* (80% knockdown efficiency) to mimic *VHL* loss-of-function in tumors, we also found increased lactate levels, which was a marker for glycolysis (**Figure RL 2B**). More importantly, we found the knockdown of *VHL* distinctly enhanced the inhibition the invasiveness of Sunitinib to cancer cells through the transwell assay (**Figure RL 2C**). In addition to the transwell assay, in this round of revision, we also verified the effect of *VHL* mutation in Sunitinib treatment with cell proliferation rate through CCK-8 assay. The results showed that knockdown of *VHL* significantly inhibited the cell proliferation in the Sunitinib treatment (**Figure RL 2D**). Overall, all these results suggested that the *VHL* inhibition potentially increased the vulnerability to Sunitinib treatment in ccRCC.

Figure RL 2 **A** Transfection efficiency of VHL siRNA detected by qRT-PCR and western blot. **B** The lactate's concentration in cells after VHL knockdown. **C** Transwell detected the effect of VHL knock down and Sunitinib treatment on cell invasiveness. **D** CCK-8 detected the effect of *VHL* knockdown and Sunitinib treatment on cell proliferation. **Please also see in the revised Fig. 4g-h and Supplementary Fig. 7d-e.**

The reviewer is correct. Both the proteomic bioinformatic analysis and *VHL* loss-of-function mutations assay indicated that *VHL* mutation were associated with glycolysis, however, enhanced glycolysis is not the same as glycolysis addiction. According to the reviewer's suggestions, we rephrased glycolysis addiction as enhanced glycolysis flux in the revised manuscript (**lines 35, pages 2; lines 270, pages 9; lines 428, pages 14; lines 486, pages 16; lines 672, pages 21**). We thank the reviewer again for making our manuscript more rigorous.

Q2. In figure 3e, why are only three genes marked as cis-regulated? The text was not clear on why these three genes were picked for further analysis. Also, even though LAMTOR4 was classified as an mTOR pathway gene, all LAMTOR4, MDH2 and CALU had the same effect on pS6K level. Was this result expected? Did the authors only try these three genes, or did some genes have different effects?

Response: We thank the reviewer for the comments.

About the three genes cis-regulated by 7q in out cohort

In our tyrosine kinase inhibitor therapy cohort, there are only three *cis*-regulated genes on 7q chromosome. In detail, to explore the CNA landscape at gene level between the Responder and Non-Responder groups, we compared the amplified genes between the two groups and found genes that more frequently amplified in Non-Responder group were located in 7q (Fisher's exact test, $p < 0.05$) (**Figure RL3A**). To find out the effects of 7q copy number (CN) on sunitinib therapy at proteome level, we calculated the spearman's correlation of 7q CN and proteome data. As a result, there were 485 proteins showed significantly positive correlations with 7q CN ($p < 0.05$). These proteins were enriched in pathways including lysosome (LARP1, LARP2), innate immune system (PTGES), mTOR signaling (RRAGB, RRAGD, LAMTOR2, LAMTOR4, MLST8), TCA Cycle (CS, SDHA, MDH2), oxidative phosphorylation (COX6C, NDUFS6), and fatty acid biosynthesis (ACACA, MCAT).

Notably, there were 29 out of 485 proteins encoded by 7q genes. Genomic alterations that affect gene expression levels at the same locus are defined to act in *cis*. We observed three (LAMTOR4, MDH2, CALU) out of 29 proteins showed *cis* effects with their encoding genes on 7q, in which LAMTOR4, participating mTOR signaling, showed the most significant *cis* effects ($p = 1.88E-4$) (**Figure RL 3B**). We have updated the description in the revised manuscript as follows: "Comparing the CNA landscape at the gene level between the Responder and Non-Responder groups, we found that genes, more frequently amplified in the Non-Responder group than in the Responder group, were located in 7q (Fisher's exact test, $p < 0.05$) (**Fig. 3d**). To find out the effects of 7q copy number (CN) on sunitinib therapy at proteome level, we calculated the spearman's correlation of 7q CN and proteome data. As shown in **Fig. 3e**, there were 485 proteins showed significantly positive correlations ($p < 0.05$). These proteins were enriched in pathways including lysosome (LARP1, LARP2), innate immune system (PTGES), mTOR signaling (RRAGB, RRAGD, LAMTOR2, LAMTOR4, MLST8), TCA Cycle (CS, SDHA, MDH2), oxidative phosphorylation (COX6C, NDUFS6), and fatty acid biosynthesis (ACACA, MCAT). Genomic alterations that affect gene expression levels at the same locus are defined to act in *cis*, whereas the impacts of another locus are defined as a *trans* effect. The diagonal patterns in **Supplementary Fig. 6** represent the *cis* effects of CNAs, whereas vertical patterns indicate the *trans* effects. We found three gene LAMTOR4, MDH2, and CALU on 7q showed *cis* effect among the 485 genes".

For the connection between LAMTOR4, MDH2, CALU and mTOR signaling

In our validation experiments, we found overexpressed LAMTOR4, MDH2, and CALU increased the phosphorylation of S6K, indicating activation of mTORC1 (*Biochem J.*, 2012, PMID: 22168436). Specifically, LAMTOR4, as a part of the Ragulator complex, is directly involved in amino acid sensing and activation of mTORC1. MDH2, utilizing the NAD/NADH cofactor system in the citric acid cycle, participate in mitochondrial metabolism. The mitochondrial metabolism is closely related to the mTOR pathway (*Nature*, 2007, PMID: 18046414; *Nature*, 2007, PMID: 18046414; *J Aging Res*, 2011, PMID: 21629705). In our study, we found MDH2 *cis*-regulated by 7q gain. To further verify the impacts of chromosome of 7q gain, we found overexpressed MDH2 increased the phosphorylation of S6K, indicating activation of mTORC1. Based on the published studies and the experiment results, we hypothesize MDH2 might participate in mTOR signaling activation by mitochondrial metabolism. The product of CALU is a calcium-binding protein localized in the endoplasmic reticulum (ER) and it is involved in such ER functions as protein folding and sorting. It is rarely reported that CALU is directly connected to mTOR signaling in the previously published study. However, the direct relationship between these proteins have not been extensively reported. The specific mechanism needs further research in the future. We discussed the association between these proteins and mTOR signaling in the revision and added this part in the **Discussion** in the revised manuscript.

As for the selection of three 7q *cis* genes for validation and two random selected genes for the control

To further validate the impacts of chromosome 7q gain on Sunitinib treatment, we overexpressed five *cis*-regulated genes including LAMTOR4, MDH2 and CALU (encoded by 7q) and two other random genes as control (TBL2 and POR2, not encoded by 7q) to mimic the effects of 7q gain (**Figure RL3 C-I**) in cell line 786-O and 769-P, separately. The results showed that the overexpression of LAMTOR4, MDH2 and CALU, rather than TBL2 and POR2, increased the phosphorylation of S6K (**Figure RL 3C**), indicating activation of mTORC1 (*Biochem J.*, 2012, PMID: 22168436). Next, we treated the candidate genes overexpressed cells and normal cells with or without Sunitinib. By monitoring the invasion with transwell assay, we found increased

expression of LAMTOR4, MDH2 or CALU abrogated the effects of Sunitinib, while TBL2 and POR2 not (**Figure RL 3D, E, G, H**). In the revision, in addition to the transwell assay, we performed cell proliferation assay to directly evaluate the effects of chromosome 7q gain on Sunitinib therapy (**Figure RL 3F, I**). In conclusion, chromosome 7q gain would activate mTOR signaling and lead to poor Sunitinib treatment effectiveness. The related figures were updated in the **Fig. 3f, 3g, 3h** and **Supplementary Fig. 6c, 6e, 6f, 6g** in the revision.

Figure RL3 A Comparison of gene-level CNAs between Responders and Non-Responder in this cohort. The upper plot illustrates the frequency of CNA events, the lower plot illustrates the $-\log_{10}$ (p value) of each gene for the comparison of Responders and Non-Responder (two-sided Fisher's exact test). **B** The plot depicting the Spearman correlations between protein expression and 7q gain. **C** Effects of overexpression of LAMTOR4, MDH2, CALU, TBL2 and POR on phosphorylation of S6K. **D** Transwell detected the effect of LAMTOR4, MDH2, CALU, TBL2 and POR overexpression and Sunitinib treatment on cell invasiveness for 786-O cell line. **E**. Quantification of transwell results for 786-O cell line. **F** CCK-8 detected the effect of LAMTOR4, MDH2, CALU, TBL2 and POR overexpression and Sunitinib treatment on cell proliferation for 786-O cell line. **G** Same layout as Figure RL 3D, but for 769-P cell line. **H** Same layout as Figure RL 3E, but for 769-P cell line. **I** Same layout as Figure RL 3F, but for 769-P cell line. **Please also see in the revised Fig. 3f, h and Supplementary Fig. 6c, 6e-g.**

Q3. The idea of using a trans-well migration assay to address a question about a drug being curative (my original comment) or having anti-tumor effects more generally is poorly justified. While the migratory phenotype might have something to do with drug resistance, it is not clear to the reader whether this is the case here. This comment relates to the data shown in 2f, 3g, 4h and 6m. The images are hard to interpret, and no quantification is shown. But the biggest issue is the underlying

logic. Why use this assay in this context?

Response: Thank you for your professional suggestions. The reviewer is correct that the transwell migration assay could not directly reflect anti-tumor effects. In order to more intuitively reflect the anti-tumor effects, we have verified each corresponding experiment 2f, 3g, 4h and 6m with cell proliferation rate through CCK-8 assay in this round of revision (**Figure RL 4**). The quantitative results of 2f, 3g, 4h and 6m transwell assay were also provided in this revision. All these results were added in the revised **Fig. 2e, 2f, 2g, 2h, 2j, 2m, 3f, 3h, 4h, 6m, Supplementary Fig. 5e, 6c, 6e, 6f, 6g, 7e, 9i** and the source data of transwell quantitative results were also provided in the **Source Data** part.

Specifically, **as for the Fig. 2f**, to prove that AA improved the prognosis of ccRCC patients under Sunitinib treatment, we performed the cell proliferation assay *in vitro* and found that AA treatment did not impact cell proliferation directly, but enhanced the inhibition to cell proliferation of Sunitinib (**Figure RL 2f**). The transwell assay, which may reflect the degree of malignancy and treatment effect of tumor to a certain extent, showed that AA treatment did not impact cell invasiveness directly, but enhanced the inhibition of Sunitinib to cell invasiveness. For the sake of reliability, we performed the cell proliferation assay, and transwell assay in both ACHN and 786-O cell lines in the revision. Please also see in the revised **Fig. 2e-f** and **Supplementary Fig. 5e**.

As for the Fig. 3g, to investigate the impacts of 7q gain on Sunitinib therapeutic outcomes, we overexpressed LAMTOR4, MDH2 and CALU, *cis*-regulated by 7q gain, to mimic chromosome 7q gain in ACHN and 786-O cell lines. We found overexpressed LAMTOR4, MDH2 and CALU increased the phosphorylation of S6K, indicating activation of mTORC1 (*Biochem J.*, 2012, PMID: 22168436). Next, we treated the candidate genes overexpressed cells and normal cells with or without Sunitinib. By monitoring the cell proliferation rates, we found increased expression of LAMTOR4, MDH2 or CALU abrogated the effects of Sunitinib. In conclusion, chromosome 7q

gain would activate mTOR signaling and lead to poor Sunitinib treatment effectiveness. Please also see in the revised **Fig. 3h** and **Supplementary Fig. 6e-g**.

As for the **Fig. 4h**, proteomic data revealed that patients with *VHL* mutation showed higher expression of glycolytic enzymes, such as CA9, PGK1, ENO1 and LDHA, indicating the enhanced glycolysis of *VHL* mutated tumors. In the clinic samples, we found *VHL* mutation tumors exhibited accumulated HIF and upregulation of glycolysis. To mimic *VHL* loss-of-function mutations, we

knocked down *VHL* in ACHN cells and found increased lactate levels, which is a marker for glycolysis. More importantly, we verified the effect of *VHL* mutation in Sunitinib treatment with cell proliferation rate through CCK-8 assay. The results showed that knockdown of *VHL* significantly inhibited the cell proliferation in the Sunitinib treatment. All these results suggested that the *VHL* inhibition potentially increased the vulnerability to Sunitinib treatment in ccRCC. Please also see in the revised **Fig. 4h** and **Supplementary Fig. 7e**.

4h

As for the **Fig. 6m**, TGFB1 was elevated in the Progenitor-cell infiltrated cluster and was well correlated with the Sunitinib response. We observed that TGFB1 was co-expressed with proteins involved in angiogenesis and tumor immune escape in ccRCC, indicating that the alternative angiogenesis driven by TGFB1 resulted in Sunitinib resistance. To verify the association between TGFB1 and Sunitinib resistance, we added TGFB1 in the culture medium to simulate the impact of tumor microenvironment-derived TGFB1 on cancer cells. We tested the cell proliferation rate through CCK-8 assay. The results showed that TGFB1 could enhance the invasiveness and proliferation of ccRCC cell and abrogate the impact of Sunitinib on ccRCC cells. In summary, we considered that the tumor microenvironment, comprising abundant progenitors, led to insufficient Sunitinib therapy response by TGFB1 signaling. Please also see in the revised **Fig. 6m-n** and **Supplementary Fig. 9i**.

6m

Figure RL 4 2f (Left) Transwell detected the effect of AA and Sunitinib treatment on invasiveness (AA-100 μ M, Sunitinib-200nM) (Middle), Quantification of transwell results (Right), the proliferation of cells was detected by CCK-8 assay (AA-100 μ M, Sunitinib-200nM). **3g** Transwell detected the effect of LAMTOR4, MDH2, CALU, TBL2 and POR overexpression and Sunitinib treatment on cell invasiveness. **4h** (Left) CCK-8 detected the effect of *VHL* knockdown and Sunitinib treatment on cell proliferation. (Middle) quantification of transwell results, (Right) CCK-8 detected the effect of *VHL* knockdown and Sunitinib treatment on cell proliferation. **6m** (Left) Transwell detected the effect of TGFBI1 intervention and Sunitinib treatment on cell invasiveness, (Middle) quantification of transwell results, (Right) CCK-8 detected the effect of TGFBI1 intervention and Sunitinib treatment on cell proliferation. **Please also see in the revised Fig. 2e-f, 3h, 4h, 6m-n, and Supplementary Fig. 5e, 6e-g, 7e, 9i.**

Q4. The original version of this manuscript linked AA exposure to higher mutation burden, and the authors also identified some pathway-level characteristics of the AA-exposed tumors. This seemed meaningful and the role of AA as a mutagen has been reported elsewhere too. The current version has expanded this by suggesting that AA exposure would directly affect cell signaling. While this is likely to be the case, as would with any chemical, it is less clear how the increased level of mutations is linked to the acute signaling events and possible alterations in drug responsiveness. How close are the AA concentrations used in the experimental assays to those seen in patients with dietary AA exposure? The thinking and model related to AA should be clarified.

Response: We thank the reviewer for the professional comments. To clearly response the question, we split it into two parts.

For the AA signature and its linkage to mutations and cell signaling in drug responsiveness

AA exposure may cause mutagenesis characteristic of predominant T>A transversions, which match the Catalogue of Somatic Mutations in Cancer (COSMIC) SBS22 (*Nature*, 2013, PMID: 23945592; *Nat Prod Rep*, 2014, PMID: 24691743). Several studies have demonstrated the potential association between AA exposure and ccRCC oncogenesis (*Br J Cancer*, 2014, PMID: 26657656, *Nature communications*, 2022, PMID: 35440542). In this study, we decomposed the mutation

spectra of ccRCC tumor samples using COSMIC database (*Nucleic Acids Res*, 2019, PMID: 30371878) and found that there were 19 patients carried the SBS22 signature.

Furthermore, these patients showed smaller tumor size and better survival in Sunitinib treatment (**Figure RL 5A**), suggesting the specific mechanism of underlying AA exposure. Therefore, we assessed the proteomic impact of AA signature, the results showed that proteins involved in the pentose phosphate pathway (PPP), immune system, adhesion-PI3K-AKT-mTOR signaling, and DNA replication were significantly downregulated (**Figure RL 5B**). We found that the attenuated pentose phosphate pathway (PPP) and enhanced glycolysis in tumors with AA signature, manifested as upregulation of PFKFB3, PFKFB1, GPI (t-test, $p < 0.05$) and downregulation of G6PD, PGD, TKT (t-test, $p < 0.05$) (**Figure RL 5C**). Thus, we supposed that AA improved the prognosis of ccRCC patients under Sunitinib treatment by downregulation PPP. These findings allowed us to design the *in vitro* experiments to verify the AA effects on Sunitinib treatment (**Figure RL 5D**). We found that AA treatment did not impact cell proliferation directly, but enhanced the inhibition to cell proliferation of Sunitinib, the similar results were also found in transwell assay.

It is well-established that the AA exposure impact the mutagenesis characteristic of predominant T>A transversions, but rare study focused on the impact of cell signaling. Here, based on the multi-omics data and bioinformatical analysis, in addition to the genomic impact of AA, we described the proteomic differences between AA patients and non-AA patients, and observed PPP and glycolysis alteration which connected to the Sunitinib treatment. These results indicated AA not only affect the mutation at genomic level, but also alter the cell signaling at proteomic level. Furthermore, it is still uncertain that the connection between two events at genomic and proteomic level. Hence, the intrinsic linkage between AA exposure, mutagenesis characteristic, and signaling need to be further investigated.

Figure RL 5 **A** Comparisons of tumor size between patients with or without the AA signature (two-sided t test). Data are shown as mean \pm SD. **B** Pathways enriched by differentially expressed proteins between patients with or without the AA signature. **C** Glucose metabolism alteration caused by the AA signature. **D** CCK-8 detected the effect of AA on Sunitinib treatment on cell proliferation. Please also see in the revised Fig. 2b-e.

For the AA concentrations used in the experimental assays

The AA concentration in human blood after taking Chinese herbal medicine has not been reported in literature. To determine the optimal concentration of AA *in vitro* experiments, based on the product manual, we did the gradient experiments. In detail, patients with AA signature showed smaller tumor size and better survival in Sunitinib treatment. By the functional enrichment analysis, we found that the attenuated pentose phosphate pathway (PPP) and enhanced glycolysis in tumors with AA signature, manifested as downregulation of G6PD, PGD, TKT ($p < 0.05$) (key enzymes of PPP). To verify this finding, *in vitro* experiments were performed with renal cell lines, to evaluate the expression level of G6PD, PDG, and TKT undergoing a gradient of AA concentration (including 0 μ M, 5 μ M, 25 μ M, and 100 μ M). As shown in **Figure RL 6**, the gradually increased AA concentration decreased the expression level of the three proteins in both 786-O and ACHN cell lines. When the concentration of AA was set as 100 μ M, we observed the dramatic inhibition effect of the protein expression. Once the concentration exceeds 100 μ M, it will adversely impact the cell state. Hence, the concentration of AA in our experiment was set as 100 μ M and this AA concentration is consistent with the previously published study about the AA on renal epithelial

cells (*Kidney International*, 2005, PMID: 15840026). In the revision, we updated the description about the AA concentration used in our *in vitro* experiments in the **Methods** part.

Figure RL 6 AA treatment inhibited the expressions of G6PD, PGD and TKT in 786-O and ACHN cells in dose-dependent manners. **Please also see in the revised Fig. 2g-h.**

Minor:

Q5. Fig. 1B still doesn't have units for the tumor size, PFS or age.

Response: We apologize for the negligence of submitting the not-updated version. We confirmed that **Fig. 1b** was updated in this revision (**Figure RL 7**).

Figure RL 7 The summary of the clinical parameters in this cohort. This cohort includes 27 Responders and 88 Non-Responders undergoing Sunitinib treatment. Their clinical parameters are shown in the heatmap. **Please also see in the revised Fig. 1b.**

Reviewer #3 (Remarks to the Author):

The authors can be congratulated to their excellent study with further improvements after the review process. My comments have been fully addressed in a satisfactory fashion and the paper merits its

publication.

Response: Thank you for your positive comments and suggestions, which help to improve the quality of this manuscript.

Just some very minor comments:

Q6. Line 138: "Other clinical information, including...": Hypertension and diabetes could be added.

Response: Thanks for the comment. According to reviewer's suggestion, we revised this sentence as "Other clinical information, including gender, age, chronic disease status such as hypertension, diabetes, Karnofsky performance score (KPS, a clinical assessment tool used to assess the overall health of patients, **Methods**), neutrophilic granulocyte (GRAN) count, lymphocyte (LYM) count, eosinophil (ESO) count, platelet (PLT) level, haemoglobin (Hb) level, serum lactate dehydrogenase (LDH) level, serum calcium (Ca) level, serum creatinine (CRE) level, blood urea nitrogen (BUN) level, urine protein state, estimated glomerular filtration rate (eGFR) (**Methods**) were also collected (**Supplementary Fig. 2**)."

Q7. Line 142: "... level, urine protein state, estimated glomerular filtration rate (eGFR)": Formula used to calculate eGFR should be stated (e.g. CKD-EPI).

Response: Thank the reviewer for the comment. We used the CKD-EPI equation (*Ann Intern Med*, **2009**, PMID: 19414839) to estimate the eGFR and the equation is listed as below: $eGFR = 141 \times \min(Cr/\kappa, 1)^\alpha \times (\max(Cr/\kappa, 1)^{-1.209}) \times (0.993^{Age}) \times 1.018$ [if female] $\times 1.159$ [if black]. Cr is serum creatinine in mg/dL, κ is 0.7 for females and 0.9 for males, α is -0.329 for females and -0.411 for males, min indicates the minimum of Cr/ κ or 1, and max indicates the maximum of Cr/ κ or 1. In the revision, we added the description about the Formula used to calculate eGFR in the **Methods** part in lines 1191, pages 38. Thank you again for your professional suggestion.

Q8. Could not find the Supplementary Table 1 among the uploaded files.

Response: Thanks for the comment. Now, all the Supplementary files were submitted in the revision.

Q9. Obtained findings on miRNA could be briefly mentioned as data not shown in the paper also.

Response: Thanks for reviewer's comments. According to the reviewer's suggestion, we added the

finding of miRNA in the **Discussion** part in **lines 787, pages 25** in the revised manuscript as follows.

“Functional studies have confirmed that miRNA dysregulation is causal in many cases of cancer (*Nat Rev*, 2017, PMID: 28209991). Insights into the roles of miRNAs in development and disease, particularly in cancer, have made miRNAs attractive tools and targets for novel therapeutic approaches. In our study, we analyzed the miRNAs which might regulate the proteins associated with the response to Sunitinib. Based on the transcriptome sequencing data in our cohort, we identified 348 miRNAs in this cohort, among which 4 miRNA showed significant differences between Responders and Non-Responders. Specifically, MIR3939, MIR4635, and MIR578 showed higher expression levels in Responders, while MIR27A was higher in Non-Responders. This result suggested that miRNA treatment might contribute to increase the response to Sunitinib for ccRCC patients.”

Reviewer #4 (Remarks to the Author):

Qu et al. have substantially enlarged the datasets in their study (e.g., phosphoproteomics, RNA-seq, more experiments). In general, this has vastly improved the rationale of their argumentation. The conclusions seem to be mostly backed up with data. However, I have some concerns with the rebuttal letter and the data that is presented there (some of this is also part of the main figures in the manuscript).

Q10. Reviewer 1 indicated that it should not be possible to know whether the markers identified would be associated with therapy. In response, the authors state that they recruited 37 cases of ccRCC patients, who did not receive any treatment after surgery and this cohort would support their claims. While it seems ideal to have such a control-cohort, I am unsure if it was ethical to not provide patients/human beings with any treatment other than surgery in the conditions of advanced and metastasized cancer. The authors should comment on this.

Response: Thank the reviewer for the comments. This study is retrospective. We objectively recorded the treatment status of the patients during the follow-up. It was noted that, the price of Sunitinib was about 50,000 CNY (~7,000 US dollar) ten years ago, which brought unaffordable economic pressure for a Chinese family. Some patients refused to receive treatment due to economic

reasons. All patients in the non-treatment group voluntarily gave up treatment, which were recorded literally in the electronic medical records. All procedures were compliant with ethical standards. With the development of economy and the declined price of Sunitinib (covered by health insurance in China), this condition almost disappeared.

In the following, I focus on the points that I had in my initial review and on the specific responses having been provided.

Q11-1. Q22: The authors knocked down the VHL gene and performed trans-well assays to test whether this knockdown would impinge on the migration capacity of cells. The achieved knockdown efficiency of 50% is not convincing. The authors should use some other means of knocking down/out VHL and seek to achieve an efficiency of at least 70-80%. Use of two or three independent si/shRNAs should be performed to rule out off-target effects.

Response: Thank you for your professional suggestions. In order to achieve higher knockout efficiency, we simultaneously used three siRNAs (5'- GCUCUACGAAGAUCUGGAATT -3'; 5'- GGCUCAACUUCGACGGCGA-3'; 5'- CUGCCAGUGUAUACUCUGA -3') to exclude off-target effects. Finally, we were surprised to find that the inhibitory effect reached nearly 80% (**Figure RL 8 A-B**). Besides, in order to make the experimental results more observable, we replaced the phenotype with the cell proliferation rate and tested it through CCK-8 assay in this round of revision (**Figure RL 8 D**, also see in the **Fig. 4h**). In addition, the quantitative results of transwell were also provided in this revision (**Figure RL 8 C**). All the results further verified that inhibiting VHL could promote the efficacy of sunitinib. Thank the reviewer again for the comments.

Figure RL 8 **A** Transfection efficiency of VHL siRNA detected by qRT-PCR and western blot. **B** The lactate concentration in cells after VHL knockdown. **C** Transwell detected the effect of VHL knock down and Sunitinib treatment on cell invasiveness. **D** CCK-8 detected the effect of *VHL* knockdown and Sunitinib treatment on cell proliferation. **Please also see in the revised Fig. 4g-h and Supplementary Fig. 7d-e.**

Q11-2. In panel D of Figure RL17/Supplementary Figure 5, the authors present the outcome of this experiment. In the photograph, they see a reduction in the number of cells particularly in the VHL-knockdown cells having been treated with sunitinib. The data should be quantified and presented together with the photograph in the Supplementary Fig.. Quantification of image data should be done also for other figures, like effects of overexpressing LAMTOR4, MDH2 and CALU, presented in Figure 3.

Response: Thanks for the professional suggestions. We have provided the transwell quantification results in the revision (**Figure RL 9 A-E**, also see in the revised **Supplementary Fig. 5e, 6c, 6e, 6f, 6g, 7e, 9i**). The source data of transwell quantitative results were also provided in the **Source Data** part in the revision.

In order to more intuitively reflect the anti-tumor effects, we have verified each corresponding experiment with cell proliferation rate through CCK-8 assay in this round of revision.

For the knockdown *VHL* in ACHN cells in Figure RL17/Supplementary Figure 5

In the revision, we added the quantification results of transwell and also performed cell proliferation experiments. Specifically, to mimic *VHL* loss-of-function mutations, we knocked down *VHL* in ACHN cells, which belong to *VHL*-proficient cells. we found the knockdown of *VHL* distinctly enhanced the inhibition the invasiveness of Sunitinib to cancer cells through the transwell assay (**Figure RL 9A**). In addition to the transwell assay, in this round of revision, we also preformed cell proliferation assay to verify the effect of *VHL* mutation in Sunitinib treatment (**Figure RL 9B**). The results showed that knockdown of *VHL* significantly inhibited the cancer cell proliferation in the Sunitinib treatment. **Please also see in the revised Fig. 4h and Supplementary Fig 7e.**

For the quantification of overexpression of LAMTOR4, MDH2, CALU

According to the reviewer's suggestion, we added the quantification results of the transwell assay for the effects of overexpressing LAMTOR4, MDH2, and CALU in the revision. Specifically, to further validate the impacts of chromosome 7q gain on Sunitinib treatment, we overexpressed five cis-regulated genes including LAMTOR4, MDH2 and CALU (encoded by 7q) and two other random genes as control (TBL2 and POR2, not encoded by 7q) to mimic the effects of 7q gain in cell line 786-O and 769-P, separately. Next, we treated the candidate genes overexpressed cells and normal cells with or without Sunitinib. By monitoring the invasion with transwell assay, we found increased expression of LAMTOR4, MDH2 or CALU abrogated the effects of Sunitinib, while TBL2 and POR2 not (**Figure RL 9C-D, 9F-G**). In addition, to directly evaluate the effect of 7q gain on Sunitinib therapy, we also performed CCK-8 cell proliferation assay in the revision (**Figure RL 9E, H**). In conclusion, chromosome 7q gain would lead to poor Sunitinib treatment effectiveness. Please also see in the revised **Fig 3h** and **Supplementary Fig. 6e-g**.

Figure RL 9 **A** Transwell detected the effect of *VHL* knock down and Sunitinib treatment on cell invasiveness. **B** CCK-8 detected the effect of *VHL* knockdown and Sunitinib treatment on cell proliferation. **C** Transwell detected the effect of LAMTOR4, MDH2, CALU, TBL2 and POR overexpression and Sunitinib treatment on cell invasiveness for 786-O cell line. **D** Quantification of transwell results for 786-O cell line. **E** CCK-8 detected the effect of LAMTOR4, MDH2, CALU, TBL2 and POR overexpression and Sunitinib treatment on cell proliferation for 786-O cell line. **F** Transwell detected the effect of LAMTOR4, MDH2, CALU, TBL2 and POR overexpression and Sunitinib treatment on cell invasiveness for 769-P cell line. **G** Quantification of transwell results for

769-P cell line. **H** CCK-8 detected the effect of LAMTOR4, MDH2, CALU, TBL2 and POR overexpression and Sunitinib treatment on cell proliferation for 769-P cell line. **Please also see in the revised Fig. 3h, 4h and Supplementary Fig 6e-g, 7e.**

Q12. Q23: The authors did a series of experiments to verify a link between VHL deficiency and metabolic phenotypes in the context of sunitinib-treatment. In panel D of Figure RL20, the authors show that the mutation status in VHL correlates with response to sunitinib in patients of the Immotion151 study. There seem to be two groups that are most different between the two cohorts: patients with partial response (PR) and patients not evaluated (NE). Were patients who could not be evaluated included in the calculation of statistical difference between the cohorts (Fisher's exact test)? If so, this group should likely be removed prior to statistical testing.

Response: We apologize for the confusion. Indeed, we compared the Responder (CR+PR) and Non-Responder (SD+PD) between patients with and without *VHL* mutation in the Immotion151 (**Figure RL 10**). The patients which were not evaluated (NE) were not included in the statistical testing. As the results shown, Responders (CR+PR) were significantly overrepresented in patients with *VHL* mutations (Fisher's exact test, $p = 0.013$) (**Figure RL 10**). In the revision, we have updated this information in the **Supplementary Fig. 7a** to avoid any confusion in the revision. Thank the reviewer again for pointing it out.

Figure RL 10 Stacked graph bars represent the number of responses for patients treated with Sunitinib in Immotion151 study (Fisher's exact test). R, Responder; NR, Non-Responder. **Please also see in the revised Supplementary Fig 7a.**

Q13. Q24: The authors describe very well the experimental details in the sequencing experiment. However, the numbers of reads per sample, the numbers of mapped reads, and the coverages of exomes should be provided as well. In the revised manuscript the author states that :’The WES target region was 33 M.’ Do the authors mean 33 Mbp? The xGen 1.0 panel which was applied for exome enrichment had been designed in 2015 to span 39 Mb of the human genome and to provide coverage of 19,396 genes (https://genome.ucsc.edu/cgi-bin/hgTables?db=hg38&hgta_group=map&hgta_track=exomeProbesets&hgta_table=xGen_Research_Targets_V2&hgta_doSchema=describe+table+schema, accessed 22/06/03). Why are the numbers not consistent?

Response: Thank the reviewer for the professional comment. We apologize for not providing enough complete sequencing information. The number of reads among samples ranged from 30.51M to 93.43M, with the average value was 43.85M. The number of mapped reads among samples ranged from 23.34M to 86.19M, with the average value was 38.28M and the average mapping rate was 86.52%. The numbers of reads per sample, the numbers of mapped reads, and the coverages of exomes were all provided in the revised **Supplementary Data 2**. The reviewer is correct that the targeted region of the xGen 1.0 panel was 39 Mb. We apologize for the mistake. We have revised it in the **Methods** part in the revision.

Q14. Q27/Figure RL25: In the original manuscript, it was not clear how many proteins were recurrently identified in the 115 samples (88 responders and 27 non-responders). If I get it right from panel B in Figure RL25, the number of proteins that were identified in more than 1 sample was 52 specifically identified in responders, and less than 10 proteins were identified in more than 2 different samples. The numbers seem to be better for proteins that were specifically identified in non-responders. Do the authors have some idea why the re-identification of proteins was so poor in the responders?

Response: Thanks for the comments. We apologize for the unclear description. Actually, there were 27 responder (R) samples and 88 non-responder (NR) samples in our cohort. According to the reviewer’s suggestion, we added the number of identified samples of each protein and updated the **Supplementary Data 4**.

About the number of the re-identification of proteins in the R/NR group

In the last round of revision, to response the reviewer's comments that how many of the exclusively identified proteins in R/NR group were recurrently identified, we added the number of identified samples of each protein in the **Supplementary Data 4**. There were 442 proteins identified exclusively in R group, and among the 442 proteins, there were 53 proteins were recurrently identified. There were 2,226 proteins identified exclusively in NR group, and among the 2,226 proteins, there were 903 proteins were recurrently identified (**Figure RL 11 A**). Compared with the NR group, there were fewer samples in R group, which might result in the fewer exclusively detected proteins and re-identification of proteins in responder group.

Figure RL 11 A Barplots showing the numbers of recurrently identified proteins in Responders and Non-Responders.

About the clarification of our data used for the analysis

In our cohort, the data used for downstream analysis were carefully screened. Firstly, peptide identification stringency was set at a maximum 1% FDR at peptide level. Then, proteins with 1% FDR were select for further analysis. The same cutoff strategies of FDR at protein/peptide level have been widely used in recently published researches (*Cell*, 2021, PMID: 34534465; *Cancer cell*, 2021, PMID: 33577785). To improve the reliability of our analysis results, proteins with low identification frequency were excluded from the analysis. The analysis in our cohort focused on the proteins identified in > 25% of the samples (exclusive proteins in R and NR were excluded in our analysis) which had been applied in the previous published studies (*Cell*, 2020, PMID:33212010, *Cell*, 2021, PMID: 34358469; *Nat Commun*, 2018, PMID: 29739932). In the revision, we updated

the description about the data used for the analysis in in the **Methods** part. Thank the reviewer again for the comments.

Q15. Q27-3: The authors write that they used the match-between-runs algorithm. In the manuscript, they cite Tyanova et al., 2016 (reference 52). In the rebuttal letter, they cite the original article for this algorithm (Cox et al., 2014). This should be corrected in the manuscript, even if both manuscript come from the Mann-lab. The match-between-runs algorithm was originally implemented in the MaxQuant software. Did the authors use this software or, if not, how did they implement the algorithm?

Response: Thanks for the reviewer's comments. To clearly response this question, we split it into two parts.

For the correctness of the match-between-runs (MBR) algorithm citation

We apologize for the misleading in the previous manuscript and thank the reviewer for pointing out. In fact, the match-between-runs algorithm was described in the Cox et al., (*Mol Cell Proteomics*, **2014**, **PMID: 24942700**). We have corrected it in the revised manuscript.

For the implement of MBR algorithm

We did not use the MaxQuant software, instead we used the mascot engine to search the raw data. We implemented the MBR algorithm using Python in Firmiana (*Nat Biotechnol*, **2017**, **PMID: 28486446**). The idea and parameter setting of MBR algorithm were same as Tyanova et al (*Nat. Protoc*, **2016**, **PMID: 27809316**) in principle. In detail, we built a dynamic regression function based on those common identified peptides in grouping experiments. According to correlation value R^2 , Firmiana will choose linear or quadratic function for regression to calculate RT of corresponding hidden peptides, and check the existence of the XIC based on the m/z and calculated RT. Finally, the program could evaluate the peak area values of those existed XICs. These peak area values should be considered as parts of corresponding proteins. The scripts were uploaded to GitHub: <https://github.com/FirmianaPlatform/SourceCode/blob/master/Firmiana%20Frontend/gardener/iews.py> (line 3948- line 4588). The same MBR strategy was also implemented in the published study

(*Nature communications*, 2022, PMID: 35440542; *Nature communications*, 2022, PMID: 36175412;).

Q16. Below, the authors write that they did not apply any other data imputation algorithm. However, this concrete and positive statement is followed by another statement in the very next sentence, where the authors write that missing values were assigned to $1e-5$, the minimum value across their proteome data. The assignment of such value is a kind of imputation. The right way to treat missing values is to annotate them as n.a. (not analyzed) or n.d. (not detected). The way the authors treat missing values is wrong even if this has been used in previous publications. This measure does not allow discrimination between real missing values/not expressed and random dropouts (compare Supplementary Data 4, and the statement on poor re-identification of proteins in different samples). The impact of this assignment becomes obvious in Figure RL28. There, top up- and down-regulated proteins in the responder and non-responder groups are presented. The numbers of recurrently identified proteins were much higher in the non-responder group than in the responder group (see Q27/Figure RL25). A protein that was detected, for example, in just a few R-samples, however, in a much larger fraction in NR-samples, would automatically be much lower expressed in the R-samples as its imputed expression value of $1e-5$ was more prevalent in the R-group. A correct way out is to treat missing values as missing values.

Response: Thank the reviewer for the comments. According to reviewer's suggestions, the missing values were assigned to n.a. We re-analyzed data without any imputation and updated all the related figures.

For the comparison of Response group and Non-Response group without impute method at proteome level

According to reviewer's constructive suggestions, to avoid the effects of imputation on low detected protein, we treated missing values as n.a. and re-analyzed the proteome difference between Response group and Non-Response group based on the 25% protein identification threshold. As shown in **Figure RL 12 A**, 132 proteins were down-regulated (Fold change (Response / Non-Response) < 0.5 and Wilcox p-value < 0.05) and 36 proteins were up-regulated (Fold change > 2 and Wilcox p-value < 0.05) in Response group. Furthermore, we used the pathway enrichment

analysis to determine the dominant biological processes in two groups. Response group was enriched in G2M checkpoint, Antigen processing and presentation, Th17 cell differentiation, and NF-kappa B signaling pathway, Non-Response group was characterized by mTOR signaling pathway, Neutrophil degranulation, Platelet activation signaling and aggregation. The main results of pathway enrichment between R and NR were not changed.

Accordingly, we updated related figures including **Fig 1g, 2d, 3e, 4d, 4e, 4f, 4i, 4j, 4q, 5c, 6j, 6l, Supplementary Fig. 3c, 3d, 3f, 3h, 8a** (updated missing value as n.a). The main conclusions were unchanged. For example, we updated the heatmap of proteins involved in glycolysis and pentose phosphate pathway (PPP) between AA and non-AA patients (**Figure RL 12 C**); the heatmap of 7q gain correlated proteins participate in mTOR signaling, translation, proteasome, TCA cycle, and FAS signaling (**Figure RL 12 D**), the Kaplan-Meier curve of overall-survival for patients with high and low CRP protein expression group (**Figure RL 12 E**), et al.

Figure RL 12 A The volcano plot depicting the differentially expressed proteins between Response (R) group and Non-Response (NR) group based on no-imputation proteome data. P-values were calculated by Wilcox test. **B** The enriched biological processes of R and NR group by over-represented analysis. **C** Glucose metabolism alteration caused by the AA signature. **D** Heatmap of proteins positively correlated with the Chromosome 7q copy number. Two-sided Spearman's correlations are showed in the right panel. **E** the Kaplan-Meier curves of OS for patients with different CRP abundances (log-rank test).

Q17. Also the dynamic range of protein expression might be related to the assignment of an expression value of $1e-5$ for missing values as this could potentially explain the huge dynamic range (six orders of magnitude) that is described for each protein in the whole dataset. The lower plot in Figure RL29 suggests that many (it is not clear how many proteins are shown) proteins have been identified recurrently in a large number of samples (a span of \log_{10} fold-changes, and a very strong enrichment as some particular range on the Y-scale). How does this relate to the finding that so many proteins were found in ‘just’ 25% of the samples (the total number of samples having been analyzed is 115)? In response to question Q35 ‘the points in the right panel of Figure 4L which the reviewer noted were the missing values after assigned as $10e-4$ ’ points at consequences the assignment of any values to missing values could have’, the authors excluded missing values and recalculated the correlation plot. This is realistic.

Response: Thanks for the comments. The reviewer is correct. The dynamic range is based on the quantification methods.

About how many proteins shown in the Figure RL29 in the last revision

Actually, in our last version, the presentation of dynamic range was based on the raw data which did not impute expression value of $1e-5$ for missing values. There were 12,309 proteins were shown in Figure RL29 in the last revision. In this round of revision, to confirm the reliability of the dynamic range results, we assigned missing value as n.a and redraw the plot again, and the results were consistent with the previous (six orders of magnitude) (**Figure RL 13**).

Figure RL13. Boxplot of the \log_{10} (FOT) protein expression levels for the all identified proteins.

About many proteins identified in a large number of samples in our cohort

In the last version, we compared with the published papers. We compared our cohort with the Xu et al., study published in Cell (*Cell*, 2020, PMID: 32649877), and the Jiang et al., study published in Nature (*Nature*, 2019, PMID: 30814741), according to the number of the proteins detected in more than 25%, 35%, 45%, 55%, 65%, 75%, 85%, 95%, and 100% of samples, respectively (**Figure RL 14**). The proteins detected at different cutoff of the samples in the three cohort were, 25% (7,451, 8,729, 7,149), 35% (7,184, 8,042, 6,578), 45% (6,920, 7,572, 6,101), 55% (6,607, 7,065, 5,667), 65% (6,221, 6,556, 5,249), 75% (5,765, 5,984, 4,743), 85% (5,243, 5,381, 4,185), 95% (4,256, 4,432, 3,316), and 100% (2,823, 3,068, 2,226), respectively. The results showed that our cohort was comparable with the published papers.

According to the reviewer's suggestion, we assigned the missing values as n.a in our cohort in the revision, and updated all the analysis (also see the response of the **Q16**). Thanks the reviewer again for the comments.

Figure RL14. Number of proteins identified with different criterion in our cohort, Xu et al., cohort and Jiang et al., cohort.

Q18. Q30 and revised manuscript lines 203 and the following: The authors predicted transcription factors potentially involved in the regulation of transcription of G6PD, PGD, and TKT. How was this prediction done? Which algorithm was applied? Which other transcription factors were predicted? How/why were SP1, SMAD4, YY1, and SREBF2 selected?

Response: Thank you for your professional suggestions. To response the question more clearly, we split it as two parts.

For the transcription factors (TFs) prediction and applied algorithm

In order to explore the mechanism of AA inhibiting pentose phosphate pathway (PPP), we predicted the TFs that could regulate the expression of G6PD, PGD and TKT. We used the online JASPAR database (<http://jaspar.genereg.net>) and ChEA3 database (*Nucleic Acids Res*, 2019, PMID: 31114921), simultaneously. We have added this information about the TFs prediction in **Methods** part in the **lines 1096, pages 35** in the revision.

For the predicted TFs

We obtained 84 TFs that can bind to G6PD, PGD and TKT gene promoter regions by JASPAR database and 200 potential TFs by ChEA3 database. To improve the confidence of predicted TFs, we took the intersection of the two databases and obtained 18 most likely TFs including TFE3, SP1, SPI1, GATA1, NR2C1, KLF4, TCF3, TEAD4, TFAP2A, KLF5, GATA3, SMAD4, ISX, YY1, SNAI3, SREBF2, MEF2A and VDR. We had updated related information in **Supplementary Data 2**. In this study, there were 8 out of 18 TFs were identified in this ccRCC cohort. As the previous analysis showed the inhibition of AA on pentose phosphate pathway (PPP), we hypothesized the expression level of TFs that regulated PPP affected by AA were down-regulated in tumor tissue. Therefore, we assessed the expression of these 8 TFs in RCC, the results showed 4 out of 8 TFs were lower expressed including SP1, YY1, SREBF2, and SMAD4. Hence, we selected these 4 TFs for the further *in vitro* experiments (**Figure RL 15**). We have added this information in the **Result** part in **lines 292, pages 10** in the revision. Thank you again for your suggestions.

Figure RL 15 The expression level of eight TFs in RCC cohort. **Please also see in the revised Supplementary Fig 5f.**

Q19. Q34-2: I am confused by the response. 1. Why have PBMCs from normal people been sequenced as the background? Where is the data?

Response: We apologize for the confusion.

About why PBMCs from normal people were sequenced as the background and where is the data

In this study, we used the tumor adjacent tissues as the background to detect the somatic mutation of the tumors. We did not collect the blood sample of these patients. In order to response the reviewer's comments about the mutations in the adjacent tissues in the last round of revision, we re-analyzed the mutation signatures using the sequencing result from peripheral blood mononuclear cells (PBMCs) of normal people as the background (PBMCs background libraries provided by sequencing company). The strategy of using the PBMCs background libraries provided by sequencing company was also used in the previous published paper (*Nature Communication*, 2022,

PMID: 36104359). We added the detailed process analysis results in the updated **Supplementary Data 2**.

Q20. The authors find that AA leads to a strongly enhanced AA Signature in the tumor tissue, while this is not found in the adjacent normal tissue (Figure RL34). They cite Li et al. (PMID:33004515) and write that their findings would be consistent with that previous publication. Li and coworkers had analyzed somatic clonal events in morphologically normal human urothelium with and without application of AA. These authors found drastically accelerated mutation accumulation also in the normal urothelium and concluded that AA was a major mutagenic driving factor there. Based on the findings of Li et al., I would have expected to see indications of the AA signature (i.e., A>T transversions) also in normal adjacent tissue from patients having been treated with AA. The cited study does not really support the findings of Qu et al. The authors should discuss the apparent disparate findings.

Response: We apologize for the confusion. The reviewer is correct, we have detected the T>A transversions in the tumor adjacent tissues (**Figure RL 16**). In details, AA exposure may cause mutagenesis characteristic of predominant T>A transversions, which match the Catalogue of Somatic Mutations in Cancer (COSMIC) SBS22 (*Nature*, 2013, **PMID: 23945592**; *Nat Prod Rep*, 2014, **PMID: 24691743**). In this study, we decomposed the mutation spectra of ccRCC tumor samples using COSMIC database (*Nucleic Acids Res*, 2019, **PMID: 30371878**) by the SigProfiler module in python (*Cell Rep*, 2013, **PMID: 23318258**). As a result, we found that there were 19 patients carried the SBS22 signature. We compared the frequency of T>A transversions between tumor and tumor adjacent tissues. The data showed that the mutation signatures of tumor tissues were higher than the mutation signatures of tumor adjacent tissues (**Figure RL 16**), demonstrating the strong mutagenic effect of AA in tumor tissues. Besides, in our cohort, the AA signature might also affect the tumor adjacent tissue to a certain extent, which was consistent with Li et al. (*Science*, 2020, **PMID: 33004515**) study reported that AA signatures were found in normal human tissues. There were 15 out of 19 patients carried AA signature with the count number ranged from 0 - 11 in the normal tissue, which lower than the 238 - 5802 in the tumor tissue. Overall, although the number of mutations in normal tissues was far less than in the tumor tissues, it is necessary to study the somatic mutation accumulation in normal cells which is essential for understanding tumorigenesis

and development. We discussed the effect of AA in tumor adjacent tissue in the **Discussion** part in **lines 680, pages 21** in the revision.

	Tumor-AA count	Normal-AA count
ccRCC_12	4473	11
ccRCC_50	5802	10
ccRCC_22	1239	7
ccRCC_23	2700	6
ccRCC_72	2190	6
ccRCC_96	1578	6
ccRCC_112	4280	5
ccRCC_5	2475	5
ccRCC_60	2136	3
ccRCC_111	1870	2
ccRCC_6	1679	2
ccRCC_16	782	2
ccRCC_80	2993	1
ccRCC_91	1538	1
ccRCC_61	858	1
ccRCC_13	1067	0
ccRCC_11	1062	0
ccRCC_105	685	0
ccRCC_40	238	0

Figure RL 16 The AA count number in tumor tissue and normal tissue, respectively.

Q21. Q34-3: The tumor samples were inspected by an expert pathologist who confirmed sample quality according to the following standards: ‘1) histopathological defined ccRCC tumors; 2) tumor samples with tumor cell rate (tumor purity) > 90%; 3) no tumor cells in the adjacent tissues.’ This classification scheme does not make sense. I should suppose that all samples matched standard 1 – only ccRCC samples were taken into the study. All 115 tumor samples in the study should also meet criterion #2 and have >90% cellularity. #3: Why were tumors excluded that had tumor cells in adjacent tissue?

Response: Thank the reviewer for the comments. The reviewer is absolutely correct, we collected all 115 ccRCC tumor tissue samples with tumor purity >90% and matched adjacent tissue samples. For the criterion 3), we used the matched tumor adjacent tissue DNA as the background to calling

the somatic mutations and copy number alteration (CNA). Once adjacent tissues with tumor cell contaminated were used, it will cause the false negative of somatic mutation and CNA calling, which might mislead the downstream proteogenomic analysis. Therefore, the samples with tumor cell in the adjacent tissues were excluded in this study. We added this explanation in the updated **Methods** in **lines 844, pages 27** in the revision.

Q22. Some explanation of abbreviations, like 'KPS', should be warranted.

Response: Thanks for reviewer's comment. We apologize for not explaining it clearly.

Karnofsky performance score (KPS) is a clinical assessment tool used to assess the overall health of patients. KPS assesses an individual's overall functional status on an 11-point scale, in increments of 10, where a score of 0 for death and 100 for normal function, higher scores signified better functional status (*J Hepatol*, 2018, PMID: 29883596). KPS of patients in this cohort range from 70 to 100. The score 70 shows patients who, despite the inability for normal activity, can take care of themselves; the score 80 shows patients who are able to carry on normal activity with effort, and have some signs or symptoms of disease; the score 90 shows patients who are able to carry on normal activity and have minor signs or symptoms; the score 100 shows patients who are no complaints and no evidence of disease (*Nephrology*, 2007, PMID: 17995577). We have added the explanation of KPS in the **Methods** part in **lines 1197, pages 38** in the revision.

We carefully checked all the abbreviations used in the manuscript, and explained the obscure abbreviations for understanding in the revision. Thank the reviewer again for pointing it out.

Q23. The span in VAFs, in combination with the high tumor purity (90%) suggests high intra-tumor heterogeneity in most patients. This should be discussed.

Response: We thank the reviewer for the comments. The reviewer is correct. The high tumor purity (more than 90%) and a relatively broad range of VAF indicated a high intra-heterogeneity of tumor. There might be two reasons of the high heterogeneity in most patients.

As for the heterogeneity of a single tumor sample, inevitably immune cell will infiltrate tumor tissues, hence, tumor tissue usually has a complexity of the tumor microenvironment. Yasin *et al.*

reported ccRCC is a highly infiltrated by immune cells, particularly by T cells (*Genome Biol*, 2016, PMID: 27855702). Moreover, recent studies utilize single-cell analyses of patient samples to pinpoint cancer cell of origin and identify 12 lymphoid cell subsets (including B cells, natural killer T cells, regulatory T cells, etc.) in RCC (*Cancer cell*, 2021, PMID: 33711272), which further indicating the heterogeneity of the tumor cell microenvironment. In our study, immune landscape characterization also revealed diverse tumor microenvironment subsets in ccRCC patients (**Figure RL 17 A**).

As for the heterogeneity among the tumor samples, in order to illustrate the tumor heterogeneity of samples, we discuss this point at genome and proteome level, separately. For genome level, as shown in **Figure RL 17 B**, the samples showed different mutation pattern. Furthermore, by applying NMF algorithm on mutation spectrum based on the Catalogue of Somatic Mutations in Cancer (COSMIC), we identified different mutation signatures and found the characteristics of genomic signature is distinctive from each other. For the proteomic level, after using ConsensusClusterPlus on the proteome data, the cohort was clustered in three groups characterised by various biological processes (**Figure RL 17 C**).

In summary, the reviewer is correct. The high tumor purity and a relatively broad range of VAF are possibly induced by the complexity of the tumor microenvironment and the heterogeneity of tumor gene variation. We discussed the association among the VAF, tumor purity and heterogeneity in tumor and added this part in the **Discussion** in **lines 741, pages 23** in the revised manuscript. Thank you again for the comments.

Figure RL 17 **A** Immune landscape subsets in ccRCC patients. **B** Single-base substitution (SBS) signature (SBS1, SBS5, SBS22, SBS40) distribution in samples. **C** Proteomic subtypes and upregulated proteins enriched pathways of three proteomic subtypes. **Please also see in the revised Fig. 6a and Supplementary Fig 5b, 7a, 9a, 9d.**

Q24. Q35: The relevance of the regression line in Figure RL38A could be disputed. The authors might look into this.

Response: Thank the reviewer for the comments. To avoid disputation, we used another form to show the result. Concretely, we compared the plasma LDH level between patients identified with CD274 [CD274(+)] and without CD274 [CD274(-)] in mass spectrometry (MS). The result showed that patients with CD274 identification in MS showed higher level of LDH than patients without CD274 identification (**Figure RL18**). We have updated this plot in **Supplementary Fig. 3j** in the revised manuscript.

Figure RL 18 The comparison of plasma LDH levels between patients with and without CD274 identification in MS (Wilcoxon rank-sum test). * indicates $p < 0.05$. **Please also see in the revised Supplementary Fig 3j.**

Other points

Materials and Methods:

Clinical sample collection

Q25. The authors write: ‘At the last follow-up, 102 patients (88.7%) had progressive disease and 84 patients (73.0%) had died of ccRCC.’ These numbers cannot be true for a cohort of 115 patients.

Response: Thank you for your suggestion. Although our research is a retrospective study, all the information we recorded is true and reliable. We determined whether the patient has progressive disease based on the results of long-term follow-up, rather than evaluating the objective response rate. Since the median progression free survival of ccRCC patients treated with targeted therapy is usually about 10 months as previous studies reported (*Clin Genitourin Cancer*, 2019, PMID: 31601514; *Ann Oncol*, 2017, PMID:28327953), we think it is rational that 102 patients (88.7%) had progressive disease as the follow-up time is relatively long in this study (Median OS:28.8 months; Longest OS:167.2 months). We investigate two other targeted therapy cohorts (CABOSUN trial cohort and COMPARZ trial cohort). The median overall survival is 28.8 months in our cohort, 21.2 months in CABOSUN trial cohort, and 29.3 months in COMPARZ trial cohort. The follow-up time is 167.2 months in our cohort, 42 months in CABOSUN trial cohort, and 67 months in COMPARZ trial cohort. The proportion of patients died of ccRCC is 73% in this study, 70%

in CABOSUN trial cohort, and 60.6% in COMPARZ trial cohort. Since this study has a longer follow-up period than the other two cohorts (CABOSUN trial cohort and COMPARZ trial cohort), it is rational that the proportion of patients died of ccRCC in this study seemed to be higher than that in the other two cohorts. When we set censored times the same as the CABOSUN trial research as 42 months, the proportion of patients died of ccRCC in this study was 58.3% (67/115), which was lower than the mortality rate in the CABOSUN trial research (70%) (**Table RL 1**). Thank you again for your professional comments.

Study cohort name	Follow-up time (months)	Proportion of died (censored time to 42 months)
Our cohort	167.2	58.3%
CABOSUN trial	42	70%

Table RL 1 Comparison of follow-up time between two cohorts.

Q26.RNA extraction and RNA-seq

The authors now performed RNA-sequencing. In the materials and methods section, the library preparation protocol having been applied should be stated. In the results part, the mean number of reads/sample should be provided.

Response: Thank you for your suggestion. The library preparation protocol applied have been updated in the **Methods** section. The details are as follows.

RNA Extraction and Qualification

Total RNA from each tissue sample was isolated using TRIzol Reagent (Invitrogen). All RNA samples were assayed for the RNA purity and integrity. Briefly, RNA degradation and contamination were monitored with 1% agarose gels. RNA purity was checked using the NanoPhotometer spectrophotometer (IMPLEN, Los Angeles, CA, USA). RNA concentration and integrity were measured using the Qubit RNA Assay Kit with the Qubit 2.0 Fluorometer (Life Technologies, CA, USA) and the RNA Nano 6000 Assay Kit of the Agilent Bioanalyzer 2100 System (Agilent Technologies, CA, USA), respectively.

Library Preparation and Quality Control

A total amount of 3µg RNA per sample was used as input material for the RNA sample preparations. Firstly, ribosomal RNA was removed by Epicentre Ribo-zero™ rRNA Removal Kit (Epicentre,

USA), and rRNA free residue was cleaned up by ethanol precipitation. Subsequently, sequencing libraries were generated using the rRNAdepleted RNA by NEBNext® Ultra™ Directional RNA Library Prep Kit for Illumina® (NEB, USA) following manufacturer's recommendations. Briefly, fragmentation was carried out using divalent cations under elevated temperature in NEBNext First Strand Synthesis Reaction Buffer(5X). First strand cDNA was synthesized using random hexamer primer and M-MuLV Reverse Transcriptase (RNase H minus). Second-strand cDNA synthesis was subsequently performed using DNA Polymerase I and RNase H. In the reaction buffer, dNTPs with dTTP were replaced by dUTP. Remaining overhangs were converted into blunt ends via exonuclease/polymerase activities. After 3' adenylation, NEBNext Adaptors with a hairpin loop structure was ligated to prepare for hybridization. To preferentially select cDNA fragments of 150~200 bp, the library fragments were purified with AMPure XP system (Beckman Coulter, Beverly, USA). Size-selected, adaptor-ligated cDNA was treated with 3 µl of USER Enzyme (NEB) at 37 °C for 15 min followed by 5 min at 95 °C before PCR. PCR was performed with Phusion High-Fidelity DNA polymerase, Universal PCR primers and Index (X) Primer. At last, products were purified (AMPure XP system) and library quality was assessed on the Agilent Bioanalyzer 2100 system.

Clustering and Sequencing

The clustering of the index-coded samples was performed on a cBot Cluster Generation System using TruSeq PE Cluster Kit v3-cBot-HS (Illumina), according to the manufacturer's instructions. After cluster generation, the libraries were sequenced on an Illumina HiSeq 4000 platform and 125 bp paired-end reads were generated. After trimming the adaptors and removing low-quality tags, sequencing reads were mapped onto the hg19. The mapped reads were assembled into transcripts or genes by using StringTie software (v2.1.4) (*Nat Biotechnol*, 2015, PMID: 25690850).

For quantification purpose, the relative abundance of the transcript or gene was measured by a normalized metrics, FPKM (Fragments Per Kilobase of transcript per Million mapped reads). Transcripts with median FPKM > 1 were retained (**Supplementary Data 3**).

In addition, the mean number of mapped reads/samples was 36.85M, which was also added in the revision. The reads number of each sample was provided in the **Supplementary Data 3**, and we

updated the description of RNA sequencing part in the **Methods** part in **lines 919, pages 29** in the revision. Thank you again for your professional suggestions.

Q27.Protein extraction and trypsin digestion

The make and brand of the sonicator should be provided. The listed conditions (3s on and 3s off, amplitude 25%) do not make sense without knowing which instrument had been applied.

Response: Thank the reviewer for the suggestion. We provided the manufacturer and model of the sonicator (SONICS, VCX130) in the **Methods** section in **lines 963, pages 31** in the revision.

Reviewer #5 (Remarks to the Author):

the authors responded to their best abilities to my comments. in total, they had 71 pages of comments.

The reviewer appreciate that the authors made this huge and lauded effort

Response: Thank you for your positive comments and suggestions, which help to improve the quality of this manuscript.

Reviewer #6 (Remarks to the Author): Expert in renal cell carcinomas and metabolism

The authors performed multi-omics analysis of primary ccRCCs from a Chinese cohort of patients that responded or did not respond to sunitinib treatment. The analysis showed activation of mTOR pathway in nonresponders and aristolochic acid signature in responders. While the study provides some interesting and new information, there are several issues that should be addressed.

Response: Thank you for your positive comments and suggestions, which help to improve the quality of this manuscript. The detailed responses are as follows.

Major comments:

Q28. The manuscript requires major reorganization, prioritization of the results, rewriting of the sections so that they integrate in a concise form all information supporting major conclusions, rather than description of the flow of experiments. Rigorous English editing is necessary. As is, the manuscript is very long, very descriptive, and includes a lot of details that confuse and distract from

the main points and ultimately make the rigorous review process difficult to impossible. In general, figure legends lack sufficient information to evaluate what is shown.

Response: We sincerely thank the reviewer for the careful read and constructive comments. We apologize for the lengthiness and the lack of conciseness of conclusions in the original manuscript. Here, we divided the responses into two parts as follows.

1. Reorganization of the manuscript to clarify our main contributions more understandable, clearer and focused

Following reviewer's constructive suggestions, we condensed the manuscript to make the conclusion more clearly and compelling. We have also reconstructed the framework of the article. In the revised manuscript, the highlights were summarized as follows.

(1) To systematically analyze the different clinical outcomes with tyrosine kinase inhibitor therapy, we constructed ccRCC multi-omics (including genome, transcriptome, proteome, and phosphoproteome) landscape (**Figure 1**). We firstly conducted differential analysis of clinical baseline features between Responders and Non-Responders. The results showed that tumor size, PLT level and LDH level were significantly correlated with PFS/OS. Further, we elucidate biological basis of signaling pathways with significant correlations of these clinical features.

(2) We analyzed the mutation signatures on Sunitinib therapeutic outcomes (**Figure 2**). The results showed that the frequency of T > A transversions was higher in the Non-Responder group than in the Responder group and patients with SBS22, associated with aristolochic acid (AA) exposure, showed better survival in Sunitinib treatment. Furthermore, systematically biological process analysis of mutation signatures revealed that the AA signature enhanced the response to sunitinib treatment by downregulating pentose phosphate pathway and decreasing the ribose-5-phosphate. More importantly, we verified that SP1 is the key transcription factor of AA regulating pentose phosphate pathway.

(3) To explore the Copy Number alterations landscape at the gene level between the Responder and Non-Responder groups, we compared the amplified genes between the two groups and found genes

that more frequently amplified in Non-Responder group were located in 7q. We further analyzed the *cis* effects of the 7q genomic aberrations on proteomic alterations. The further *in vitro* experimental results showed that overexpression of LAMTOR4, MDH2 and CALU, *cis*-regulated by 7q gain, can increase the phosphorylation of S6K and accelerated the invasion of cancer cells. The analysis of Copy Number alterations on Sunitinib therapeutic outcomes revealed that chromosome 7q gain would activate mTOR signaling and lead to poor Sunitinib treatment effectiveness (**Figure 3**).

(4) To further investigate the effects of genetic mutation to Sunitinib treatment, we analyzed the differentially altered genes between Responder and Non-Responder groups. We found that the *VHL* mutation, leading to Warburg effect, increased the vulnerability to Sunitinib treatment in ccRCC. We next comprehensively analyzed the mutational signatures, CNAs, and somatic mutations. The results demonstrated that 3p loss and VHL mutation caused enhanced glycolysis was associated with Sunitinib therapeutic response, and the AA signature could enhance glycolysis process, these results were further confirmed by *in vitro* experiments (**Figure 4**).

(5) The comparison between Responder and Non-Responder groups was drawn as another main topic of this research at proteome and phosphoproteome levels (**Figure 5**). We analyzed the differentially activated kinases between Responders and Non-Responders and found that MAP2K1 (MEK1) and MTOR were activated in Non-Responders, while CDK1/2 were activated in Responders. Further investigation into the differentially expressed phosphosites between Responders and Non-Responders showed that mTOR and MAPK signaling pathways associated substrates phosphorylation were elevated in Non-Responders, while RNA splicing was increased in Responders. The phosphoproteome data suggested that exceeded mTOR and MAPK signaling pathways were associated with Sunitinib treatment resistance.

(6) To further investigate the heterogeneity of Sunitinib therapeutic outcomes, we proposed a rational stratification of ccRCC patients based on immune signatures as T-cell infiltrated, Cold, and Progenitor-cell infiltrated. Progenitor-cell infiltrated group had the highest proportion of Non-Responders and the worst survival among the three subtypes. The Progenitor-cell infiltrated group

exhibited upregulation of the platelet aggregate formation pathway and complement cascades, and high expression of the platelet marker CD321, which were responsible for Sunitinib failure. Further, we found that TGFB1, overexpressed in Progenitor-cell infiltrated group, was an important component for alternative angiogenic signalling in ccRCC, indicating that TGFB1 inhibitors might increase the Sunitinib response (**Figure 6**).

(7) To further apply our findings to clinical screening for TKI response of patients, we constructed classifiers to distinguish the Sunitinib Response group and Non-Response group. Our study reported a random forest model to predict the patients' response to TKI by integrate multiple omics (different layers of CNV, mutation, transcriptome, and proteome). The integrative model showed excellent performance AUROC = 0.89, and can distinguish the response to TKI with >95% sensitivity/specificity. To simplify the complexity of predicted model, we built it only based on the proteome data. It is worth noting that the single model also showed the good performance with AUROC = 0.84 and >95% sensitivity/specificity. Furthermore, the robustness of the single predictive model was validated in an independent dataset (**Figure 7**).

In conclusion, we delineated the proteogenomic landscape of Sunitinib response in Chinese patients with ccRCC and constructed a model for predicting the Sunitinib response and validate the robustness of the predictive model in an independent dataset. More importantly, our multi-level omics analysis identified the molecular mechanisms underlying the Sunitinib response and defined the genomic, proteomic, and immune signatures to stratify patients with ccRCC to develop more rational therapeutic interventions.

2 We rearranged the materials and added detailed figure legend information

To make the manuscript more concise and focused, we rearranged the supporting materials and adjusted the structure of the article. Especially in the “The Clinical Features Associated with Sunitinib Treatment Outcomes” part and “Clinical Parameters Associated Proteomic Features” part in the original manuscript, we moved the analysis of tumor size, plasma LDH levels, and PLT levels from **Fig. 5h, I, j, k, l, m, n** in the last revision to **Supplementary Fig. 3a, b, c, d, e, f, g** in this revision. At last, the original **Fig. 5** and **Supplementary Fig. 2** were integrated into **Fig. 5** and

Supplementary Fig. 3 in the revision. Overall, all the clinical correlations analysis was presented in one section. Meanwhile, we added detailed figure legend information and the quantitative results of validation assay were also provided in the revision. In order to more intuitively reflect the anti-tumor effects, we also verified the all-corresponding experiment **2f**, **3g**, **4h** and **6m** with cell proliferation rate through CCK-8 assay in this round of response.

In summary, we reorganized our story line and highlights, adjusted the structure of the article, and polished the language in revision. After revisions, we believe that the manuscript would be more concise, understandable and focused.

Q29. The cell line data used for the validation of the bioinformatics data are of not very good quality. Only one cell line was used in individual experiments (786-0 in most and ACHN for VHL KDs), and the minimum should be two. The data are often not quantified, the effects are weak and there is no proper information about doses and time courses. Methods section lists 769-P cell line but missed it being used.

Response: Thank you for your professional suggestions.

About the cell line for individual experiments and the data quantification

In order to improve the quality for the validation of the bioinformatics data, we used ACHN and 786-O cell lines for **Fig. 2e**, **2g**, **2h**, **2j**, **2m**, 786-O and 769-P cell lines for **3g-l**, 786-O and ACHN cell lines for **6m-n**, respectively, so that all the important results were verified with two cell lines. In order to more intuitively reflect the anti-tumor effects, besides the transwell assay, we have verified each corresponding experiment **2f**, **3g**, **4h** and **6m** with cell proliferation rate through CCK-8 assay. The corresponding experiment **2f**, **3g**, **4h**, and **6m** quantification results were added in the revised manuscript. All the quantification results were provided in the source data in the revision.

Specifically, **as for the Fig. 2f**, to prove that AA improved the prognosis of ccRCC patients under Sunitinib treatment, we performed the cell proliferation assay *in vitro* and found that AA treatment did not impact cell proliferation directly, but enhanced the inhibition to cell proliferation of Sunitinib

(Figure RL 2f). The transwell assay, which may reflect the degree of malignancy and treatment effect of tumor to a certain extent, showed that AA treatment did not impact cell invasiveness directly, but enhanced the inhibition of Sunitinib to cell invasiveness. For the sake of reliability, we performed the cell proliferation assay, and transwell assay in both ACHN and 786-O cell lines in the revision.

As for the Fig. 3g, to investigate the impacts of 7q gain on Sunitinib therapeutic outcomes, we overexpressed LAMTOR4, MDH2 and CALU, *cis*-regulated by 7q gain, to mimic chromosome 7q gain in ACHN and 786-O cell lines. We found overexpressed LAMTOR4, MDH2 and CALU increased the phosphorylation of S6K, indicating activation of mTORC1. Next, we treated the candidate genes overexpressed cells and normal cells with or without Sunitinib. By monitoring the cell proliferation rates, we found increased expression of LAMTOR4, MDH2 or CALU abrogated the effects of Sunitinib. In conclusion, chromosome 7q gain would activate mTOR signaling and lead to poor Sunitinib treatment effectiveness.

As for the Fig. 4h, proteomic data revealed that patients with *VHL* mutation showed higher expressions of glycolytic enzymes, such as CA9, PGK1, ENO1 and LDHA, indicating the enhanced glycolysis of *VHL* mutated tumors. In the clinic samples, we found *VHL* mutation tumors exhibited accumulated HIF and upregulation of glycolysis. To mimic *VHL* loss-of-function mutations, we knocked down *VHL* in ACHN cells and found increased lactate levels, which was a marker for glycolysis. More importantly, we verified the effect of *VHL* mutation in Sunitinib treatment with cell proliferation rate through CCK-8 assay. The results showed that knockdown of *VHL* significantly inhibited the cell proliferation in the Sunitinib treatment. All these results suggested that the *VHL* inhibition potentially increased the vulnerability to Sunitinib treatment in ccRCC.

As for the Fig. 6m, TGFB1 was elevated in the Progenitor-cell infiltrated cluster and was well correlated with the Sunitinib response. We observed that TGFB1 was co-expressed with proteins involved in angiogenesis and tumor immune escape in ccRCC, indicating that the alternative angiogenesis driven by TGFB1 resulted in Sunitinib resistance. To verify the association between TGFB1 and Sunitinib resistance, we added TGFB1 in the culture medium to simulate the impact of

tumor microenvironment - derived TGFB1 on cancer cells. We tested the cell proliferation rate through CCK-8 assay. The results showed that TGFB1 could enhance the invasiveness and proliferation of ccRCC cell and abrogate the impact of Sunitinib on ccRCC cells. In summary, we considered that the tumor microenvironment, comprising abundant progenitors, led to insufficient Sunitinib therapy response by TGFB1 signaling.

About the information of doses and time courses

The information of doses and time courses were provided in the **Methods** sections in the revision. All the related figures were updated in the revision.

Sunitinib was obtained from MedChemExpress (**SU 11248, HY-10255A**). According to the *in vitro* cell assay experimental concentration recommended by the product manual (1nM-500nM) and the range of *in vitro* experimental concentration used in previous literatures (***EMBO J*, 2011, PMID: 21317875; *J Med Chem*, 2003, PMID: 12646019**), the doses of Sunitinib we used in treating cells were 200nM. (**Figure RL 19**). Specifically, to determine the optimal concentration of Sunitinib in the *in vitro* experiments, based on the product manual and previous studies, we did the gradient experiments. We found pentose phosphate pathway (PPP) was downregulated, manifested as downregulation of G6PD, PGD, and TKT (key enzymes of PPP), in the patients with AA signature undergoing Sunitinib treatment. To determine the optimal concentration of Sunitinib in the *in vitro* experiment, we evaluated the mRNA level of G6PD, PDG, and TKT undergoing a gradient of Sunitinib concentration (including 0nM, 100nM, 200nM, and 300nM). As shown in **Figure RL 19**, the gradually increased Sunitinib concentration decreased the mRNA level of these proteins. When the concentration of the Sunitinib was set as 300uM, it can cause great damage to the cells. Finally, the concentration of Sunitinib in our experiment was set as 200nM, which was consistent with the previous studies (***EMBO J*, 2011, PMID: 21317875; *J Med Chem*, 2003, PMID: 12646019**). In the revision, we updated the description about the Sunitinib concentration used in our *in vitro* experiments in the **Methods** part in **lines 1115, pages 35** in the revision.

Figure RL 19 Effect of different concentration gradient (0nM, 100nM, 200nM, 300nM) of Sunitinib on key enzymes G6PD, PDG and TKT of pentose phosphate pathway.

The AA concentration in human blood after taking Chinese herbal medicine has not been reported in literature. To determine the optimal concentration of AA *in vitro* experiments, based on the product manual, we did the gradient experiments. In detail, patients with AA signature showed smaller tumor size and better survival in Sunitinib treatment. By the functional enrichment analysis, we found that the attenuated pentose phosphate pathway (PPP) and enhanced glycolysis in tumors with AA signature, manifested as downregulation of G6PD, PDG, TKT ($p < 0.05$) (key enzymes of PPP). To verify this finding, *in vitro* experiments were performed with renal cell lines, to evaluate the expression level of G6PD, PDG, and TKT undergoing a gradient of AA concentration (including 0 μM , 5 μM , 25 μM , and 100 μM). As shown in **Figure RL 20**, the gradually increased AA concentration decreased the expression level of the three proteins in both 786-O and ACHN cell lines. When the concentration of AA was set to 100 μM , we observed the dramatic inhibition effect of the protein expression. Once the concentration exceeds 100 μM , it will adversely impact the cell state. Hence, the concentration of AA in our experiment is 100 μM and this AA concentration is consistent with the results of the previously published study about the AA on renal epithelial cells (*Kidney International*, 2005, PMID: 15840026). In the revision, we updated the description about the AA concentration used in our *in vitro* experiments in the **Methods** part in **lines 1107, pages 35** in the revision.

Figure RL 20 AA treatment inhibited the expressions of G6PD, PGD and TKT in 786-O and ACHN cells in dose-dependent manners. **Please also see in the revised Fig. 2g-h.**

Q30. While the effects of AA are interesting there is no description how exposure to AA was determined and if it was qualitative or quantitative. It is not clear how the doses of AA used in cell line experiments correspond to the AA exposures in patients.

Response: Thank you for your professional suggestions.

For the criteria of AA exposure definition

AA exposure may cause mutagenesis characteristic of predominant T>A transversions, which match the Catalogue of Somatic Mutations in Cancer (COSMIC) SBS22 (*Nature*, 2013, PMID: 23945592; *Nat Prod Rep*, 2014, PMID: 24691743). Several studies have demonstrated the potential association between AA exposure and ccRCC oncogenesis (*Br J Cancer*, 2014, PMID: 26657656). In this study, we decomposed the mutation spectra of ccRCC tumor samples using COSMIC database (*Nucleic Acids Res*, 2019, PMID: 30371878) by the SigProfiler module in python (*Cell Rep*, 2013, PMID: 23318258). As a result, we found that there were 19 patients carried the SBS22 signature. This strategy is a qualitative, rather than exact quantitative method to evaluate AA exposure level.

For the quantification of AA content in patients' tissues or any other concrete estimation of exposure

The connection between AA exposure and T>A transversions motivate the evaluation of the AA exposure by the genome events. The detection of the SBS22 signature indicated the AA-caused mutations aggregated in patients' tumor tissues. Due to the complexity of the patient's diet, the diversity of environmental exposures, etc., direct quantification of the AA content in patients' tissues or estimation of the AA exposure is difficult. We introduced the AA signature in the **Instruction** section in the previous manuscript. To deliver the information more clearly, we emphasized the definition of AA exposure in the **Result** section the revision.

For the AA concentration in the *in vitro* experiment

The concentration of AA in our experiment is 100μM. The detailed information please also see in the response of **Q29**.

Q31. The main conclusion that 7q gain induced mTOR signaling is responsible for lack of response to sunitinib is based on (i) enrichment for mTOR pathway among proteins that were positively correlated with 7q copy number gain; (ii) GSEA analysis in transcriptome; however, the results of RNA analysis are not clearly presented. (iii) increased mTOR activity measured by KSEA. The cell line validation of these conclusions is weak. A simple convincing validation would be immunocytochemistry for S6-P between the responder and non-responders.

Response: We thank the reviewer for the comments. According to reviewer's suggestions, we performed immunohistochemical (IHC) staining for pS6K in Responders and Non-Responders. The results showed that Non-Responders expressed higher level of pS6K than Responders (**Figure RL 21**). We added the IHC results in the revised **Fig. 3g**.

Figure RL 21 The representative IHC for pS6K in the tumor and tumor adjacent tissue of Responders and Non-Responders. **Please also see in the revised Fig. 3g.**

Detailed comments:

Q32. Lanes 133-144: did the authors check combinations of the clinical parameters for differences between nonresponders and responders? These data are confusing as it is stated that clinical parameters were not different between responders and nonresponders, however, later on in the “clinical correlations of proteomic feature”, it appears that LDH and PLT have a number of correlations that are relevant for the stratification. This is confusing and perhaps all clinical correlations should be presented in one section.

Response: We appreciate the reviewer’s comments. In spite of clinical parameters showed no significant differences between Non-Responders and Responders, these parameters were associated with patient’s survival. The reviewer is correct that the split discussion of clinical correlations leads to confusion. According to reviewer’s suggestion, we moved the analysis of tumor size, plasma LDH levels, and PLT levels from **Fig. 5h, I, j, k, l, m, n** in the last revision to **Supplementary Fig. 3a, b, c, d, e, f, g** in this revision. At last, the original **Fig. 5** and **Supplementary Fig. 2** were integrated into **Fig. 5** and **Supplementary Fig. 3** in the revision. Overall, all the clinical correlations were presented in one section in “The Clinical Features Associated with Sunitinib Treatment Outcomes” part of the revised manuscript. Thank the reviewer again for pointing it out.

Q33. What are the criteria for the definition of AA exposure? Is it possible to have a quantification of AA content in patients’ tissues or any other concrete estimation of exposure?

Response: Thanks for reviewer’s comments. We response the question one by one.

For the criteria of AA exposure definition

AA exposure may cause mutagenesis characteristic of predominant T>A transversions, which match the Catalogue of Somatic Mutations in Cancer (COSMIC) SBS22 (*Nature*, 2013, PMID: 23945592; *Nat Prod Rep*, 2014, PMID: 24691743). Several studies have demonstrated the potential association between AA exposure and ccRCC oncogenesis (*Br J Cancer*, 2014, PMID: 26657656). In this study, we decomposed the mutation spectra of ccRCC tumor samples using COSMIC database (*Nucleic Acids Res*, 2019, PMID: 30371878) by the SigProfiler module in python (*Cell Rep*, 2013, PMID: 23318258). As a result, we found that there were 19 patients carried the SBS22 signature. This strategy is a qualitative, rather than exact quantitative method to evaluate AA exposure level.

For the quantification of AA content in patients' tissues or any other concrete estimation of exposure

The connection between AA exposure and T>A transversions motivate the evaluation of the AA exposure by the genome events. The detection of the SBS22 signature indicated the AA-caused mutations aggregated in patients' tumor tissues. Due to the complexity of the patient's diet, the diversity of environmental exposures, etc., direct quantification of the AA content in patients' tissues or estimation of the AA exposure is difficult. We introduced the AA signature in the **Instruction** section in the previous manuscript. To deliver the information more clearly, we emphasized the definition of AA exposure in the **Result** section the revision.

For the AA concentration in the *in vitro* experiment

The concentration of AA in our experiment is 100µM. The detailed information please also see in the response of **Q29**.

Q34. Figure 2 and the legend require additional specific information: 2e and f - cell line and doses of the drugs; 2g duration of treatment, 2h, 2l, 2m - doses of AA; 2j, 2k – what is BC and NC; 2m – not clear what the bleached parts of the blots are. It is not clear how the doses of AA used in *in vitro* experiments relate to the AA exposures of patients. Cell line validation should use a minimum of two cell lines in each experiment.

Response: Thank you for your professional suggestions.

About the Figure 2 and the legend

We apologize for the unclear description in the original manuscript. In the revision, we added the detailed information of corresponding figures and figure legends.

Fig. 2e and **2f** use 786-O cell line. In order to improve the quality for the validation of the bioinformatics data, we added ACHN cell line in the revision to verify the corresponding assay. The dose of AA we used in treating cells was 100µM. The dose of Sunitinib we used in treating cells was 200nM. The treatment time of **Fig. 2g** and **2i-2m** is 24 hours, and the working concentration of AA in **Fig. 2h** and **2i-2m** is 100µM. As for the **Fig. 2j, 2k**, BC is the abbreviation of blank control

and NC is the abbreviation of negative control. In order to avoid misunderstanding, we only retained the negative control as the control group in most of the new revised manuscript. **2m**-the bleaching part of the stain is the background of the band, which may be caused by non-specific intervention.

About the dose of AA used *in vitro* experiments

The concentration of AA in our experiment is 100 μ M. The detailed information please also see in the response of **Q29**.

About the cell line

In order to improve the quality for the validation of the bioinformatics data, we used ACHN and 786-O cell lines for **Fig. 2e, 2g, 2h, 2j, 2m**, 786-O and 769-P cell lines for **3g-1**, 786-O and ACHN cell lines for **6m-n**, respectively, so that most of the important results were verified with two cell lines. In order to more intuitively reflect the anti-tumor effects, we have verified each corresponding experiment **2f, 3g, 4h and 6m** with cell proliferation rate through CCK-8 assay. The corresponding experiment **2f, 3g, 4h and 6m** quantification results were added in the revised manuscript. All the quantification results were provided in the source data in the revision. All the quantification results were provided in the source data in the revision. The detailed information please also see in the response of the **Q29**. We thank the reviewer again for pointing it out.

Q35. Lane 187: the statement that “glucose was shunted away from pentose phosphate pathway” is not appropriate as there is no metabolomic flux data. The conclusion is implied by transcriptomic data and should be reported as such.

Response: We thank the reviewer for the comments. We tuned down our statement in the revision. The sentence was revised as “we found that the attenuated pentose phosphate pathway (PPP) and enhanced glycolysis in tumors with AA signature, manifested as upregulation of PFKFB3, PFKFB3, GPI (t-test, $p < 0.05$) and downregulation of G6PD, PGD, TKT (t-test, $p < 0.05$).”

Q36. Was there an enrichment for SP1 signature in the transcriptomics data of the responders?

Response: Thanks for the comment. We used the target genes of SP1 from DoRothEA (*Genome Res*, 2019, PMID: 31340985) to infer the SP1 activities in ccRCC tumor using VIPER (*Nat Genet*,

2016, PMID: 27322546). To evaluate the SP1 signature between Response (R) and Non-Response (NR), we split all samples into two group by the median score of SP1 signature (SP1 signature (high) and SP1 signature (low)) and performed fisher exact test. The results showed there's no significance between R and NR, but significantly different between AA and non-AA, which indicating the effect of SP1 on AA signature (**Figure RL 22**). We updated this part in the **Supplementary Fig. 5k** in the revision.

Figure RL 22 The stacked barplot depicting the distribution of SP1 signature (high) and SP1 signature (low) patients between AA, non-AA and NR, R. **Please also see in the revised Supplementary Fig. 5k.**

Q37. Lane 221: “ results indicated that AA enhanced the response to sunitinib by downregulating ppp...” is an overstatement as there is no mechanistic causative connection shown.

Response: Thanks for the comment. According to reviewer’s suggestion, we revised this sentence as “these results indicated that there were connections between AA exposure, Sunitinib treatment outcome, and the downregulated pentose phosphate pathway” to avoid overstatement. We thank the reviewer again for pointing it out.

Q38. What exactly is shown in Fig. 3e? is the expression of the genes based on RNA or CNA? Or are these protein expressions?

Response: Thanks for the comments.

As for the information of Fig. 3e

To explore the CNA landscape at gene level between the Responder and Non-Responder groups, we compared the amplified genes between the two groups and found genes that more frequently amplified in Non-Responder group were located in 7q (Fisher's exact test, $p < 0.05$). To find out the effects of 7q copy number (CN) on sunitinib therapy at proteome level, we calculated the spearman's correlation of 7q CN and proteome data. As a result, there were 485 proteins showed significantly positive correlations with 7q CN ($p < 0.05$). These proteins were enriched in pathways including mTOR signaling (RRAGB, RRAGD, LAMTOR2, LAMTOR4, MLST8), TCA Cycle (CS, SDHA, MDH2), oxidative phosphorylation (COX6C, NDUFS6), and fatty acid biosynthesis (ACACA, MCAT), etc. As shown in **Figure RL 23** (also see in the **Fig. 3e**), the proteins participate in above pathways showed positive correlation with 7q CN.

Figure RL 23 Heatmap depicting the protein expression levels positively correlated with the Chromosome 7q copy number. Two-sided Spearman's correlations are showed in the right panel. Please also see in the revised **Fig. 3e**.

As for the data used in **Fig.3e**

Fig. 3e showed the protein expression levels of these genes (**Figure RL 23**). To avoid confusion, we revised the legend as “Heatmap depicting the protein expression levels positively correlated with the Chromosome 7q copy number. Two-sided Spearman's correlations are showed in the right panel.”

Thanks again for the comments.

Q39. Fig. 3g – doses of sunitinib and quantification of the effects need to be provided. Overall, the

effects of overexpression on S6K-P are rather minor. These data would need to be quantified. How is MDH2, mitochondrial malate dehydrogenase, connected to mTOR signaling?

Response: Thank you for your professional suggestions. We apologize for the confusion.

About the dose of sunitinib and quantification of the effects

Sunitinib was obtained from MedChemExpress (SU 11248, HY-10255A). According to the *in vitro* cell assay experimental concentration recommended by the product manual (1nM-500nM) and the range of *in vitro* experimental concentration used in previous literatures (*EMBO J*, 2011, PMID: 21317875; *J Med Chem*, 2003, PMID: 12646019), the doses of Sunitinib we used in treating cells were 200nM. (**Figure RL 24**). Specifically, to determine the optimal concentration of Sunitinib in the *in vitro* experiments, based on the product manual and previous studies, we did the gradient experiments. We found pentose phosphate pathway (PPP) was downregulated, manifested as downregulation of G6PD, PGD, and TKT (key enzymes of PPP), in the patients with AA signature undergoing Sunitinib treatment. To determine the optimal concentration of Sunitinib in the *in vitro* experiment, we evaluated the mRNA level of G6PD, PDG, and TKT undergoing a gradient of Sunitinib concentration (including 0nM, 100nM, 200nM, and 300nM). As shown in **Figure RL 24**, the gradually increased Sunitinib concentration decreased the mRNA level of these proteins. When the concentration of the Sunitinib was set as 300uM, it can cause great damage to the cells. Finally, the concentration of Sunitinib in our experiment was set as 200nM, which was consistent with the previous studies (*EMBO J*, 2011, PMID: 21317875; *J Med Chem*, 2003, PMID: 12646019). In the revision, we updated the description about the Sunitinib concentration used in our *in vitro* experiments in the **Methods** part in **lines 1115, pages 35** in the revision.

Figure RL 24 Effect of different concentration gradient (0nM, 100nM, 200nM, 300nM) of Sunitinib on key enzymes G6PD, PDG and TKT of pentose phosphate pathway.

About the effects of overexpression on S6K-P

To validate the impacts of chromosome 7q gain, we overexpressed LAMTOR4, MDH2 and CALU, cis-regulated by 7q gain, to mimic chromosome 7q gain in 786-O cells, respectively. In this round of revision, we repeated the effects of overexpression on S6K-P results. The results showed that overexpressing of LAMTOR4, MDH2 and CALU greatly increased the phosphorylation of S6K by 1.6, 1.5, and 1.6 times, respectively, indicating activation of mTORC1 (*Biochem J.*, 2012, PMID: 22168436) (**Figure RL 25 A**). In conclusion, chromosome 7q gain would activate mTOR signaling and lead to poor Sunitinib treatment effectiveness. Next, we treated the candidate genes overexpressed cells and normal cells with or without Sunitinib. By monitoring the invasion of cancer cells with transwell assay, we found increased expression of LAMTOR4, MDH2 or CALU abrogated the effects of Sunitinib. In this round of revision, the quantitative results of transwell were provided (**Figure RL 25 B**) and the source data of transwell quantitative results were also provided in the **Source Data**. In order to more intuitively reflect the anti-tumor effects, we have verified each corresponding experiment with cell proliferation rate through CCK-8 assay (**Figure RL 25 B**) in the revision.

Thank the reviewer again for pointing this out. We thoroughly updated the all quantitative results of **2f**, **3g**, **4h** and **6m** transwell assay in this revision. All these results were also added in the revision **Fig. 2f**, **3g**, **4h** and **6m**, and we also provided the **Source Data** of transwell quantitative results in the revision.

Figure RL 25 **A** Effects of overexpression of LAMTOR4, MDH2, and CALU on phosphorylation of S6K. **B** Up panel, Transwell detected the effect of LAMTOR4, MDH2, and CALU overexpression and Sunitinib treatment on cell invasiveness. Middle, Quantification of transwell results. Down, CCK-8 detected the effect of LAMTOR4, MDH2, and CALU overexpression and Sunitinib treatment on cell proliferation. **Please also see in the revised Fig. 3f-h and Supplementary Fig 6e-g.**

About the relationship between MDH2 and mTOR signaling

MDH2, utilizing the NAD/NADH cofactor system in the citric acid cycle, participate in mitochondrial metabolism. The mitochondrial metabolism is closely related to the mTOR pathway (*Nature*, 2007, PMID: 18046414; *Nature*, 2007, PMID: 18046414; *J Aging Res*, 2011, PMID: 21629705). In our study, we found MDH2 *cis*-regulated by 7q gain, to further validate the impacts of chromosome of 7q gain, we found overexpressed MDH2 increased the phosphorylation of S6K, indicating activation of mTORC1. Based on the published studies and the experiment results, we hypothesis MDH2 might participate in mTOR signaling activation by mitochondrial metabolism. However, the direct relationship between MDH2 and mTOR has not been extensively reported. The specific mechanism needs further research in the future. We discussed the association between the MDH2 and mTOR signaling in the revision and added this part in the **Discussion** in the revised manuscript. Thank you again for the comments.

Q40. Fig. 3f and 3g: It is not clear why these particular genes were selected for the *in vitro* analysis. Is the putative connection between 7q gain and mTOR supported in other studies or databases?

Response: We thank the reviewer for the comment. To response the question clearly, we split this question as follows:

The particular genes selection for the *in vitro* analysis

To explore the CNA landscape at gene level between the Responder and Non-Responder groups, we compared the amplified genes between the two groups and found genes that more frequently amplified in Non-Responder group were located in 7q (Fisher's exact test, $p < 0.05$) (**Figure RL 26 A**). To find out the effects of 7q copy number (CN) on sunitinib therapy at proteome level, we calculated the spearman's correlation of 7q CN and proteome data. As shown in **Figure RL 26 B**, there were 485 proteins showed significantly positive correlations ($p < 0.05$). These proteins were enriched in pathways including lysosome (LARP1, LARP2), innate immune system (PTGES), mTOR signaling (RRAGB, RRAGD, LAMTOR2, LAMTOR4, MLST8), TCA Cycle (CS, SDHA, MDH2), oxidative phosphorylation (COX6C, NDUFS6), and fatty acid biosynthesis (ACACA, MCAT). There were 29 out of 485 proteins encoded by 7q genes. Genomic alterations that affect gene expression levels at the same locus are defined to act in *cis*. We observed three (LAMTOR4, MDH2, CALU) out of 29 proteins showed *cis* effects and LAMTOR4, participating mTOR signaling, showed the most significant *cis* effects ($p = 1.88E-4$). Therefore, we overexpressed LAMTOR4, MDH2 and CALU, *cis*-regulated by 7q gain and RBL2, POR2, *cis*-regulated by other chromosome to mimic the effects of 7q gain on mTOR signaling. The results showed that the overexpression of LAMTOR4, MDH2 and CALU, rather than TBL2 and POR2, increased the phosphorylation of S6K, indicating activation of mTORC1 (**Biochem J., 2012, PMID: 22168436**). In conclusion, chromosome 7q gain would activate mTOR signaling and lead to poor Sunitinib treatment effectiveness.

The connection between 7q gain and mTOR was further validated by TCGA cohort

According to reviewer’s suggestion, we surveyed the TCGA RPPA data of ccRCC. The result showed that 7q gain tumors had higher level of pS6K (P70S6KP T389, Wilcoxon rank-sum test, $p = 0.03$) and mTOR signaling scores than 7q WT tumors (Wilcoxon rank-sum test, $p < 0.01$) (**Figure RL 26 C**). These results indicated the chromosome 7q gain would activate mTOR signaling in ccRCC patients.

Figure RL 26 A Comparison of gene-level CNAs between Responders and Non-Responder in this cohort. The upper plot illustrates the frequency of CNA events, the lower plot illustrates the $-\log_{10}$ (p value) of each gene for the comparison of Responders and Non-Responder (two-sided Fisher’s exact test). **B** The plot depicting the Spearman correlations between protein expression and 7q gain. **C** Comparison of pS6K and mTORC1 signaling scores between 7q gain ccRCC and 7q WT ccRCC patients in the TCGA RPPA and transcriptome data (Wilcoxon rank-sum test). **Please also see in the revised Fig. 3d and Supplementary Fig 6d.**

Q41. Lane 266 – HLA-B genes are highly polymorphic; how the mutation status was altered between the two groups?

Response: We thank the reviewer for the comment. According to reviewer’s suggestion, we checked the mutation status in the two groups and presented the relevant contents in the following

table (**Table RL 2**). Concretely, a total of 4 samples had HLA-B gene missense mutation, including 3 Responders (R) (RCC_60, RCC_86 and RCC_83) and 1 Non-Responder (NR) (RCC_59). Four missense mutation sites of HLA-B were detected, including 3 in R (rs4997052, rs1051488 and rs1131500) and 1 in NR (rs1050529). It is worth noting that RCC_86 in R had two HLA-B missense mutation sites (rs1051488 and rs1131500), and RCC_86 and RCC_83 in R had an identical HLA-B missense mutation site (rs1051488). In general, the mutation status of HLA-B was different between the two groups, with R/NR of 3:1, suggesting the potential association between the mutation status of HLA-B and response of Sunitinib. We updated this information into the **Supplementary Data 2** in the revision.

Hugo Symbol	Start Position	End Position	Variant Classification	Variant Type	dbSNP_RS	Group	Sample ID
HLA-B	31324144	31324144	Missense Mutation	SNP	rs4997052	R	RCC_60
HLA-B	31322911	31322911	Missense Mutation	SNP	rs1051488	R	RCC_86
HLA-B	31322980	31322980	Missense Mutation	SNP	rs1131500	R	RCC_86
HLA-B	31322911	31322911	Missense Mutation	SNP	rs1051488	R	RCC_83
HLA-B	31324615	31324615	Missense Mutation	SNP	rs1050529	NR	RCC_59

Table RL 2 HLA-B mutation status of R and NR, including Hugo Symbol, Start Position, End Position, Variant Classification, Variant Type, dbSNP_RS, Group and Sample ID.

Q42. Fig. 4g – it is well established that loss of VHL leads to increased glycolytic activity and lactate production; the fact that that AA appears to enhance this response is not referred to in the manuscript. Again, the doses of AA are not specified. VHL KD is very weak. 4h – needs quantification.

Response: Thank you for your professional comments of our results.

About the connection between AA signature, VHL mutation and glycolytic activity and lactate production

The reviewer is correct that loss of VHL leads to increased glycolytic activity and lactate production is well-established. Actually, we discussed the result that “AA treatment significantly upregulated the VHL deficiency caused lactate increase” in **line 335-336** in the previous version of manuscript.

Specifically, in our cohort, by analyzing the impacts of mutation signatures on Sunitinib therapeutic outcomes, we found that AA improved the prognosis of ccRCC patients under Sunitinib treatment by attenuating the pentose phosphate pathway (PPP) and enhancing glycolysis in tumors. The results were also confirmed by the intracellular lactate level in the *in vitro* experiments. As shown in **Figure RL 27 A**, the lactate level was lower in the *VHL* knockdown group than the WT group, regardless of AA state. Notably, for the *VHL* knockdown group, by adding AA in the cultured medium, the lactate level was increased significantly. These results indicated the connection between AA signature, *VHL* mutation, and glycolysis. Considering the latent impact of the interactions of genetic alterations on the prognosis of Sunitinib therapy, we comprehensively analyzed the mutational signatures, CNAs, and somatic mutations. The integrative analysis revealed the co-occurrence of the AA signature, 3p loss, and *VHL* mutation. We divided our cohort into 5 groups based on their alteration status (Group1: AA/3p loss/*VHL*^{Mut}; Group2: AA/3p loss/*VHL*^{WT}; Group3: non-AA/3p loss/*VHL*^{WT}; Group4: non-AA/3p loss or *VHL*^{Mut}; Group5: non-AA/non-3p loss/*VHL*^{WT}), and found they were significantly associated with poor prognosis. Through differentially expressed protein analysis, we found that glycolysis (PFKL, LDHA) and HIF pathways (VEGFA) were progressively downregulated from Group1 to Group5.

Figure RL 27 A The impact of *VHL* knockdown and AA treatment on intracellular lactate (two-sided t test). Please also see in the revised Fig. 4g.

About the doses of AA

The concentration of AA in our experiment is 100μM. The detailed information please also see in the response of **Q29**.

About the VHL knockdown and the quantification for figure 4h

According to the reviewer’s suggestions, in the revision, in order to achieve higher knockout efficiency, we simultaneously used three siRNAs (5’- GCUCUACGAAGAUCUGGAATT-3’; 5’- GGCUCAACUUCGACGGCGA-3’; 5’- CUGCCAGUGUAUACUCUGA -3’) to exclude off-target effects. Finally, we were surprised to find that the inhibitory effect reached nearly 80% (**Figure RL 28 A**). More importantly, to make the experimental results more observable, we replaced the phenotype with the cell proliferation rate and tested it through CCK-8 assay in the revision (**Figure RL 28 C**). In addition, the quantitative results of transwell were also provided in this revision (**Figure RL 28 B**). The results further verified that inhibiting VHL could promote the efficacy of sunitinib. We provided these results in the revised **Fig. 4h** and **Supplementary Fig. 7d**.

Figure RL 28 A Transfection efficiency of VHL siRNA detected by qRT-PCR and western blot. **B** Transwell detected the effect of VHL knock down and Sunitinib treatment on cell invasiveness. **C** CCK-8 detected the effect of *VHL* knockdown and Sunitinib treatment on cell proliferation. **Please also see in the revised Fig. 4h and Supplementary Fig 7d-e.**

Q43. Lanes 328-332 – it is not specified in which group what pathways are changed.

Response: We thank the reviewer for the comment and we apologize for not explaining it clearly. In this section, we divided our cohort into 5 groups based on their alteration status (Group1: AA/3p loss/*VHL*^{Mut}; Group2: AA/3p loss/*VHL*^{WT}; Group3: non-AA/3p loss/*VHL*^{WT}; Group4: non-AA/3p loss or *VHL*^{Mut}; Group5: non-AA/non-3p loss/*VHL*^{WT}), and found they were significantly associated with poor prognosis (**Figure RL 29 A-B**). Through differentially expressed protein analysis, we found that glycolysis (PFKL, LDHA) and HIF pathways (VEGFA) were progressively downregulated from Group1 to Group5, and glucose transport (SLC2A3), the pentose phosphate pathway (TKT, PGD), and inflammatory response (C1QC, C9, S100A8) were progressively upregulated from Group1 to Group5 (**Figure RL 29 C**).

To make this description more precise, we revised the sentence as “Differentially expressed protein analysis among the five types of events showed the attenuation of glycolysis (PFKL, LDHA) and HIF pathways (VEGFA), and enhancement of pathways such as glucose transport (SLC2A3), the pentose phosphate pathway (TKT, PGD), and inflammatory response (C1QC, C9, S100A8) from Group1 to Group 5 (**Fig. 4q**)” in **lines 475, pages 16** in the revision. Thank you again for your professional suggestions.

Figure RL 29 A Comprehensive analysis of mutation signatures, CNAs, and gene mutations revealed the co-occurrence of AA signature, 3p loss, and *VHL* mutations. This cohort was divided into five groups based on their alteration status (Group1: AA/3p loss/*VHL*^{Mut}; Group2: AA/3p loss/*VHL*^{WT}; Group3: non-AA/3p loss/*VHL*^{WT}; Group4: non-AA/3p loss or *VHL*^{Mut}; Group5: non-AA/non-3p loss/*VHL*^{WT}). **B** Kaplan-Meier curves of PFS for five genomic subgroups (log-rank test for trend). **C** Differentially expressed proteins in the five groups and their associated biological pathways. **Please also see in the revised Fig. 4o-q.**

Q44. The section “Differential analysis between sunitinib... proteome and phosphoproteome levels” (lane 341-342) is written in a very confusing manner. It starts with phosphoproteome, then moves to the GSEA analysis of the transcriptome, and then to proteome and phosphoproteome again. If the main conclusion is that sunitinib resistance is mediated by augmented mTOR signaling, the data should be presented to support that main conclusion. I am not sure where proper transcriptome analysis is presented.

Response: Thanks for the comments. We apologize for the confusion. Actually, we performed GSEA analysis using proteome data but not transcriptome (line 352). We only used the proteome and phosphoproteome data in the section “Differential Analysis between Sunitinib Therapeutic Responders and Non-Responders at Proteome and Phosphoproteome Levels”. We have revised this part as follows “Next, we compared the proteome of the Responder and Non-Responder groups, using gene set enrichment analysis (GSEA). It was observed that G2M checkpoint, antigen processing and presentation, Th17 cell differentiation, and NF-kappa B signaling pathway were enhanced in the Responders, while mTOR signaling pathway, neutrophil degranulation, and platelet activation signaling and aggregation were upregulated in the Non-Responders (**Fig. 5c**). The differentially expressed proteins (t-test < 0.05, FC >1.5) between Responder and Non-Responder groups in tumor tissues were shown in the **Supplementary Fig. 8b (Supplementary Data 5)**.” in **lines 500, pages 16** in the revision.

For the analysis of transcriptome data in this study

We performed transcriptome sequencing in 94 samples in this cohort and identified 12,276 protein-coding genes with median fragments per kilobase of transcript per million fragments mapped (FPKM) of more than 1 (**Supplementary Data 2** in the revision). Our data showed that the mRNA-protein correlation was moderate with sample-wise median Spearman’s correlation of 0.39 (**Fig. 1g**). In the “Impacts of Mutation Signatures on Sunitinib Therapeutic Outcomes” part in the revised manuscript, we found patients with aristolochic acid (AA) signature were characterized by enhanced glycolysis and attenuated pentose phosphate pathway (PPP) (G6PD, PGD, and TKT). Further in vitro experiments also showed the inhibition of AA on PPP enzymes (G6PD, PGD, and TKT). In order to investigate how AA inhibited the expressions of those genes, by predicting the potential transcriptional factors (TFs) which were involved in the regulation of those genes and using VIPER

tools to predict TFs activity based on transcriptome data, we found SP1 activity was significantly active in patients with AA signature (**Supplementary Fig. 5I**), which indicating the effect of SP1 on AA signature. Thanks again for the reviewer's comments.

Q45. It is not clear why only proteome classifier was determined and not classifiers that would integrate multiple omics.

Response: Thanks for the reviewers' suggestions. According to the reviewer's suggestion, we built a random forest model to predict the patients' response to TKI by combining data from different layers of CNV, mutation, transcriptome, and proteome. It is worth noting that the integrative multiple omics features model showed better performance (area under the receiver operating characteristics (AUROC)= 0.89, **Figure RL 30 B**) than the previous model only using proteome data (AUROC= 0.84, **Figure RL 30 A**). In detail, the multi-omics feature combination can distinguish the response to TKI with >85% sensitivity/specificity (**Figure RL 30 C**). We have updated the multi-omics predictive model section in the revised manuscript.

Figure RL 30 A Proteomic ROC–AUC statistics in fivefold cross-validation repeated five times in a random forest model for classification of Response group and Non-Response group. **B** Multi-omics ROC–AUC statistics in fivefold cross-validation repeated five times in a random forest model for classification of Response group and Non-Response group. **C** Multi-omics Confusion matrix indicating model performance when predicted on the test split of the cohort. **Please also see in the revised Figure 7c-f.**

REVIEWER COMMENTS

Reviewer #1 (Remarks to the Author):

The authors have revised the manuscript extensively, providing additional data and information. This has partially addressed my previous comments. The most significant unresolved question is related to the AA concentration used in experiments. There is literature on the AA content of herbal medicines (e.g. PMID 11877594), which has been used to estimate possible peak plasma AA concentrations (PMID 21546538). These values are orders of magnitude lower than the 100uM used in the present study, making it hard to interpret the results. The underlying logic of this section is also difficult to follow, as the presence of an AA mutation signature does not necessarily mean that the patients have had significant AA levels in their plasma during sunitinib treatment.

Reviewer #4 (Remarks to the Author):

The authors present an exceptionally long response to rather many reviewer comments.

The response to Q10 is clear enough: the patients simply could not afford to pay for the drug (i.e., Sunitinib) and thus did not receive any treatment. Those patients are not mentioned in the revised manuscript, 'Clinical sample collection' in lines 768 and following. The authors should indicate there whether or not those patients, who had been in the clinics 10 years ago had consented to use of their samples for the study, and assess whether this was in agreement with the general principles of the WMA Declaration of Helsinki.

In response to Q11-1, the authors state that they used a pool of three siRNAs to downregulate VHL mRNA. They state that they were surprised to find that the inhibitory effect reached nearly 80%. I am similarly surprised. Their siRNA #1 contains both uracil and thymidine residues. The latter are uncommon in RNA. Two of the three siRNAs have a length of just 19 nucleotides, while siRNAs commonly have a length of 21-24 nt. One siRNA 5'-GGCUCAACUUCGACGGCGA-3' has a perfect match to the plus strand of the VHL gene (ENST00000696153.1) at positions 333-351 at the 5'-end (after replacement of uracil to thymidine in that sequence). It is surprising that this siRNA should bind and downregulate an mRNA that has the same sequence (and is not the reverse complement). The other two siRNA sequences do not match ENST00000696153.1 (VHL) in either plus or minus strand. The authors should state which reference sequence for VHL they used and how the design of siRNAs was done.

'Independent' and not 'simultaneous' use, as specified by the authors, of siRNAs should be done to exclude off-target effects -> testing redundancy of effects with independent tools.

In lines 80-82 of the revised manuscript the authors write that, 'due to the complexity of the patient's diet, the diversity of environmental exposures, etc., direct quantification of the AA content in patients' tissues or estimation of the AA exposure is difficult.' With such statement, any causal relation between

AA exposure and molecular properties of that compound in patients should be challenging. The authors should tone down their claims accordingly – e.g., in line 617-624, where they discuss of ‘proteomic differences between AA patients and non-AA patients’.

In Supplementary Figure 3C the authors show proteins and related pathways they found enriched in their proteomic dataset. The term ‘translation’ is covered by just two proteins, EIF2B2 and EIF5. Inspection of Supplementary Figure 3C does not really indicate any correlation of EIF2B2 expression with tumor size. What are significance levels of this and the other enriched pathways?

In lines 685 and following, the authors discuss tumor heterogeneity in the context of high tumor purity and an observed broad range of VAF. That combination does not necessarily infer immune cell infiltration but rather intra-tumor heterogeneity of the tumor cells.

Based on four miRNAs having been found differentially expressed in responders vs. non-responders the authors conclude (lines 740-741) that ‘miRNA treatment might contribute to increase the response to Sunitinib for ccRCC patients.’ Such claim would require extensive experimentation and verification of causalities before it could potentially be made. The authors should tone down their final statement on those miRNAs and their potential use in therapy.

Line 932: the type and brand of the mass spectrometer used to acquire mass spectra should be indicated.

Reviewer #7 (Remarks to the Author): Expert in clear cell renal cell carcinoma genomics and metabolism

After careful consideration, I find the scope of the study by Zhang et al. that includes a comprehensive multi-omics profiling of ccRCC patients in Chinese populations treated with Sunitinib, and the strength of evidence supporting the observed differences between responders and non-responders, as sufficiently compelling to make this a noteworthy contribution to be judged in full by the scientific community.

At the same time, in my opinion the manuscript requires substantial improvements in 3 main areas: i) overall structure, clarity of exposition, and better prioritization/integration of overabundant minor points; ii) better integration of the relevant clinical, mutation, CNV, transcriptomic, proteomic and phosphoproteomic data for the studied cohort to be shared in a harmonized and easily digestible form with the community without infringing on privacy or IP issues; iii) detailed discussion of the proposed statistical and machine learning models for the prediction of responders vs. non-responders, including the analysis of feature importance to suggest a small set of proteins (and other features/predictors) that can be used to discriminate between responders vs. non-responders.

I would consider these changes minor, but essential. Minor in the sense that no new data need to be generated and no major methodological changes are required to further analyze these data to support conclusions. Essential, on the other hand, since without them the manuscript remains difficult to follow, data sets generated are hard to access and re-analyze, and the proposed prediction model used and tested directly, likely limiting the overall impact of the study.

Regarding the first point, multiple sections of the manuscript, individual paragraphs and sentences are very hard to follow. A perfect example is the following sentence in the Discussion section, lines 631-633, which reads: “There were 15 out of 19 patients carried AA signature with the count number ranged from 0 - 11 in the normal tissue, which lower than the 238 – 5,802 in the tumor tissue.”

What are the ‘counts’ the authors are referring to in that sentence? Since the next sentence concerns mutations, one might presume these are mutation counts. If so, however, how 0-11 ‘counts’ found in adjacent normal tissue of 15 out 19 patients would qualify as a signature of AA? I believe that a wholesome editing by a professional scientific editor is warranted to improve the grammar, clarity, and overall exposition.

Perhaps even more importantly, a better organization and prioritization/integration of overabundant minor points is required. I am in full agreement with Reviewer 5 who raised this issue before, and I still find the manuscript lacking in this regard. I would suggest focusing on main findings summarized in Figure 7, which summarizes the results of the proposed prediction model to discriminate between responders vs. non-responders using proteomics and other data, and shows clear trends in terms of the size of the tumor, PLT, 7q gain, VHL/KMT2C mutational status, immune subtypes, and proteomics-based pathway activation status.

These main findings and critical validation steps could then represent the main arc of the paper, while all other minor or non-central observations and analyses should be relegated to Supplementary Materials. I believe that one prime example of the latter is RNA-seq transcriptional profiling that does not seem to be central to the main conclusions and can be summarized in the main paper by a short statement of the overall correlation between mRNA and protein levels. Similarly, many methodological aspects can be briefly summarized in the main paper, while remaining details, such as details of RNA-seq library prep and sequencing, can be moved to Supplementary Data.

On the second point, I can appreciate privacy and IP related constraints, especially with regards to WES and the final prediction model. However, since only VHL and other frequent mutations are used for the prediction of putative responders, the WES data would not need to be shared. Similarly, any sensitive clinical data could be removed, while providing the most relevant variables, such as the tumor size. I believe that once stripped of sensitive information, the data should be made publicly available following

the TCGA model of multi-domain data and meta-data integration, ideally through a user-friendly interface. Since this intersects with editorial policies regarding data sharing, I defer here to the editor.

On the last point, the manuscript does not provide sufficient details regarding the proposed prediction models.

In particular, 'the linear model' referred to in line 591 is never defined. It is mentioned again in the methods sections (line 1136) but without a definition, searching Supplementary Materials for 'linear model' finds no hits. There are many types of linear classifiers, linear perceptron or linear (L1 norm) SVM being two well known examples. So, assuming it is a linear model, what is the classification algorithm that is used, what are the input features, and since these are apparently derived from protein abundance levels for 'high confidence proteins' are those the same for all samples, or are they sample specific? Is the reference set used to define the 'outliers' always the same? How does this affect feature selection in 5-fold cross-validation?

For the random forest-based classifier, is the set of 'highly confident proteins' selected in each run of the 5-fold cross-validation? What are the most informative proteins based on the feature importance analysis in random forest? What are other predictors/features that contribute to the multi-omics model?

Since random forest is an ensemble of decision trees, it would be of interest to the community and important for clinical applications to extract a simple rule-based model that combines the size of the tumor, PLT, 7q gain, VHL/KMT2C mutational status, immune subtypes, and proteomics-based pathway activation status to predict responders vs. non-responders.

The sentence in lines 605-606 reads: "In detail, the multiomics feature combination can distinguish the response to TKI with >95% sensitivity/specificity (Fig. 7f)." Yet, the inspection of the figure indicates that about 50% sensitivity is achieved for 95% specificity. Incidentally, this is where the multi-omics model shows better performance compared to 'linear model' in terms of TPR at 0% FPR, which is arguably more important than the overall AUROC, especially in this relatively imbalanced classification problem.

Some other minor points:

Please harmonize the often referred to Spearman correlation coefficient by introducing SCC early on and using it consistently throughout the paper (as opposed to 'Spearman's'), provide the values of SCC consistently when referring to it, rather than just providing p-values.

Please harmonize and refer consistently to mRNA vs. protein vs. phosphoprotein data wherever applies.

The following lines contain unclear passages, bad grammar or other issues that need to be fixed (too many to list specific issues in each):

181, 184, 187, 189, 227, 230, 234, 260, 261, 262-267, 272, 285, 304, 309, 311, 321, 330, 340-343, 361, 366, 374, 379-382, 390, 462, 490-495, 500, 506, 514, 539, 557, 571, 591, 604-606, 610, 616, 621, 622-623, 625, 745-746.

REVIEWER COMMENTS

Reviewer #1 (Remarks to the Author):

Q1. The authors have revised the manuscript extensively, providing additional data and information. This has partially addressed my previous comments. The most significant unresolved question is related to the AA concentration used in experiments. There is literature on the AA content of herbal medicines (e.g. PMID 11877594), which has been used to estimate possible peak plasma AA concentrations (PMID 21546538). These values are orders of magnitude lower than the 100uM used in the present study, making it hard to interpret the results. The underlying logic of this section is also difficult to follow, as the presence of an AA mutation signature does not necessarily mean that the patients have had significant AA levels in their plasma during sunitinib treatment.

Response: We sincerely thank the reviewer for the careful read and constructive comments. To answer the reviewer's questions clearly, we divided the responses into two parts as follows.

As for the AA concentration

Sung-Sen Yang et al's study is a case report (*Am J Kidney Dis*, 2002, PMID: 11877594), in which it is mentioned that some Chinese herbal medicines contain aristolochic acid, and the aristolochic acid contained in Chinese herbal medicines is 3.1-77 (mg/g herb), but the concentration of aristolochic acid in patients' blood is not mentioned. Kathleen G Dickman et al. (*J Pharmacol Exp Ther*, 2011, PMID:21546538) provided abundant AA concentration data, especially in Supplemental Table 1 (*J Pharmacol Exp Ther*, 2011, PMID:21546538), which summarized the reported and inferred values of plasma aristolochic acid concentration related to nephropathy in human and animal models (**Table RL 1**). Notably, in a Phase I Clinical Study about Aristolochic Acid (Nsc-50413), the patient's daily dose was 1mg/kg, and the peak plasma concentration of AA was 67uM (*Cancer Chemother Rep*, 1964, PMID: 14226128). The content of aristolochic acid in most Chinese herbs is more than 1mg/kg, so it is speculated that the AA concentration in the blood can reach 100uM. In our experiment, the concentration of 100uM may be able to simulate the real situation *in vivo*, which also benefits from the guidance of the product manual of AA reagent we purchased (<https://www.medchemexpress.cn/aristolochic-acid-a.html>), and is supported by the corresponding reference (*Toxicol Lett*, 2018, PMID:29655784). In addition, in the last round of the response, we also provided the exploration and verification data of the optimal concentration gradient *in vitro*. Thanks again for your valuable advice.

Supplemental Table 1. Reported and extrapolated values for plasma aristolochic acid concentrations associated with nephropathy in humans and in animal models.

Model	Daily dose	Route	Duration	Peak plasma concentration	Reference
Human	1 mg/kg	IV	3 d	67 μ M	S1
Human	25 μ g/kg	PO	13 mo	2 μ M	S2
Human	6.7 μ g/kg [#]	PO	5 mo	391 nM	S3
Human	3.9 μ g/kg [#]	PO	12 mo	226 nM	S4
Human	3.9 μ g/kg [#]	PO	72 mo	226 nM	S4
Human	1.5 μ g/kg [#]	PO	8 mo	88 nM	S3
Human	1.5 μ g/kg [#]	PO	22 mo	88 nM	S5
Human	0.8 μ g/kg [#]	PO	2 mo	49 nM	S6
Human	0.3 μ g/kg [#]	PO	10 mo	18 nM	S5
Human	0.3 μ g/kg [#]	PO	24 mo	18 nM	S3
Mouse	10 mg/kg	IP	Once	100 μ M	S7
Mouse	10 mg/kg	IP	Once	65 μ M	S8
Mouse	10 mg/kg	IP	Once	26 μ M	S9
Rat	12 mg/kg	PO	Once	30 μ M	S10
Rat	10 mg/kg	IG	Once	21 μ M	S11
Rabbit	0.4 mg/kg	IV	Once	12 μ M	S12

Human peak plasma concentration values are extrapolated and represent peak levels based on full distribution into a 3 L plasma volume. [#]based on 60 kg as an average body weight. IP, intraperitoneal; IV, intravenous; IG, intragastric; PO, oral.

Table RL 1 Summary of the reported and inferred values of plasma aristolochic acid concentration related to nephropathy in human and animal models (*J Pharmacol Exp Ther*, 2011, PMID:21546538).

As for the underlying logic of the AA validation assay

Aristolochic acid (AA) is prevalently used in traditional herbal medicine in Asia (*Biomed Res Int*, 2014, PMID: 25431765; *J Natl Cancer Inst*, 2010, PMID: 20026811); AA exposure may cause mutagenesis characteristic of predominant T>A transversions, which match COSMIC SBS22 (*Nature*, 2013, PMID: 23945592). In our cohort, we detected the mutation characteristics caused by AA from the patient's renal cell carcinoma tissue, patients with SBS22, associated with AA exposure, showed better survival in Sunitinib treatment (GB-Wilcoxon test, $p = 0.038$) (**Figure RL 1A**). Consistently, compared with the non-AA group, patients with the AA signature showed smaller tumor size (t test, $p = 0.0062$) (**Figure RL 1B**), suggesting the specific mechanism of AA exposure. The reviewer is absolutely correct that the presence of an AA mutation signature does not necessarily mean that the patients have had significant AA levels in their plasma during sunitinib treatment. Our study is retrospective and the tissue samples from patients we used in the current study are collected in the past several years. However, we did not obtain the blood samples from those patients. Therefore, we cannot quantify the concentration of AA in blood of our cohort of patients, which is really a regret of this research. In view of ethical problems, it is also impossible to prescribe aristolochic acid to patients alone and then test their blood for verification.

Later, we need to conduct further studies with AA close to physiological concentration in animal primary cells, which is the direction we will do in the future.

In the revision, we toned down the statement about the association between the AA levels and sunitinib treatment as “based on the multi-omics data and bioinformatical analysis, in addition to the genomic impact of AA, we described the proteomic differences between AA patients and non-AA patients, and observed attenuated pentose phosphate pathway and enhanced glycolysis alteration which connected to the Sunitinib treatment. These results indicated AA not only affect the mutation at genomic level, but also bring about changes at proteomic level”. Furthermore, the AA concentration used in experiments was discussed in **Discussion** part in the revised manuscript. Thank you again for your careful reminder.

Figure RL1 Impacts of AA Exposure on Sunitinib Therapeutic Outcomes **A**, Kaplan–Meier curves of progression-free survival (PFS) for patients with or without the AA signature (GB-Wilcoxon test). **B**, Comparisons of tumor size between patients with or without the AA signature (two-sided t test). Data are shown as mean \pm SD.

Reviewer #4 (Remarks to the Author):

Q2. The authors present an exceptionally long response to rather many reviewer comments. The response to Q10 is clear enough: the patients simply could not afford to pay for the drug (i.e., Sunitinib) and thus did not receive any treatment. Those patients are not mentioned in the revised manuscript, ‘Clinical sample collection’ in lines 768 and following. The authors should indicate there whether or not those patients, who had been in the clinics 10 years ago had consented to use of their samples for the study, and assess whether this was in agreement with the general principles of the WMA Declaration of Helsinki.

Response: Thank you for your suggestions. All the patients enrolled for analysis in this research had signed consent forms (**Figure RL 2**) before they underwent surgical treatment at Fudan University Shanghai Cancer Center (FUSCC), which allows us to use their samples for scientific

research. FUSCC will also perform an ethical review (Figure RL 3) before researches begin and the collection of the patients' samples is following general principles of the WMA Declaration of Helsinki. Thank you again for your suggestions.

**复旦大学附属肿瘤医院生物样本库
样本收集知情同意书**

患者姓名: _____ 床号: _____ 住院号: _____
肿瘤部位: _____ 生物样本库编号: _____

一、知情部分

您受邀参加复旦大学附属肿瘤医院将您的部分组织组织、体液（包括血液）和其他样本、及样本相关信息捐献给科研项目所用的行动。样本将被保存在复旦大学附属肿瘤医院徐汇院区和浦东院区西区（东安路270号和红曲路688号，电话021-64175590）。

收集样本及相关信息本身不会给您带来任何痛苦。血液和其他样本收集是在各项检查的同时收集的。组织和其他样本包括病理科常规样本的收集的前提是诊断和治疗必须进行切除或活组织检查，是在标本离体后，充分保证病理诊断所需才进行的。所收集的样本将在低温下或通过福尔马林固定石蜡包埋保存。

这些样本及相关信息是战略性收集，将会被用于与肿瘤治疗相关的生物医药研究、寻找有助于判断预后好坏的因子，可能包括一些基因表达研究和遗传学研究。这些研究可能有助于为临床选择最为合适的治疗方案或为药物治疗效果提供预测等。因此，一些与临床疾病治疗相关的您的信息也会与样本一同收集。

样本及相关信息收集是公益性的、非盈利性的，同时也不需支付任何费用。用于研究的样本及相关信息的保存和使用将是长期的，目前还不能准确推测保存及使用的时间，因为人类攻克癌症是长期目标。

样本及相关信息用于对人类健康有益的研究项目。使用样本及相关信息有严格的审查程序，保证科研的合理性和可行性，以及符合伦理法律规范。复旦大学附属肿瘤医院还设有符合伦理的样本最终销毁的具体规范。

样本及相关信息的采集收集是公益性的、非盈利性的，除医院投入的成本外，没有获益，对您本人也没有经济获益。但未来研究的结果会为您以及与您相似的患者提示新的治疗方法，这可能会给您和与您类似的其他患者带来益处。

您具有充分的隐私权，样本和所有信息的采集都将在法律允许的范围内实现全面的保密。您的传统身份识别信息在交付样本使用者时均被隐去。您的生物样本以及医学信息将用编码进行标记。在任何研究报告和出版物中您将不会被辨认出来。

捐献样本的举动是互助互利的，您的参与是自愿的。您的参加会为别的学生带来更多治愈的可能。当然如果您选择不参加，不会对您的治疗有任何不良影响。您可以选择在任何时间退出这一行动，将不会影响您的治疗。

如果您有任何疑问，您有权向我们提出问题（请致电：021-64175590复旦大学附属肿瘤医院伦理委员会）

二、同意部分

我已经阅读了本知情同意书。
我有机会提问而且所有问题均已得到解答。
我理解参加本活动完全是自愿的。
我知道我的样本和所有信息的采集都将在法律允许的范围内实现全面的保密。
我也可以选择在任何时候退出这一举动，我的任何医疗待遇与权益不会因此而受到影响。
我知道签名并不意味着可以免去任何费用、应尽的事项和药品费用。
复旦大学附属肿瘤医院会提供给我一份经过签名并注明日期的知情同意书副本。

患者或法定代理人签名: _____ 医务人员签名: _____
日期: 2014.05.20 日期: 2014.5.20

.1.

Figure RL 2. Consent form.

复旦大学附属肿瘤医院医学伦理委员会审查意见通知
(适用于非初始审查项目)

伦理编号: 050432-4-1911D

审查日期	2019.11.4
审查会议地点	2号楼5楼会议五室
研究项目名称	复旦大学附属肿瘤医院生物样本库样本收集知情同意书 原标题: 复旦大学附属肿瘤医院组织库样本收集知情同意书
审查文件	相关资料(每单项必须填写,提交资料标记为√,未提交资料的标记为×,如无版本号标记为—) [x] 国家食品药品监督管理局批件,批件文号: _____ [x] 方案,版本号: _____ [x] 项目,编号: _____ [x] 药品生产许可证及检测报告/医疗器械注册证及检验报告 [x] 知情同意书样本,版本号: 3.1版(2019.10.21) [x] 研究者履历、临床研究经历 [x] 其他(请说明): _____
研究科室	组织库
主要研究者	孙孟红
申办者	复旦大学附属肿瘤医院组织库
伦理审查方式	[x] 会议审查 [ ] 快速审查
审查投票结果	应到人数: 17人(其中列席1人) 投票人数: 16人 弃权人数: 0人 回避委员: 无
审查意见	复旦大学附属肿瘤医院医学伦理委员会于2019年11月4日会议审查了组织库孙孟红教授递交的“复旦大学附属肿瘤医院组织库样本收集知情同意书”模板修正申请。 本次会议应到人数17人,实到人数17人(其中1人列席),投票人数16人。审查结果: 知情同意书修改: 同意。 伦理委员会批准3.1版(2019.10.21)组织库样本收集知情同意书模板,知情同意书模板标题改为: 复旦大学附属肿瘤医院生物样本库样本收集知情同意书。 主任或副主任委员签字: _____ 复旦大学附属肿瘤医院医学伦理委员会(盖章) 日期: 2019年11月5日
注意:(请仔细阅读)	1. 该研究进行过程中将接受伦理委员会的持续审查,请根据初始审查伦理批准函要求及时提交持续审查报告。 2. 本通知将在各中心机构及其伦理委员会备案。 3. 已批准项目须遵循本伦理委员会批准的方案执行,须符合CFDA/GCP和《赫尔辛基宣言》的原则。 4. 暂停/提前终止临床研究,请及时通知伦理委员会。 5. 发生严重不良事件及影响研究风险受益比的非预期事件,须及时报告本伦理委员会。 6. 对已批准的临床研究方案、知情同意书等材料的任何修改及主要研究者更换等,须及时通知本伦理委员会重新审查,获得批准后执行。 7. 发现违反方案情况须及时报告伦理委员会。 8. 请根据伦理委员会对持续审查频度的意见,无论试验开始与否,请在持续审查日到期前1个月提出持续审查的申请。 9. 完成临床研究,须提交结题报告供伦理委员会审查。 10. 所有伦理委员会审查的内容都已经涵盖在本通知上。

附件: 伦理委员会签到及保密协议、伦理委员会组成人员名单(适用于会议审查项目)

Figure RL 3. The Ethical review

Q3. In response to Q11-1, the authors state that they used a pool of three siRNAs to downregulate VHL mRNA. They state that they were surprised to find that the inhibitory effect reached nearly 80%. I am similarly surprised. Their siRNA #1 contains both uracil and thymidine residues. The latter are uncommon in RNA. Two of the three siRNAs have a length of just 19 nucleotides, while siRNAs commonly have a length of 21-24 nt. One siRNA 5'-GGCUCAACUUCGACGGCGA-3' has a perfect match to the plus strand of the VHL gene (ENST00000696153.1) at positions 333-351 at the 5'-end (after replacement of uracil to thymidine in that sequence). It is surprising that this siRNA should bind and downregulate an mRNA that has the same sequence (and is not the reverse complement). The other two siRNA sequences do not match ENST00000696153.1

(VHL) in either plus or minus strand. The authors should state which reference sequence for VHL they used and how the design of siRNAs was done. ‘Independent’ and not ‘simultaneous’ use, as specified by the authors, of siRNAs should be done to exclude off-target effects -> testing redundancy of effects with independent tools.

Response: We thank the reviewer for the comments. We apologize for the unclear description in the last version. Previously, we only listed the information of VHL-siRNA sense chain. The details of three VHL-siRNA are as follows (**Table RL 2**).

VHL-siRNA-1	Target sequence	GCTCTACGAAGATCTGGAA
	sense-F (5-3)	GCUCUACGAAGAUCUGGAATT
	antisense-R (5-3)	UCCAGAUUCGUAGAGCTT
VHL-siRNA-2	Target sequence	GGCTCAACTTCGACGGCGA
	sense-F (5-3)	GGCUCAACUUCGACGGCGA (dT)(dT)
	antisense-R (5-3)	UCGCCGUCGAAGUUGAGCC (dT)(dT)
VHL-siRNA-3	Target sequence	CTGCCAGTGTATACTCTGA
	sense-F (5-3)	CUGCCAGUGUAUACUCUGA (dT)(dT)
	antisense-R (5-3)	UCAGAGUAUACACUGGCAG (dT)(dT)

Table RL 2 The detailed information of the three siRNA sequences

For detailed sequence information, please also see the link file (**Figure RL 4**):

<https://www.jianguoyun.com/p/DUO51DsQkcKnCxiz3PIEIAA>. In the revision, we updated the detailed siRNA sequences in the **Source data**.

Figure RL 4 The details position of three VHL-siRNA used in our cohort.

As for why we use pool instead of a single one, it was because we have tried various combinations and finally found that three together have the best knockdown effect, and similar practices have been reported in previous articles (*Journal of Drug Targeting*, 2008, PMID: 18274934). The specific experimental verification results are as follows (**Figure RL 5**), and this is based on the actual experimental results, which is why we chose this combination. In the revision, we have added this results in the **Source data** parts. Thank the reviewer again for pointing it out.

Figure RL 5 The display of *VHL*-siRNA knockdown experimental efficiency.

Q4. In lines 80-82 of the revised manuscript the authors write that, ‘due to the complexity of the patient’s diet, the diversity of environmental exposures, etc., direct quantification of the AA content in patients’ tissues or estimation of the AA exposure is difficult.’ With such statement, any causal relation between AA exposure and molecular properties of that compound in patients should be challenging. The authors should tone down their claims accordingly – e.g., in line 617-624, where they discuss of ‘proteomic differences between AA patients and non-AA patients’.

Response: Thanks for the considerate comments. According to reviewer’s suggestion, we tone down the statement in the **Introduction** part by removing the sentence “due to the complexity of the patient’s diet, the diversity of environmental exposures, etc. direct quantification of the AA content in patients’ tissues or estimation of the AA exposure is difficult”. We also toned down our statement in the **Discussion** part to make the results more accurate. For example, line 617-624 was revised from “These results indicated AA not only affect the mutation at genomic level, but also

alter the cell signaling pathway at proteomic level” to “These results indicated AA not only affect the mutation at genomic level, but also bring about changes at proteomic level”.

Q5. In Supplementary Figure 3C the authors show proteins and related pathways they found enriched in their proteomic dataset. The term ‘translation’ is covered by just two proteins, EIF2B2 and EIF5. Inspection of Supplementary Figure 3C does not really indicate any correlation of EIF2B2 expression with tumor size. What are significance levels of this and the other enriched pathways?

Response: Many thanks for your comments and pointing this out. We apologize for the incorrect presentation. The term ‘translation’ is covered by two proteins EIF1AX (but not the protein EIF2B2), and EIF5 which were significant correlation with tumor size in our data. As the **Table RL 3** showed that all proteins in the **Supplementary Figure 3C** were significant correlated with the tumor size in our data. We have revised the **Supplementary Figure 3C** in the revision (**Figure RL 6**). Thank you again for your kind reminding.

Symbol	correlation r for tumor size	$-\text{Log}_{10}(\text{p-value})$	annotation
ORM1	0.407992311	5.221021556	Acute phase proteins
CRP	0.289427773	2.726900593	Acute phase proteins
SAA1	0.240710762	1.990453429	Acute phase proteins
VCAN	0.24659495	2.10287812	Angiogenesis
VTN	0.210438202	1.620079322	Angiogenesis
NRP1	0.196508838	1.452193495	Angiogenesis
SERPINA1	0.355816659	4.021412599	Complement
C5	0.318221409	3.276850105	Complement
FGA	0.279092365	2.597975074	Complement
FN1	0.332742461	3.553267521	EMT
BMP1	0.282232365	2.471444665	EMT
CD44	0.250795204	2.163522513	EMT
THBS2	0.226580153	1.827070609	EMT
EIF1AX	0.339847738	3.665201854	Translation
EIF5	0.207065824	1.578532339	Translation

Table RL 3 The detailed information for the proteins correlated with the tumor size

Figure RL 6 Heatmap showed the proteins significant correlated with the tumor size

Q6. In lines 685 and following, the authors discuss tumor heterogeneity in the context of high tumor purity and an observed broad range of VAF. That combination does not necessarily infer immune cell infiltration but rather intra-tumor heterogeneity of the tumor cells.

Response: Thanks for reviewer’s constructive comments. The reviewer is absolutely correct. The high tumor purity and a relatively broad range of variant allele frequency does not necessarily infer immune cell infiltration but rather intra-tumor heterogeneity of the tumor cells. We have revised **Discussion** on this part as follows in the revision. “The high tumor purity (more than 90%) and a relatively broad range of variant allele frequency (VAF) indicated a high intra-heterogeneity of tumor. In our study, by applying NMF algorithm on mutation spectrum based on the COSMIC, we identified different mutation signatures and found the characteristics of genomic signature is distinctive from each other. Furthermore, consensus clustering identified three ccRCC proteomic subtypes with distinct features. Moreover, immune landscape characterization also revealed diverse tumor microenvironment subsets in ccRCC patients. In summary, the high tumor purity and a relatively broad range of VAF are possibly induced by the heterogeneity of tumor gene variation.”

Q7. Based on four miRNAs having been found differentially expressed in responders vs. non-responders the authors conclude (lines 740-741) that ‘miRNA treatment might contribute to increase the response to Sunitinib for ccRCC patients.’ Such claim would require extensive experimentation and verification of causalities before it could potentially be made. The authors should tone down their final statement on those miRNAs and their potential use in therapy.

Response: Thanks for reviewer’s comments. The reviewer is correct that extensive experimentation and verification of causalities are needed to support this conclusion. Following

the reviewer's comments, we toned down this by removing the statement "This result suggested that miRNA treatment might contribute to increase the response to Sunitinib for ccRCC patients" in the revised manuscript.

Q8. Line 932: the type and brand of the mass spectrometer used to acquire mass spectra should be indicated.

Response: Thanks for the reviewer's comments. All samples in our cohort were analyzed on a Q Exactive HF-X mass spectrometer (Thermo Fisher Scientific). Following the reviewer's comments, we revised the sentence as "Detection was performed using Orbitrap (Q Exactive HF-X mass spectrometer, Thermo Fisher Scientific) and data were acquired using Xcalibur software (Thermo Fischer Scientific)" in the revised manuscript. Thank the reviewer again for pointing it out.

Reviewer #7 (Remarks to the Author): Expert in clear cell renal cell carcinoma genomics and metabolism

After careful consideration, I find the scope of the study by Zhang et al. that includes a comprehensive multi-omics profiling of ccRCC patients in Chinese populations treated with Sunitinib, and the strength of evidence supporting the observed differences between responders and non-responders, as sufficiently compelling to make this a noteworthy contribution to be judged in full by the scientific community.

Response: We appreciate the reviewer for the positive evaluation and constrictive comments. We have revised the manuscript according to the comments. The point to point responses were as follows.

Q9. At the same time, in my opinion the manuscript requires substantial improvements in 3 main areas:

i) overall structure, clarity of exposition, and better prioritization/integration of overabundant minor points;

Response: We thank the reviewer for the careful read and valuable suggestions to present the statement of our manuscript clearly and concisely. According to reviewers' constructive comments, firstly, we restructured the story line, removed the redundant information and modified the imprecise conclusions. Then, we prioritized and integrated overabundant minor points, making

the manuscript more clearly and focused. At last, after compiling all the materials, we adjusted arrangement of all kinds of supporting materials, including figures, tables, et al., according to our story line to make the article more coherent and logical.

As for the overall structure

In the revised manuscript, the story line and our improvements were as follows:

1) To systematically reveal the molecular basis of differential clinical outcomes with tyrosine kinase inhibitor (TKI) therapy, we constructed 115 patients with clear cell renal cell carcinoma (ccRCC) undergoing Sunitinib treatment multi-omics landscape. Patients were divided into Responders (n = 27) and Non-Responders (n = 88) (**Figure 1**). After presenting the basic characteristics of our cohort and clinicopathologic data, we compared somatic mutations differences between tumor tissues and NATs. In this part, we moved the original **Fig 1d, f and g** to Supplementary material, which made the revised manuscript more concise.

2) To explore the impacts of mutation signatures on Sunitinib therapeutic outcomes, we further decomposed the mutation spectra using the COSMIC database. The results showed that patients with SBS22, associated with aristolochic acid (AA) exposure, showed smaller tumor size and better survival in Sunitinib treatment (**Figure 2**). Therefore, we assessed the proteomic impact of AA signature, the results showed that the attenuated pentose phosphate pathway (PPP) and enhanced glycolysis in tumors with AA signature, manifested as upregulation of PFKFB3, PFKFB1, GPI and downregulation of G6PD, PGD, TKT. Thus, we supposed that AA improved the prognosis of ccRCC patients under Sunitinib treatment by downregulation PPP. By conducting *in vitro* experiments, we found that AA treatment did not impact cell proliferation directly, but enhanced the inhibition to cell proliferation of Sunitinib, the similar results were also found in transwell assay. Additionally, in this round of revision, according to the reviewer's suggestion, to verify our conclusions in multiple dimensions, we also added additional omics hierarchical data evidence wherever applies, such as the Fig 2d in the revised manuscript.

3) To further elucidate the therapeutic outcomes of Sunitinib from genomic perspective, the copy number alterations (CNA) were analyzed. We found that 7q gain was associated with shorter survival in this Sunitinib treatment cohort. In additions, genes that were more frequently amplified in the Non-Responder group than in the Responder group were located in 7q. To further evaluate the proteomic consequences of 7q gain in ccRCC, we performed *cis-/trans*-effect analysis. We observed 3 (LAMTOR4, MDH2, CALU) proteins showed *cis* effects with their encoding genes on

7q, in which LAMTOR4, participating mTOR signaling, showed the most significant *cis* effects on 7q. Further *in vitro* experiments indicated the 7q gain would activate mTOR signaling and lead to poor Sunitinib treatment effectiveness in ccRCC patients (**Figure 3**). In this part, we moved the original **Fig 3b-c** to Supplementary material, which made the revised manuscript more concise.

4), In addition to elucidating the CNA effect on Sunitinib therapeutic outcomes, we further evaluated the gene mutation effects in Responders and Non-Responders (**Figure 4**). We found the *VHL* mutation, considered as a truncal genetic alteration event, was significantly associated with good prognostic outcomes in Sunitinib treatment. To further investigate the impacts of *VHL* mutations on protein expressions and related biological functions, we examined the significantly altered proteins in patients with or without *VHL* mutation. The enrichment analysis results showed glycolysis and HIF-1 signaling were found to be elevated in patients with *VHL* mutation. In addition, through the transwell assay and cell proliferation assay, we found that *VHL* knockdown distinctly enhanced the inhibition of cell proliferation and invasiveness of Sunitinib to cancer cells. Based on the mutation status of *VHL* and *KMT2C*, we divided patients into four genotypes. Notably, the different genotypes showed different tumor sizes and distinct clinical outcomes. To make the revised manuscript more concise, in this part, we moved the original **Fig 4o-q** to Supplementary material.

5), By comparing the phosphorylation levels of all substrates of receptor tyrosine kinases (RTK), we evaluate the global activities of Sunitinib targeted RTKs. We found that the global activities of Sunitinib targeted RTKs showed no significant differences between Responders and Non-Responders. These results suggested that the abundances and activities of the targeted proteins might not be effective indicators for the TKI response. Kinase-substrate enrichment analysis (KSEA) was conducted to probe the differentially activated kinases between Responders and Non-Responders. We found that MAP2K1 (MEK1) and MTOR were activated in Non-Responders, while CDK1/2 were activated in Responders. Evaluation of kinase activities by ssGSEA further confirmed that MTOR was activated in Non-Responders while CDK2 was activated in Responders. The activities of MAP2K1 was significantly associated with poorer survival. In conclusion, our phosphoproteome revealed that exceeded mTOR and MAPK signaling pathways were associated with Sunitinib treatment resistance (**Figure 5**).

6), To further investigate the heterogeneity of ccRCC patients among response mechanisms and therapeutic targets, we used xCell to perform cell type deconvolution analysis. We identified three

subtypes of ccRCC. Consensus clustering based on the inferred cell proportion identified three sets of tumors defined as T-cell infiltrated, Cold, and Progenitor-cell infiltrated. Moreover, among the three subtypes, we observed higher Responder proportions in the T-cell infiltrated group and lower Responder proportions in the Progenitor-cell infiltrated group, which could be leveraged to predict therapeutic response (**Figure 6**).

7), To further apply our findings to clinical screening for ccRCC patients, we constructed multi-omics classifiers to distinguish the samples between Responders and Non-Responders. We used the ensemble random forest model algorithm to build the predictive model with the 18 proteins as the input features (ROC-AUC = 0.85). To construct an advanced classifier for improving the predicted performance, we enrolled the multi-omics features (clinical features, mutation features, mutational signatures, copy number alteration features, transcriptome features, and proteome features) rather than the only proteome features and built the random forest-based classifier (ROC-AUC = 0.98). The predictive model offers an opportunity to expedite translation of basic research to more precise diagnosis and treatment in the clinic (**Figure 7**). In summary, we used an ensemble approach that inputs multi-omics features to derive predictors of the TKI Responders. The predictive model offers an opportunity to expedite translation of basic research to more precise diagnosis and treatment in the clinic.

In summary, we firstly delineated the proteogenomic landscape of Sunitinib response in Chinese patients with ccRCC. We found that *VHL* mutation and the AA signature synergistically improved the clinical outcomes of Sunitinib treatment in Chinese patients with ccRCC. Multiple results repetitively showed that mTOR signaling was an intrinsic pathway for Sunitinib resistance. Our study further defined three immune subsets as T-cell infiltrated, Cold, and Progenitor-cell infiltrated, and showed that the Progenitor-cell infiltrated cluster was significantly correlated with Sunitinib resistance, which may be caused by activation of platelet signaling and secretion of TGF β 1. We summarized the features of Responders in multi-dimension. We also constructed a model for predicting the Sunitinib response and validate the robustness of the predictive model in an independent dataset. Overall, our multi-level omics analysis identified the molecular mechanisms underlying the Sunitinib response and defined the genomic, proteomic, and immune signatures to stratify patients with ccRCC to develop more rational therapeutic interventions.

As for the clarity of exposition, and better prioritization/integration of overabundant minor points

In the revised version, to represent the data and manuscript logically and readably, we streamlined the manuscript and re-arranged the figures and supplementary figures in the revised version. In addition, we moved 10 panels in the original manuscript to Supplementary Figures in the revision. In detail, in the original manuscript, we moved **Fig 1d, 1f, and 1g** panel to the Supplementary information; We moved the analysis about the occurrence of four CNA events (gains of 3q, 7p, 7q, and 8q) (**Fig 3b-3c**) to Supplementary information. The integrative analysis in the original **Fig 4a-4q**, revealed co-occurrence of the AA signature, 3p loss, and VHL mutation was also moved to Supplementary Figure. Additionally, in this round of revision, according to the reviewer's suggestion, to verify our conclusions in multiple dimensions, we also added additional omics hierarchical data evidence wherever applies. For example, the proteomic and transcriptomic data further highlighted the enhanced glycolysis and attenuated pentose phosphate pathway (PPP) in patients with AA signature (**Fig 2d** in the revised manuscript). In summary, we reorganized our story line and highlights, adjusted the structure of the article, and polished the language in revision. After revision, we believe that the manuscript would be more concise, understandable and focused.

ii) better integration of the relevant clinical, mutation, CNV, transcriptomic, proteomic and phosphoproteomic data for the studied cohort to be shared in a harmonized and easily digestible form with the community without infringing on privacy or IP issues;

Response: Thank you for your constructive comments and suggestions, which help to improve the quality of this manuscript. In order to improve the accessibility of the dataset and the usage of the predictive model, in this revision, we provided more details about the machine learning model including features, feature importance, model hyperparameters etc. These details could help readers better understand method and results in this study, and also facilitate other researchers to reproduce or improve our model in the future. Moreover, we also provided a detailed description and statistics of the dataset in the Supplementary Material including genome affections on the proteome data (VHL mutation, KMT2C mutation, copy number alteration), multi-omics alteration between Responders and non-Responders, the immune clustering, etc. Besides, in order to better integrating the relevant clinical and multi-omics data, we are building up a harmonized, interactive, and accessible cloud platform based on Firmiana (*Nature Biotechnology*, 2017, PMID: 28486446) to facilitate data sharing and benefit for the further research. This information makes it easier for the readers to access and use our dataset and model, and also increase the transparency and credibility of our research.

In addition, genome, transcriptome, proteome, and phosphoproteome data in our cohort were uploaded according to the requirements of the Nature Communications journal. In detail, the Proteome and phosphoproteome raw datasets have been deposited to the iProX partner repository (<https://www.iprox.cn/page/PSV023.html?url=1683359926130ZtHc>, with a password: BuPs) under Project ID: IPX0002932000. The raw WES and RNA data are available in the Genome Sequence Archive (GSA) under restricted access HRA003490. The user can register and login to the GSA database website (<https://ngdc.cncb.ac.cn/gsa-human/>) and follow the guidance of “Request Data” to request the data step by step (https://ngdc.cncb.ac.cn/gsa-human/document/GSA-Human_Request_Guide_for_Users_us.pdf). As publicly sharing of the raw genomic data is restricted by the regulation of the Human Genetic Resources Administration of China, detailed results of whole-exome sequencing were included in Supplementary Materials. The raw sequencing data are available for non-commercial purposes under controlled access because of data privacy laws, and access can be obtained by request to the corresponding authors. In addition, the clinical data and multi-omics data can be found in the **Supplementary Data 1, 2, and 3**. The more detailed information about the data shared form could be found in **the response of Q13**.

iii) detailed discussion of the proposed statistical and machine learning models for the prediction of responders vs. non-responders, including the analysis of feature importance to suggest a small set of proteins (and other features/predictors) that can be used to discriminate between responders vs. non-responders.

Response: Thanks for your advice about the machine learning model building of responders vs. non-responders. In this research, we used the step forward method which add features (from proteome to multi-omics) in the model to build the predictive classifier and similar practices have been used in previous study (*Nature*, 2022, PMID: 34875674). According to the reviewer’s suggestion, we have clearly described the predictive model construction pipeline including feature selection, model training, and performance evaluating etc. We have constructed a proteome-based model (logistic regression) with 18 proteins as the input features including CCDC132, COTL1, EIF3C, EPB41L3, GPR89C, HEATR3, HNRNPH3, HNRNPU, HOGA1, LAMTOR4, NBEAL2, NPM1, PMM1, RPS7, SMARCA5, SNRPE, TNS1, TRIO. The logistic regression model had AUC = 0.77. To increase the performance of the predictive model for this imbalanced cohort, the random forest (RF) model was constructed with AUC = 0.85. To improve the robustness of the model, according to the reviewer’s suggestion, we enrolled more multi-omics feature (clinical and genomic features) to build up a RF model. As a result, the model showed great performance

evaluated by different scores including AUC, balanced accuracy, precision, recall, and F1. The more detailed methods and results could be found **in the response of Q14**.

Q10. I would consider these changes minor, but essential. Minor in the sense that no new data need to be generated and no major methodological changes are required to further analyze these data to support conclusions. Essential, on the other hand, since without them the manuscript remains difficult to follow, data sets generated are hard to access and re-analyze, and the proposed prediction model used and tested directly, likely limiting the overall impact of the study.

Response: Thank you for your constructive suggestions. According to reviewers' constructive comments, firstly, we restructured the story line, removed the redundant information and modified the imprecise conclusions. Then, we prioritized and integrated overabundant minor points, making the manuscript more clearly and focused. At last, after compiling all the materials, we adjusted arrangement of all kinds of supporting materials, including figures, tables, et al., according to our story line to make the article more coherent and logical. Furthermore, in order to better integrate the relevant clinical and multi-omics data, we are building up a harmonized, interactive, and accessible cloud platform based on Firmiana to facilitate data sharing and benefit for the further research.

Reorganization of the manuscript to clarify our main contributions more understandable, clearer and focused

According to reviewer's suggestions, we have reconstructed the framework of the article. The storyline was now organized as below. In this study, we firstly delineated the proteogenomic landscape of Sunitinib response in Chinese patients with ccRCC (**Figure 1**). We found that *VHL* mutation and the AA signature synergistically improved the clinical outcomes of Sunitinib treatment in Chinese patients with ccRCC (**Figure 2,4**). Multiple results repetitively showed that mTOR signaling was an intrinsic pathway for Sunitinib resistance (**Figure 3, 5**). Our study further defined three immune subsets as T-cell infiltrated, Cold, and Progenitor-cell infiltrated, and showed that the Progenitor-cell infiltrated cluster was significantly correlated with Sunitinib resistance, which may be caused by activation of platelet signaling and secretion of TGF β 1 (**Figure 6**). We summarized the features of Responders in multi-dimension. We used an ensemble approach that inputs multi-omics features to derive predictors of the TKI Responders. The multi-omics random forest model showed good performance with ROC-AUC = 0.98 on the test cohort (**Figure 7**). Overall, our multi-level omics analysis identified the molecular mechanisms underlying the Sunitinib response and defined the genomic, proteomic, and immune signatures to

stratify patients with ccRCC to develop more rational therapeutic interventions. For the more detailed response of this question, please also see the response of the **Q9**.

As for the data-friendly presentation and the proposed prediction model

In order to improve the accessibility of the dataset and the usage of the predictive model, in this revision, we provided more details about the machine learning model including features, feature importance, model hyperparameters etc. These details could help readers better understand method and results in this study, and also facilitate other researchers to reproduce or improve our model in the future. Moreover, we also provided a detailed description and statistics of the dataset in the **Supplementary Material** including genome affections on the proteome data (*VHL* mutation, *KMT2C* mutation, copy number alteration), multi-omics alteration between Responders and non-Responders, the immune clustering, etc. This information makes it easier for the readers to access and use our dataset and model, and also increase the transparency and credibility of our research. The more detailed response of this question, please also see the response of the **Q9**.

Q11. Regarding the first point, multiple sections of the manuscript, individual paragraphs and sentences are very hard to follow. A perfect example is the following sentence in the Discussion section, lines 631-633, which reads: “There were 15 out of 19 patients carried AA signature with the count number ranged from 0 - 11 in the normal tissue, which lower than the 238 – 5,802 in the tumor tissue.” What are the ‘counts’ the authors are referring to in that sentence? Since the next sentence concerns mutations, one might presume these are mutation counts. If so, however, how 0-11 ‘counts’ found in adjacent normal tissue of 15 out 19 patients would qualify as a signature of AA? I believe that a wholesome editing by a professional scientific editor is warranted to improve the grammar, clarity, and overall exposition.

Response: We thank the reviewer for the comment, and apologize for not expressing it clearly.

The counts of AA signature were referring to the frequency of T>A transversions, because AA signature displayed predominant T>A transversions with conspicuous biases in the local sequence context, and this signature matched COSMIC SBS22 with the underlying etiological factor being aristolochic acid (AA). Therefore, the meaning of the sentence in lines 631-633 is that: there were 19 out of 115 patients in this cohort carried AA signature, among which 15 patients also carried AA signature in the normal tissue; however, the frequency of T>A transversions ranged from 0 – 11 in the normal tissue, which was far lower than the 238 – 5,802 in the tumor tissue. The founding revealed that the AA signature also affect the tumor adjacent tissue to a certain extent, which was consistent with Li et al. study reported that AA signatures were found in

morphologically normal human urothelium (MNU).

In general, this part of the **Discussion** section revealed that the AA signature might affect both tumor tissues and normal adjacent tissues in our cohort, while the frequency of T>A transversions was far less in the latter, suggesting that the AA signature might be the underlying mutagenic driving factor and was essential for understanding the development and evolution of ccRCC. According to reviewer's comments, we thoroughly checked the language errors in the revision. The entire manuscript was also edited by professional manuscript services. Thanks again for pointing it out.

Q12. Perhaps even more importantly, a better organization and prioritization/integration of overabundant minor points is required. I am in full agreement with Reviewer 5 who raised this issue before, and I still find the manuscript lacking in this regard. I would suggest focusing on main findings summarized in Figure 7, which summarizes the results of the proposed prediction model to discriminate between responders vs. non-responders using proteomics and other data, and shows clear trends in terms of the size of the tumor, PLT, 7q gain, VHL/KMT2C mutational status, immune subtypes, and proteomics-based pathway activation status.

Response: We sincerely thank the reviewer for the careful read and comments. As for the brevity and clarity, we refined our statements in the "**Results**", and "**Discussion**" to avoid unduly long and repetitive. In the "**Results**", we removed some overabundant stories. In detailed, to represent the data and manuscript logically and readably, we streamlined the manuscript and re-arranged the figures and supplementary figures in the revised version. In addition, we removed the redundant and unrelated information. The examples were shown as follows. In detail, in the original manuscript, we moved Fig 1d, 1f, and 1g panel to the Supplementary information; We moved the analysis about the occurrence of four CNA events (gains of 3q, 7p, 7q, and 8q) (Fig 3b-3c) to Supplementary information. The integrative analysis in the original Fig 4o-4q, revealed co-occurrence of the AA signature, 3p loss, and VHL mutation was also moved to Supplementary Figure. Additionally, in this round of revision, according to the reviewer's suggestion, to verify our conclusions in multiple dimensions, we also added additional omics hierarchical data evidence wherever applies. For example, the proteomic and transcriptomic data further highlighted the enhanced glycolysis and attenuated pentose phosphate pathway (PPP) in patients with AA signature (**Fig 2d** in the revised manuscript). In the revision, we focused on the main findings summarized in **Fig 7**, and the detailed information please see in **the response of Q9-Q10**. Thank the reviewer for the constructive suggestions.

Q13. These main findings and critical validation steps could then represent the main arc of the paper, while all other minor or non-central observations and analyses should be relegated to Supplementary Materials. I believe that one prime example of the latter is RNA-seq transcriptional profiling that does not seem to be central to the main conclusions and can be summarized in the main paper by a short statement of the overall correlation between mRNA and protein levels. Similarly, many methodological aspects can be briefly summarized in the main paper, while remaining details, such as details of RNA-seq library prep and sequencing, can be moved to Supplementary Data.

Response: Thank you for your professional suggestions. According to the reviewer's suggestion, we condensed all other minor or non-central observations and analyses as well as the content of *materials and methods*. The details are as follows.

As for the analysis of RNA-seq transcriptional profiling, we briefly stated that “the transcriptome and proteome showed moderate correlation with sample-wise median SCC of 0.39 in this study (**Fig. 1g**), consistent with the previous study” in **Result** part. Next, we streamlined the findings of miRNA in the **Discussion** part of in the revision. The details of these analyses were relegated to **Supplementary Materials**.

As for the content of *materials and methods*, we summarized many of them and moved remaining details to **Supplementary Data**. For example, we summarized the section of *RNA extraction and RNA-seq* as follows: “Total RNA from each tissue sample was isolated using TRIzol Reagent (Invitrogen). All RNA samples were assayed for the RNA purity and integrity. After RNA samples were qualified, the RNA was reverse-transcript into cDNA and constructed library and conducted sequencing. The clustering of the index-coded samples was performed on a cBot Cluster Generation System using TruSeq PE Cluster Kit v3-cBot-HS (Illumina), according to the manufacturer's instructions. After cluster generation, the libraries were sequenced on an Illumina HiSeq 4000 platform and 125 bp paired-end reads were generated. After trimming the adaptors and removing low-quality tags, sequencing reads were mapped onto the hg19. The mapped reads were assembled into transcripts or genes by using StringTie software (v2.1.4). For quantification purpose, the relative abundance of the transcript or gene was measured by a normalized metrics, FPKM (Fragments Per Kilobase of transcript per Million mapped reads). Transcripts with median FPKM > 1 were retained (**Supplementary Data 3**).”

We carefully checked all the non-central contents, and condensed them to make the manuscript shorter and more readable. Thank the reviewer again to pointing them out.

Q14. On the second point, I can appreciate privacy and IP related constraints, especially with regards to WES and the final prediction model. However, since only *VHL* and other frequent mutations are used for the prediction of putative responders, the WES data would not need to be shared. Similarly, any sensitive clinical data could be removed, while providing the most relevant variables, such as the tumor size. I believe that once stripped of sensitive information, the data should be made publicly available following the TCGA model of multi-domain data and meta-data integration, ideally through a user-friendly interface. Since this intersects with editorial policies regarding data sharing, I defer here to the editor.

Response: We thank the reviewer for the kindly comment and suggestions. Here, we divided the response into two parts as follows.

As for WES and sensitive clinical data

In the analysis of WES data, it is true that only *VHL* and other frequent mutations were used for our subsequent story lines. In accordance with the requirements of the *Nature Communications*, we still uploaded the all WES related data to NGDC. In the revision, according to the reviewer's suggestions, we removed sensitive clinical data that may infringe the privacy of patients, such as date of birth, height, weight etc. and retained only the variables most relevant to data analysis, such as gender, tumor size, TNM stage and so on (**Supplementary Data 1**). Importantly, all data provided in this study were in strict compliance with legal regulations and requirements of *Nature Communications*.

As for user-friendly data sharing

To make the data public friendly, we deposited the data of this study on a cloud platform, named Firmiana (*Nat Biotechnol*, 2017, PMID: 28486446). This is a one-stop proteomic data processing and integrated omics analysis cloud platform developed by our laboratory, that allows scientists to deposit mass spectrometry (MS) raw files, perform proteome identification and quantification online, carry out bioinformatics analyses, extract knowledge, and visualize results using a biologist-friendly web interface without the need for programming expertise. Furthermore, in addition to the data related to this study, the data of all other tumor multi-omics studies published by our laboratory were also stored on this platform, involving a variety of malignant tumors and diseases, such as pituitary neuroendocrine tumors (*Cell Res*, 2022, PMID: 36307579), urothelial

carcinoma (*J Hematol Oncol*, 2022, PMID: 35659036), cholangiocarcinoma (*Hepatology*, 2023, PMID: 35716043), diffuse gliomas (*Nature Communications*, 2023, PMID: 36720864), early duodenal cancer (*Nature Communications*, 2023, PMID: 36991000), esophageal cancer (*Nature Communications*, 2023, PMID: 36966136) and so on. Ultimately, we anticipated to provide an online resource for the proteomics and biology research community to take advantage of proteomics in the big data era. Thank you again for your professional suggestions.

Q15. On the last point, the manuscript does not provide sufficient details regarding the proposed prediction models. In particular, ‘the linear model’ referred to in line 591 is never defined. It is mentioned again in the methods sections (line 1136) but without a definition, searching Supplementary Materials for ‘linear model’ finds no hits. There are many types of linear classifiers, linear perceptron or linear (L1 norm) SVM being two well known examples. So, assuming it is a linear model, what is the classification algorithm that is used, what are the input features, and since these are apparently derived from protein abundance levels for ‘high confidence proteins’ are those the same for all samples, or are they sample specific? Is the reference set used to define the ‘outliers’ always the same? How does this affect feature selection in 5-fold cross-validation? For the random forest-based classifier, is the set of ‘highly confident proteins’ selected in each run of the 5-fold cross-validation? What are the most informative proteins based on the feature importance analysis in random forest? What are other predictors/features that contribute to the multi-omics model?

Response: Thanks for the comment. We apologize for the typo of “linear classification” in line 591 and the unclear statements of the machine learning model part. To response the question clearly, we split this question in two parts.

For the feature selection pipeline and the result of the predictive model

In order to build the proteome-based predictive model on this imbalanced proteome (88 non-responders and 27 Responders), we used two different models including the linear model (logistic regression) and the ensemble bagging (random forest) on this TKI therapy cohort.

For the feature selection, based on the model construction pipeline (**details as the following response part**), we chose a more reasonable and commonly used feature selection method F test and chi-square test in this revision to select the features on the train cohort (**response to the point of the detail of the feature selection part including 5-fold cross-validation**). As a result, the 18 proteins which showed the robust performance were selected for the predictive model construction

including CCDC132, COTL1, EIF3C, EPB41L3, GPR89C, HEATR3, HNRNPH3, HNRNPU, HOGA1, LAMTOR4, NBEAL2, NPM1, PMM1, RPS7, SMARCA5, SNRPE, TNS1, TRIO **(response to the point of the input feature of the “linear model” and “random forest model”)**.

After the feature selection, we used the logistic regression (LR) to construct the predictor **(response to the point of the “linear model” definition)**. As a result, the LR model showed good performance with ROC-AUC = 0.88 on the train cohort, but with ROC-AUC = 0.77 on the test cohort **(Figure RL 7A, H)**. Notably, the sensitivity is 0.625 and the specificity is 0.72 **(Figure RL 7D)**, which indicated the LR model does not have the best performance to handle the imbalance in this cohort.

To enhance the predictive model performance on this imbalanced cohort, next, we used the ensemble random forest (RF) model algorithm to build the predictive model **(as described in the last version, but typo as “linear classification” in line 591)** with the above 18 proteins as the input features. As shown in **Figure RL 7B**, the ROC plot showed the ROC-AUC of the repeated cross validation on the train cohort. The sensitivity and specificity of the test cohort was 0.75 and 0.85, separately **(Figure RL 7E)**. This result showed the advantages of RF model than the LR in this cohort. Furthermore, we calculated the feature importance for this RF model **(Figure RL 7G)**. As a result, the RPS7, SMARCA5, and HNRNPH3 showed higher feature importance **(response to the point of the most informative proteins based on the feature importance analysis in random forest)**.

To construct the advanced classifier for improving the predicted performance, according to the reviewer’s suggestion, we added the clinical and genomic features as the input of the multi-omics model in the revised manuscript. In detail, there are clinical features (LDH, tumor size, and PLT), mutation features (*VHL* and *KMT2C*), mutational signature features (AA signature), copy number alteration features (7q, 3p), transcriptome features (MIR3939, ALDH1A3, LPAR1, FBLN5, C7), and proteome features (same as the proteome-based model) were selected for the multi-omics classifier construction (response to the point of the other predictors/features that contribute to the multi-omics model). As shown in **Figure RL 7C**, the multi-omics classifier showed ROC-AUC = 0.86 for the repeated cross-validation on the train cohort. The sensitivity and specificity of the test cohort was 1 and 0.90 with the great improvement than the proteome-based model, separately **(Figure RL 7F)**. By comparing the ROC-AUC of the test cohort among the proteome-based LR model, proteome-based RF model, and multi-omics RF model, the multi-omics RF model with the

AUC-ROC = 0.98 which greater than the other models (**Figure RL 7H**). Furthermore, the balanced accuracy, ROC-AUC, precision, recall, and F1 score were evaluated on the test cohort. The result showed the good generalized performance on the test cohort (**Figure RL 7I**). The feature importance of the multi-omics RL model indicated the importance of proteome and transcriptome in the prediction processing (**Figure RL 7J**). These results of the predictive model were updated in **Results** parts of the revised manuscript.

Figure RL 7 A-C. The five repeated five cross validated ROC on the train cohort for proteome-based linear regression (LR), proteome-based random forest (RF), multi-omics-base RF, separately. **D-F.** The confusion matrix of test cohort for proteome-based LR, proteome-based RF, and multi-omics-base RF, separately. **G.** The feature importance of proteome-based RL model. **H.** The comparison of AUC on the test cohort for proteome-based LR, proteome-based RF, and multi-omics-base RF. **I.** Bar plot depicting the different evaluation score of multi-omics-base RF model including balanced accuracy, ROC AUC, precision, recall, and F1 score. **J.** The feature importance of multi-omics-based RF model. The blue, orange, green, red rectangle indicated proteome, transcriptome, clinical, and genomic features, respectively.

For the detail of the machine learning construction framework

In this research, we used the step forward method which add features in the model to build the predictive classifier (from proteome to multi omics) (*Nature*, 2022, PMID: 34875674). To illustrate the model construction framework clearly, we split this part into three parts as Feature selection, Model construction, and model performance evaluation. The machine learning

framework was built on Python (version 3.9.0) using the following libraries: scikit-learn (version 1.2.1), numpy (version 1.16.4), scipy (version 1.3), pandas (version 1.5.2).

Feature selection. For the feature selection of the predictive model, we applied the following pipeline: the first step removed all features with a mutual Pearson correlation above 0.8, retaining only the one with the highest correlation with the response variable. The second step used the chi-square test, and F test which embedded in SelectKBest function of scikit-learn library to select the best proteome features. The third step applied z-score scaling to the remaining features.

Model construction. Considering the model complexity and the interpretability, we chose the logistic regression and random forest model to construct the predictive model. We split the dataset into train cohort (70%) and test cohort (30%). Hyperparameters were optimized using five-fold cross-validation in the training set to maximize the area under the receiver operating characteristic (AUC ROC) curve. As for the proteome-based model, after applying the model construction pipeline, logistic regression was implemented with l2 regularization and lbfgs solver, with C parameter was 8.5. Random forest model was implemented with class_weight was balanced, max_features was 1, min_samples_leaf was 14, min_samples_split was 14, n_estimators was 150. As for the multi-omics-based random forest model class_weight was balanced, max_features was 2, min_samples_leaf was 8, min_samples_split was 5, n_estimators was 90.

Model performance evaluation. The model performance was evaluated by the different evaluation scores including precision, recall, ROC AUC, F1 score, and balanced accuracy for the classification. These indispensable description of predictive model methods were updated in **Materials and Methods** parts of the revised manuscript.

Q16. Since random forest is an ensemble of decision trees, it would be of interest to the community and important for clinical applications to extract a simple rule-based model that combines the size of the tumor, PLT, 7q gain, VHL/KMT2C mutational status, immune subtypes, and proteomics-based pathway activation status to predict responders vs. non-responders. The sentence in lines 605-606 reads: “In detail, the multiomics feature combination can distinguish the response to TKI with >95% sensitively/specificity (Fig. 7f).” Yet, the inspection of the figure indicates that about 50% sensitivity is achieved for 95% specificity. Incidentally, this is where the multi-omics model shows better performance compared to ‘linear model’ in terms of TPR at 0% FPR, which is arguably more important than the overall AUROC, especially in this relatively

imbalanced classification problem.

Response: Thank you for your comments. The imbalanced dataset was a challenging problem in this study. To handle this problem, we applied a proteome-based and a multi-omics-based random forest algorithm. In this revision, as shown in **Figure RL 8A**, the multi-omics model outperformed proteome-based the ‘linear model’ logistic regression model and random forest model in terms of TPR at 0% FPR (TPRs were 0.25, 0.25, 0.5 for proteome LR, proteome RF, and multi-omics RL, separately). To better evaluate the performance of the multi-omics model in dealing with the imbalanced dataset, we computed the balanced accuracy, precision, recall, and F1 score, separately. As a result (**Figure RL 8B**), the multi-omics model exhibited good performance. These results demonstrate that our method can effectively utilize multi-omics data to improve the recognition and classification of Responders, and provide valuable insights for clinical diagnosis and treatment.

Figure RL8 A. The comparison of AUC on the test cohort for proteome-based LR, proteome-based RF, and multi-omics-base RF. **B.** Bar plot depicting the different evaluation score of multi-omics-base RF model including balanced accuracy, ROC AUC, precision, recall, and F1 score.

Some other minor points:

Q17. Please harmonize the often referred to Spearman correlation coefficient by introducing SCC early on and using it consistently throughout the paper (as opposed to 'Spearman's'), provide the values of SCC consistently when referring to it, rather than just providing p-values.

Response: Thanks for the comments. According to the reviewer’s suggestion, we referred to Spearman correlation coefficient by abbreviation SCC in the revision. In describing the correlation, we provide both the correlation coefficient and the significance p-value.

For example as follows. Line 127, “The transcriptome and proteome showed moderate correlation with sample-wise median Spearman correlation coefficient (SCC) of 0.39 in this study”. Line 180, “To elucidate the biological basis, we calculated SCC of clinical features and ssGSEA scores of pathways in the Hallmark, KEGG, and Reactome databases”. Line 183-189, “As a result, the proteomic pathways, complement and coagulation cascades (SCC = 0.24, $p < 0.05$), epithelial-mesenchymal transition (EMT) (SCC = 0.23, $p < 0.05$), angiogenesis (SCC = 0.18, $p < 0.05$), and inflammatory response (SCC = 0.21, $p < 0.05$) were positively correlated with tumor size”. Line 203, “Furthermore, we found that CRP abundance was significantly correlated with inflammatory response scores (SCC = 0.31, $p = 8.1E-4$)”.

Q18. Please harmonize and refer consistently to mRNA vs. protein vs. phosphoprotein data wherever applies.

Response: Thanks for the comments. In this study, we mainly elucidated the molecular basis of differential clinical outcomes with tyrosine kinase inhibitor (TKI) therapy from proteomic level. To verify our conclusions in multiple dimensions, we also added additional omics hierarchical data evidence. For example, the proteomic and phosphoproteomic data further highlighted the responsibility of mTOR signaling for non-response to Sunitinib. We added evidence of integrated analysis of multiple omics data in the revision (**Figure RL 9**).

Figure RL9 **A.** Glucose metabolism alteration caused by the AA signature at transcriptome and proteome level separately. **B.** The boxplots depicting the distribution of LAMTOR4, MDH2, and

CALU between 7q gain and non 7q gain cohort at transcriptome and proteome level, separately. **C.** Differentially level of RNAs and proteins between VHL mutation and WT groups. The values were transformed by z-score. **D.** The distribution of MTOR and CDK2 protein abundance between R and NR group. P-values were derived by the ranksums test.

Q19. The following lines contain unclear passages, bad grammar or other issues that need to be fixed (too many to list specific issues in each):

181, 184, 187, 189, 227, 230, 234, 260, 261, 262-267, 272, 285, 304, 309, 311, 321, 330, 340-343, 361, 366, 374, 379-382, 390, 462, 490-495, 500, 506, 514, 539, 557, 571, 591, 604-606, 610, 616, 621, 622-623, 625, 745-746.

Response: Thanks for the comments. We're sorry for any unclear descriptions and bad grammar in our previous manuscript. We have sent our manuscript to professional proofreading agency to revise the text to avoid typos and grammatical errors. The detailed information as follows.

Line 181 were revised from “we calculated Spearman’s correlations of clinical features and ssGSEA scores of pathways in the Hallmark, KEGG, and Reactome databases” to “we calculated SCC of clinical features (tumor size, PLT and LDH) and ssGSEA scores of pathways terms”.

Line 184 were revised from “As a result, the proteomic pathways, complement and coagulation cascades, epithelial-mesenchymal transition (EMT), angiogenesis, and inflammatory response were positively correlated with tumor size (Spearman’s correlation, $p < 0.05$)” to “As a result, the proteomic pathways, complement and coagulation cascades (SCC = 0.24, $p < 0.05$), epithelial-mesenchymal transition (EMT) (SCC = 0.23, $p < 0.05$), angiogenesis (SCC = 0.18, $p < 0.05$), and inflammatory response (SCC = 0.21, $p < 0.05$) were positively correlated with tumor size”.

Line 187 were revised from “inflammatory response, innate immune system, neutrophil degranulation, and RAB regulation of trafficking were positively correlated with the PLT level (Spearman’s correlation, $p < 0.05$).” to “inflammatory response (SCC = 0.21, $p = 0.02$) were positively correlated with tumor size. Inflammatory response (SCC = 0.22, $p = 0.02$), innate immune system (SCC = 0.27, $p < 0.01$), neutrophil degranulation (SCC = 0.27, $p < 0.01$), and RAB regulation of trafficking (SCC = 0.25, $p < 0.01$) were positively correlated with the PLT level.”

Line 189 were revised from “Energy dependent regulation of mTOR by LKB-AMPK, and fatty acid metabolism were positively correlated with the LDH level (Spearman’s correlation, $p < 0.05$)” to “Energy dependent regulation of mTOR by LKB-AMPK (SCC = 0.24, $p < 0.01$), and fatty acid metabolism (SCC = 0.24, $p < 0.01$) were positively correlated with the LDH level”.

Line 227 were revised from “Despite there was no difference of MSKCC risk and IMDC risk” to “Despite there was no significant difference of MSKCC risk and IMDC risk”.

Line 230 were revised from “Gene set enrichment analysis (GSEA) revealed that poor MSKCC risk tumors showed elevated...” to “Gene set enrichment analysis (GSEA) revealed that poor MSKCC risk patients showed elevated...”.

Line 234 were revised from “downregulated glycolysis and apoptosis was observed in poor IMDC risk tumors” to “downregulated glycolysis and apoptosis was observed in poor IMDC risk patients”.

Line 260 were revised from “were upregulated in tumors with the AA signature” to “were upregulated in tumors with the AA signature compared with the tumors without AA signature”.

Line 261 were revised from “The downregulated DNA replication might accountable for the smaller tumor size of tumors with AA signature.” to “The downregulated DNA replication might accountable for the smaller tumor size of pathents with AA signature”.

Line 262-267 were revised from “Interestingly, we found that the enhanced glycolysis and attenuated pentose phosphate pathway (PPP) in tumors with AA signature, manifested as upregulation of PFKP, PFKL, GPI (t-test, $p < 0.05$) and downregulation of G6PD, PGD, TKT (t-test, $p < 0.05$) (Fig. 2d). Further, downregulation of PGD and TKT was associated with improved survival in Sunitinib treatment group, but not in the non-treated control group” to “Interestingly, we found that the enhanced glycolysis and attenuated pentose phosphate pathway (PPP) in patients with AA signature, manifested as upregulation of PFKP, PFKL, GPI (t-test, $p < 0.05$) and downregulation of G6PD, PGD, TKT (t-test, $p < 0.05$) both on proteome and transcriptome level (Fig. 2d). Further, low expression of PGD and TKT was associated with better survival in Sunitinib treatment group, which was not observed in the non-treated control group”.

Line 272 were revised from “We found that AA treatment did not impact cell proliferation directly, but enhanced the inhibition to cell proliferation of Sunitinib” to “We found that in vitro treatment of ccRCC cells with proper concentration of AA had no direct effect on the proliferation of ccRCC cells, but could enhance the inhibitory effect of sunitinib on ccRCC cells”.

Line 285 were revised from “To improve the confidence of predicted TFs, we took the intersection of the two databases and obtained 18 most likely TFs” to “To improve the confidence of predicted TFs, we took the intersection of the 84 TFs in JASPAR and 200 TFs in ChEA3 databases.”.

Line 304 were revised from “indicating AA led to SP1 degradation in the post-translation stage” to “might indicate AA led to SP1 degradation in the post-translation stage”.

Line 309 were revised from “The results showed there’s no significance between R and NR” to “The results showed there’s no significance difference between R and NR”.

Line 330 were revised from “We found the occurrence of these four CNA events (gains of 3q, 7p, 7q, and 8q), and 7p gain and 7q gain were even totally overlapped” to “We found the occurrence of these four CNA events (gains of 3q, 7p, 7q, and 8q). Notably, 7p gain and 7q gain were even totally overlapped”.

Line 340-343 were revised from “To find out the effects of 7q copy number (CN) on sunitinib therapy at proteome level, we calculated the spearman’s correlation of 7q CN and proteome data. Totally, there were 485 proteins showed significantly positive correlations with the 7q copy number (CN) (Spearman’s correlation, $p < 0.05$).” to “To find out the effects of 7q copy number (CN) on sunitinib therapy at proteome level, we calculated the SCC of 7q CN and proteome data. Totally, there were 485 proteins showed significantly positive correlations with the 7q CN ($SCC > 0$, $p < 0.05$)”.

Line 361 were revised from “increased the phosphorylation of S6K” to “increased the phosphorylation level of S6K”.

Line 366 were revised from “These results indicated the 7q gain would activate mTOR signaling in ccRCC patients.” to “These results indicated the 7q gain was associated with the activation of mTOR signaling in ccRCC patients”.

Line 374 were revised from “In conclusion, chromosome 7q gain would activate mTOR signaling and lead to poor Sunitinib treatment effectiveness” to “In conclusion, chromosome 7q gain would activate mTOR signaling and link to the poor Sunitinib treatment effectiveness”.

Line 379-382 were revised from “In total, we identified ten genes (VHL, KMT2C, SFT2D1, COL5A3, DYNC2H1, EYS, STAG3, HEATR1, PABPC5, and HLA-B) that were significantly differentially altered in the Responder and Non-Responder groups (Fisher’s exact test, $p < 0.05$). Interestingly, all ten genes showed higher” to “In total, we identified ten genes (VHL, KMT2C, SFT2D1, COL5A3, DYNC2H1, EYS, STAG3, HEATR1, PABPC5, and HLA-B) that were significantly differentially mutated in the Responder and Non-Responder groups (Fisher’s exact test, $p < 0.05$). Interestingly, all ten genes showed higher”.

Line 390 were revised from “An important finding was that VHL mutation, considered as a truncal genetic alteration even, was significantly associated with good prognostic outcomes in Sunitinib treatment” to “Notably, we found that VHL mutation, considered as a truncal genetic alteration event, was significantly associated with good prognostic outcomes in Sunitinib treatment”.

Line 490-495 were revised from “Further investigation into the differentially expressed phosphosites between Responders and Non-Responders, it was showed that mTOR and MAPK signaling pathways associated substrates phosphorylation were elevated in Non-Reponders, such

as DEPTOR at S265, MAPK1(ERK2) at Y187, while SNRNP70 at S226 which was involved in RNA splicing was increased in Responders (Supplementary Fig. 8c)” to “Furthermore, we performed the differentially expressed phosphosites analysis between Responders and Non-Responders. As a result, the mTOR and the MAPK signaling pathways associated substrates phosphorylation were elevated in Non-Responders, such as DEPTOR at S265, MAPK1 (ERK2) at Y187, while SNRNP70 at S226 which was involved in RNA splicing was increased in Responders (Supplementary Fig. 8c)”.

Line 500 were revised from “In conclusion, our phosphoproteome revealed that exceeded mTOR and MAPK signaling pathways were associated with Sunitinib treatment resistance” to “In conclusion, our phosphoproteome revealed that excessive mTOR and MAPK signaling pathways were associated with Sunitinib treatment resistance”.

Line 506 were revised from “Consensus clustering identified three proteomic subtypes with distinct features, containing 51, 37, and 27 patients respectively (Supplementary Fig. 9a), reflecting the intertumoral heterogeneity of ccRCC” To “We performed the consensus clustering algorithm to identify three proteomic subtypes with distinct features, containing 51, 37, and 27 patients respectively”.

Line 571 were revised from “In summary, we considered that the TME, comprising abundant progenitors, led to insufficient Sunitinib therapy response by platelet activation and TGFBI signaling” to “In summary, we considered that the connection of platelet activation with TME, comprising abundant progenitors, which might lead to insufficient Sunitinib therapy response”.

Line 604-606 were revised from “In detail, the multi-omics feature combination can distinguish the response to TKI with >95% sensitivity/specificity” to “In detail, the multi-omics based on RF model showed >95% sensitivity/specificity on the test cohort”.

Line 610 were revised from “Mutation signatures revealed that the AA signature, generated by ingestion of herbs containing aristolochic acid, was associated with smaller tumors and better survival” to “By analyzing the mutation signature, we observed AA signature, generated by ingestion of herbs containing aristolochic acid, was associated with smaller tumor size and better survival outcomes”.

Line 616 were revised from “Here, based on the multi-omics data and bioinformatical analysis, in addition to the genomic impact of AA, we described the proteomic differences between AA patients and non-AA patients, and observed attenuated pentose phosphate pathway and enhanced glycolysis alteration which connected to the Sunitinib treatment” to “Here, in addition to observing the genomic impact of AA, we illustrated the multi-omics alterations between AA patients and non-AA patients and concluded the attenuated pentose phosphate pathway and

enhanced glycolysis alteration which connected to the Sunitinib treatment”.

Line 621 were revised from “These results indicated AA not only affect the mutation at genomic level, but also alter the cell signaling pathway at proteomic level” to “These results indicated AA not only affect the mutation at genomic level, but also bring about changes at proteomic level.”.

Line 622-623 were revised from “Furthermore, it is still uncertain that the connection between two events at genomic and proteomic level” to “Furthermore, it is still unclear about the contribution of mutation signature by AA in the multi-omics alteration”.

Line 625 were revised from “There were 19 patients carried the SBS22 signature in our cohort” to “In this cohort, 19 out of 115 patients carried AA signature which defined in the COSMIC SBS22”.

Line 745-746 described as “To our knowledge, the classifier presented in our research is the first model constructed based on large-scale proteome data” were removed in this revision.

REVIEWERS' COMMENTS

Reviewer #1 (Remarks to the Author):

I have no further comments.

Reviewer #4 (Remarks to the Author):

The authors have sufficiently addressed the remaining points I had raised.

Reviewer #7 (Remarks to the Author):

I believe that the manuscript by Zhang et al. presents noteworthy results, the methodology appears to be sound, and should be published to inform future studies of ccRCC.

The revised manuscript addresses my concerns and comments, especially with regards to machine learning models. Logistic regression and Random Forest models were apparently retrained using a more rigorous feature selection process that separates training and control sets, and evaluated in a more comprehensive way with multiple accuracy metrics and feature importance analysis. The advantage of using multi-omics models is now clear.

While the overall exposition and grammar have been improved quite a bit, I still find many awkward sentences and unclear statements.

Here is one example:

"The feature importance result indicated the more contribution of proteome and transcriptome in the prediction processing (Fig. 7f)."

I think this sentence could be rephrased as follows:

"Proteomic and transcriptomic features, i.e., protein/phosphoprotein abundance and gene expression levels, contribute most to the success of the prediction model, as revealed by the feature importance analysis included in Fig 7f."

Many other instances like that suggest that another round of professional editing would be desirable. This is a minor point though, and I leave it to the discretion of the editorial team.

Reviewer #1 (Remarks to the Author):

I have no further comments.

Response: Thank you for your positive comments and suggestions!

Reviewer #4 (Remarks to the Author):

The authors have sufficiently addressed the remaining points I had raised.

Response: Thank you for your positive comments and suggestions!

Reviewer #7 (Remarks to the Author):

I believe that the manuscript by Zhang et al. presents noteworthy results, the methodology appears to be sound, and should be published to inform future studies of ccRCC. The revised manuscript addresses my concerns and comments, especially with regards to machine learning models. Logistic regression and Random Forest models were apparently retrained using a more rigorous feature selection process that separates training and control sets, and evaluated in a more comprehensive way with multiple accuracy metrics and feature importance analysis. The advantage of using multi-omics models is now clear.

While the overall exposition and grammar have been improved quite a bit, I still find many awkward sentences and unclear statements. Here is one example: "The feature importance result indicated the more contribution of proteome and transcriptome in the prediction processing (Fig. 7f)." I think this sentence could be rephrased as follows: "Proteomic and transcriptomic features, i.e., protein/phosphoprotein abundance and gene expression levels, contribute most to the success of the prediction model, as revealed by the feature importance analysis included in Fig 7f." Many other instances like that suggest that another round of professional editing would be desirable. This is a minor point though, and I leave it to the discretion of the editorial team.

Response: Thank you for your positive comments and suggestions! We have rephrased the sentence from "The feature importance result indicated the more contribution of proteome and transcriptome

in the prediction processing (Fig. 7f)” as “Proteomic and transcriptomic features, i.e., protein/phosphoprotein abundance and gene expression levels, contribute most to the success of the prediction model, as revealed by the feature importance analysis included in Fig. 7f” as reviewer suggested in the revised manuscript. According to reviewer’s comments, we thoroughly checked the language errors in the revision. The entire manuscript was also edited by professional manuscript services. Thanks again for pointing it out.